# A Formal Framework for Understanding Length Generalization in Transformers

**Xinting Huang**[1]* **Andy Yang**[2]* **Satwik Bhattamishra**[3] **Yash Sarrof**[1]
**Andreas Krebs**[4] **Hattie Zhou**[5] **Preetum Nakkiran**[6] **Michael Hahn**[1]†
[1]Saarland University [2]University of Notre Dame [3]University of Oxford
[4]University of Tübingen [5]Mila, Université de Montréal [6]Apple

## Abstract

A major challenge for transformers is generalizing to sequences longer than those observed during training. While previous works have empirically shown that transformers can either succeed or fail at length generalization depending on the task, theoretical understanding of this phenomenon remains limited. In this work, we introduce a rigorous theoretical framework to analyze length generalization in causal transformers with learnable absolute positional encodings. In particular, we characterize those functions that are identifiable in the limit from sufficiently long inputs with absolute positional encodings under an idealized inference scheme using a norm-based regularizer. This enables us to prove the possibility of length generalization for a rich family of problems. We experimentally validate the theory as a predictor of success and failure of length generalization across a range of algorithmic and formal language tasks[1]. Our theory not only explains a broad set of empirical observations but also opens the way to provably predicting length generalization capabilities in transformers.

## 1 Introduction

A key problem in neural sequence modeling is generalization from shorter sequences seen during training to longer sequences afterwards – *length generalization*. A wide range of empirical research has found that transformers sometimes succeed and sometimes fail at length-generalization (e.g. Bhattamishra et al., 2020; Anil et al., 2022; Wang et al., 2024a; Kazemnejad et al., 2023; Zhou et al., 2024b; Awasthi & Gupta, 2023; Jelassi et al., 2023; 2024; Chang & Bisk, 2024) with no theoretical explanation as to why. For instance, while transformer decoders can easily copy long strings (Bhattamishra et al., 2024), length generalization is substantially more brittle: Transformers trained to copy short strings often do not generalize well to longer strings when the input string includes repeated substrings (Zhou et al., 2024a; Jelassi et al., 2024). Similarly, while transformers can in principle simulate many finite-state automata (Liu et al., 2023), their success at length generalization in practice varies widely across different automata (Liu et al., 2023; Bhattamishra et al., 2020). Theoretical understanding of these phenomena is largely lacking, making it difficult to anticipate on which problems transformers will succeed or fail at length-generalization.

An important step towards theoretical understanding was made in the RASP-L Conjecture (Zhou et al., 2024a). This conjecture states that transformers are likely to length-generalize exactly on those algorithmic tasks that can be solved by simple programs in RASP-L, a fragment of the RASP language (Weiss et al., 2021) with substantial restrictions on the ways in which positional information can be used. While Zhou et al. (2024a) provided empirical evidence in support of this idea, two important gaps remain: First, while compelling empirical evidence supports a link between definability in RASP fragments and length generalization, no formal proof exists. Second, the RASP-L language has not been fully formalized, so it is largely open how to prove that a certain problem is not representable in it.

---

*XH and AY are co-first authors.

†Lead senior author. Contact: mhahn@lst.uni-saarland.de

[1]Code is available at https://github.com/lacoco-lab/length_generalization

We address both these gaps by presenting a general theoretical framework analyzing length generalization as ultimate identifiability in the limit for a well-defined class of functions: When the input-output behavior of a given function is observed at increasing input lengths, can a learner converge on inferring the ground-truth function? We give a positive answer for a specific idealized learning strategy: in this setting, transformers are guaranteed to length generalize given any function in the aforementioned class. Later, we rigorously characterize the expressivity of this class.

We define an idealized inference procedure in which a sequence of transformers is fitted to reproduce a target function on successively longer inputs while minimizing a specific norm-based regularizer, producing an infinite sequence of transformers operating on longer and longer context windows. Our results provide conditions under which the computations represented by these transformers converge to a single underlying algorithm reproducing the target function. In this case, generalization is guaranteed for sufficiently long inputs.

Our results apply to multilayer transformers, focusing on causal transformers trained from scratch with absolute positional encodings (APE) or without positional encodings (NoPE). A key technical challenge in analyzing length generalization for absolute positional encodings is the scaling of the transformer's parameter count with the input length. To address this, we define a transformer-like limiting object, the *Limit Transformer*, which encapsulates the computations of a sequence of transformers operating on longer and longer inputs into a single object. Our main theoretical result states that the inference procedure will ultimately lead to length generalization for sufficiently long training inputs, provided the ground-truth function is expressible by a single Limit Transformer across all input lengths:

**Theorem 1** (Informal Version of Theorem 7). *Let $f$ be the target function expressible by a single Limit Transformer at all input lengths, subject to restrictions on the use of positional information. Choose transformers $T_n$ ($n = 1, 2, 3, \dots$) with context size $n$, where $T_n$ reproduces the behavior of $f$ up to length $\frac{n}{2}$, while minimizing a norm-based regularizer. Then, for large $n$, $T_n$ will match the output of the target function $f$ up to length $\leq n$.*

We then show that the expressivity of Limit Transformers can be understood for many previously studied algorithmic tasks, as well as new ones. We extend a recently introduced RASP variant (**C-RASP**, Yang & Chiang, 2024) to provide lower bounds, showing that, under the idealized inference procedure, transformers will succeed at length generalization on various concrete problems. Conversely, we employ communication complexity to obtain upper bounds on the class of functions for which length generalization is predicted. While we have not proven this class is complete, experiments confirm that the theory predicts both success and failure at length generalization across various algorithmic tasks and formal languages. Overall, our results formalize the RASP-L Conjecture and take a step toward a theoretical understanding of length generalization.

## 2 MODEL OF TRANSFORMERS

**Positional Encoding Scheme**   We study two positional encoding schemes. One uses no positional encoding at all; we refer to this as NoPE (No Positional Encoding). The other one uses Absolute Positional Encodings (APE), with learned per-position embedding vectors $\boldsymbol{p}_1, \dots, \boldsymbol{p}_N$. We follow Zhou et al. (2024a) in requiring transformers to be able to perform a task at different offsets within a longer context. Whereas Zhou et al. (2024a) concatenated different examples of a task, we simply encode an input $x$ of length $|x| = k \leq N$ using positional encodings $\boldsymbol{p}_{1+o}, \dots, \boldsymbol{p}_{k+o}$ where $o$ is an offset such that $k + o \leq N$, and require that the transformer correctly performs the task independently of the offset $o \geq 0$. This mimics the computations in language models, where the same reasoning task can typically appear at different places in a long context. For simplicity, we treat positions outside of the input, including those preceding the offset, as empty.

**Parameterization**   We focus on transformers with causal masking; for simplicity, we will use the term "transformer" for these throughout. A transformer $T$ is parameterized by a finite alphabet $\Sigma$, a width $d \in \mathbb{N}$, a token embedding matrix $\boldsymbol{E} \in \mathbb{R}^{|\Sigma| \times d}$, a context width $N(T) \in \mathbb{N} \cup \{+\infty\}$, positional encodings $\{\boldsymbol{p}_i \in \mathbb{R}^d : 1 \leq i < N(T) + 1\}$, a depth $L \in \mathbb{N}$, head count $H \in \mathbb{N}$, key, query, and value matrices $\{\boldsymbol{K}_{l,h}, \boldsymbol{Q}_{l,h}, \boldsymbol{V}_{l,h} \in \mathbb{R}^{d \times d} : 1 \leq l \leq L, 1 \leq h \leq H\}$, MLP matrices and biases $\{\boldsymbol{A}_l \in \mathbb{R}^{d \times d}, \boldsymbol{B}_l \in \mathbb{R}^{d \times d}; \boldsymbol{b}_l \in \mathbb{R}^d : 1 \leq l \leq L\}$, and an unembedding matrix $\boldsymbol{U} \in \mathbb{R}^{|\Sigma| \times d}$. For matrices, we use $\| \cdot \|$ and $\| \cdot \|_F$ to denote the spectral and Frobenius norm respectively.

**Computation of Activations and Outputs**   We assume a standard causal transformer, with a few technical points: We explicitly scale attention logits with the logarithm of the input length, omit layer norm, allow Heaviside activations in addition to ReLU activations, and assume that, while the transformer may overall compute at infinite precision, attention logits are computed at fixed fractional precision. We next define all computations formally. Reserving a special SOS symbol not in $\Sigma$, written as "\$", we take the set of input strings to be $\mathfrak{S}$, the set of strings $x \in \Sigma^*$ where $x_1 = \$$ and \$ does not occur in $x_{2\ldots|x|}$. We now define the computation of the transformer $T$ on an input $x \in \mathfrak{S}$ where $1 \le |x| < N(T) + 1$ (that is $- |x| \le N(T)$ if $N(T) < \infty$, and $|x|$ is any finite length otherwise). If $L$ is the number of layers, then we write the output of layer $l = 1, \ldots, L$ at position $i = 1, \ldots, N(T)$ as $\boldsymbol{y}_i^{(l)} \in \mathbb{R}^d$. Let $o \ge 0$ be any offset such that $|x| + o < N(T) + 1$ – that is, $x$ still fits into the transformer's context width if encoded at offset $o$. Given this offset, we set

$$\boldsymbol{y}_i^{(0)} = \boldsymbol{E}_{x_i} + \boldsymbol{p}_{i+o} \quad i = 1, \ldots, |x| \tag{1}$$

where $x_i \in \Sigma$ is the input symbol at position $i$. Attention logits, at query position $i$ and key position $j$ are computed as

$$a_{i,j}^{(l,h)} = (\boldsymbol{y}_j^{(l-1)})^T \boldsymbol{K}_{l,h}^T \boldsymbol{Q}_{l,h} \boldsymbol{y}_i^{(l-1)} \ \text{ for } 1 \le j \le i \le |x|; \ l = 1, \ldots, L; \ h = 1, \ldots, H \tag{2}$$

We assume standard softmax attention, but incorporate scaling with $\log |x|$ following prior work finding it necessary to theoretically represent sparse functions and circumvent theoretical limitations of soft attention (Chiang & Cholak, 2022; Edelman et al., 2022):

$$\boldsymbol{Y}_i^{(l)} := \boldsymbol{y}_i^{(l-1)} + \sum_{h=1}^{H} \frac{\sum_{j=1}^{i} \exp\left(\log |x| \cdot a_{i,j}^{(l,h)}\right) \boldsymbol{V}_{l,h} \boldsymbol{y}_j^{(l-1)}}{\sum_{j=1}^{i} \exp\left(\log |x| \cdot a_{i,j}^{(l,h)}\right)} \tag{3}$$

After each attention block, the activations are passed through a one-layer MLP:

$$\boldsymbol{y}_i^{(l)} := \boldsymbol{Y}_i^{(l)} + \boldsymbol{B}_l \cdot \psi_l(\boldsymbol{A}_l \boldsymbol{Y}_i^{(l)} + \boldsymbol{b}_l) \tag{4}$$

where we allow the activation function $\psi_l$ to be, in each coordinate, either ReLU or Heaviside (see Appendix D.1 for discussion of this). We omit layer norm, as it plays no important role in our results, but it can be accounted for (See Appendix D.3). We assume an infinite-precision setup for the activations, with the restriction that attention logits (2) and the output of the $\exp(\cdot)$ function are both rounded to $p$ fractional bits of precision before further processing. This is a mild restriction preventing tiny changes in attention patterns from potentially snowballing into large changes in the output due to infinite precision (See Appendix D.2).

A transformer $T$ maps strings $x \in \mathfrak{S}$ ($|x| \le N(T)$) to vectors of next-token prediction logits, $T(x, o) \in \mathbb{R}^{|x| \times |\Sigma|}$, where $T(x, o)_i = \boldsymbol{U} \boldsymbol{y}_i^{(L)}$ ($i = 1, \ldots, |x|$) for the unembedding matrix $\boldsymbol{U} \in \mathbb{R}^{|\Sigma| \times d}$, and $o$ is the offset. Let $\mathcal{F}(\Sigma)$ be the set of all maps $f$ mapping $x \in \mathfrak{S}$ to $f(x) \in \mathbb{R}^{|x| \times |\Sigma|}$.

## 3 THEORETICAL FRAMEWORK

### 3.1 LIMIT TRANSFORMERS

Our theory addresses the setting of transformers with absolute positional encodings, where the width may grow with the input length. Importantly, we cannot view the ground-truth function as realized by a single transformer: Even if one assigned such a transformer an infinite number of positional encodings, it would still effectively only be able to distinguish between a bounded number of positions, because the width of positional encodings within a single transformer is bounded. Instead, we will derive a parameterization of transformers that allows us to convert sequences of transformers operating on longer and longer sequences to a single limiting transformer-like object. Our key technical idea is to reparameterize the transformer in terms of **product functions**, inner products of parameter vectors as mediated by parameter matrices, such as

$$\begin{aligned} \boldsymbol{E}_\sigma^T \boldsymbol{K}_{1,h}^T \boldsymbol{Q}_{1,h} \boldsymbol{E}_\tau \quad & \boldsymbol{p}_i^T \boldsymbol{K}_{1,h}^T \boldsymbol{Q}_{1,h} \boldsymbol{p}_j \\ \boldsymbol{p}_i^T \boldsymbol{K}_{2,h}^T \boldsymbol{Q}_{2,h} \boldsymbol{V}_1 \boldsymbol{p}_j \quad & \boldsymbol{U}_\sigma \boldsymbol{V}_2 \boldsymbol{V}_1 \boldsymbol{E}_\sigma \end{aligned} \tag{5}$$

and various others; see Appendix F.1 for the full formal definition. We first note that the transformer's computations are uniquely specified by such products. The number of products as in (5) depends, among others, on $|\Sigma|$, $L$, $H$, $N(T)$, but crucially not on $d$. We will use this parameterization to translate sequences $T_1, T_2, T_3, \ldots$ of transformers running on inputs of length $1, 2, 3, \ldots$ to limiting transformer-like objects that are applicable at all input lengths, while keeping width $d$ bounded even if the widths of $T_n$ diverge to infinity. This limiting object, a **Limit Transformer**, differs from an ordinary transformer, as defined in Section 2, just in a few respects. Formally:

**Definition 2.** *A* Limit Transformer *is a transformer $T$ where:*

1. $N(T) = +\infty$

2. *All parameters (including positional encodings $\boldsymbol{p}_i$, and the output of $\phi_{l,h}$) are expressed in $p$-bit precision, for some $p \in \mathbb{N}$*

3. *Attention logits on input length $N$ are computed as*
$$a_{i,j}^{(l,h)} = (\boldsymbol{y}_j^{(l-1)})^T \boldsymbol{K}_{l,h}^T \boldsymbol{Q}_{l,h} \boldsymbol{y}_i^{(l-1)} + \phi_{l,h}(j,i) \qquad (6)$$
*where $\phi_{l,h} : \mathbb{N} \times \mathbb{N} \to \mathbb{R}$.*

The most important point is the third one, which augments the attention logits (Eq. 2) with a second term, $\phi_{l,h}(j,i)$, where $\phi_{l,h}$ is a potentially arbitrary function. Thus, a Limit Transformer can use positional information in two ways: through bounded-width and bounded-precision positional encodings $\boldsymbol{p}_i$, and additionally through potentially more complicated functions $\phi_{l,h}$. Our main result will link length generalization to expressibility by Limit Transformers satisfying specific properties:

**Definition 3.** *A Limit Transformer satisfies*

1. PERIODIC *if $\boldsymbol{p}_i = \boldsymbol{p}_{i+\Delta}$ for all $i$ for some $\Delta > 0$*
2. LOCAL *if each $\phi_{l,h}$ is translation-invariant and local.*

*Here, a function $f : \mathbb{N} \times \mathbb{N} \to \mathbb{R}$ is "translation-invariant" if $f(i,j) = f(i+\tau, j+\tau), \forall i \leq j, \forall \tau \geq 0$, and "local" if there is $\tau$ such that $f(i,j) = 0$ when $j > i + \tau$.*

The parameterization in terms of inner products permits a translation from a transformer $T$ to a bounded-width Limit Transformer satisfying PERIODIC and LOCAL (Lemma 52 in the Appendix). In this translation, products of the form $\boldsymbol{p}_i^T \boldsymbol{K}_{l,h}^T \boldsymbol{Q}_{l,h} \boldsymbol{p}_j$ are encoded into the functions $\phi_{l,h}(i,j)$, permitting the Limit Transformer to distinguish infinitely many different positions even while keeping its width bounded. We note that Limit Transformer are a mathematical construct helping us prove statements about standard transformers, and are not themselves trained or implemented.

## 3.2 DEFINITION OF INFERENCE PROCEDURE

To define the inference procedure, we specify the following hypothesis class at each input length $n$:

**Definition 4** (Hypothesis Class). *For each $n = 1, 2, 3, \ldots$, define the hypothesis class $\Theta_n$ as the set of transformers $T$ (as defined in Section 2) where (1) $N(T) = n$, (2) each parameter vector and matrix of $T$ is represented at $p$ bits of precision, for some $p \in \mathbb{N}$, (3) each product function involving positional encodings is translation-invariant. That is, every product function involving exactly one positional encoding is constant across positions, and for every $1 \leq i, j, i + \Delta, j + \Delta \leq n$,*
$$\boldsymbol{p}_i^T \boldsymbol{M}_1 \ldots \boldsymbol{M}_k \boldsymbol{p}_j = \boldsymbol{p}_{i+\Delta}^T \boldsymbol{M}_1 \ldots \boldsymbol{M}_k \boldsymbol{p}_{j+\Delta} \qquad (7)$$
*whenever $\boldsymbol{M}_1 \ldots \boldsymbol{M}_k$ is a product of parameter matrices linking the input layer.[2]*

Note that the width $d$ of the transformers $T \in \Theta_n$ is unconstrained. The most interesting requirement here is the third one: We ask that, while the positional encodings $\boldsymbol{p}_i$ will typically vary with position, their contributions to the transformer's computations are offset-independent. This is a stronger requirement than for the input-output behavior to be offset-independent: we ask for the transformer's "algorithm" itself to be the same across offsets. This is a substantive condition, but we believe it to be a natural requirement in the context of length generalization (Appendix G.6). Our inference procedure will use a regularizer $\mathcal{R}$ favoring simpler hypotheses. The following will be sufficient:

---

[2]Such as $\boldsymbol{K}_{1,h}^T \boldsymbol{Q}_{1,h}$, $\boldsymbol{V}_{2,h}^T \boldsymbol{K}_{3,h'}^T \boldsymbol{Q}_{3,h'} \boldsymbol{V}_{1,h''}$, and similar. See Appendix F.2 for a formal definition.

**Definition 5** (Regularizer). *Let $T \in \Theta_n$, thus $N(T) = n$. Define $\mathcal{R}(T)$ as the sum of (1) $L + H$; (2) the precision $p$ used in Definition 4; the precision $p$ used for rounding attention logits and the output of $\exp(\cdot)$ (Section 2); (3) $\max_{l,h} rank(\mathbf{V}_{l,h})$; (4) $\max_{l,h} \|\mathbf{K}_{l,h}^T \mathbf{Q}_{l,h}\|$; $\max_{l,h} \|\mathbf{V}_{l,h}\|$; $\max_l \|\mathbf{A}_l\|_F$, $\|\mathbf{B}_l\|_F$; $\|\mathbf{U}\|$; (5) $\max_i \|\mathbf{p}_i\|_2$, $\max_\sigma \|\mathbf{E}_\sigma\|_2$, $\max_l \|\mathbf{b}_l\|_2$; (6) the term*

$$\sum_{l=1}^{L} \sum_{h=1}^{H} \sum_{j=1}^{N(T)} \left| \mathbf{p}_1^T \mathbf{K}_{l,h}^T \mathbf{Q}_{l,h} \mathbf{p}_j \right|^2 \tag{8}$$

The idea of (8) is to discourage accidental attention between far-away positions that do not appear together during training, which could hamper length generalization. Due to translation invariance, this term entails a bound on products for all pairs $\mathbf{p}_i, \mathbf{p}_j$ ($i \leq j$) entering causal attention. While such a regularizer is not part of standard training, standard initialization tends to lead to bounded values for (8) when $d$ is large (Appendix G.1); it thus captures an implicit bias of standard initialization and training. Importantly, the width $d$ does not explicitly enter $\mathcal{R}$; as a consequence, for any sufficiently large $C$, the number of transformers $T_n \in \Theta_n$ with $\mathcal{R}(T_n) \leq C$ is infinite, simply because $d$ is not constrained. Nonetheless, this regularizer will be sufficient for identification under our idealized inference procedure, which observes the input-output behavior of the target function $f$ on inputs of length $\leq \frac{n}{2}$ and selects a transformer $T$ with maximal context window $n$, $T \in \Theta_n$ that exactly fits that input-output behavior while minimizing the regularizer $\mathcal{R}(T)$:

**Definition 6** (Inference Procedure). *Given a function $f \in \mathcal{F}(\Sigma)$, the Inference Procedure obtains a sequence of transformers $T_1 \in \Theta_1, T_2 \in \Theta_2, \ldots$ as follows. Define $U_n$ as the set of $T \in \Theta_n$ matching the behavior of $f$ on all inputs of length $\leq \frac{n}{2}$. Then choose $T_n \in U_n$ such that*

$$\mathcal{R}(T_n) \leq \frac{1}{n} + \inf_{T \in U_n} \mathcal{R}(T) \tag{9}$$

In (9), we do not simply ask for minimizing the regularizer, as the set of elements of $U_n$ with $\mathcal{R}(T)$ smaller than a given value need not be finite and thus a minimum need not be attained by any $T_n$. Importantly, we only ask $T_n$ to match the behavior of $f$ up to length $\frac{n}{2}$, formalizing the idea of training on shorter inputs and testing on longer ones; our identifiability guarantee will provide conditions under which $T_n$ will end up matching $f$ correctly up to length $n$ – representing length generalization. While we take the testing length to be twice the training length, there is nothing special about this; our analysis works whenever the training length diverges to infinity.

### 3.3 MAIN RESULT: CONVERGENCE OF INFERENCE PROCEDURE

Our main result asymptotically characterizes length generalization under the inference procedure from Definition 6. For functions representable by Limit Transformers satisfying LOCAL and PERIODIC, we guarantee that *any* run of the Inference Procedure will ultimately achieve length generalization, so that transformers with context length $n$ chosen to fit the target function on inputs with length $\leq \frac{n}{2}$ will, when $n$ is sufficiently large, also perform correctly at all lengths $\leq n$. Formally,

**Theorem 7** (Guaranteed Length Generalization in the Limit). *Let $f \in \mathcal{F}(\Sigma)$. Then the following are equivalent:*

1. *$f$ is expressible by a Limit Transformer satisfying PERIODIC and LOCAL.*

2. *(Guaranteed Length Generalization) Applying the Inference Procedure from Definition 6 to $f$ generates a sequence $T_1, T_2, \ldots$ with $\sup_{n=1,2,3,\ldots} \mathcal{R}(T_n) < \infty$, for which there is some $N_0$ such that, for all $m > N_0$, $T_m$ matches $f$ on all inputs of any length $k \leq m$.*

The formal proof is in Appendix B.1. Intuitively, if $f$ is expressible by a Limit Transformer satisfying PERIODIC and LOCAL, then, even though the Inference Procedure produces infinitely many distinct *transformers* $T_1, T_2, \ldots$, these can only traverse a finite set of underlying *algorithms*, each described by some Limit Transformer. PERIODIC and LOCAL ensure that the Limit Transformer's parameter count effectively remains finite, as its position-related parameters can be fully specified in terms of $\mathbf{p}_1, \ldots, \mathbf{p}_\Delta$ and $\phi(1,1), \ldots, \phi(1, \tau)$. The regularizer bounds width, depth, and precision of the Limit Transformers; this keeps the set of algorithms traversed finite. Each of these finitely many algorithms will either be ruled out at some input length $n$, or else match the behavior of $f$ at

all input lengths. At some finite $N_0$, only the latter type of algorithm remains; hence, transformers produced after this point will match the target function. The proof also entails a result on NoPE length generalization: Applying the inference procedure to $f$ while constraining $\boldsymbol{p}_i \equiv 0$ will lead to length generalization when $f$ is expressible by a Limit Transformer where all $\boldsymbol{p}_i$ and all $\phi_{l,h}$ are zero (Corollary 18). While Theorem 7 guarantees length generalization from length $\frac{n}{2}$ to length $n$ for expressible problems, it does not rule out length generalization for inexpressible problems. Such a statement becomes possible if we allow arbitrary scaling of training vs. testing lengths (Appendix B.4). Besides length generalization guarantees, Limit Transformers are also useful in providing *expressiveness* results for transformers with absolute positional encodings (Appendix B.5).

# 4 WHICH FUNCTIONS ARE IDENTIFIABLE? EXPRESSIVENESS OF LIMIT TRANSFORMERS AND **C-RASP**

We have found that, if a target function $f$ is expressible by a Limit Transformer satisfying PERIODIC and LOCAL, then Theorem 7 implies length generalization under our Inference Procedure. In order to understand the ramifications of this result, we now study what functions Limit Transformers can express – for these functions, Theorem 7 will then guarantee length generalization.

## 4.1 SIMPLE EXAMPLE: INDUCTION HEAD

We consider the task of predicting the next token in proportion to the frequency at which different tokens had previously followed tokens matching the current one:

$$f(x_1 \ldots x_N)_{i,\sigma} = \frac{\#\{k < i : x_k = x_i, x_{k+1} = \sigma\}}{\#\{k < i : x_k = x_i\}} \tag{10}$$

A Limit Transformer solves this with the *Induction Head* construction (Olsson et al., 2022), in two layers and one head: In the first layer, $\phi(i,j) = 1$ if $i + 1 = j$ and 0 else; the head copies the preceding symbol. In the second layer, attention focuses on positions with the same symbol. The transformer outputs next-token predictions in proportion to bigram frequencies in the context, up to approximation error $O(\frac{1}{n})$ (due to logit scaling). Hence, Theorem 7 guarantees that the Inference Procedure will length-generalize on (10). A special case of (10) occurs when each symbol occurs exactly once; here, such an induction head circuit suffices to copy a string (Zhou et al., 2024a), we thus obtain a length generalization guarantee for copying such strings (see Section 5).

## 4.2 LENGTH GENERALIZATION FOR **C-RASP**

We next present a large class of functions for which Theorem 7 guarantees length generalization. We extend the **C-RASP** formalism (Yang & Chiang, 2024) with positional information, and then show that any function defined by a **C-RASP** program is expressible by a Limit Transformer; hence, transformers will, by Theorem 7, length-generalize on those functions. We first define **C-RASP**:

**Definition 8** (**C-RASP**). *Let $\Sigma$ be an alphabet, let $\Phi$ be a set of* unary relations $\phi : \mathbb{N} \to \{0,1\}$, *and let $\Psi$ be a set of* binary relations $\psi : \mathbb{N} \times \mathbb{N} \to \{0,1\}$. *A **C-RASP**$[\Phi, \Psi]$ program $P$ is defined as a sequence $P_1, \ldots, P_k$ of **C-RASP** operations. There are two sorts of operations:*

| **Boolean-Valued Operations** | | | **Count-Valued Operations** | |
|---|---|---|---|---|
| *Initial* | $P(i) := Q_\sigma(i)$ 
 for $\sigma \in \Sigma$ | | *Counting* | $C(i) := \#\left[j \leq i, \psi(i,j)\right] \ P(j)$ 
 for $\psi \in \Psi \cup \{\top\}$ |
| *Boolean* | $P(i) := \neg P_1(i)$ 
 $P(i) := P_1(i) \wedge P_2(i)$ | | *Conditional* | $C(i) := P(i) \ ? \ C_1(i) : C_2(i)$ |
| *Constant* | $P(i) := \top$ | | *Addition* | $C(i) := C_1(i) + C_2(i)$ |
| *Positional* | $P(i) := \phi(i)$ 
 for $\phi \in \Phi$ | | *Subtraction* | $C(i) := C_1(i) - C_2(i)$ |
| | | | *Constant* | $C(i) := 1$ |
| *Comparison* | $P(i) := C_1(i) \leq C_2(i)$ | | | |

A Counting operation returns the number of positions $j \leq i$ where $P(j)$ and $\psi(i,j)$ hold. A conditional operation returns $C_1(i)$ if $P(i)$, and $C_2(i)$ otherwise. We use the value of the *last*

Boolean-valued operation, at the last position of the string, to determine acceptance using a **C-RASP** program. That is, if the program is run on input $w$ with final operation $L$, then we accept $w$ if and only if $L(|w|)$ is true. **C-RASP**[periodic, local] is the class of **C-RASP** programs where each $\phi(i)$ is periodic in $i$, and each $\psi(i, j)$ is translation-invariant and local (Definition 3). We also write **C-RASP**[periodic, local] for the class of all languages accepted by some **C-RASP**[periodic, local] program. As an example, we present a program recognizing $L = \Sigma^* ab\Sigma^*$ over $\Sigma = \{a, b\}$:

---

**C-RASP** program for $L = \Sigma^* ab\Sigma^*$ over $\Sigma = \{a, b\}$

| | | |
|---|---|---|
| $C_{a-}(i) := \# [j \leq i, j = i - 1] \; Q_a(j)$ | # of immediately preceding $a$ | (1) |
| $P_{a-}(i) := C_{a-}(i) \geq 1$ | Position $i - 1$ holds an $a$ | (2) |
| $Q_{ab}(i) := Q_b(i) \wedge P_{a-}(i)$ | A substring $ab$ ends at position $i$ | (3) |
| $C_{ab}(i) := \# [j \leq i] \; Q_{ab}(j)$ | # of substrings $ab$ | (4) |
| $L(i) := C_{ab}(i) \geq 1$ | At least one $ab$ precedes position $i$ | (5) |

---

Any **C-RASP**[periodic, local] program can be translated to a Limit Transformer with corresponding positional functions. We say a Limit Transformer $T$ *accepts* an input if the value in the last dimension in the last position of the output is greater than 0, and rejects otherwise.

**Theorem 9.** *For every **C-RASP**$[\Phi, \Psi]$ program $P$ with local functions $\Psi$ and periodic functions $\Phi$ there exists a Limit Transformer $T_\infty$ that satisfies* PERIODIC *and* LOCAL *such that for all $w \in \Sigma^*$, $P$ accepts $w$ iff $T_\infty$ accepts $\$w$. If $P$ uses no local or periodic relations, then $T$ requires no functions $\phi_{l,h}$ or positional encodings $\boldsymbol{p}_i$.*

The proof is in Appendix B.6. As a consequence, the Inference Procedure will ultimately length-generalize on inputs from a function $f$ expressible by a **C-RASP**[periodic, local] program. If the **C-RASP** program requires no positional functions (i.e., it is in **C-RASP**$[\emptyset]$), then length generalization will succeed even with NoPE transformers. We establish that various functions are in **C-RASP**:

**Theorem 10.** *Membership in the following languages is definable in **C-RASP**$[\emptyset]$: (1) MAJORITY, (2) DYCK-1, (3) $a^n b^n c^n$.*

The proof is in Appendix C.1. By Theorem 9, these positive results translate into length generalization guarantees. For these tasks, length generalization even with NoPE is empirically already well-documented (Bhattamishra et al., 2020). Further, **C-RASP**[local] can implement versions of the Induction Head task from Section 4.1 (Appendix C.2.1–C.2.2). **C-RASP** also helps understand why transformers show varying abilities even on simple finite-state languages (Bhattamishra et al., 2020; Liu et al., 2023; 2024), a fact poorly understood theoretically. For instance, we find:

**Lemma 11.** *Consider the alphabet $\Sigma = \{a, b, e\}$.*

1. *$PARITY := b^*(ab^*ab^*)^* \notin$ **C-RASP**$[periodic, local]$*
2. *$(aa)^* \in$ **C-RASP**$[periodic, local]$ and $(aa)^* \notin$ **C-RASP**$[\emptyset]$*
3. *$\Sigma^* be^* \notin$ **C-RASP**$[periodic, local]$*
4. *$L \in$ **C-RASP**$[\emptyset]$ for piecewise testable $L$*

The proof is in Appendix C.3. Notably, all of these languages are recognizable by simple finite-state automata which are expressible by transformers (Liu et al., 2023), but empirical length generalization behavior differs in line with **C-RASP** expressiveness (Section 5). PARITY (1) has long been found difficult for transformers (e.g. Hahn, 2020; Bhattamishra et al., 2020; Anil et al., 2022; Chiang & Cholak, 2022; Delétang et al., 2023; Hahn & Rofin, 2024). Result (2) exemplifies the effect of different positional relations. The language $\Sigma^* be^*$ (3) is a simple model of FlipFlop (Liu et al., 2024), a language on which transformers empirically struggle to generalize perfectly despite its simplicity for recurrent models (Liu et al., 2024; Sarrof et al., 2024). The class (4) is useful for determining expressibility of languages in **C-RASP**$[\emptyset]$, as in Section E.1.2.

### 4.3 LIMITATIONS: LOGARITHMIC COMMUNICATION COMPLEXITY

Having shown that various functions are definable by Limit Transformers, we now provide a simple technique for showing that various functions are *not* definable by Limit Transformers. Informally, any function satisfying the conditions in Theorem 7 has logarithmic communication complexity:

**Theorem 12.** *Let $T$ be a Limit Transformer satisfying* PERIODIC *and* LOCAL. *On an input $x \in \Sigma^{2N}$, assume Alice has access to $x_{1...N}$ and Bob has access to $x_{N+1...2N}$. Then Alice can communicate $C \log N$ bits to Bob, where $C$ depends on $T$ but not $N$, so that Bob can compute each activation in the second half, $\boldsymbol{y}_i^{(l)}$ ($N + 1 \le i \le 2N$).*

The proof is in Appendix B.3. In principle, computing activations in the second half of the input requires full knowledge of the first half of the input, because positions in the second half can freely attend to positions in the first half. In this situation, one would expect Bob to need full knowledge of $\frac{N}{2}$ input symbols from Alice's part, exponentially more than the $C \log N$ claimed in the theorem. This is indeed needed if $T$ performs the task of, say, checking if $x_{1...,N}$ and $x_{N+1...2N}$ are identical. However, if $T$ satisfies PERIODIC and LOCAL, attention must largely be determined by the presence of tokens and token sequences; when an attention head's behavior is determined by positional information, it can only focus its attention on a local neighborhood or equally distribute it over a periodic pattern. Intuitively, in such cases, the set of possible queries and keys can be grouped into a finite partitioning, of size bounded independently of $N$. It then suffices for Alice to communicate, for each possible group of keys, an aggregate of the value vectors at the positions where a matching key is computed. The proof (Appendix B.3) formalizes this. As a corollary:

**Corollary 13.** *The following problems are not expressible by Limit Transformers satisfying* PERIODIC *and* LOCAL: *(1) copying arbitrary strings, (2) addition of $n$-digit numbers.*

This is proven in Appendix B.3. As a consequence, any run of the Inference Procedure on these functions will output solutions $T_1, T_2, T_3, \ldots$ for which the depth, number of heads, parameter norms or ranks, MLP dimensions, or precision $p$, must increase with the input length $n$; indeed, length generalization is empirically challenging for these functions (Section 5).

## 5 EXPERIMENTS

We evaluate the expressiveness of Limit Transformers and **C-RASP** as predictors of empirical length generalization. Based on Theorems 7 and 9, we expect that APE transformers should length-generalize on problems with a **C-RASP**[periodic,local] program, and that NoPE transformers will be successful on problems with a **C-RASP**[∅] program. We test this prediction on a suite of algorithmic problems and formal languages, largely from prior empirical work on length generalization (Bhattamishra et al., 2020; Zhou et al., 2024a), but evaluated within a uniform framework.

For each task, the model is trained on inputs whose LEN is in the range $[l_{min}, 50]$, where $l_{min}$ is the minimum length for this task. LEN is the length of the input in the algorithmic tasks (Appendix E.2), and the overall sequence length in the formal language tasks. The model is tested on 3 test sets, where LEN is in the range $[l_{min}, 50]$, $[51, 100]$, $[101, 150]$; these lengths are based on the source of the regular languages benchmark (Bhattamishra et al., 2020). We trained using a standard AdamW setup; see details in Appendix E.3. Hyperparameters are selected by searching in order of increasing complexity until we find a setting that performs well up to length $\le 100$. We interpret results on lengths $[101, 150]$ as a measure of length generalization. Each model has as many positional encodings as needed to encode the longest inputs (at least 150); each input is presented with a random offset in agreement with the theoretical setup. On algorithmic sequence-to-sequence tasks, we train with cross-entropy loss on the output. On formal languages, where next-symbol predictions are generally not deterministic, we instead train the model to predict the *set of legal next symbols*, with each such set coded as an atomic symbol (as in Bhattamishra et al., 2020; Sarrof et al., 2024). At test time, predictions are considered correct on a sequence if and only if the output at every step is correct; the random baseline is thus very low on the formal language benchmark. We report accuracy as the fraction of test sequences where the predictions are correct at each step.

We first evaluate on 8 algorithmic problems, which largely overlap with Zhou et al. (2024a), but are tailored to those where **C-RASP** expressiveness can be clearly settled. Tasks are defined formally in Appendix E.2.1. A new problem here is BINARY MAJORITY INTERLEAVE, which interleaves multiple MAJORITY functions and can be solved by **C-RASP** using periodic functions. Length generalization behavior matches **C-RASP** expressiveness; **C-RASP**[∅] expressiveness predicts the success of NoPE (see Figure 1). In agreement with prior empirical results (Zhou et al., 2024a; Jelassi et al., 2023), COPY is difficult in the presence of repetition and easy when it is avoided; these findings match **C-RASP** expressiveness (Corollary 13 and Section 4.1).

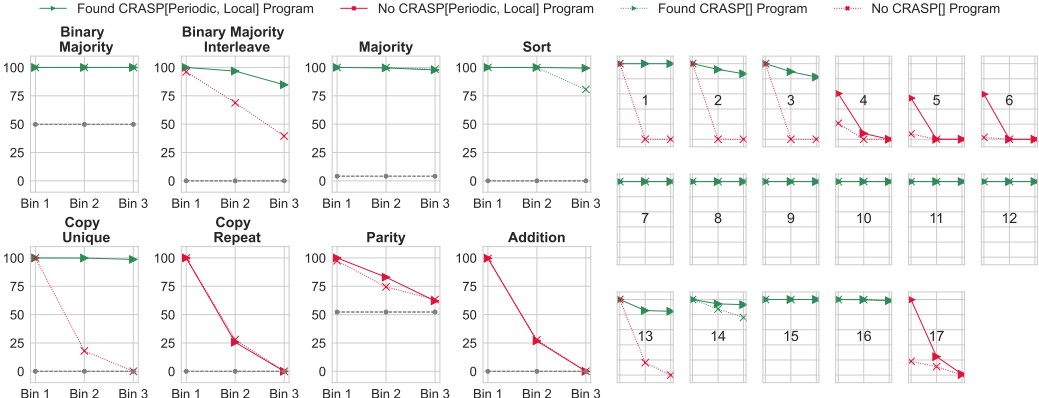

Figure 1: Experimental results (y axis: accuracy), at lengths $\leq 50$ (Bin 1, training), $[51, 100]$ (Bin 2), and $[101, 150]$ (Bin 3, generalization), for APE (solid) and NoPE (dotted). Green lines indicate that we found a **C-RASP** program (**C-RASP**[periodic, local] for APE, **C-RASP**[$\emptyset$] for NoPE), red lines indicate that we proved nonexistence, or found no program. Random baselines are indicated in gray in (left), and very close to zero in (right). On the algorithmic problems (left), we replicate prior empirical findings; **C-RASP** expressiveness predicts observed length generalization. On the regular languages (right, with same $x$ and $y$-axes as left, Table 2), length generalization tracks **C-RASP** expressiveness established in Lemma 11 ((1) = $(aa)^*$, (17) = $\Sigma^* be^*$) and other results (see Appendix E.1). **C-RASP** expressiveness performs much better than circuit complexity and standard notions of regular language complexity in predicting length generalization (Appendix, Figures 3–4).

We next applied the experimental framework to 17 regular languages assembled by Bhattamishra et al. (2020), who evaluated length generalization in transformers and LSTMs. Whereas LSTMs perform strongly across the board, the behavior of transformers on these regular languages has so far eluded theoretical understanding. While it is known that transformers struggle with PARITY, it has remained unclear why they would struggle to length-generalize on some seemingly very simple languages. We found **C-RASP**[periodic,local] programs for 13 of the languages and proved nonexistence for the others (Appendix E.1.2). Length generalization succeeded in those cases where we had found a **C-RASP**[periodic,local] program (see Figure 1 right). In those cases where a **C-RASP**[$\emptyset$] program exists, generalization also succeeded with NoPE. Generalization failed for languages where no **C-RASP** program exists, such as $\Sigma^* be^*$ (#17 in Figure 1; Lemma 11).

## 6  DISCUSSION

Prior work has empirically found that transformers' length generalization capabilities differ between tasks, but theoretical understanding has been lacking. We have introduced a formal framework analyzing length generalization in an idealized inference procedure. The framework explains the commonality across diverse tasks where successful length generalization has been observed in prior research, in terms of expressiveness in two simple mathematical formalisms, Limit Transformers and **C-RASP**. We also proved that various problems, on which length generalization is less successful empirically, are not expressible in one or both of these formalisms. Beyond length generalization, the framework further sheds light on the expressiveness of APE transformers. Our results on length generalization study an idealized regularizer and assume perfect fitting of the training distribution. Making the guarantee from Theorem 7 more realistic by incorporating SGD training dynamics and subsampling of training data is an interesting problem for future research.

Our results can be viewed as formalizing the RASP-L Conjecture (Zhou et al., 2024a). Both Limit Transformers and **C-RASP**[periodic,local] formalize intuitions underlying RASP-L in restricting how positional information can be used. An important advance over Zhou et al. (2024a) is that we settle the expressiveness of these formalisms for many problems, and are able to explicitly prove a variety of problems with poor empirical length generalization, such as copying with repeated strings, to be inexpressible by Limit Transformers. Our results provide a step towards rigorously confirming the idea that expressiveness in such restricted formalisms predicts length generalization.

**Expressiveness of Transformers**   A substantial line of research has studied the in-principle expressiveness of transformers (Strobl et al., 2024). Transformers express a subset of the class $\mathbf{TC}^0$ (Merrill & Sabharwal, 2023b; Strobl, 2023), but it is unknown if this inclusion is proper. *All* problems considered in Section 5 are in $\mathbf{TC}^0$, but empirical length generalization behavior largely tracks **C-RASP**[periodic,local] expressiveness, which defines a proper subclass of $\mathbf{TC}^0$ (Appendix C.3.1). While it remains open if the expressive power of transformers exhausts $\mathbf{TC}^0$, our results suggest a separation between $\mathbf{TC}^0$ and those problems for which length generalization is possible with absolute positional encodings. In particular, our results suggest that the existence of APE transformers that perform a task across larger ranges of input lengths is linked to the expressiveness of Limit Transformers (Section C). It is an open question how far new, yet-to-be-discovered positional encoding schemes may increase the range of length generalization; empirical evidence indicates that NoPE and APE may be hard to beat by other general-purpose encodings (Kazemnejad et al., 2023). The proof of Theorem 12 is closely linked to previous communication-complexity bounds for transformer layers (Sanford et al., 2023; 2024; Peng et al., 2024; Bhattamishra et al., 2024), which importantly were shown only for individual layers, not multilayer transformers. Indeed, Bhattamishra et al. (2024) showed that such a logarithmic bound is not in general possible for arbitrary multilayer transformers. In contrast, our result applies even at multilayer models, which is enabled by the restrictions on the ways in which positional information can be used in a Limit Transformer.

**Length Generalization of Transformers**   Various studies have empirically evaluated length generalization in transformers. Our work is most closely related to Zhou et al. (2024a), discussed above. Bhattamishra et al. (2020) study length generalization on formal languages; we find that **C-RASP**[periodic,local] expressiveness explains behavior on their benchmark well (Section 5). Anil et al. (2022) show that language models, finetuned on various reasoning problems, do not length-generalize well. Wang et al. (2024a) evaluate length generalization of NoPE transformers on real-world tasks. Kazemnejad et al. (2023) explore length generalization across different positional encoding schemes, finding NoPE to perform surprisingly well. Zhou et al. (2024b) show that length generalization for addition improves with specific encoding schemes and input formats. Jelassi et al. (2024) show that transformers can succeed in length generalization on copying when inputs avoid $n$-gram repetition. Chang & Bisk (2024) empirically find limitations in generalization in counting. In contrast to the rich landscape of empirical studies, theoretical understanding of length generalization has been limited. Most relevant, Ahuja & Mansouri (2024) study length generalization in simple neural architectures, including a one-layer transformer setup with linear (not softmax) attention. Our results, in contrast, apply to multi-layer softmax transformers and make statements about many concrete problems that have been studied empirically. Some other works (e.g. Hou et al., 2024; Xiao & Liu, 2023) provide length-generalizing constructions for certain problems, but leave open whether learning would lead to such constructions. Wang et al. (2024b) show that GD training leads to length generalization on a specific token selection task.

**Limitations**   We study idealized asymptotic identification of a global minimum with perfect knowledge of behavior on the training distribution (cf. Q.4 in App. A for more discussion). Extending Theorem 7 to account for subsampling of the training data and learning dynamics is an important problem for future research. In particular, providing a practical upper bound on the threshold $N_0$ at which length generalization is expected is an interesting problem. Our study focuses on absolute positional encodings; extending it to other positional encodings (e.g. Su et al., 2024; Press et al., 2021; Ruoss et al., 2023) is another important problem for future research.

## 7   CONCLUSION

We have introduced a theoretical framework that unifies a broad array of empirical findings about successes and failures of length generalization in transformers with absolute positional encodings. Our framework is based on the analysis of an idealized inference procedure, for which length generalization provably happens whenever the ground-truth function is expressible with only limited access to positional information. By providing upper and lower bounds on the expressiveness of transformers trained using this inference procedure, we accurately predict the success and failure of length generalization across a wide set of algorithmic tasks and formal languages.

## CONTRIBUTIONS

MH coordinated the project. MH and XH developed the conceptual framework of Theorem 7, with input from the other authors. XH and YS contributed Section 5. AY contributed Section 4.2 with input from MH and AK. MH, XH, AY, YS jointly developed the translation to Limit Transformers; MH worked out the formalization. SB contributed Proposition 54, Lemma 56, and provided conceptual input throughout the project. AK contributed to settling the **C-RASP** expressiveness of formal languages. PN and HZ provided conceptual and writing-level input over the course of the project. MH drafted the remaining portions of the paper and the proof of Theorem 7, including definitions and lemmas.

## ACKNOWLEDGMENTS

Funded by the Deutsche Forschungsgemeinschaft (DFG, German Research Foundation) – Project-ID 232722074 – SFB 1102. MH thanks Lena Strobl, Dana Angluin, David Chiang, Mark Rofin, Anthony Lin, and Georg Zetzsche for conversations on related topics. We thank Entang Wang for comments on the draft. We also thank anonymous reviewers for their close reading and detailed feedback.

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

# Contents

## A   FAQ

**(1)** *What is the point of introducing Limit Transformers?*

Limit Transformers are a mathematical formalism helping us prove a length generalization guarantee (Theorem 7) for a broad class of functions, not just one specific function. They thus serve as an object that can help us prove things about standard transformers.

**(2)** *What is the relation between Limit Transformers and **C-RASP**? Why use two different formalisms?*

They serve two distinct purposes - one is easier to prove a length-generalization guarantee with, and the other is easier to prove expressivity results with. Limit Transformers are closely connected to standard transformers, and provide a convenient formalism for formalizing a length generalization guarantee in our inference setup (Theorem 7); they also provide bounds on APE transformer expressiveness as a side result (Appendix C). **C-RASP** is a formalism based on the RASP language (Weiss et al., 2021), intended to provide a formal abstraction of the kinds of computations that transformers can perform in a human-readable format. Limit Transformers with PERIODIC and LOCAL express all the functions definable in **C-RASP**[periodic,local], though it is open if this inclusion is strict. We provide rigorous tools for understanding the expressiveness of both formalisms. For Limit Transformers, we prove a logarithmic communication complexity bound (Theorem 12). **C-RASP** brings additional use in understanding expressiveness from two angles. First, one can conveniently prove functions expressible by writing down programs, as we did in Section 4.2. Second, to prove negative results, we can bring to bear a set of deep results about logics using majority quantifiers (Krebs, 2008), which allow us to provably settle expressiveness of many problems. Positive results translate into positive results about Limit Transformer expressiveness and hence, under our idealized learning setup, length generalization. While it is open if problems not expressible in **C-RASP** cannot in principle show length generalization, experimental results suggest that such an implication might hold in many cases.

**(3)** *Why are Limit Transformers needed – can't one just consider transformers whose parameters have infinite precision and hence can accommodate infinitely many different positional encodings $p_i$?*

The key advantage of Limit Transformers is that they effectively have finite parameter counts whenever they satisfy LOCAL and PERIODIC, which is useful in establishing Theorem 7. In an ordinary transformer, due to fixed width, effectively distinguishing unboundedly many positions requires in-

finitely many parameters $\boldsymbol{p}_i$. Even then, a function as simple as $\phi(i,j) = \delta_{ij}$ cannot be exactly represented for infinitely many $i, j$ by a product $\boldsymbol{p}_i \boldsymbol{Q}^T \boldsymbol{K} \boldsymbol{p}_j$ at bounded width $d$.

We used Limit Transformers as a tool specifically to prove results about APE transformers. We note in Corollary 18 that the analogous results for the special case of NoPE transformers do not require the use of Limit Transformers. Understanding what is needed to derive results for other positional encoding schemes (such as relative positional encodings) is left for future work.

**(4)** *Why is the idealized setup considered for the analysis, as opposed to more practical frameworks of learning?*

Proving guarantees in a more practical setting (SGD training dynamics, subsampling of training data) would, of course, be ideal. However, such guarantees have been notoriously difficult to establish for deep learning models (Neyshabur et al., 2017). Standard frameworks for learning, such as PAC-learning, assume that the training and test distributions are the same, which precludes out-of-distribution guarantees such as length generalization. Even within the PAC-learning framework, obtaining nontrivial guarantees for deep neural networks remains challenging without making strong assumptions. Instead of analyzing the learning and generalization of Transformers trained with gradient-based methods, our work aims to understand the length generalization properties of Transformers from an architectural perspective. A substantial body of work (cf. Section 6) has empirically investigated the length generalization properties of Transformers and found a complex array of empirical behavior, while theoretical understanding has been very limited. Hence, consolidating the theoretical relation between these empirical observations and the computational model of Transformers seems like an important direction. Our work provides a formal framework, based on an idealized model of learning, that separates the tasks on which Transformers succeed and those on which they fail to length-generalize. The learning model considered in our work is closely related to the "identification in the limit" setting, which has been widely studied for decades in the context of learning automata and grammars (De la Higuera, 2010). Our framework is successful in explaining a wide range of empirical observations (Figure 1). This is a substantial advance, as no prior theoretical framework has been able to explain the empirical patterns in Figure 1 to the extent that our framework can. We hope that further work can build on these insights to establish guarantees that reproduce this success while narrowing the gap between theoretical analysis and practical learning.

**(5)** *Why does the length generalization condition in Theorem 7 ask for $\sup_n \mathcal{R}(T_n) < \infty$? Isn't asking for length generalization sufficient?*

If $\sup_n \mathcal{R}(T_n) = \infty$, a transformer $T$ minimizing $\mathcal{R}(T)$ while fitting behavior at some length will be unlikely to work at substantially longer lengths, because performing the task correctly at longer and longer lengths requires unbounded increase in $\mathcal{R}(T)$. It might still happen that generalization from length $\frac{n}{2}$ to length $n$ is possible in certain problems not expressible by Limit Transformers. However, this will depend on the problem and the specific scaling of test lengths relative to training lengths; for problems not satisfying the conditions in Theorem 7, length generalization will fail when the test length is made sufficiently longer than the training length, even as the training length diverges to infinity. We make this formal in Section B.4.

**(6)** *Given a task, how can one settle Limit Transformer and **C-RASP** expressiveness?*

Showing that a task is definable by Limit Transformers or **C-RASP** simply requires providing an explicit construction, as we exemplify for various tasks (Section C.1). For showing that a task is not definable in these formalisms, we provide a battery of methods which allow us to provide an answer for many tasks: communication complexity (Theorem 12) applies to both formalisms; for showing non-definability in **C-RASP**, reduction to specific languages already proven not to be expressible (such as Parity and $L_{bb}$, see Appendix E.1.2) is frequently useful. Although Theorem 4.2 shows that every function expressible in **C-RASP** is expressible by Limit Transformers, it is not known whether or not this inclusion is strict.

**(7)** *Why does the guarantee specifically apply to* PERIODIC *and* LOCAL *Limit Transformers? What is special about such positional relations?*

Local positional relations are important because, if a product function of the form $\boldsymbol{p}_i^T \boldsymbol{Q}^T \boldsymbol{K} \boldsymbol{p}_j$, where the rank of $\boldsymbol{Q}, \boldsymbol{K}$ is not constrained, takes nonzero values at unboundedly long distances $j - i$, there is no general reason why the function should length-generalize. Independent initialization of the $\boldsymbol{p}_i$'s tends to lead to values close to zero for most of these products (Appendix G.1); our Inference

Procedure incorporates this via the term (8). Given this, one expects a learned model to still exhibit small products at distances not present in the training distribution, and hence a failure of length generalization in the presence of nonlocal product functions. In the translations between standard transformers and Limit Transformers, such local positional relations correspond to the functions $\phi_{l,h}$.

The situation is different for products involving $\boldsymbol{V}_{l,h}$ matrices, whose rank is penalized by $\mathcal{R}(\cdot)$; these are able to represent not local, but periodic functions. Periodicity falls out of the setup: In the finite-precision setup, a translation-invariant product function of the form $\boldsymbol{p}_i^T \boldsymbol{M}_1 \dots \boldsymbol{M}_k \boldsymbol{p}_j$ must be periodic in $j - i$ whenever one of the matrices $\boldsymbol{M}_1 \dots \boldsymbol{M}_k$ has bounded rank as the number of positions considered diverges to infinity, with period bounded in terms of the rank (Lemma 48). Hence, in a transformer $T \in \Theta_n$, any product function involving one or more $\boldsymbol{V}_{l,h}$ matrices needs to be periodic with period bounded in terms of $\mathcal{R}(T)$. In the translations between standard transformers and Limit Transformers, such periodic relations are encoded into bounded-width bounded-precision positional encodings $\boldsymbol{p}_i$ of the Limit Transformer; finite width and precision are sufficient due to periodicity.

## B    PROOFS ABOUT LIMIT TRANSFORMERS

### B.1    PROOF OF THEOREM 7

We re-state and then prove Theorem 7:

**Theorem 14** (Guaranteed Length Generalization in the Limit, restated from Theorem 7)**.** *Let* $f \in \mathcal{F}(\Sigma)$. *Then the following are equivalent:*

1. *$f$ is expressible by a Limit Transformer satisfying* PERIODIC *and* LOCAL.

2. *(Guaranteed Length Generalization) Consider the inference procedure from Definition 6 applied to $f$ with $\mathcal{R}$, generating a sequence $T_1, T_2, \dots$. For* any *such sequence, there is some $N_0$ such that, for all $m > N_0$, $T_m$ matches $f$ on all inputs of any length $k \leq m$, and* $\sup_{n=1,2,3,\dots} \mathcal{R}(T_n) < \infty$.

**Remark 15.** *We note that a limit transformer $T_\infty$ representing $f$ need not itself be offset-invariant. It is sufficient to have*

$$T_\infty(x, 0) = f(x) \tag{6}$$

*Lemma 47 shows that such a function has a sequence of transformers $T_n \in \Theta_n$ which are offset-invariant, even without assuming $T_\infty$ to be offset-invariant.*

**High-Level Proof Sketch**    The key to the proof is a compactness property: Any sequence $T_1, T_2, \dots$ ($T_i \in \Theta_i$) where $\sup_i \mathcal{R}(T_i) < \infty$ has a subsequence of transformers whose behavior across inputs can be summarized into a single Limit Transformer. For 1⇒2, given a sequence generated by the Inference Procedure, we show that $\mathcal{R}$ stays bounded and use the compactness property to show that a subsequence exhibits behavior equivalent to $f$. To show that, in fact, *all* possible sequences $T_n$ generated by the Inference Procedure ultimately exhibit behavior equivalent to $f$, when $n$ is large, we show that subsequences failing to length-generalize would exhibit increasing attention dot products between far-away positions as input length increases. However, due to the penalty on attention dot products in $\mathcal{R}$, any such sequence would, for large $n$, need to have a higher value of $\mathcal{R}$ than sequences avoiding such an increase. For 2⇒1, we obtain the Limit Transformer from the compactness property applied to the sequence generated by the Inference Procedure. The penalty on attention dot products enforces that it satisfies LOCAL; the bounds on the MLP and value matrices enforce that the positional encodings in the Limit Transformer can be taken to be periodic.

**Preliminaries and Formal Proof**    We now proceed to the formal proof. We make crucial use of the two technical Lemmas 47 and 52, which provide translations between ordinary transformers and Limit Transformers.

The following definition will be used:

**Definition 16.** *If $T \in \Theta_i$, then define $\mathcal{R}_-(T)$ to be $\mathcal{R}(T)$ minus the term in Eq. (8). That is,*

$$\mathcal{R}(T) = \mathcal{R}_-(T) + \sum_{l,h} \sum_{1 \leq j \leq N(T)} |\boldsymbol{p}_1^T \boldsymbol{K}_{l,h}^T \boldsymbol{Q}_{l,h} \boldsymbol{p}_j|^2 \tag{7}$$

The following lemma will be used for both directions of the main theorem:

**Lemma 17.** *Let $T_1, T_2, \ldots$, where $T_n \in \Theta_n$, be a sequence generated by the Inference Procedure based on the functional behavior of a function $f \in \mathcal{F}$, and such that*

$$\sup_{n=1,2,3,\ldots} \mathcal{R}(T_n) < \infty \tag{8}$$

*Then $f$ is expressible by a Limit Transformer satisfying* PERIODIC *and* LOCAL*, and there is some $N_0$ such that, for all $m > N_0$, $T_m$ matches $f$ on all inputs of length $k \leq m$.*

*Proof.* As we will be reasoning over the $\phi_{l,h}$ functions of different limit transformers, we use a superscript, $\phi_{l,h}^{(T)}$, to indicate the relevant limit transformer.

From the sequence $T_1, T_2, \ldots$ generated by the Inference Procedure, we obtain, using Lemma 52, Limit Transformers $\tilde{T}_1, \tilde{T}_2, \ldots$ such that $\sup_i \mathcal{R}_\infty(\tilde{T}_i) < \infty$ where

$$\tilde{T}_i(x, o) = T_i(x, o), \quad \forall i, o, x; |x| + o \leq i \tag{9}$$

and, in each $T_n, \tilde{T}_n$,

$$\phi_{l,h}^{(\tilde{T}_n)}(i, j) = \boldsymbol{p}_i^T \boldsymbol{K}_{l,h}^T \boldsymbol{Q}_{l,h} \boldsymbol{p}_j \tag{10}$$

Each limit transformer $\tilde{T}_i$ consists of two parts: (i) the collection of its parameter vectors and matrices, (ii) the functions $\phi_{l,h}^{(\tilde{T}_i)}$. We will write $\mathfrak{P}(\tilde{T}_i)$ for the collection of parameter vectors and matrices of $\tilde{T}_i$. Let $A := \sup_i \mathcal{R}_\infty(\tilde{T}_i) < \infty$. Then $\{\mathfrak{P}(\tilde{T}) : \mathcal{R}_\infty(\tilde{T}) \leq A\}$ is finite, because $A$ bounds (1) the number of parameters, (2) their magnitudes, (3) the precision at which they are represented. Hence, only a finite number of Limit Transformer parameter settings $\mathfrak{P}(\tilde{T}_i)$ will be traversed as $i \to \infty$. The remainder of the proof is devoted to showing that, in fact, the number of limit transformers $\tilde{T}_i$ itself is finite. For this, we need to show that each $\phi_{l,k}^{(\tilde{T}_i)}$ only traverses a finite set of distinct functions as $i \to \infty$. Each function $\phi_{l,k}^{(\tilde{T}_i)}$ is local; however, a priori, they might not be local for any single finite $\tau$ across the different $\tilde{T}_n$. We will show that this is not possible, i.e., we will show that all $\phi_{l,k}^{\tilde{T}_n}$ are local for a single finite $\tau$. This will occupy us for the remainder of the proof.

Now note that

$$\mathcal{R}(T_n) \in \left[ \inf_{T \in U_n} (\mathcal{R}(T)), \frac{1}{n} + \inf_{T \in U_n} (\mathcal{R}(T)) \right]$$

Because $\inf_{T \in U_n} \mathcal{R}(T)$ is bounded and monotonically increasing in $n$, $\inf_{T \in U_n}(\mathcal{R}(T)))$ converges to a limit, say $\tilde{R}$. Since $1/n \to 0$, the width of the interval converges to 0. The Squeeze Theorem then implies that $\mathcal{R}(T_n) \to \tilde{R}$.

For each $\tau$ and each $n$, we consider

$$D_n(\tau) = \sum_{l,h} \sum_{i=1}^{\min(n,\tau)} |\phi_{l,h}^{(\tilde{T}_n)}(1, i)|^2 \leq \mathcal{R}(T_n)$$

As $\phi_{l,h}^{(\tilde{T}_n)}$ has precision bounded in terms of $\mathcal{R}(T_n)$, $\{D_n(\tau) : n \in \mathbb{N}\}$ is a discrete set. An important consequence is that any accumulation point of the sequence $(D_n(\tau))_{n \in \mathbb{N}}$ must equal $D_n(\tau)$ for infinitely many $n$.

Consider $\mathcal{R}_-(T_n)$ (Equation 7). Let

$$R_0 := \liminf_{n \to \infty} \mathcal{R}_-(T_n) \tag{11}$$

and let $\nu_1, \nu_2, \nu_3, \ldots$ be such that

$$\lim_{i \to \infty} \mathcal{R}_-(T_{\nu_i}) = R_0 \tag{12}$$

Then, for some $D_0$,

$$\lim_{i \to \infty} D_{\nu_i}(\nu_i) = D_0 \tag{13}$$

and

$$\tilde{R} = \lim_{n \to \infty} \mathcal{R}(T_n) = \lim_{i \to \infty} \mathcal{R}(T_{\nu_i}) = R_0 + D_0 \tag{14}$$

Indeed,

$$D_0 = \limsup_{n \to \infty} D_n(n) \tag{15}$$

because[3]

$$D_0 + R_0 = \lim_{n \to \infty} \left( \mathcal{R}_-(T_n) + D_n(n) \right) = \liminf_{n \to \infty} \mathcal{R}_-(T_n) + \limsup_{n \to \infty} D_n(n) \tag{16}$$

Define, for each $\tau \in \mathbb{N}$,

$$D_\infty(\tau) = \liminf_{i \to \infty} D_{\nu_i}(\tau) \tag{17}$$

As this function is monotonically increasing, and as each $\phi_{l,h}^{(\tilde{T}_n)}$ has precision bounded in terms of $\mathcal{R}(T_n)$, there must be $\tau_\infty$ such that $D_\infty(\tau_\infty) = \lim_{\tau \to \infty} D_\infty(\tau)$.

Now define a sequence $T_n'$ as follows. For each $n$, recall from the definition of $D_\infty$ that

$$\liminf_{j \to \infty} D_{\nu_j}(n) = D_\infty(n) \leq D_\infty(\tau_\infty)$$

As $\phi_{l,h}$ has bounded precision, there are infinitely many $\nu_i$ such that $D_{\nu_i}(n) = \liminf_{j \to \infty} D_{\nu_j}(n)$. Hence, we can select $i(n) \in \mathbb{N}$ such that $\nu_{i(n)} \geq n$ and

$$D_{\nu_{i(n)}}(n) = \liminf_{j \to \infty} D_{\nu_j}(n)$$

Then, for each $n$, define $T_n'$ as the restriction of $T_{\nu_{i(n)}}$ to positions up to $n$. As $T_{\nu_{i(n)}}$ agrees with the behavior of $f$ up to length $\frac{\nu_{i(n)}}{2} \geq \frac{n}{2}$, we also find that $T_n'$ agrees with the behavior of $f$ up to length $\frac{n}{2}$. Then

$$\begin{aligned}
\limsup_{n \to \infty} \mathcal{R}(T_n') &= \limsup_{n \to \infty} \mathcal{R}_-(T_n') + D_{\nu_{i(n)}}(n) \\
&= \limsup_{n \to \infty} \mathcal{R}_-(T_{\nu_{i(n)}}) + D_\infty(\tau_\infty) \\
&= R_0 + D_\infty(\tau_\infty)
\end{aligned}$$

Since $T_n$ was created by the Inference Procedure, we have

$$\limsup_{n \to \infty} \mathcal{R}(T_n') \geq \lim_{n \to \infty} \mathcal{R}(T_n) \tag{18}$$

On the other hand, since $\mathcal{R}(T_n') \leq \mathcal{R}(T_{\nu_{i(n)}})$, we also have

$$\limsup_{n \to \infty} \mathcal{R}(T_n') \leq \lim_{n \to \infty} \mathcal{R}(T_n) \tag{19}$$

giving

$$\limsup_{n \to \infty} \mathcal{R}(T_n') = \lim_{n \to \infty} \mathcal{R}(T_n) = D_0 + R_0 \tag{20}$$

Hence,

$$\begin{aligned}
R_0 + D_\infty(\tau_\infty) &= \limsup_{n \to \infty} \mathcal{R}(T_n') \\
&= \lim_{n \to \infty} \mathcal{R}(T_n) \\
&= R_0 + D_0
\end{aligned}$$

---

[3]In general, if $a_n + b_n$ converges and $a_n, b_n$ are bounded, then the limit $\lim(a_n + b_n)$ equals $\limsup a_n + \liminf b_n$. For, assume $\limsup a_n + \liminf b_n > \lim(a_n + b_n)$ (similar if $>$ is replaced by $<$). Then let $i(n)$ be a subsequence such that $a_{i(n)} \to \limsup a_n$. Then $\lim(a_n + b_n) = \lim(a_{i(n)} + b_{i(n)}) = \limsup a_n + \lim b_{i(n)} \geq \limsup a_n + \liminf b_n > \lim(a_n + b_n)$, contradiction.

and $D_\infty(\tau_\infty) = D_0$. Now assume there are infinitely many $n$ such that $\phi_{l,h}^{(T_n)}$ is not $\tau_\infty$-local, hence, infinitely many $n$ such that $D_n(n) \geq D_n(\tau_\infty) + 2^{-2p}$. Then:

$$\begin{aligned}
D_0 &= \limsup_{n\to\infty} D_n(n) \\
&\geq \limsup_{n\to\infty} D_n(\tau_\infty) + 2^{-2p} \\
&\geq \liminf_{i\to\infty} D_{\nu_i}(\tau_\infty) + 2^{-2p} \\
&= D_0 + 2^{-2p}
\end{aligned}$$

This is a contradiction. Here, the first inequality holds because $D_n(n) \geq D_n(\tau_\infty)$ whenever $n \geq \tau_\infty$, simply because $D_n(\cdot)$ is monotonically increasing for each individual $n$. The second inequality holds because $(\nu_i)_{i\in\mathbb{N}}$ is a subsequence of $(n)_{n\in\mathbb{N}}$; hence a $\limsup$ over the larger sequence upperbounds the $\liminf$ over the subsequence. Overall, and as announced at the beginning of the proof, we thus have shown that the functions $\phi_{l,h}^{(\tilde{T}_n)}$ must be local for a uniform $\tau_\infty$.

Because each $\phi_{l,h}^{(\tilde{T}_n)}(i,j)$ equals an inner product $\boldsymbol{p}_i^T \boldsymbol{K}_{l,h}^T \boldsymbol{Q}_{l,h} \boldsymbol{p}_j$, all values are expressed at precision bounded by $(R_0)^4$, and are bounded in absolute value by $\leq \|\boldsymbol{p}_i\|_2 \|\boldsymbol{K}_{l,h}^T\| \|\boldsymbol{Q}_{l,h}\| \|\boldsymbol{p}_j\|_2 \leq (R_0)^4$. There are only a finite set of functions that satisfy these properties and are are local for this $\tau_\infty$.

Above, we have remarked that each limit transformer $\tilde{T}$ consists of (i) the collection $\mathfrak{P}(\tilde{T})$ of parameter matrices and vectors, (ii) the collection of functions $\phi_{l,h}^{(\tilde{T})}(i,j)$. Hence, we know that,

$$\mathcal{Q} := \{\tilde{T}_i : i \in \mathbb{N}\} \tag{21}$$

is finite. Let $\mathcal{Q}_\infty \subseteq \mathcal{Q}$ be the set of Limit Transformers that equal $\tilde{T}_i$ for infinitely many different $i$. By definition of the Inference Procedure, every element of $\mathcal{Q}_\infty$ is functionally equivalent to $f$ at all input lengths. Because $\mathcal{Q}$ is finite, there is $N_0$ such that $\tilde{T}_i \in \mathcal{Q}_\infty$ for each $i \geq N_0$. Hence, $T_i$ is functionally equivalent to $f$ at all lengths $\leq i$ as soon as $i$ exceeds the threshold $N_0$. $\qquad\square$

We now prove the theorem.

*Proof of the Theorem.* Both directions are corollaries of Lemma 17.

**2⇒1:** This directly follows from Lemma 17.

**1⇒2:** By Lemma 47, for each $i = 1, 2, 3, \ldots$, there are $\widehat{T}_i \in \Theta_i$ such that $R := \sup_i \mathcal{R}(\widehat{T}_i) < \infty$ such that

$$\widehat{T}_i(x, o) = f(x, o), \quad \forall i, o, x; |x| + o \leq i \tag{22}$$

such that

$$\boldsymbol{p}_i^T \boldsymbol{K}_{l,h}^T \boldsymbol{Q}_{l,h} \boldsymbol{p}_j = \phi_{l,h}^{(T)}(i,j) \tag{23}$$

By LOCAL,

$$\mathcal{R}(\widehat{T}_i) < \infty \tag{24}$$

and we conclude

$$\limsup_{i\to\infty} \mathcal{R}(T_i) \leq \limsup_{i\to\infty} \mathcal{R}(\widehat{T}_i) < \infty \tag{25}$$

where $T_i$ refers to the sequence generated by Inference Procedure in Lemma 17. Lemma 17 now provides $N_0 > 0$ and a function $g$ such that for all $m > N_0$,

$$T_m(x, o) = g(x), \forall x : |x| + o \leq m \tag{26}$$

On the other hand, for any string $x \in \mathfrak{S}$, we have

$$f(x) = T_n(x, 0), \forall n \geq 2|x| \tag{27}$$

Hence, $f \equiv g$ and for all $m > N_0$,

$$T_m(x, o) = f(x), \forall x : |x| + o \leq m \tag{28}$$

$$\square$$

## B.2 RESULT FOR NoPE TRANSFORMERS

**Corollary 18.** *For ease of the reader, we mark the differences to Theorem 7 in* blue font.

*Let $f \in \mathcal{F}(\Sigma)$. Then the following are equivalent:*

1. *$f$ is expressible by a Limit Transformer where all $\boldsymbol{p}_i \equiv 0$, $\phi_{l,h} \equiv 0$.*

2. *(Guaranteed Length Generalization) Consider the inference procedure from Definition 6 applied to $f$ with $\mathcal{R}$ while constraining all $\boldsymbol{p}_i \equiv 0$, generating a sequence $T_1, T_2, \dots$. For any such sequence, there is some $N_0$ such that, for all $m > N_0$, $T_m$ matches $f$ on all inputs of any length $k \leq m$, and $\sup_{n=1,2,3,\dots} \mathcal{R}(T_n) < \infty$.*

Note that a Limit Transformer where all $\phi_{l,h} \equiv 0$ is also an *ordinary transformer* as defined in Section 2. Hence, in the special case of NoPE transformers, our proof boils down to an argument using standard transformers, effectively without Limit Transformers. In contrast, Limit Transformers are key to our proof in the more general case of APE transformers.

*Proof.* Retracing the proof of Lemma 47 shows that, when translating a Limit Transformer to an ordinary transformer, the positional encodings can be taken to be zero when $\boldsymbol{p}_i \equiv 0$, $\phi_{l,h} \equiv 0$ in the Limit Transformer. Retracing the proof of Lemma 52 shows that, when $\boldsymbol{p}_i \equiv 0$ in a transformer, the resulting Limit Transformer will have zero positional encodings and zero outputs for all $\phi_{l,h}$. The proof of Theorem 7 then applies equally to show Corollary 18. □

## B.3 LOGARITHMIC COMMUNICATION COMPLEXITY FOR LIMIT TRANSFORMER

**Theorem 19** (Restated from Theorem 12). *Let $T$ be a Limit Transformer satisfying* PERIODIC *and* LOCAL. *On an input $x \in \Sigma^{2N}$, assume Alice has access to $x_{1\dots N}$ and Bob has access to $x_{N+1\dots 2N}$. There is a communication protocol in which Alice and Bob exchange at most $C \log N$ bits, where $C$ depends on $T$ but not $N$ or $x$, and Bob can compute each activation in the second half, $\boldsymbol{y}_i^{(l)}$ ($N + 1 \leq i \leq 2N$). Further, $C$ is bounded linearly by $\mathcal{R}_\infty(T)$.*

*Proof.* We first establish that all activations $\boldsymbol{y}_i^{(l)}$ are computed at $O(\log N)$ precision. This holds because (i) parameters are at fixed precision, (ii) the output of $\exp(\cdot)$ in the softmax attention computation is computed at fixed fractional precision, (iii) and hence attention weights can be represented at $O(\log N)$ precision.

We first consider the attention logits, in the case where $j < N \leq i$:

$$a_{i,j}^{(l,h)} = \text{Round}_p[(\boldsymbol{y}_j^{(l-1)})^T \boldsymbol{K}_{l,h}^T \boldsymbol{Q}_{l,h} \boldsymbol{y}_i^{(l-1)} + \phi_{l,h}(i,j)]$$

where $\text{Round}_p[\dots]$ rounds each entry to the closest number with $p$ fractional bits. It is certainly sufficient to have access to

$$a_{i,j}^{(l,h)} = \left(\text{Round}_{p'}[\boldsymbol{y}_j^{(l-1)}]\right)^T \boldsymbol{K}_{l,h}^T \boldsymbol{Q}_{l,h} \boldsymbol{y}_i^{(l-1)} + \text{Round}_{p'}[\phi_{l,h}(i,j)]$$

where $p'$ depends on $p$ and the largest singular value of $\boldsymbol{K}_{l,h}^T \boldsymbol{Q}_{l,h}$, which is a finite constant. We can thus partition the positions $j = 1, \dots, N - 1$ into a bounded number of sets, indexed by

1. $\text{Round}_{p'}[\boldsymbol{y}_j^{(l-1)}]$

2. $\max(N - j, N - L)$ where $L = \max\{k : \phi_{l,h}(1, k) \neq 0\}$.

Let $\mathcal{K} \in \mathbb{N}$ be the number of these sets; we then write these sets as $\mathcal{A}_1, \dots, \mathcal{A}_{\mathcal{K}}$. Note that, while these sets are always disjoint, their elements are input-dependent as they depend on the activations $\boldsymbol{y}_j^{(l-1)}$.

Due to the finite precision rounding of logits and the locality of positional relations, we can maintain a finite set of keys and queries (though not values). This is fundamental to getting a logarithmic communication bound.

We show the claim by induction over the layers.

We can write

$$\boldsymbol{Y}_i^{(l)} = \boldsymbol{y}_i^{(l-1)} + \sum_{h=1}^{H} \frac{\sum_{j=1}^{i} \exp(\log|x| \cdot a_{i,j}^{(l,h)}) \boldsymbol{V}_{l,h} \boldsymbol{y}_j^{(l-1)}}{\sum_{j=1}^{i} \exp(\log|x| \cdot a_{i,j})}$$

The residual stream is known to Bob by inductive hypothesis. We need to understand the term inside the sum. The green terms are fully known to Alice, the blue ones are fully known to Bob by inductive hypothesis:

$$\frac{\sum_{j=1}^{i} \exp(\log|x| \cdot a_{i,j}^{(l,h)}) \boldsymbol{V}_{l,h} \boldsymbol{y}_j^{(l-1)}}{\sum_{j=1}^{i} \exp(\log|x| \cdot a_{i,j})}$$

$$= \frac{\sum_{j=1}^{N-1} \exp(\log|x| \cdot a_{i,j}^{(l,h)}) \boldsymbol{V}_{l,h} \boldsymbol{y}_j^{(l-1)}}{\sum_{j=1}^{N-1} \exp(\log|x| \cdot a_{i,j}) + \sum_{j=N}^{i} \exp(\log|x| \cdot a_{i,j})}$$

$$+ \frac{\sum_{j=N}^{i} \exp(\log|x| \cdot a_{i,j}^{(l,h)}) \boldsymbol{V}_{l,h} \boldsymbol{y}_j^{(l-1)}}{\sum_{j=1}^{N-1} \exp(\log|x| \cdot a_{i,j}) + \sum_{j=N}^{i} \exp(\log|x| \cdot a_{i,j})}$$

Alice thus needs to communicate the green terms. More formally, for every set $r = 1, \dots, \mathcal{K}$, Alice communicates

1. the number of relevant positions, $|\mathcal{A}_r| \in \{1, \dots, N-1\}$

2. $\sum_{i \in \mathcal{A}_r} \boldsymbol{V}_{l,h} \boldsymbol{y}_j^{(l-1)}$.

For each $r$, communicating (1) takes $O(\log N)$ bits. The vectors in (2) are bounded in norm by the $l$-th power of the maximum spectral norm of any parameter matrix, times the maximum $\ell_2$ norm of any parameter vector. Overall, this is bounded in terms of $R_\infty(T)$, thus $O(1)$ for fixed $T$. Further, each $\boldsymbol{y}_j^{(l-1)}$ is expressed in $O(\log N)$ precision as shown above. Expressing (2) requires $O(\log N)$ fractional bits (as the sum will not need more fractional bits than the individual vectors) and $O(\log N)$ integer bits (due to the bound on the norm). Overall, Alice needs to communicate $\mathcal{K} \cdot O(\log N)$ bits. As $\mathcal{K}$ is independent of $N$, this is $O(\log N)$ for a fixed Limit Transformer $T$.

Alice can partition the positions into a bounded number of partitions, and for each of them needs to transfer the number of positions in that partition.

$\square$

**Corollary 20** (Restated from Corollary 13). *The following problems are not expressible by Limit Transformers satisfying* PERIODIC *and* LOCAL*: (1) copying strings with repeated n-grams, (2) addition of $n$-digit numbers.*

*Proof.* Formally, we define *copying* as the task of, given a prefix \$$x$\#, autoregressively predicting $x$. Copying *with repeated n-grams* means that there is no restriction on the repetition of consecutive subspans of $x$ of any length; this is in contrast to copying tasks with restrictions on the repetition of n-grams (for some $n$) in $x$ (Jelassi et al., 2024; Zhou et al., 2024a), which we study separately (Appendix E.2).

Formally, we define *addition* as the task of, given a prefix \$$x + y =$, where $x, y$ are binary strings, to output the sum of the numbers denoted by $x, y$ in binary.

The communication complexity lower bound for copying follows from a standard communication complexity lower bound for determining string equality. The bound follows for addition since the special case of adding 0 to a number amounts to copying. $\square$

**Remark 21.** *Analogous bounds follow for various other algorithmic and formal language problem. For instance, the special case of multiplying with 1 amounts to copying; hence, such a bound holds for multiplication. For the unbounded-depth Dyck over two bracket types, we can consider a word of the form $(_{i_1} \ldots (_{i_N})_{j_N} \ldots)_{j_1}$, which is in the Dyck language if and only if $i_k = j_k$ for all $k$, again allowing a reduction to the communication complexity lower bound for determining string equality.*

### B.4 STATEMENT OF MAIN THEOREM FOR ARBITRARY TRAINING LENGTHS

Our main theorem considers generalization from length $\frac{n}{2}$ to length $n$. Here, we discuss an alternative version applying to arbitrary scaling of training vs testing lengths. In particular, in such a setup, we explicitly obtain *failure of length generalization for inexpressible functions*, though potentially requiring testing on lengths more than twice the lengths used in training. We use the following definition:

**Definition 22.** *A* training length *is a function* $t : \mathbb{N} \to \mathbb{N}$ *satisfying* $\lim_{t \to \infty} t(n) = +\infty$ *and* $t(n) \leq n$ *for all* $n$.

*If $t(n)$ is a training length, then the $t(n)$-Inference Procedure determines $T_n \in \Theta(n)$ to match $f$ at all inputs of lengths $\leq t(n)$ while minimizing $\mathcal{R}(T_n)$ up to $\frac{1}{n}$.*

*The special case of $t(n) = \frac{n}{2}$ is the Inference Procedure from Definition 6.*

We then state:

**Theorem 23.** *Let $f \in \mathcal{F}(\Sigma)$. The following are equivalent:*

1. *$f$ is expressible by a Limit Transformer satisfying* PERIODIC *and* LOCAL.

2. *Let $t(n)$ be any training length. Then the $t(n)$-Inference Procedure will output solutions $T_1, T_2, \ldots$ such that, for some $N_0$, for all $m > N_0$, $T_m$ matches $f$ at all lengths $\leq m$.*

   *Intuitively, this says that, when selected to fit the behavior of $f$ on sufficiently long inputs of length $t(n)$, the output of the Inference Procedure will generalize to unboundedly longer inputs of length $n$, where $n$ can be arbitrarily larger than $t(n)$.*

**Corollary 24.** *Assume $f \in \mathcal{F}(\Sigma)$ is not expressible by a Limit Transformer satisfying* PERIODIC *and* LOCAL. *Then, for some training length $t(n)$, the $t(n)$-Inference Procedure outputs a sequence $T_n$ where infinitely many $T_n$ fail to match $f$ at length $n$.*

**Remark 25.** *There are two important differences compared to Theorem 7. First, the second condition refers to length generalization for all arbitrary training lengths $t(n)$, not specifically training length $\frac{n}{2}$. Second, the second condition does not ask for $\sup_i \mathcal{R}(T_i) < \infty$, but simply asks for $T_n$ to ultimately length generalize.*

*Proof of Theorem 23.* 1$\Rightarrow$2 The proof of Theorem 7 remains valid in this direction without any changes, as it does not specifically rely on the training lengths being half the overall context size.

2$\Rightarrow$1 We show the contrapositive. Assume $f$ is not expressible by a Limit Transformer satisfying PERIODIC and LOCAL. Then, using the same arguments as in the proof of Lemma 17[4], any sequence $T_n \in \Theta_n$ that matches $f$ will have $\liminf_{n \to \infty} \mathcal{R}(T_n) = \infty$ (†). Now consider $k \in \mathbb{N}$; we will assign every $k$ a number $n_k > k$, starting with $n_0 = 0$. For each $n > k$, there is $\hat{T}_{k,n} \in \Theta_n$ that matches $f$ up to length $k$ while $U_k := \sup_n \mathcal{R}(\hat{T}_{k,n}) < \infty$ for every fixed $k$. Now select $n_k > n_{k-1}$ such that no $T \in \Theta_{n_k}$ with $\mathcal{R}(T) \leq U_k + 1$ matches $f$ at length $n_k$; this is possible because of (†). We thus obtain a sequence $(k, n_k) \in \mathbb{N} \times \mathbb{N}$. By construction, there are infinitely many distinct different values $n_k$. Then define

$$t(n) := \max\left(\{k : n_k \leq n\}\right) \tag{29}$$

Then $t(n)$ is a training length. By definition, the $t(n)$-Inference Procedure will, whenever $n$ is one of the $n_k$'s, find a transformer $T_{n_k}$ with $\mathcal{R}(T_{n_k}) \leq U_k + \frac{1}{n_k}$ that fails to match $f$ at length $n = n_k$. $\square$

---

[4]Assume there is a sequence $T_n \in \Theta_n$ that matches $f$ and has $\liminf_{n \to \infty} \mathcal{R}(T_n) < \infty$. Translating each element to a Limit Transformer leads to a sequence where, except perhaps for the functions $\phi_{l,h}$, only a finite number of settings will be traversed. Now, as in the proof of Lemma 17, one can use $D_\infty(\tau)$ to construct a sequence of Limit Transformers that are local for a single $\tau$. The important difference to Lemma 17 is that here we are not assuming the sequence $(T_n)_n$ to be constructed by the inference procedure, but we nonetheless obtain such a sequence.

### B.5 Corollary about Expressivity

We have introduced Limit Transformers as a formalism for distilling computations of transformers performing on longer and longer sequences into a single limiting object, helping understand length generalization. Here, we show that they also provide a simple lower bound for the expressiveness of causal transformers across input lengths:

**Corollary 26.** *Let $f \in \mathcal{F}(\Sigma)$. Assume $f$ is expressible by a Limit Transformer satisfying* PERIODIC *and* LOCAL. *Then at each input length $N$, there exists a transformer $T_N$ performing $f$ on all inputs of length up to $N$ such that:*

1. *The parameters of $T_N$ are expressed at $p$ bit precision, with $p$ independent of $N$*

2. *The number of heads and layers of $T_N$ is bounded independently of $N$.*

3. *The width $d$ of $T_N$ is bounded as $O(N)$.*

We note that an important aspect is that $T_N$ performs correctly not just at length $N$, but at *all lengths up to $N$*. This distinguishes the result from constructions guaranteeing the existence of a transformer at a fixed length. For instance, Bhattamishra et al. (2024) provide a transformer for testing equality between length $N$-strings (which could also be used for copying), but this construction uses specific positional encodings that depend on the input length. In contrast, the result here provides conditions under which a transformer can perform a task at all lengths up to a given bound; in this stronger setup, no APE transformer for copying with uniform complexity bounds as provided by Corollary 26 is known, and the problem is indeed not expressible by Limit Transformers satisfying PERIODIC and LOCAL (Corollary 13). In contrast, Corollary 26 provides APE constructions performing correctly *up to* any given length for a wide class of problems including **C-RASP**[periodic,local].

Another important feature is that the construction provides a fixed precision for the parameters, as is the case in real-world implementation. We note that, if parameters are at fixed precision, it is generally not possible to find a single transformer across all input lengths in the APE setting; hence, it is unavoidable that the width of the transformers will need to increase as the input length increases. Importantly, many other aspects of the transformer's complexity, such as the number of heads and layers, remain bounded.

*Proof.* The statement is an immediate corollary of Lemma 47, which provides transformers $T_1, T_2, \ldots$ with bounded $\mathcal{R}(T_N)$, which by Definition 5 entails a uniform bound on precision, heads, and layers. The construction provided in the proof of Lemma 47 provides a width bounded as $O(N)$. $\square$

### B.6 From **C-RASP** to Limit Transformers

The proofs are adaptations of the proofs from Yang & Chiang (2024).

**Theorem 27** (Restated from Theorem 9). *For every* **C-RASP**$[\Phi, \Psi]$ *program $P$ with local functions $\Psi$ and any periodic functions $\Phi$ there exists a Limit Transformer $T_\infty$ that satisfies* PERIODIC *and* LOCAL *such that for all $w \in \Sigma^*$, $P$ accepts $w$ iff $T_\infty$ accepts $\$w$. If $P$ uses no local or periodic relations, then $T$ requires no functions $\phi_{l,h}$ or positional encodings $\mathbf{p}_i$.*

**Remark 28.** *We note that the Limit Transformer $T_\infty$ provided by the proof of Theorem 27 emulates the* **C-RASP** *program $P$ at zero offset: That is, $P$ accepts $w$ iff a predetermined entry in the last output dimension of $T_\infty(\$w, 0)$ is above some threshold. In principle, its computations may not be offset-invariant, i.e., for the constructed $T_\infty$, the output $T_\infty(\$w, o)$ may depend on $o$. Importantly, the proof of Theorem 7 does not require a Limit Transformer computing $f$ to be offset-invariant, but just requires it to compute $f$ when the offset is zero. This is because Lemma 47 ensures that, for any Limit Transformer $T_\infty$ satisfying* LOCAL *and* PERIODIC, *even if it is not offset-invariant, there are transformers $T_n \in \Theta_n$ whose behavior matches $T_\infty(\cdot, 0)$.*

*Proof of Theorem 27.* **C-RASP** has two sorts of operations, a Boolean sort and a Count sort. We will simulate each operation in the transformer by storing the Boolean values as $\{0, 1\}$, and storing the counts as $\frac{c}{i+1}$. That is, we say that a Limit Transformer $T_\infty$ simulates a **C-RASP** program $P$ if for every operation $P_k$ of $P$ there is a dimension $d_k$ in $T$ such that when $P_k(i)$ when run on $w$ is true iff

$T_\infty(\$w)_{i+1,d_k} = 1$ (and 0 otherwise) for Boolean operation and $P_k(i) = c$ iff $T_\infty(\$w)_{i+1,k} = \frac{c}{i+1}$ for count operations.

The theorem will be shown by induction on the length of $P$. As a clarifying note, we use 0-indexing everywhere in this proof. If $P$ is of length 0, we only have initial $Q_\sigma(i)$ vectors, which can be simulated by appropriately setting the word embedding. Otherwise, assume all programs of length $\leq k$ are simulated by some transformer, and we have cases for each type of operation $P_{k+1}(i)$ can be. All cases are identical to Yang & Chiang (2024) except for comparison, conditional, and counting.

First, we must address the SOS token $. There exists an transformer layer which sets the entire vector to $\mathbf{0}$ in the initial position, while leaving all other layers untouched. For instance, we may use a conditional operation, as described later in the proof.

If $P_{k+1}(i) := \phi(i)$, a periodic positional function in $\Phi$, then it is simulated in $T_\infty$ by appropriately setting $\boldsymbol{p}_i$ in the positional encoding.

If $P_{k+1}(i) := P(i) ? C_1(i) : C_2(i)$, we can implement the following function: for $P \in \{0,1\}$ and $V \in [0,1]$

$$f(P, V) = \begin{cases} V & P = 0 \\ 0 & P = 1 \end{cases}$$

This is achieved by $f(P, V) = \mathrm{ReLU}(V - P)$. Thus, the desired Conditional Output can be defined in a single FFN as $f(P, V_1) + f(-P, V_2)$, where the first layer and $\mathrm{ReLU}$ compute each $f$ term and the second layer adds them together.

If $P_{k+1}(i) := C_1(i) \leq C_2(i)$. By the inductive hypothesis $C_1(i)$ and $C_2(i)$ are stored in dimensions $d_1$ and $d_2$ as the value $\frac{C_1(i)}{i+1}$ and $\frac{C_2(i)}{i+1}$. It suffices to check that $\frac{C_2(i)}{i+1} - \frac{C_1(i)}{i+1} \geq 0$.

To compute this, we use the heaviside activation function, which we used in our model of MLPs as discussed in D.1.

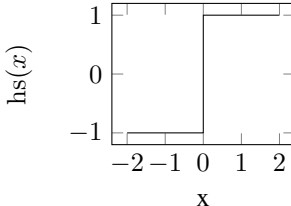

Thus, there exists an MLP which, letting $x_1$ and $x_2$ be the values in dimensions $d_1$ and $d_2$, computes $(\mathrm{hs}(x_2 - x_1) + 1)/2$ in the dimension reserved for $P_{k+1}$, which will be the Boolean value in $\{0,1\}$ corresponding to $C_1(i) \leq C_2(i)$.

If $C(i) := \#[j \leq i] \; P(j)$ (using $\psi(i,j) = \top$), then the desired sum is computed using uniform attention, since the boolean representation of $P(j)$ is just 0 or 1. We enforced that $P(0)$ is false, so it does not contribute to the sum. This is described in more detail in Yang & Chiang (2024), though the case here is simpler.

If $C(i) := \#[j \leq i, \psi(i,j)] \; P(j)$, we can think of it as implementing $\#[j \leq i] \; \psi(i,j) \wedge P(j)$. Suppose $\psi$ is a local function of the following form

$$\psi(i,j) = \begin{cases} 1 & j = i - \ell \\ 0 & \text{else} \end{cases}$$

Then $C(i)$ will either be 1 or 0 depending if $P(i - \ell)$ is true or false. If we set the query and key matrices to 0 we get

$$s_{ij} = \log N \cdot \psi(i,j)$$

We assume the $\log$ is base 2, but the argument is similar for others. Then we can have attention compute

$$c_{i,k} = \frac{\displaystyle\sum_{j \leq i} \exp\left(\log N \cdot \psi(i,j)\right) \cdot P(j)}{\displaystyle\sum_{j \leq i} \exp\left(\log N \cdot \psi(i,j)\right)} = \frac{\displaystyle\sum_{j \leq i} N^{\left(\frac{\psi(i,j)}{\ln 2}\right)} \cdot P(j)}{\displaystyle\sum_{j \leq i} N^{\left(\frac{\psi(i,j)}{\ln 2}\right)}}$$

If $P(i - \ell)$ and $\neg P(j)$ for $j \neq i - \ell$, then we have a lower bound:

$$\frac{N^{\left(\frac{1}{\ln 2}\right)}}{N^{\left(\frac{1}{\ln 2}\right)} + i - 1} \leq c_{i,k}$$

If $\neg P(i - \ell)$ and $P(j)$ for $j \neq i - \ell$ then we have an upper bound:

$$c_{i,k} \leq \frac{i - 1}{N^{\left(\frac{1}{\ln 2}\right)} + i - 1}$$

Since $N^{\frac{1}{\ln 2}} \geq i$, and we know that $P(i-\ell) \iff c_{i,k} \geq \frac{1}{2}$, we can construct an MLP that computes the correct value. It will output either $\frac{0}{i+1}$ or $\frac{1}{i+1}$, in the dimension reserved for $P_{k+1}(i)$, for instance by using a conditional operation that checks that the output of the attention layer $c_{i+1,k} \geq \frac{1}{2}$, which was shown in an earlier case. $\qquad\square$

## C    EXPRESSIVITY PROOFS FOR **C-RASP**

### C.1    **C-RASP** CONSTRUCTIONS

#### C.1.1    MAJORITY

MAJORITY is the language of strings over $\Sigma = \{0, 1\}$ with at least as many 1's as 0's.

| MAJORITY | |
|---|---|
| $C_1(i) := \#\,[j \leq i]\;Q_1(j)$ | (1) |
| $C_0(i) := \#\,[j \leq i]\;Q_0(j)$ | (2) |
| $M(i) := C_1(i) \geq C_0(i)$ | (3) |

#### C.1.2    DYCK-1

Dyck-1 is the language of strings over $\Sigma = \{0, 1\}$ with at least as many 1's as 0's.

| Dyck-1 | | |
|---|---|---|
| $C_{(}(i) := \#\,[j \leq i]\;Q_{(}(j)$ | The number of ( up to position $i$ | (1) |
| $C_{)}(i) := \#\,[j \leq i]\;Q_{)}(j)$ | The number of ) up to position $i$ | (2) |
| $V(i) := C_{(}(i) < C_{)}(i)$ | *Violation*: there are more ) than ( | (3) |
| $C_V(i) := \#\,[j \leq i]\;V(j)$ | The number of *Violations* | (4) |
| $M(i) := C_V(i) = 0$ | *Matched*: zero *Violations* | (5) |
| $B(i) := C_{(}(i) = C_{)}(i)$ | *Balanced*: same number of ( and ) | (6) |
| $D(i) := M(i) \wedge B(i)$ | String is *Matched* and *Balanced* | (7) |

### C.1.3 $a^n b^n c^n$

Let $\Sigma = \{a, b, c\}$. This is another example of a counter language which **C-RASP** can express and which transformers have been observed to length generalize on (Bhattamishra et al., 2020).

---

**$a^n b^n c^n$**

$$C_a(i) := \#\,[j \leq i]\ Q_a(j) \tag{1}$$
$$C_b(i) := \#\,[j \leq i]\ Q_b(j) \tag{2}$$
$$C_c(i) := \#\,[j \leq i]\ Q_c(j) \tag{3}$$
$$A(i) := C_b(i) + C_c(i) = 0 \tag{4}$$
$$B(i) := C_c(i) = 0 \tag{5}$$
$$C_A(i) := \#\,[j \leq i]\ Q_a(j) \wedge A(j) \tag{6}$$
$$C_B(i) := \#\,[j \leq i]\ Q_b(j) \wedge B(j) \tag{7}$$
$$G_a(i) := C_A(i) = C_a(i) \tag{8}$$
$$G_b(i) := C_B(i) = C_b(i) \tag{9}$$
$$G_{abc}(i) := C_a(i) = C_b(i) = C_c(i) \tag{10}$$
$$L(i) := G_a(i) \wedge G_b(i) \wedge G_{abc}(i) \tag{11}$$

---

### C.1.4 Existential Quantification

This is generally a useful primitive, so to save a little space we can add a macro for existential quantification towards the left in **C-RASP**. This is easily defined using counting:

---

**$P(i) := \overleftarrow{\exists}\, A(i)$**

$$C(i) := \#\,[j \leq i]\ A(j) \tag{1}$$
$$P(i) := C(i) \geq 1 \tag{2}$$

---

And we abbreviate this using $P(i) := \overleftarrow{\exists}\, A(i)$. We demonstrate its use below.

### C.1.5 Piecewise Testable Languages

Piecewise testable languages are Boolean combinations of languages of the form $\Sigma^* a_1 \Sigma^* a_2 \Sigma^* \ldots \Sigma^* a_n \Sigma^*$. This allows us to check for the presence of noncontiguous substrings, which contrasts with the proof in C.3.2 that implies the presence of contiguous substrings cannot be expressed in **C-RASP**$[\emptyset]$.

It suffices to show programs for languages of the form $L = \Sigma^* a_1 \Sigma^* a_2 \Sigma^* \ldots \Sigma^* a_n \Sigma^*$, since Boolean combinations are recognizable using Boolean operations of **C-RASP**. For $L$ we have the following **C-RASP** program which has the final accepting operation $L_n$:

---

**$\Sigma^* a_1 \Sigma^* a_2 \Sigma^* \ldots \Sigma^* a_n \Sigma^*$**

$$L_1(i) := \overleftarrow{\exists}\, Q_{a_1}(i) \qquad\qquad a_1 \text{ occurred} \tag{1}$$
$$L_2(i) := \overleftarrow{\exists}\, Q_{a_2}(i) \wedge L_1(i) \qquad\qquad a_2 \text{ occurred, preceded by } a_1 \tag{2}$$
$$\vdots \tag{3}$$
$$L_n(i) := \overleftarrow{\exists}\, Q_{a_n}(i) \wedge L_{n-1}(i) \qquad\qquad a_n \text{ occurred, preceded by } a_{n-1}, \ldots, \text{ preceded by } a_1 \tag{4}$$

---

## C.2 **C-RASP**[PERIODIC, LOCAL] CONSTRUCTIONS

### C.2.1 INDUCTION HEAD (ARGMAX VERSION)

As an example consider $\Sigma = \{a, b, c\}$. Predicate $NEXT_a(i)$ is true iff the next token should be an $a$. First we can define predecessor

$$CP_a(i) := \#[j \leq i, j = i - 1] \ Q_a(j)$$
$$PRED_a(i) := CP_a(i) \geq 1$$

Then we can count bigram occurence by counting

$$C_{ab} := \#[j \leq i] \ Q_b(j) \wedge PRED_a(j)$$

Then each $NEXT_a(i)$ predicate can be defined by checking the current symbol and finding the most frequently occuring bigram.

---

$NEXT_a(i)$ (Argmax) over $\Sigma = \{a, b, c\}$

$$\vdots \tag{1}$$
$$MORE_{aa,ab}(i) := C_{aa}(i) \geq C_{ab}(i) \tag{2}$$
$$MORE_{aa,ac}(i) := C_{aa}(i) \geq C_{ac}(i) \tag{3}$$
$$NEXT_a(i) := Q_a(i) \wedge MORE_{aa,ab}(i) \wedge MORE_{aa,ac}(i) \tag{4}$$

---

This corresponds to testing, for the $f$ in Equation 10, for which $\sigma$ the entry $f(x_1 \ldots x_N)_{N,\sigma}$ is maximal.

### C.2.2 INDUCTION HEAD (ALL POSSIBLE NEXT SYMBOLS)

Consider $\Sigma$. For $a \in \Sigma$, predicate $NEXT_a(i)$ is true iff the next token can possibly be an $a$. As in Section C.2.1, first, we can define predecessor

$$CP_a(i) := \#[j \leq i, j = i - 1] \ Q_a(j)$$
$$PRED_a(i) := CP_a(i) \geq 1$$

Then we can check for bigram occurence by counting

$$CBIGRAM_{ab} := \#[j \leq i] \ Q_b(j) \wedge PRED_a(j)$$
$$EXISTS_{ab} := CBIGRAM_{ab}(i) \geq 1$$

If a bigram $\sigma a$ ever occured previously in the string, nonzero probability is assigned to predicting $a$ when at symbol $\sigma$. Then each $NEXT_a(i)$ predicate can be defined as follows

---

$NEXT_a(i)$ (All Possible) over $\Sigma = \{a, b, c\}$

$$\vdots \tag{1}$$
$$NEXT_a(i) := \bigvee_{\sigma \in \Sigma} [Q_\sigma(i) \wedge EXISTS_{\sigma a}(i)] \tag{2}$$

---

where $\bigvee_{\sigma \in \Sigma}$ can be expressed using the Boolean operations $\wedge$ and $\neg$ as defined in Section 4.2. This corresponds to testing, for the $f$ in Equation 10, for which $\sigma$ we have $f(x_1 \ldots x_N)_{N,\sigma} > 0$.

**Generating based on this program**  Consider an input prefix of the form $\#x\#$, where $\#$ denotes a separator symbol. If we iteratively generate the next symbol $a$ by selecting $a \in \Sigma$ where $NEXT_a$ holds at the last position, we generate a string $\#x\#y$ where all bigrams in $\#y$ had already occurred in $\#x$, a simple version of the in-context Markov chains studied by (Edelman et al., 2024).

**Special Case: Unique Copying**  In the special case of an input prefix where each symbol occurs at most once in $x$, the generation procedure defined above will copy $x$, and (assume we stop at $\#$) resulting in an overall string of the form $\#x\#x\#$. This is essentially the RASP-L construction of unique copying noted by Zhou et al. (2024a).

**Necessity of Positional Relations**  Intuitively, an induction head circuit requires positional information; indeed, we observe length generalization in unique copying with APE but not with NoPE (Figure 1). Formally, we can prove as follows that the predicate $NEXT_a$ defined above for each $a \in \Sigma$, while definable in **C-RASP**[local], is not definable in **C-RASP**[∅]. Consider $\Sigma = \{a, b\}$; then the predicate $NEXT_b$ can be used to define the (disjoint) union of the languages $\Sigma^* ab\Sigma^* a$, $\Sigma^* bb\Sigma^* b$. As the first one is definable in **C-RASP**[∅][5] and the union is disjoint, the second would be definable if $NEXT_b$ is. This contradicts the fact that $\Sigma^* bb\Sigma^* \not\in$ **C-RASP**[∅], because $\Sigma^* bb\Sigma^* \not\in \widehat{MAJ}_2[<]$ (Lemma 6.11 in Krebs (2008)) and the inclusion **C-RASP**[∅] $\subseteq \widehat{MAJ}_2[<]$ (see Section C.3.1).

### C.2.3  $(aa)^*$

The following function that checks parity of a position mod 2 is a periodic function.

$$\phi(i) := i \equiv 0 \mod 2$$

So the following program recognizes $(aa)^*$

---
**$(aa)^*$**

$$C_{\neg a}(i) := \#[j \leq i] \; \neg Q_a(j) \tag{1}$$
$$A(i) := C_{\neg a}(i) = 0 \tag{2}$$
$$D(i) := \phi(i) \wedge A(i) \tag{3}$$
---

The Boolean value of the last operation in the last position of the string is the accepting value. This is true if the string is of even length and contains only $a$'s. Overall, we have constructed a program in **C-RASP**[periodic, local].

### C.3  EXPRESSIBILITY OF REGULAR LANGUAGES IN **C-RASP**[PERIODIC,LOCAL]

**Lemma 29** (Restated from Lemma 11). *Consider the alphabet $\Sigma = \{a, b, e\}$.*

1. *$PARITY := b^*(ab^*ab^*)^* \not\in$ **C-RASP**[periodic, local]*

2. *$(aa)^* \in$ **C-RASP**[periodic, local] and $(aa)^* \not\in$ **C-RASP**[∅]*

3. *$(a|b|e)^* be^* \not\in$ **C-RASP**[periodic, local]*

4. *$L \in$ **C-RASP**[∅] for piecewise testable $L$*

*Proof.* 1–3 are shown in Lemmas 36 (for 3.), 38 (for 2.), 41 (for 1.), and Appendix C.2.3 (for 2.). 4. is shown in Appendix C.1.5. □

---
[5]It is sufficient to check whether $a$ and $b$ both are present, and whether one $b$ has a $a$ in its preceding context; as $\Sigma = \{a, b\}$, this is equivalent to $ab$ being a substring

### C.3.1  LINK TO MAJORITY LOGIC

In understanding the exressiveness of **C-RASP**, we draw on an established body of work on logics using MAJORITY quantifiers. Merrill & Sabharwal (2023a); Strobl (2023) show that the expressiveness of transformers is upper-bounded by uniform $\mathbf{TC}^0$, which can be defined as the logic **FOM**[BIT]. This logic is defined in terms of MAJORITY quantifiers and various predicates. **C-RASP**[periodic,local] can be viewed as a highly restricted fragment of this logic. Specifically, it is contained in $\widehat{MAJ}_2[<, +1, \text{MOD}]$, which was studied by Krebs (2008); Behle et al. (2007; 2009); results about that logic help understand the expressiveness of **C-RASP**:

**Definition 30.** $\widehat{MAJ}_2[<, +1, MOD]$ *is the logic defined by the constructs*

  1. *The construct*

$$\widehat{Maj}\, x\, \langle \phi_1, \ldots, \phi_c \rangle$$

  2. *The predicates $Q_a(x)$ for $a \in \Sigma$*

  3. *Numerical predicates $Succ(y, x)$ and $Mod_{m,r}(x)$ for $m, r \in \mathbb{N}$:*

  4. *Boolean connectives*

  5. *First-order quantifiers*[6]

*such that only two variables (say, $x$ and $y$) can appear within a formula. We define the semantics, when $x \in \Sigma^*$, by defining*

  1. *for the majority quantifier:*

$$w \models \widehat{Maj}\, x\, \langle \phi_1, \ldots, \phi_c \rangle \Leftrightarrow 0 < \sum_{i=1}^{n} \sum_{j=1}^{c} \begin{cases} 1 & \text{if } w|_{x=i} \models \phi_j \\ -1 & \text{else} \end{cases}$$

  2. *for the predicates:*

$$\begin{aligned} w|_{x=i} &\models Q_a(x) & \Leftrightarrow & \quad w_i = a \\ w|_{x=i} &\models Mod_{m,r}(x) & \Leftrightarrow & \quad i \equiv r \pmod{m} \\ w|_{x=i, y=j} &\models Succ(y, x) & \Leftrightarrow & \quad j + 1 = i \end{aligned}$$

*Semantics of Boolean connectives and first-order quantifiers follow the standard definition.*

*A language $\mathcal{L} \subseteq \Sigma^*$ is* definable *in $\widehat{MAJ}_2[<, +1, Mod]$ if there is a formula $\phi$ without free variables such that $w \in \mathcal{L}$ if and only if $w \models \phi$.*

The logic $\widehat{MAJ}_2[<]$ results by omitting the numerical predicates defined under (3).

It is straightforward to convert **C-RASP** programs into formulas of $\widehat{MAJ}_2[<, +1, \text{MOD}]$. As we shall see later in Section C.3.3, the inclusion is strict because $PARITY$ is expressible even in $\widehat{MAJ}_2[<]$.

**Proposition 31.** *C-RASP[periodic, local]* $\subseteq \widehat{MAJ}_2[<, +1, MOD]$

We make two remarks about corollaries of the result:

**Remark 32.** *First, the proof, simply by omitting the positional relations, also yields a corresponding inclusion without positional relations: **C-RASP**$[\emptyset] \subseteq \widehat{MAJ}_2[<]$.*

*Second, the result implies that **C-RASP**[periodic, local] defines a subclass of $\mathbf{TC}^0$, in fact, all **C-RASP**[periodic, local] programs translate into uniform $\mathbf{TC}^0$ circuits with a linear number of gates by results in Krebs (2008, Theorem 4.33 and Figure 4.4). The inclusion is strict, e.g., PARITY has a linear-size $\mathbf{TC}^0$ circuit but is not definable by **C-RASP**[periodic, local], as we show below.*

---

[6]These can be simulated by majority quantifiers with two variables by Proposition 5.5 in Krebs (2008), which is based on Corollary 3.3 in Lange (2004). Nonetheless, as the simulation is unobvious, they are useful for writing formulas in $\widehat{MAJ}_2[<, +1, MOD]$.

*Proof of Proposition 31.* First, every periodic positional function $\phi(i)$ is a Boolean function $Mod_{m,r}(i) \iff i = r \mod m$. For local functions $\psi(i,j)$, it suffices to only consider functions of the form $\psi(i,j) \iff j = i + c$ for $c \in \mathbb{Z}$. This is because the counting operation $C(i) := \#\left[j \leq i, |i-j| \leq c\right]\ P(j)$ is equivalent to $\hat{C}(i)$ where

$$\hat{C}(i) := \#\left[j \leq i, j = i - c \vee j = i - (c-1) \vee \ldots \vee j = i + c\right]\ P(j)$$

And it is possible to further reduce this by distributing the disjunctions over many counting operations so that each one only contains a single disjunct as positional function. It helps that a predicate $\text{fst}(i)$ is definable in $\widehat{MAJ}_2[<, +1, \text{MOD}]$ which is true iff $i = 0$. For instance $\text{fst}(i) := \neg\widehat{Maj}\ j\ \langle j \leq i, \top\rangle$

For each Boolean **C-RASP** operation $P(i)$, there exists a $\widehat{MAJ}_2[<, +1, \text{MOD}]$ formula $\hat{P}(i)$ with one free variable that is equivalent to $P(i)$ at all positions of all words. By induction, all cases are straightforward except for comparisons of counts.

For comparison operations, we will first show a formula that is equivalent for all nonempty strings. Accounting for the empty string is easy, depending on the constants in the comparison. WLOG we are able to rewrite the formula (not in standard **C-RASP** notation) as the following, where $\alpha_k, \beta \in \mathbb{Z}$

$$\left(\sum_{k \leq K} \alpha_k \cdot (\#\left[j \leq i\right]\ P_k(j))\right) + \left(\sum_{m \leq M} \alpha_m \cdot (\#\left[j \leq i, j = i + c_m\right]\ P_m(j))\right) + \beta > 0$$

We've grouped the uniform counting operations that have $\psi = \top$ together. Then using a case disjunction, we can rewrite it all the local counting operations as the following (using $\mathbb{I}[\phi]$ as notational convenience to turn $\phi(j) \in \{\bot, \top\}$ to the corresponding value in $\{0, 1\}$):

$$\bigvee_{\tau \in \{0,1\}^M\ :\ \mathbb{I}[P_m(j - c_m)] = \tau_m} \left(\sum_{k \leq K} \alpha_k \cdot (\#\left[j \leq i\right]\ P_k(j))\right) + \left(\sum_{m \leq M} \alpha_m \tau_m\right) + \beta > 0$$

We can see that for nonempty strings within each case, the additive constant $\beta$ can be reformulated as $\left(\beta + \sum_{m \leq M} \alpha_m \tau_m\right) \cdot \#\left[j \leq i\right]\ \text{fst}(j)$, and we can just add it to the summation using $\alpha_{k+1} = \left(\beta + \sum_{m \leq M} \alpha_m \tau_m\right)$. Then, it is possible to define formulas for each case disjunction using a series of existential quantifiers and $succ(j, i)$. For instance:

$$\phi(j - 2) \equiv \exists i.\, (Succ(i, j) \wedge \exists j.\, (Succ(j, i) \wedge \phi(j)))$$

Thus, we are able to rewrite the entire formula where we only ever compute sums of **C-RASP** counts without any positional functions. This means it now suffices to focus on the summations of counting terms

$$\sum_{k \leq K} \alpha_k \cdot (\#\left[j \leq i\right]\ P(j)) > 0$$

and simulate these that using a $\widehat{Maj}$ formula. For $k \leq (K+1)$, if $\alpha_k > 0$ consider the list of formulas

$$L_k := [\underbrace{\hat{P}_k(j), \hat{P}_k(j), \ldots, \hat{P}_k(j)}_{\alpha_k \text{ many}}, \underbrace{\top, \top, \ldots, \top}_{\alpha_k \text{ many}}]$$

Intuitively, the $\widehat{Maj}\ j$ quantifier can only check if the total count is greater than half the possible positions, so to check if a count is $> 0$ we need to pad the quantifier with a bunch of trivially true formulas to ensure the total count is at least half by default. And if $\alpha_k < 0$ we use

$$L_k := [\underbrace{\neg\hat{P}_k(j), \neg\hat{P}_k(j), \dots, \neg\hat{P}_k(j))}_{\alpha_k \text{ many}}, \underbrace{\bot, \bot, \dots, \bot}_{\alpha_k \text{ many}}]$$

Let $L = L_1 ++ L_2 ++ \dots ++ L_{K+1}$ be the concatenation of all these lists, and let $\varphi_1, \varphi_2, \dots, \varphi_{|L|}$ list out the formulas in $L$. Then we claim the following formula will compute the correct value of the comparison for nonempty strings.

$$\phi_1(i) := \widehat{Maj}\ j\langle \varphi_1, \varphi_2, \dots, \varphi_{|L|}\rangle$$

For empty strings, if $\left(\beta + \sum_{m\leq M} \alpha_m \tau_m\right) > 0$ then define $\phi_0(i) := \neg\widehat{Maj}\ j\ \langle\top\rangle \wedge \top$. Otherwise, use $\phi_0(i) := \neg\widehat{Maj}\ j\ \langle\top\rangle \wedge \bot$. Then we can define

$$\hat{P}(i) := \phi_0(i) \vee \phi_1(i)$$

$\square$

### C.3.2 Inexpressibility of $\Sigma^* b e^*$

Krebs (2008); Behle et al. (2007; 2009) used infinite groups to establish results about the expressiveness of $\widehat{MAJ}_2[<]$; by Corollary 31, these results entail results on **C-RASP**[periodic, local]. In particular, Lemma 6.11 in Krebs (2008) shows that $L_{bb} \notin \widehat{MAJ}_2[<]$; this result turns out to have profound consequences for **C-RASP** expressiveness.

**Definition 33.** *Let $\Sigma = \{a, b, e\}$. Define $L_{bb} := \Sigma^* b e^* b \Sigma^*$.*

As a minor note, the exact statement of Lemma 6.11 used $L_{bb} = \Sigma^* bb\Sigma^*$, but the addition of the neutral letter $e$ is inconsequential.

**Lemma 34.** *Let $\varphi$ be a $\widehat{MAJ}_2[<, +1, MOD]$ formula. There exists a morphism $h(\sigma) = e^s \sigma e^{s-1}$ and a $\widehat{MAJ}_2[<]$ formula $\psi$ such that for every $w \in \Sigma^*$, $h(w) \vDash \phi \iff w \vDash \psi$*

*Proof.* Let $M$ be the least common multiple of all occurring moduli in $\varphi$. Let $C$ be the maximum nesting depth of $Succ(x, y)$ predicates (which must be bounded by the quantifier depth of $\varphi$). Intuitively, we can think of $C$ as the largest number where a subformula $\varphi(x + C)$ occurs in $\varphi$. Let $s = MC$, and define the morphism $h(\sigma) = e^s \sigma e^{s-1}$ for $\sigma \in \Sigma$. Here we will use the notation $\phi^c(x)$ which will be true at position $x$ in $w$ whenever $\phi$ is true at position $x + c$ in $h(w)$, for $c \in [-s, s-1]$.

We will show that for every formula $\varphi(x)$ of $\widehat{MAJ}_2[<, +1, MOD]$ with at most one free variable, we can define $\varphi^{-s}(x), \varphi^{-(s-1)}(x) \dots \varphi^{(s-1)}(x)$ such that for all $i \in [0, |w|-1]$ and $c \in [-s, s-1]$

$$h(w) \vDash \varphi(i + c) \iff w \vDash \varphi^c(i)$$

Intuitively, what this does is it takes every interval of $[x - s, x + (s-1)]$ around each position in $h(w)$ and stores it in a "vector" at that position in $w$. We will induct on the complexity of $\varphi$. If $\varphi(x)$ is $Q_e(x)$, then $\varphi^0(x) := Q_e(x)$, and then $\varphi^c(x) = \top$ for every other $c \neq 0$, since the morphism $h$ pads neutral symbols $e$ in $h(w)$ between every symbol from $w$. If $\varphi(x)$ is $Q_\sigma(x)$ for $\sigma \neq e$, we have that $\varphi^0 := Q_\sigma(x)$ and $\varphi^{+c}(x) = \bot$ for every other $c \neq 0$. If $\varphi(x)$ is $Mod_{m,r}(x)$, the $\varphi^c$ can also be "hardcoded" similarly, as every position in $h(w)$ that has a symbol from $w$ is going to be $= 0$ mod $s$.

Boolean formulas are also straightforward. The only hard case is if we have a formula $\varphi(x) = \widehat{MAJ}\ y\ \langle\varphi_1(x, y), \dots, \varphi_k(x, y)\rangle$. We can think of $\varphi$ specifying the constraint

$$\left( \sum_{i \leq k} \# \, y \, [\varphi_i(x,y)] \right) > k \cdot \frac{|w|}{2}$$

Where $\# \, y \, \phi$ indicates the number of positions in the string satisfying $\phi$ (with variables bound appropriately). The idea here is to rewrite $\psi_i(x,y)$ in terms of its unary formulas (which we can apply the inductive hypothesis to) and then split $h(w)$ into some intervals, upon which evaluating $\varphi^{+c}(x)$ will be simpler. First we can rewrite each $\varphi_i(x,y)$ as

$$F_i(\alpha_1(x), \ldots, \alpha_q(x), \beta_1(y), \ldots, \beta_r(y), \chi_1(x,y), \ldots, \chi_p(x,y))$$

Where $F_i$ is a Boolean function, the $\alpha$ are unary in $x$, the $\beta$ are unary in $y$, and the $\chi(x,y)$ are inequalities of $x$ and $y$, possibly with $+1$'s, of the form $x \leq y + 1$, for example. To save space, we will abbreviate the above expression by grouping the $\alpha, \beta, \chi$ formulas together notationally $F_i(\overline{\chi}_i(x,y), \overline{\alpha_i}(x), \overline{\beta_i}(y))$.

The $\chi$ formulas are not unary, but we can "eliminate" the $\chi$ terms by casework over intervals of the string. We will show this by example for a summation with only one $\# \, y$ term. This argument works identically if we had many of $\# \, y$ terms, but it would add notational clutter. So if we had a formula $\varphi(x) = \widehat{MAJ} \, y \, \langle \varphi_1(x,y) \rangle$ we could think of it as in the form

$$\varphi(x) = \# \, y \, \left[ F(\overline{\chi}(x,y), \overline{\alpha}(x), \overline{\beta}(y)) \right] \geq \frac{|w|}{2}$$

Then we can construct the formula $\varphi^c(x)$ for $c \in [-s, s-1]$ by using the following partition of intervals of the string. Let $\Xi$ be the set of inequalities

$$\Xi = \{y < x - s, y = x - s, \ldots, y = x, \ldots, y = x + (s-1), y > x + (s-1)\}$$

And we define some notation. For $\xi \in \Xi$, let $\chi^{\xi}(x,y) \in \{\top, \bot\}$ evaluate $\chi$ in the case $\xi$ holds. For instance if $\chi(x,y)$ is $y < x + 1$, then $\chi^{y>x+2}(x,y) = \bot$. Since the intervals defined by $\xi \in \Xi$ disjoint and cover the entirety of the string, every $\chi$ can be evaluated in this manner. Then, we can essentially compute the sum in each interval, and only precision is needed in the interval $[x - s, x + (s-1)]$, so $\varphi(x)$ is equivalent to

$$\# \, y \left[ y < x \wedge F(\overline{\chi}^{y<x-s}(x,y), \overline{\alpha}^c(x), \overline{\beta}^{-s}(y)) \right]$$

$$+ \# \, y \left[ y < x \wedge F(\overline{\chi}^{y<x-s}(x,y), \overline{\alpha}^c(x), \overline{\beta}^{-(s-1)}(y)) \right]$$

$$\vdots$$

$$+ \# \, y \left[ y < x \wedge F(\overline{\chi}^{y<x-s}(x,y), \overline{\alpha}^c(x), \overline{\beta}^{+(s-1)}(y)) \right]$$

$$+ \# \, y \left[ y < x \wedge F(\overline{\chi}^{y<x-s}(x,y), \overline{\alpha}^c(x), \overline{\beta}^{s}(y)) \right]$$

$$+ \# \, y \left[ x = y \wedge F(\overline{\chi}^{y=x-s}(x,y), \overline{\alpha}^c(x), \overline{\beta}^{-s}(y)) \right]$$

$$+ \# \, y \left[ x = y \wedge F(\overline{\chi}^{y=x-(s-1)}(x,y), \overline{\alpha}^c(x), \overline{\beta}^{-(s-1)}(y)) \right]$$

$$\vdots$$

$$+ \# \, y \left[ x = y \wedge F(\overline{\chi}^{y=x+(s-1)}(x,y), \overline{\alpha}^c(x), \overline{\beta}^{(s-1)}(y)) \right]$$

$$+ \# \, y \left[ x = y \wedge F(\overline{\chi}^{y=x+s}(x,y), \overline{\alpha}^c(x), \overline{\beta}^{s}(y)) \right]$$

$$+ \# \, y \left[ x < y \wedge F(\overline{\chi}^{x+s<y}(x,y), \overline{\alpha}^c(x), \overline{\beta}^{-s}(y)) \right]$$

$$+ \# \, y \left[ x < y \wedge F(\overline{\chi}^{x+s<y}(x,y), \overline{\alpha}^c(x), \overline{\beta}^{-(s-1)}(y)) \right]$$

$$\vdots$$

$$+ \# \, y \left[ x < y \wedge F(\overline{\chi}^{x+s<y}(x,y), \overline{\alpha}^c(x), \overline{\beta}^{(s-1)}(y)) \right]$$

$$+ \# \, y \left[ x < y \wedge F(\overline{\chi}^{x+s<y}(x,y), \overline{\alpha}^c(x), \overline{\beta}^{s}(y)) \right]$$

$$\geq \frac{|w|}{2}$$

By the inductive hypothesis, all $\alpha^c$ and $\beta^c$ are definable solely in terms of $\widehat{MAJ}_2[<]$, so the entire formula is equivalent to a $\widehat{MAJ} \, y$ formula that quantifies over all the bracketed formulas above, as well as equally many trivially true formulas, as described more clearly in Proposition C.3.1. As mentioned before, since this argument also applies to a summation of $\# \, y$ terms, this completes the proof. Then for any $\varphi(x)$ in $\widehat{MAJ}_2[<, +1, MOD]$, after performing the above translation the resulting formula $\varphi^0(x)$ is our desired formula in $\widehat{MAJ}_2[<]$. $\qquad\square$

**Lemma 35.** $L_{bb} \notin \widehat{MAJ}_2[<, +1, MOD]$

*Proof.* Assume for sake of contradiction that $L_{bb}$ is definable by a formula $\varphi$ of $\widehat{MAJ}_2[<, +1, MOD]$. Let $h$ and $\psi$ be as guaranteed by the above lemma. Then for $w \in \Sigma^*$, $w \in L_{bb} \iff h(w) \in L_{bb}$. This means $\psi$ defines $L_{bb}$ which contradicts Lemma 6.11 in Krebs (2008), which has shown that $L_{bb} \notin \widehat{MAJ}_2[<]$. $\qquad\square$

**Lemma 36.** *For $\Sigma = \{a, b, e\}$, it holds that*

$$\Sigma^* b e^* \notin \textbf{C-RASP}[periodic, local] \tag{4}$$

*Proof.* To get a contradiction, note that a **C-RASP**[periodic, local] program $\Phi$ for $\Sigma^* b e^*$ could be used to construct one for $L_{bb}$, as:

$$C_1(i) := \#\,[j \le i, j = i - 1]\ \Phi(j)$$
$$PREV_\Phi(i) := C_1(i) \ge 1$$
$$C_2(i) := \#\,[j \le i]\ Q_b(j) \wedge PREV_\Phi(j)$$
$$L_{bb} := C_2(i) \ge 1$$

$\square$

### C.3.3 INEXPRESSIBILITY OF PARITY

First, let the *depth* of a **C-RASP** operation be the maximum depth of nesting of counting operations in it. For instance if $C(i) := \#\,[j \le i]\ P(j)$, the depth of the $C$ is depth of $P(i)$ plus one. None of the other operations are greater than the depth of its dependencies. We will induct on program depth for the following proof:

**Lemma 37.** *Let $\Sigma = \{a\}$. For any **C-RASP**$[\emptyset]$ program $P$ there exists an $n$ such that for all $w$ where $|w| \ge n$, either all such $w$ are accepted by $P$ or all are rejected*

*Proof.* If $P$ is depth 0, it is equivalent to either $Q_a(i)$ or $\neg Q_a(i)$, which either rejects every string or accepts every string.

Otherwise, let all **C-RASP** programs of depth $k$ give constant output for strings above length $n$, and then consider a program $P$ of depth $k + 1$. $P$ will be equivalent to a Boolean combination of linear constraints. We will see that each linear constraint becomes constant for strings above a certain length. Consider any linear constraints over $X$ many counts $C_x$ of depth $k$:

$$L(i) := \left( \sum_{x \le X} \alpha_x \#\,[j \le i]\ C_x(j) \right) \ge c$$

For string of length $i \ge n$, this is equivalent to

$$L(i) := \left( \sum_{x \le X} \alpha_x \left( (i - n)\mathbb{I}[C_x(n)] + \#\,[j \le n]\ C_x(j) \right) \right) \ge c$$

Where $\mathbb{I}[C_x(n)]$ denotes the truth value of $C_x(n) \in \{0, 1\}$. Rearrange this to

$$L(i) := \left( (i - n) \sum_{x \le X} \alpha_x \left( \mathbb{I}[C_x(n)] \right) \right) + \left( \sum_{x \le X} \alpha_x \left( \#\,[j \le n]\ C_x(j) \right) \right) \ge c$$

The sums $c_1 = \sum_{x \le X} \alpha_x \left( \mathbb{I}[C_x(n)] \right)$ and $c_2 = \sum_{x \le X} \alpha_x \left( \#\,[j \le n]\ C_x(j) \right)$ are constants depending on the formula and $n$.

$$(i - n)c_1 + c_2 \ge c$$

Depending on if $c_1, c_2$ are positive or negative, we either derive a lower bound $m$ after which the linear constraint $L(i)$ is always true, or always false for $i \ge m$. Since any formula of depth $k + 1$ is a Boolean combination of these linear constraint, we take the max of all the $m$'s from them, and any string larger than this will always be accepted or always rejected by $P$. $\square$

**Lemma 38.** $(aa)^* \notin$ **C-RASP**$[\emptyset]$

That is, no **C-RASP**$[\emptyset]$ program can determine if a general string has even length. The same proof applies to testing whether the string length is a multiple of any other fixed integer.

*Proof.* Using the previous lemma, for every **C-RASP**$[\emptyset]$ program there exists an $n$ such that the program accepts $(aa)^n$ iff it accepts $(aa)^n a$. So no program can recognize $(aa)^*$. $\qquad\square$

We will use this to show that **C-RASP**$[\text{periodic}, \text{local}]$ program cannot recognize $PARITY$, as the extra positional operations do not give sufficient expressive power. We start with an observation that simplifies the proof

**Proposition 39.** *As syntactic sugar we allow $j < i$ as a mask in **C-RASP** counting operations.*

*Proof.* Consider the counting operation $C(i) := \#\,[j \leq i]\ P(j)$. We can define the program

$$
\begin{aligned}
I(i) &:= P(i)\ ?\ 1 : 0 \\
C(i) &:= \#\,[j \leq i]\ P(j) \\
C'(i) &:= C(i) - I(i)
\end{aligned}
$$

And essentially, this operation will compute the count

$$C'(i) := \#\,[j < i]\ P(j)$$

$\qquad\square$

**Lemma 40.** *Let $P$ be a **C-RASP**$[\text{periodic}, \text{local}]$ program over $\Sigma = \{a, b\}$. There is some $s > 0$ and a morphism $h(a) = b^s a b^{s-1}$ and a **C-RASP**$[\emptyset]$ program $\hat{P}$ over $\Sigma = \{a\}$ such that for all $w \in a^*$, if $P$ accepts $h(w)$ iff $\hat{P}$ accepts $w$.*

*Proof.* Choose $s$ to be the least multiple of all moduli occurring in $P$ that is also greater than all the $|c|$ in local functions $j = i + c$. For every operation $P(i)$ of $P$, we will define $\hat{P}^c(i)$ for $c \in [-s, s-1]$ such that $\hat{P}^c(i)$ when run on $w$ is equivalent to $P(s + i(2s) + c)$ when run on $h(w)$.

If $P(i)$ is $Q_a(i)$ or $Q_b(i)$, it is straightforward, as $\hat{P}^c(i)$ is true iff $c = 0$. Modular predicates are also capable of being "hardcoded", as positions in $w$ are always $0 \mod s$ in $h(w)$, and so the offset $c$ determines the value modulo. All other kinds of operations are also straightforward using the inductive hypothesis. The only ones that need care are counting operations. First, consider a counting operation without positional functions:

$$C(i) := \#\,[j \leq i]\ A(j)$$

We can define each $\hat{C}^c(i)$ using a program like the following. The idea is that the entire window of $[j - s, j + (s - 1)]$ around each $j < i$ can be counted up completely, but around $i$ we only consider the interval $[-s, i + c]$:

$$
\begin{aligned}
C_{-s}(i) &:= \#\,[j < i]\ \hat{A}^{-s}(j) \\
C_{-(s-1)}(i) &:= \#\,[j < i]\ \hat{A}^{-(s-1)}(j) \\
&\;\;\vdots \\
C_{(s-1)}(i) &:= \#\,[j < i]\ \hat{A}^{(s-1)}(j) \\
I_{-s}(i) &:= \hat{A}^{-s}(j)\ ?\ 1 : 0 \\
I_{-(s-1)}(i) &:= \hat{A}^{-(s-1)}(j)\ ?\ 1 : 0 \\
&\;\;\vdots \\
I_c(i) &:= \hat{A}^c(j)\ ?\ 1 : 0 \\
\hat{C}^c(i) &:= \sum_{t \in [-s, s-1]} C_t(i) + \sum_{t \in [-s, c]} I_t(i)
\end{aligned}
$$

Otherwise, if we have a counting operation that involves a local positional function

$$C(i) := \#\left[j \leq i, j = i + d\right] \ A(j)$$

Then the operation returns either the count 1 or 0 and we can just use

$$\hat{C}^c(i) := (c = d \wedge \hat{A}^c(i)) \ ? \ 1 : 0$$

Since $d$ will not exceed $\pm s$, $\hat{A}^d$ exists. Using these constructions we can see that $(b^s a b^{s-1})^n$ is accepted by operation $P(i)$ in $P$ iff $a^n$ is accepted by the constructed operation $\hat{P}^{(s-1)}(i)$. $\qquad \square$

**Lemma 41.** $PARITY \notin$ **C-RASP**$[periodic, local]$

*Proof.* If such a program existed, it would be able to distinguish between $(b^s a b^{s-1})^{2n}$ and $(b^s a b^{s-1})^{2n+1}$ for all $n$ (using the $s$ guaranteed by the previous lemma). However, this implies the existence of a **C-RASP**$[\emptyset]$ program over $\Sigma = \{a\}$ that recognizes $(aa)^*$. This contradicts Lemma 38. $\qquad \square$

# D DISCUSSION OF DESIGN CHOICES

## D.1 MLP ACTIVATION FUNCTIONS

Our analysis allows ReLU and the Heaviside function as activation functions in MLPs. ReLU is a standard choice in theoretical studies of neural networks and transformers (e.g. Bhattamishra et al., 2024; Sanford et al., 2023). Modern LLMs also use other functions such as SwiGLU (Shazeer, 2020), but universal approximation theorems guarantee that ReLU networks can approximate smooth functions on bounded domains well. While the choice of ReLU is not necessarily key to our results, it is important that the number of active units provides a meaningful upper bound on the complexity of the function expressed. Our results would continue to go through if $\phi$ is an arbitrary activation function but operates at $p$-bit precision.

We also allow the Heaviside function as a second activation function. Heaviside allows exactly performing threshold computations at arbitrary input lengths, which is relevant to simulating **C-RASP** at arbitrary input lengths. This includes simple problems such as MAJORITY, on which transformers empirically do well. Real-world transformers generally do not include this function, though ReLU MLPs can approximate it arbitrarily closely.

## D.2 FIXED PRECISION

As described in Section 2, we assume that attention logits and the exponentials inside softmax are rounded to $p$ fractional bits of precision before further processing. This allows us to cluster keys and queries into finite numbers of clusters, and compute all activations at logarithmic precision, used for proving the logarithmic communication complexity bound (Theorem 12). We note that logarithmic precision of the intermediate activations is also key to upper bounds of transformers in terms of $\mathbf{TC}^0$ shown by Merrill & Sabharwal (2023b).

We also assume that the parameters in transformers and Limit Transformers are expressed at fixed precision (Definitions 4 and 2), and penalize the precision $p$ used of representing the parameters as part of the regularizer used in our inference procedure (Definition 5). Indeed, penalizing unbounded precision of parameter values is necessary to enable full identification of a transformer algorithm from behavior at finite lengths. For any real number $\alpha \geq 0$, a one-layer transformer with real-valued parameters can express the function

$$F_\alpha(x) = \begin{cases} 1 & \text{if } \alpha \cdot \#_1(x) \geq \#_0(x) \\ 0 & else \end{cases}$$

| Grammar | Star-Free | Definition |
|---|---|---|
| 1 | Yes | 1* |
| 2 | Yes | (10)* |
| 3 | No | strings without odd-length strings of ones followed by odd-length strings of zeros (i.e., no $01^{2n+1}0^{2m+1}1$ substrings) |
| 4 | Yes | strings without any 000's substrings |
| 5 | No | strings of even length with an even number of 1's |
| 6 | No | strings where number of 0's - number of 1's is divisible by 3 |
| 7 | Yes | 0*1*0*1 |

Table 1: Tomita Grammars (originally due to Tomita, 1982), following Bhattamishra et al. (2020).

For any two distinct $\alpha, \beta$, the functions $F_\alpha$ and $F_\beta$ are distinct on sufficiently long inputs (though, when $\alpha$ and $\beta$ are close, very long inputs will be needed to distinguish them). Thus, there are *uncountably* many distinct functions implemented by transformers; however, their distinction relies on infinite precision. In an infinite precision setup, one cannot hope to identify algorithms implemented by transformers from finite data, no matter the input length and the regularization applied to the model size. In contrast, when parameters are representable in finite precision (as in real computers), the number of distinct algorithms expressed by Limit Transformers is countable, and ultimate identification from long inputs is possible when the precision required for representing the parameters is penalized.

### D.3 LAYER NORM

Real-world transformers use Layer Norm or RMSNorm, whereby activations $\boldsymbol{y}_i^{(l)}$ are rescaled to have norm or standard deviation $\sqrt{d}$. Layer norm can be incorporated into the translation to Limit Transformers (Lemma 52) by recording terms of the form

$$\boldsymbol{v}^T \left( \prod_{S \in \mathcal{S}_1} S \right)^T \left( \prod_{S \in \mathcal{S}_2} S \right) \boldsymbol{w}^T \tag{5}$$

when $\boldsymbol{v} \in \mathcal{VO}_{l_1}$ and $\boldsymbol{w} \in \mathcal{VO}_{l_2}$ and $\mathcal{S}_1, \mathcal{S}_2 \in \mathcal{P}$. We can record these products in further dimensions of $\widehat{\boldsymbol{y}_i^{(l)}}$, so that $\|\boldsymbol{y}_i^{(l)}\|_2^2$ is recoverable from $\widehat{\boldsymbol{y}_i^{(l)}}$. The simplest approach is then to modify the definition of Limit Transformers by normalizing $\widehat{\boldsymbol{y}_i^{(l)}}$ based on this recovered norm.

The original translation from **C-RASP** to fixed-precision transformers (Yang & Chiang, 2024) capitalized on layer norm; it would be no problem to translate from **C-RASP**[periodic,local] to a version of Limit Transformers incorporating layer norm, and we would be able to remove the Heaviside function from Limit Transformers.

## E    ADDITIONAL DETAILS FOR EXPERIMENTS

### E.1    REGULAR LANGUAGES FROM THE BHATTAMISHRA ET AL 2020 BENCHMARK

#### E.1.1    LANGUAGE DEFINITIONS

Descriptions follow Bhattamishra et al. (2020).

**Tomita Grammars.** Definitions are shown in Table 1.

$\boldsymbol{D_n}$ are defined on the alphabet $\Sigma = \{a, b\}$ by the recursion $D_n = (aD_{n-1}b)^*$.

**PARITY.** PARITY is $b^*(ab^*ab^*)^*$. It is contained in the set of algorithmic tasks.

**Others.** Other languages: $(aa)^*$, $(aaaa)^*$ and $(abab)^*$ (not star-free), $aa^*bb^*cc^*dd^*ee^*$, $\{ab\}^*d\{b, c\}^*$, and $\{0, 1, 2\}^*02^*$ (star-free).

| # | Language | C-RASP expressiveness [∅] | [periodic, local] | Star-Free? | Dot-Depth | $\mathbf{AC}^0$? |
|---|---|---|---|---|---|---|
| 1 | Tomita 1 | yes | yes | yes | 1 | yes |
| 2 | Tomita 2 | yes | yes | yes | 1 | yes |
| 3 | Tomita 3 | no | no | no | – | yes |
| 4 | Tomita 4 | no | yes | yes | 1 | yes |
| 5 | Tomita 5 | no | no | no | – | no |
| 6 | Tomita 6 | no | no | no | – | no |
| 7 | Tomita 7 | yes | yes | yes | 1 | yes |
| 8 | $D_2$ | yes | yes | yes | 2 | yes |
| 9 | $D_3$ | yes | yes | yes | 3 | yes |
| 10 | $D_4$ | yes | yes | yes | 4 | yes |
| 11 | $D_{12}$ | yes | yes | yes | 12 | yes |
| – | PARITY | no | no | no | – | no |
| 12 | $(aa)^*$ | no | yes | no | – | yes |
| 13 | $(aaaa)^*$ | no | yes | no | – | yes |
| 14 | $(abab)^*$ | no | yes | no | – | yes |
| 15 | $aa^*bb^*cc^*dd^*ee^*$ | yes | yes | yes | 1 | yes |
| 16 | $\{a,b\}^*d\{b,c\}^*$ | yes | yes | yes | 1 | yes |
| 17 | $\{0,1,2\}^*02^*$ | no | no | yes | 2 | yes |

Table 2: The finite-state languages in the benchmark from Bhattamishra et al. (2020), with the numbering from Figure 1, **C-RASP** expressiveness properties, and three established notions of complexity (star-freeness, dot depth, and membership in the circuit complexity class $\mathbf{AC}^0$). In the **C-RASP** columns, "yes" means we found a **C-RASP** program; "no" means we proved that no **C-RASP** program can exist. Note that $\{0,1,2\}^*02^*$ is equivalent to the language $\Sigma^*be^*$ from Lemma 11. Note also that we discuss PARITY in the algorithmic benchmark, as it is included in Zhou et al. (2024a). See discussion in Appendix E.1.2. See Figure 2 for a version of Figure 1 (right) with languages labeled.

### E.1.2 C-RASP EXPRESSIVENESS

All languages in the benchmark are in $\mathbf{TC}^0$, and all are expressible in principle by transformers (Liu et al., 2023). We were able to provably settle the **C-RASP** expressiveness for all languages, with results shown in Table 2. We compare **C-RASP** expressiveness with the standard notions of complexity of finite-state languages considered by Bhattamishra et al. (2020): whether languages are star-free, and (among the star-free ones) their dot-depth. While many star-free languages in the sample show length-generalization, star-freeness does not overall account for the observed behavior (Figure 4). Within the star-free languages, a standard complexity metric is dot-depth (Figure 2); this again does not accurately predict length-generalization: it succeeds for a language with dot depth 12 but fails for a language with dot depth 2. We also considered circuit complexity of regular languages (Barrington et al., 1992). All regular languages included in the sample are in the class $\mathbf{TC}^0$; most are also in $\mathbf{AC}^0$, a smaller class sometimes compared to transformers (Hao et al., 2022; Barcelo et al., 2024). Transformers show poor length generalization on the non-$\mathbf{AC}^0$ regular languages[7], but also fail on various languages that are in $\mathbf{AC}^0$. On algorithmic problems, transformers succeed on some non-$\mathbf{AC}^0$ problems such as Majority (Table 3). Overall, **C-RASP** expressiveness is much more successful than previously considered notions of complexity in accounting for empirical length generalization behavior of transformers.

**Proof Sketches for Membership Claims** We sketch proofs for all **C-RASP** expressiveness claims in Table 2. We first note that **C-RASP**[periodic, local] is closed under the inverse images of morphisms where each symbol is mapped to a string of the same length. That is:

**Lemma 42.** *C-RASP*[*periodic*, *local*] *is closed under the inverse images of morphisms where each symbol is mapped to a string of some fixed length $n$. That is, if $h : \Sigma_1 \to \Sigma_2^n$ (for some*

---

[7]By the results of Barrington et al. (1992), regular languages outside of $\mathbf{AC}^0$ are all, informally speaking, at least as hard as PARITY, and indeed they provably are not in **C-RASP**[periodic, local].

*fixed n) is extended to a map $\Sigma_1^* \to \Sigma_2^*$ and $\mathcal{L} \in$ **C-RASP**[periodic, local], then $h^{-1}(\mathcal{L}) \in$ **C-RASP**[periodic, local].*

*Proof.* Let $h$ be such a morphism. We we take any **C-RASP**[periodic, local] program operation $P(i)$ that operates on $h(w)$ into an equivalent program which operates on $w$. We use similar notation to Lemma 40. In this case for every operation $P(i)$, we create an operation $\hat{P}^c(i)$ that, evaluated on word $w$, evaluates $P(i + c)$ on the word $h(w)$.

For $P(i) = Q_a(i)$, we can use $\hat{P}^0(i) := \bigvee_{\sigma \in \Sigma : h(\sigma)_0 = a} Q_\sigma(i)$, $\hat{P}^1(i) := \bigvee_{\sigma \in \Sigma : h(\sigma)_1 = a} Q_\sigma(i)$, and so on.

For $P(i) = Mod_{m,r}(i)$, then for $c \le n$ we can define (relying on the fact that $m, r$ are fixed). This boils down to defining $Mod_{m,r}(ni + c)$ in terms of $Mod$ terms which only reference $i$. This is simple, despite being lengthy to write out, because we can simply "hard-code" what needs to happen for all $i \le m$. The cases will repeat after that, since $Mod_{m,r}(ni + c) \equiv Mod_{m,r}(n(m + i) + c)$, so we can check $i \mod m$, and enumerate the cases from there.

For $P(i) = \#[j \le i, \psi(i,j)] \ V(j)$, where $\psi(i,j) = \top$, then like before we can define $\hat{P}^c(i)$ as

$$C^0(i) := \#[j < i] \ \hat{V}^0(j)$$
$$C^1(i) := \#[j < i] \ \hat{V}^1(j)$$
$$\vdots$$
$$C^{n-1}(i) := \#[j < i] \ \hat{V}^{n-1}(j)$$
$$I^0(i) := \hat{V}^0(j) \ ? \ 1 : 0$$
$$I^1(i) := \hat{V}^1(j) \ ? \ 1 : 0$$
$$\vdots$$
$$I^c(i) := \hat{V}^c(j) \ ? \ 1 : 0$$
$$\hat{P}^c(i) := \sum_{t \in [0,n-1]} C^t(i) + \sum_{t \in [0,c]} I^t(i)$$

Otherwise, as described before, we only need to consider the case of $\psi(i,j) := j = i - d$ since counts over any local function can be rewritten as counts over local functions with this form. Here it takes a little bit more care, as the single position $j$ upon which to check $V(j)$ may occur far behind the window of $n$ around each symbol.

So for $c \le d$, we need to find the value of some $V(j)$ at position $j = ni - d$, where we can check the value of $V(nj + c)$ using $V^c(j)$. Intuitively, the idea is to use a $\#[j \le i] \ \phi(j)$ operation in order to find a $j \le i$ such that $nj + c = ni - d$. We can use a local positional function to find the right $j$ modulo $n$, and then find the correct $c$ to retrieve $V^c(j)$.

This can be done by looking pack to position $i - \lfloor \frac{d}{n} \rfloor$, using a local positional function. Then, we need to find the offset which is equal to the position $(ni + c) - d$. Thus we can write the following summation, where only one of the count terms will be nonzero. Let $\tau_c = \mathbb{I}[(c = (n - (d \mod n)))] \in \{\top, \bot\}$ be a boolean value indicating we've found the correct $c$ value.

$$C^0(i) := \#\left[j \le i, i = j + \left\lfloor \frac{d}{n} \right\rfloor\right] \tau_0 \wedge \hat{V}^0(j)$$

$$C^1(i) := \#\left[j \le i, i = j + \left\lfloor \frac{d}{n} \right\rfloor\right] \tau_1 \wedge \hat{V}^1(j)$$

$$\vdots$$

$$C^{n-1}(i) := \#\left[j \le i, i = j + \left\lfloor \frac{d}{n} \right\rfloor\right] \tau_{n-1} \wedge \hat{V}^{n-1}(j)$$

$$\hat{P}^c(i) := \sum_{t \in [0, n-1]} C^t(i)$$

For $c > d$, the position to check occurs in the same window as $i$, so we only need to check the finitely many cases up to $i + c$.

$$I^0(i) := c - d = 0 \wedge \hat{V}^0(i) \ ? \ 1 : 0$$

$$I^1(i) := c - d = 1 \wedge \hat{V}^0(i) \ ? \ 1 : 0$$

$$\vdots$$

$$I^c(i) := c - d = c \wedge \hat{V}^c(i) \ ? \ 1 : 0$$

$$\hat{P}^c(i) := \sum_{t \in [0, c]} I^t(i)$$

All other cases are straightforward. $\qquad\qquad\qquad\qquad\qquad\qquad\qquad\qquad\qquad\quad \square$

**Tomita 1 $\in$ C-RASP[$\emptyset$]**   A **C-RASP** program can detect the presence of a symbol other than 1 and flag a violation.

**Tomita 2 $\in$ C-RASP[$\emptyset$]**   A **C-RASP** program expresses: At each position, either the current symbol is a 0 and the count of ones and zeros is balanced; or the current symbol is a 1 and the count of ones is one more than the count of zeros.

**Tomita 3 $\notin$ C-RASP[$\emptyset$]**   For a given string of the form $010^K$, the outputs will converge as $K \to \infty$ by the same argument as for $(aa)^*$ in Lemma 38. Hence, Tomita 3 $\notin$ **C-RASP**[$\emptyset$].

**Tomita 3 $\notin$ C-RASP[periodic, local]**   Informally, the only way periodic and local predicates are likely to help is if the lengths of contiguous blocks of zeros and ones were bounded (local), or the parity of the lengths of 1 and 0 substrings were globally linked to the parity of the positions of transitions 10 and 01 (periodic), but neither is the case. Sketching a formal proof, assume a **C-RASP**[periodic, local] program is given for Tomita 3. First, we eliminate the periodic predicates by labeling every symbol with the position modulo $s$, where $s$ is a multiple of 2 and all moduli appearing in periodic functions; giving an extended alphabet $1_1, \ldots, 1_s; 0_1, \ldots, 0_s$. For sufficiently large $c$ that is a co-prime with $s$, we can then also eliminate local functions by merging an adjacent block of length $c$ around every transition between ones and zeros into a single symbol $\Lambda$; indexed by the first symbol inside the block and whether the transition happens at the $floor(s/2)$-th (second part has even length) or $ceil(s/2)$-th (second part has odd length) position in the block. The resulting language, over an extended alphabet, is recognized by a **C-RASP**[$\emptyset$] program capable of determining whether a string of the form

$$\Lambda_{1_1, \ldots} \Lambda_{0_{(1+c)\%s}, \ldots} \Lambda_{1_{(1+2c)\%s}, \ldots} \cdots \tag{6}$$

contains a substring of the form

$$\Lambda_{0_{\ldots}, even} \Lambda_{1_{\ldots}, even} \Lambda_{0_{\ldots}, even} \qquad \text{or} \qquad \Lambda_{0_{\ldots}, odd} \Lambda_{1_{\ldots}, odd} \Lambda_{0_{\ldots}, odd} \tag{7}$$

This is impossible for the same reasons that $\Sigma^* bbb \Sigma^*$ (Tomita-4) is not in **C-RASP**$[\emptyset]$.[8]

**Tomita 4** $\notin$ **C-RASP**$[\emptyset]$   By Lemma 6.11 in Krebs (2008) (discussed in Appendix C.3.2), $\Sigma^* bb \Sigma^* \notin \widehat{MAJ}_2[<]$. In analogy, $\Sigma^* bbb \Sigma^* \notin \widehat{MAJ}_2[<]$.

**Tomita 4** $\in$ **C-RASP**$[$**periodic**, **local**$]$   A **C-RASP** program tests whether there is a position with a 0 where the preceding position also holds a 0 and the position preceding that also holds a 0.

**Tomita 5** $\notin$ **C-RASP**$[$**periodic**, **local**$]$   This language is the intersection of PARITY with the strings of even length. **C-RASP**$[$periodic, local$]$ inexpressiveness follows from the same arguments as for PARITY (Lemma 41).

**Tomita 6** $\notin$ **C-RASP**$[$**periodic**, **local**$]$   Consider first the language $\mathcal{L}_3$ where the number of 1's is divisible by 3. This is not in **C-RASP**$[$periodic, local$]$ in analogy to PARITY. Now consider the length-preserving morphism $h(1) = 001$, $h(0) = 000$. Then $h(w) \in \mathcal{L}_{Tomita\ 6} \Leftrightarrow w \in \mathcal{L}_3$.

**Tomita 7** $\in$ **C-RASP**$[\emptyset]$   Tomita 7 is equivalent to $\{\epsilon\} \cup a^+ \cup b^+ \cup a^+ b^+ \cup b^+ a^+ \cup a^+ b^+ a^+ \cup a^+ b^+ a^+ b^+$. It can be shown this is equivalent to $\Sigma^* \setminus \Sigma^* b \Sigma^* a \Sigma^* b \Sigma^* a \Sigma^*$, and this was constructed in section C.1.5. Interestingly, directly implementing this **C-RASP**$[\emptyset]$ construction in a transformer appears to require at least four layers; this is in contrast to the **C-RASP**$[$periodic, local$]$ construction discussed next, where two layers are sufficient.[9]

**Tomita 7** $\in$ **C-RASP**$[$**periodic**, **local**$]$   A **C-RASP** program can count the number of positions with the bigrams 01, 10 to detect a violation.

$D_n \in$ **C-RASP**$[\emptyset]$   Similar to Tomita 2.

**PARITY** $\notin$ **C-RASP**$[$**periodic**, **local**$]$   See Lemma 41.

$(aa)^* \notin$ **C-RASP**$[\emptyset]$   See Lemma 38.

$(aa)^* \in$ **C-RASP**$[$**periodic**, **local**$]$   See Example C.2.3.

$(aaaa)^* \notin$ **C-RASP**$[\emptyset]$, $\in$ **C-RASP**$[$**periodic**, **local**$]$   Analogous to $(aa)^*$.

$(abab)^* \notin$ **C-RASP**$[\emptyset]$, $\in$ **C-RASP**$[$**periodic**, **local**$]$   Analogous to $(aa)^*$.

$aa^* bb^* cc^* dd^* ee^* \in$ **C-RASP**$[\emptyset]$   A **C-RASP** program indicates, first, the presence of "a", "b", "c", "d", "e", and, second, that every "a" is preceded only by "a"; every "b" is preceded only by "a" or "b"; every "c" is preceded only by "a", "b", "c"; and analogously for "d", "e".

$\{a, b\}^* d \{b, c\}^* \in$ **C-RASP**$[\emptyset]$   There is a single $d$; $a$ can only appear before it; $c$ can only appear after it.

$\{0, 1, 2\}^* 02^* \notin$ **C-RASP**$[$**periodic**, **local**$]$   See Lemma 36. Note that this is equivalent to the language $\Sigma^* be^*$ over the alphabet $\Sigma = \{a, b, e\}$.

---

[8]Formally, Theorems 6.10 and 6.12 in Krebs (2008) show that the regular languages in $\widehat{MAJ}_2[<]$ are contained in **DA** $*$ **G**. The second component in this product can capture the first subscript of each $\Lambda$, but not the second. Since the language $\Sigma^* aaa \Sigma^* \cup \Sigma^* bbb \Sigma^*$ over $\Sigma = \{a, b\}$ is not in **DA** (shown, e.g., via the Theorem 2c in Tesson & Thérien (2002), which would entail syntactic congruence of $(aabb)^\omega bb(aabb)^\omega$ and $(aabb)^\omega$), the claimed **C-RASP**$[\emptyset]$ program cannot exist.

[9]The first layer collects bigrams, the second layer compares the count of each bigram to the count of SOS tokens (which is known to be one) to count the bigrams.

### E.2 ALGORITHMIC TASKS

#### E.2.1 TASK DEFINITIONS FOR ALGORITHMIC PROBLEMS

The tasks are generally from Zhou et al. (2024a), except for Binary Majority Interleave. Here, we define each formally.

**Binary Majority.** The binary majority problem identifies the most frequent bit in a sequence of random bits. An example is `SOS 0 1 ... 0 SEP 1 EOS` The part `0 1 ... 0` is the sequence of random bits. We define LEN to be the length of this part. We constrain the sequences such that the number of 0s and 1s are always not equal. The model is trained with the language modeling loss on the part `1 EOS`, in other words, it is only trained to predict the most frequent bit and EOS token. The minimum length $l_{min}$ of this task is 1.

**Binary Majority Interleave.** The sequences in this problem are created by interleaving multiple binary majority (see above) inputs while avoiding repeating special tokens (e.g., SOS). We use 3 binary majority sequence to compose one sequence in this task. Formally speaking, given 3 binary sequences of the same length, $x_1^1, \cdots x_n^1, x_1^2, \cdots x_n^2$, and $x_1^3, \cdots x_n^3$, and their corresponding labels (most frequent bits) $y^1, y^2, y^3$, the interleaved input is SOS $x_1^1, x_1^2, x_1^3, x_2^1, x_2^2, x_2^3, \cdots, x_n^1, x_n^2, x_n^3$. SEP $y^1, y^2, y^3$ EOS.

An example is `SOS 1 0 1 1 0 0 ... SEP 1 0 0 EOS` LEN in this problem refers to length between SOS and SEP (excluding). The model is trained with the language modeling loss on the part `1 0 0 EOS` and $l_{min} = 3$.

**Majority.** This problem is similar to binary majority problem except that the vocabulary is bigger. An example is `SOS c b a b SEP b EOS`, where `c b a b` is a sequence of random tokens, each of which is sampled independently from an alphabet of 26 symbols. The LEN is defined as the length of of this part. We constrain the sequences such that there is always a unique answer. The model is trained with the language modeling loss on the part `b EOS`. $l_{min} = 1$.

**Sort.** In sort problem, the model outputs a sorted version of the given sequence. An example is `SOS 14 23 6 9 SEP 6 9 14 23 EOS` where `14 23 6 9` is a sequence of unique numbers. LEN in this problem refers to length this part. The model is trained with the language modeling loss on the part `6 9 14 23 EOS`. In this problem, $l_{min} = 1$. The total vocabulary size of tokens except for special tokens is equal to the maximum testing length, i.e., 150.

**Copy (unique).** In this problem, the model outputs the same sequence as the given sequence, which consists of unique tokens. An example is `SOS 14 23 6 9 SEP 14 23 6 9 EOS` where the first `14 23 6 9` is a sequence of unique numbers. LEN in this problem refers to length this part. The model is trained with the language modeling loss on the second part `14 23 6 9 EOS`. In this problem, $l_{min} = 1$. The total vocabulary size of tokens except for special tokens is equal to the maximum testing length, i.e., 150.

**Copy (repeat).** In this problem, the model outputs the same sequence as the given sequence, which can contain repeated tokens. An example is `SOS b a b SEP b a b EOS` where the first `b a b` is a sequence of random symbols. As in Zhou et al. (2024a), we use an alphabet of only 2 symbols. LEN in this problem refers to length this part. Each symbols is sampled independently and uniformly. The model is trained with the language modeling loss on the second part `b a b EOS`. In this problem, $l_{min} = 1$.

**Parity.** In the parity problem, the model recognizes whether the given sequence contains even number of 1s and outputs a corresponding token. An example is `SOS 1 0 0 1 0 SEP e EOS` The bits before SEP and after SOS is a random sequence of bits. LEN in this problem refers to length this part. The token between SEP and EOS is the label, it can be either "e" or "o", meaning even or odd number of 1s. The model is trained with the loss on the part `e EOS`. $l_{min} = 0$ for this problem. The bits are randomly sampled in a way such that the number of 1s is distributed uniformly given a fixed LEN.

| | **C-RASP**[∅] | **C-RASP**[periodic, local] | **AC**$^0$? |
|---|---|---|---|
| Binary Majority | yes | yes | no |
| Binary Majority Interleave | none found | yes | no |
| Majority | yes | yes | no |
| Sort | yes | yes | yes |
| Copy (unique) | no | yes | yes |
| Copy (repeat) | no | no | yes |
| Parity | no | no | no |
| Addition | no | no | yes |

Table 3: Expressiveness properties of algorithmic tasks as defined in Appendix E.2.1 and discussed in Appendix E.2.2. In the **C-RASP** columns, "yes" means we found a **C-RASP** program; "no" means we proved that no **C-RASP** program can exist; "none found" means that we found no program despite best efforts. All problems are expressible in **TC**$^0$, the tightest known upper-bound on the expressiveness of transformers. We also show membership in the circuit complexity class **AC**$^0$, a smaller class sometimes compared to transformers (Hao et al., 2022; Barcelo et al., 2024); it is not predictive of length generalization either here or in the regular languages benchmark (e.g., Addition is in **AC**$^0$ but Majority is not).

**Addition.** In addition problem, the model does binary addition. An example is `SOS 1 0 1 + 1 0 = 1 1 1 EOS` The two operands are sampled randomly. LEN in this problem refers to the total length of them, including "+" and "=". The model is trained with the loss on the part `1 1 1 EOS` . $l_{min} = 4$ for this problem. Note that we do not pad zeros in the front of operands to make them of equal length. The length of first operand is sampled uniformly in [1, LEN-2], and the remaining length is for the second operand. After determing the lengths, random bits are sampled uniformly.

### E.2.2 LIMIT TRANSFORMERS AND **C-RASP** EXPRESSIVENESS ON ALGORITHMIC TASKS

See Table 3. We provide proof sketches.

**Binary Majority** ∈ **C-RASP**[∅]    A single count operation is sufficient.

**Binary Majority Interleave and C-RASP**[∅]    We did not find a **C-RASP**[∅] program, though we do not have a rigorous proof of nonexistence. Note that, by Lemma 38, even the (seemingly easier) task of determining whether a given input is well-formed (input length is a multiple of the number of different majority sequences) cannot be solved by **C-RASP**[∅].

**Binary Majority Interleave** ∈ **C-RASP**[periodic, local]    Periodic functions can be used to separately implement each count operation.

**Majority** ∈ **C-RASP**[∅]    Similar to Binary Majority.

**Copy (unique)** ∈ **C-RASP**[periodic, local]    If character (or $n$-gram) repetition is prevented, then the sequence length is bounded by the alphabet, so that the space of possible inputs becomes finite, seemingly precluding the asymptotic analysis done in Theorem 7. To overcome this (apparent) challenge, we consider two formalizations of this task as operating on unbounded-length inputs.

First, as explained in Zhou et al. (2024a) (and relatedly by Jelassi et al. (2023)), the unique copying task can be realized with an induction head circuit (Section 4.1 and Appendix C.2.2). More specifically, each position first records whether SEP has already appeared. An induction circuit then predicts new tokens in proportion to how frequently they have previously followed appearances of the current token before SEP (Section 4.1). Copying without repetition is a special case where each token occurs at most once, so the output of $f$ in (10) is always 0 or 1. We show in Appendix C.2.2 that the induction head construction from Section 4.1 is expressible in **C-RASP**[periodic, local] but not in **C-RASP**[∅].

Another formalization of the task is in terms of *repeated* copying, where, given an input such as SOS a c b SEP, the model repeatedly copies the string, always predicting the next character, leading to an unbounded sequence SOS a c b SEP a c b SEP a c b SEP.... This turns the copying task into a function $f \in \mathcal{F}(\Sigma)$ that operates on unboundedly long sequences, outputting next-token predictions at each position. This alternative formalization is also expressed in **C-RASP**[periodic, local], by essentially the same induction head algorithm.

**Copy (repeat)** $\notin$ **C-RASP**[**periodic**, **local**]    One proof proceeds via communication complexity: By Corollary 13, copying of general strings is not expressible by Limit Transformers and hence not in **C-RASP**[periodic, local]. While valid, this proof does not make transparent why length generalization is much easier if repetition is avoided. A different approach, not using communication complexity and crucially using the *presence of repetition* proceeds from the fact that, over the alphabet $\Sigma \supseteq \{a, b, e\}^*$, the language $\Sigma^* be^* b\Sigma^* \notin$ **C-RASP**[periodic, local] (Lemma 35), and uses it to deduce that, given an input of the form $vbe^k bw\#vbe^k$ ($v, w \in \Sigma^*$, $k$ large), no **C-RASP**[periodic,local] program can reliably determine whether a $b$ should follow.

**Sort** $\in$ **C-RASP**[$\emptyset$]    As explained in Zhou et al. (2024a), this can be realized by selecting the smallest number in the input that is larger than the last output symbol. This algorithm does not require local or periodic positional information.

Note that, as in COPY (unique), the input length is bounded by the alphabet size in this case, but we can view it as a task defined with unbounded length by the same trick as for COPY (unique), whereby an initial sequence such as SOS a c b SEP is repeatedly sorted, leading to an unbounded sequence SOS a c b SEP a b c SEP a b c SEP....

**Parity** $\notin$ **C-RASP**[**periodic**, **local**]    See Lemma 41.

**Addition** $\notin$ **C-RASP**[**periodic**, **local**]    Addition is at least as hard as copying, because the special case of adding zero to a number amounts to copying (Corollary 13).

### E.3    DETAILS OF EXPERIMENTAL SETUP

As mentioned in the main paper, at train time, we add random offsets to position indices so that all position embeddings are trained. The offsets are sampled uniformly at random in the range $[0, N - |x|]$ (see Section 2). Like Zhou et al. (2024a), we sample independent training batches on the fly instead of using a finite-size training set. In contrast, each test set contains 2000 samples that are sampled at the beginning of each experiment.

For the problems where we train models with language modeling loss, the length of inputs is sampled uniformly from minimum up to maximum length in the specified range. This is true for training data and all test sets. As mentioned before, we also use predictive modeling. At each step, the model outputs a label indicating the set of possible next characters, including EOS. The models are trained on a whole sequence of tokens. In decoder-only models, standard predictive modeling approaches are less straightforward. Therefore, we assess predictions by combining the input and output spaces. For every position in the sequence, we evaluate the predicted character by comparing the output space (where each embedding represents a subset of possible next tokens) against the expected value.

We train decoder-only transformer from scratch, using implementations from Hugging Face Transformers[10]. We train models for maximum 30k steps with a batch size of 64. We stop training early once the model's accuracy reaches 100% on the in-distribution test set (the one in range $[l_{min}, 50]$). The model is trained with a dropout rate of 0.0. We use AdamW, with a weight decay rate of 0.01.

In preliminary experiments, we found that different model architectures, while achieving 100% accuracy on in-distribution data, may perform differently on out-of-distribution data. To draw a conclusion about how the model performs on a problem in general, we determine the hyperparameters as follows: We consider configurations of $\{1, 2, 4\}$ layers, $\{1, 2, 4\}$ heads and model dimension of

---

[10] https://huggingface.co/docs/transformers/en/model_doc/gpt2# transformers.GPT2LMHeadModel

| Problem | Model Size | LR | Max Steps |
|---|---|---|---|
| Tomita-1, 2 | 1 layer; 1 head; 16 dim | 1e-3 | 30k |
| $D_2, D_3, D_4, D_{12}$ | 1 layer; 4 head; 128 dim | 1e-4 | 30k |
| Tomita-4, 7 | 4 layer; 2 head; 64 dim | 1e-3 | 30k |
| $\{a, b\}^* d\{b, c\}^*, aa^*bb^*cc^*dd^*ee^*$ | 6 layer ; 4 head; 64 dim | 1e-4 | 30k |
| $(aa)^*, (aaaa)^*, (abab)^*$ | 6 layer; 4 head; 256 dim | 1e-4 | 60k |
| Tomita-3, 5, 6 | 6 layer; 4 head; 256 dim | 1e-4 | 60k |
| $\{0, 1, 2\}^* 02^*$, Parity | 6 layer; 4 head; 256 dim | 1e-4 | 60k |

Table 4: Experimental Hyperparameters for testing NoPE on the Regular Languages

| Problem | Model Size | LR | Max Steps |
|---|---|---|---|
| Tomita-1, 2 | 1 layer; 1 head; 16 dim | 1e-3 | 30k |
| $D_2, D_3, D_4, D_{12}$ | 1 layer; 4 head; 128 dim | 1e-4 | 30k |
| Tomita-4, 7 | 4 layer; 2 head; 128 dim | 1e-3 | 30k |
| $\{a, b\}^* d\{b, c\}^*, aa^*bb^*cc^*dd^*ee^*$ | 4 layer ; 4 head; 64 dim | 1e-4 | 30k |
| $(aa)^*, (aaaa)^*, (abab)^*$, Parity | 4 layer; 4 head; 128 dim | 1e-4 | 40k |
| $\{0, 1, 2\}^* 02^*$, Tomita-3, 5, 6 | 6 layer; 4 head; 128 dim | 1e-3 | 30k |

Table 5: Experimental Hyperparameters for testing APE on the Regular Languages.

$\{16, 64, 256\}$, and learning rate of $\{0.001, 0.0001\}$. We sweep all the configurations by iterating over every combination and choose the one that achieves the highest accuracy on $[51, 100]$ among those configurations whose accuracy on $[l_{min}, 50]$ is 100%. When there are multiple such options, e.g., their accuracy on $[51, 100]$ is 100%, the one with the simplest architecture is selected (when estimating complexity, we assume the following priority: number of layers > number of heads > model dimension). The final hyperparameters we used for each task are shown in Table 6 and 7.

When no configuration from the search space defined above can achieve accuracy of 100% on $[l_{min}, 50]$, e.g., in the case of ADDITION, we use an extra configuration, where the number of layers is 12, number of heads is 12, model dimension is 768, learning rate is 1e-4 or 3e-5 (if 1e-4 does not work), and a bigger maximum number of iterations, 60k, we also use the first 3k steps as warm-up steps.

After we determine the hyperparameter configuration, we run the experiments with multiple random seeds and report the average accuracy of 5 successful runs (those runs where the model achieves 100% accuracy on in-distribution data). We do not select successful runs in cases where we use the biggest architecture (the 12-layer configuration), because we find in many cases the accuracy on in-distribution data stops at around 99%.

The random baseline plotted for the algorithmic tasks (Figure 1, left) is computed using a 2-layer MLP with a token embedding layer; hence, the model predicts the next token solely based on the current token. It is trained with the same hyperparameters as transformers, the learning rate is 1e-3.

| Problem | Model Size | LR | Max Steps |
|---|---|---|---|
| Binary Majority | 1 layer; 1 head; 16 dim | 1e-3 | 30k |
| Binary Majority Interleave | 2 layer; 4 head; 256 dim | 1e-4 | 30k |
| Majority | 1 layer; 2 head; 256 dim | 1e-3 | 30k |
| Sort | 1 layer; 2 head; 256 dim | 1e-4 | 30k |
| Copy (unique) | 2 layer; 1 head; 64 dim | 1e-3 | 30k |
| Copy (repeat) | 4 layer; 4 head; 256 dim | 1e-3 | 30k |
| Parity | 4 layer; 2 head; 256 dim | 1e-4 | 30k |
| Addition | 12 layer; 12 head; 768 dim | 1e-4 | 60k (3k) |

Table 6: Experimental hyperparameters for testing APE on each problem. In the last column, numbers in parenthesis mean the warm-up steps, which is 0 when there is no number in parenthesis.

| Problem | Model Size | LR | Max Steps |
|---|---|---|---|
| Binary Majority | 1 layer; 1 head; 16 dim | 1e-3 | 30k |
| Binary Majority Interleave | 12 layer; 12 head; 768 dim | 1e-4 | 60k (3k) |
| Majority | 1 layer; 1 head; 64 dim | 1e-3 | 30k |
| Sort | 1 layer; 1 head; 256 dim | 1e-3 | 30k |
| Copy (unique) | 4 layer; 4 head; 256 dim | 1e-3 | 30k |
| Copy (repeat) | 4 layer; 4 head; 256 dim | 1e-3 | 30k |
| Parity | 12 layer; 12 head; 768 dim | 3e-5 | 60k (3k) |
| Addition | 12 layer; 12 head; 768 dim | 1e-4 | 60k (3k) |

Table 7: Experimental hyperparameters for testing NoPE on each problem. In the last column, numbers in parenthesis mean the warm-up steps, which is 0 when there is no number in parenthesis.

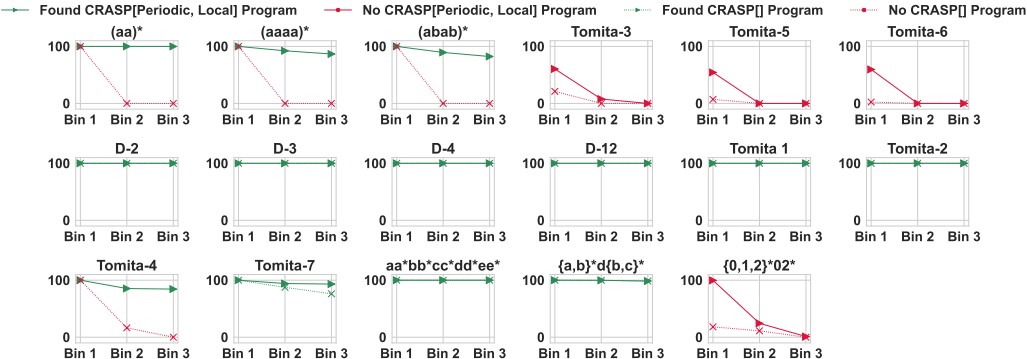

Figure 2: Detailed results for regular languages with language names, corresponding to the right part of Figure 1 but with individual languages labeled.

# F  TRANSLATING BETWEEN TRANSFORMERS AND LIMIT TRANSFORMERS

Here, we formally introduce the product parameterization, formally state the hypothesis class, and state and prove the technical lemmas establishing the correspondence between ordinary transformers and limit transformers.

## F.1  PRODUCT PARAMETERIZATION

This parameterization is defined as follows:

**Definition 43** (Product Parameterization). *For $l = 1, \ldots, L$, set*

$$
\begin{aligned}
\mathcal{VO}_0 &= \{\boldsymbol{p}_i : i\} \cup \{\boldsymbol{E}_\sigma : \sigma\} \\
\mathcal{VO}_l &= \{(\boldsymbol{B}_l)_{\cdot,s} : s = 1, \ldots, d\} \\
\mathcal{VI}_0 &= \emptyset \\
\mathcal{VI}_l &= \{(\boldsymbol{A}_l)_{s,\cdot} : s = 1, \ldots, d; \boldsymbol{U}_\sigma : \sigma \in \Sigma\} \\
\mathcal{VO} &= \bigcup_{l=0,1,\ldots,L} \mathcal{VO}_l \\
\mathcal{VI} &= \bigcup_{l=0,1,\ldots,L} \mathcal{VI}_l \\
\mathcal{P} &= \{\{V_{l_1,h_1}, \ldots, V_{l_k,h_k}\} \ : \ 0 \leq k \leq L; \quad l_1 < \cdots < l_k; \quad 1 \leq h_i \leq H\}
\end{aligned}
$$

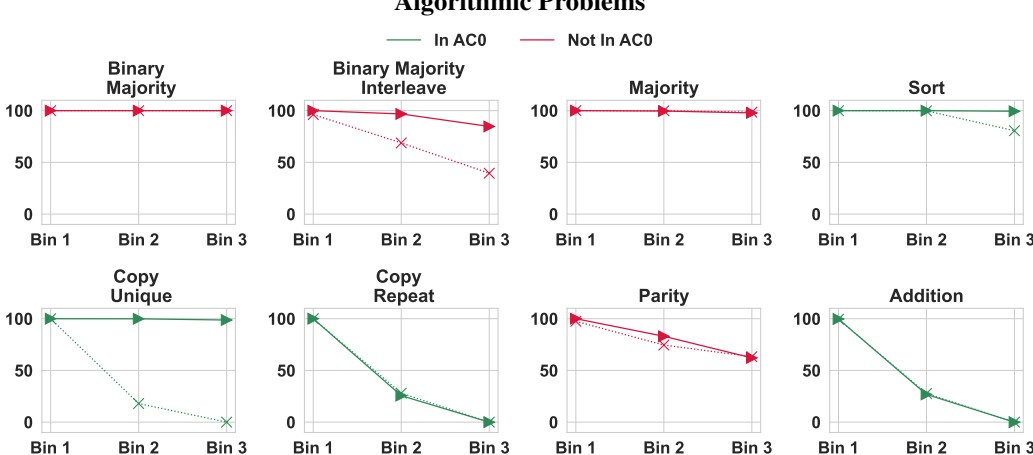

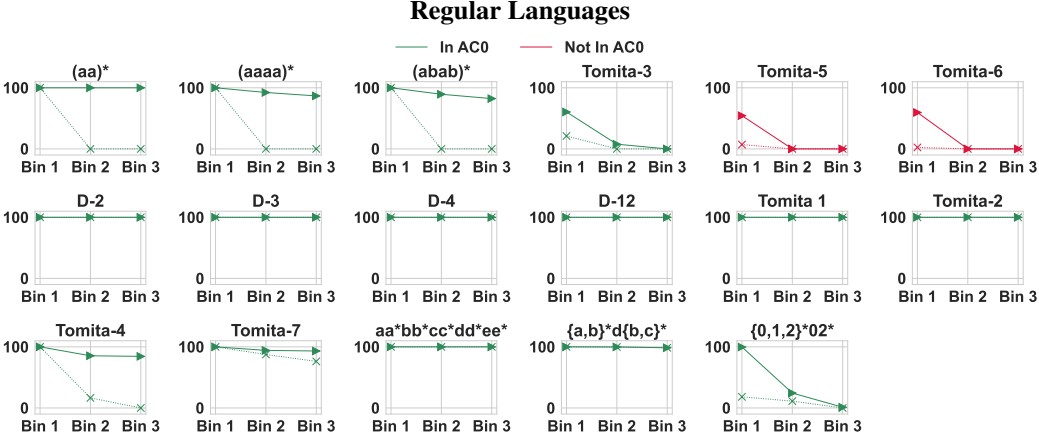

Figure 3: Membership in the circuit complexity class $\mathbf{AC}^0$ does not predict transformers' length generalization on algorithmic problems (top) or regular languages (bottom). Prior work has often linked the expressiveness of transformers to circuit complexity (e.g. Hahn, 2020; Hao et al., 2022; Merrill & Sabharwal, 2023c; Strobl, 2023; Barcelo et al., 2024). All tasks included in our experiments are in the class $\mathbf{TC}^0$, the tightest known upper bound on transformers' expressiveness. A well-known circuit complexity class within $\mathbf{TC}^0$ is $\mathbf{AC}^0$, known to upper-bound the power of certain hard-attention models of transformers (Hao et al., 2022; Barcelo et al., 2024), which may raise hopes that it helps understand transformers' practical abilities. However, membership in this class does not predict transformers' length generalization behavior. On the algorithmic problems, there is no apparent correlation at all; majority-type problems, which the attention mechanism can easily implement, are not in $\mathbf{AC}^0$, but problems with super-logarithmic communication complexity such as copying and addition (Corollary 13) are contained. On the regular languages, $\mathbf{AC}^0$ exactly covers the class $\mathbf{FO}[reg]$. This class can be proven to include all regular languages in $\mathbf{C\text{-}RASP}$, but it also includes various languages that transformers length-generalize poorly on, such as Tomita-3. A natural subclass, obtained by restricting the size of $\mathbf{AC}^0$ circuits to a linear number of wires, yields the class $\mathbf{FO}_2[Reg]$ (Cadilhac & Paperman, 2022), which does not match transformers' behavior well either, e.g. it includes $\{0,1,2\}^*02^*$ (bottom right, equals $\Sigma^*be^*$ from Lemma 11) but does not include D-12. Taken together, established circuit complexity classes do not account for Transformers' length generalization behavior. Compare to $\mathbf{C\text{-}RASP}$ results in Figures 1 and 2.

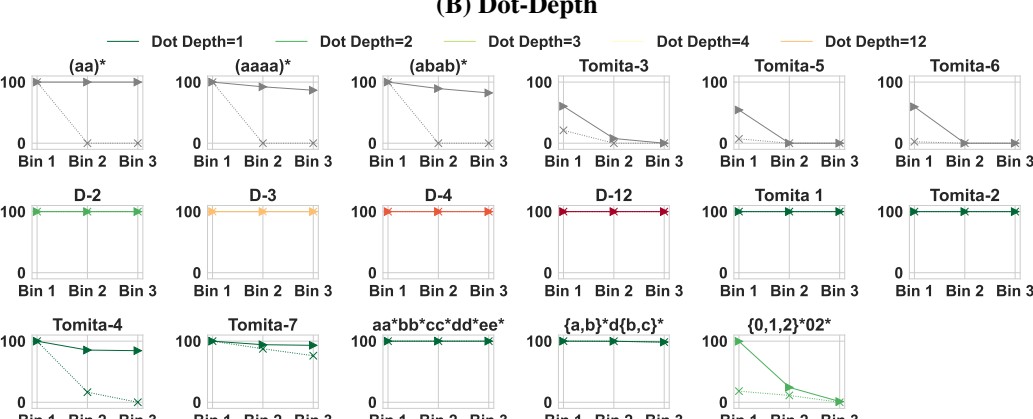

Figure 4: (1) Comparing length-generalization with a standard notion of the complexity of finite-state languages: Star-free languages (green) do not require modular counting (McNaughton & Papert, 1971), have simpler algebraic representations in terms of group-free monoids (Schützenberger, 1965), are easily represented by modern state-space models (Sarrof et al., 2024), and match the expressiveness of a formal model of hard attention Transformers (Yang et al., 2023). However, they do not consistently lead to length generalization in transformers, which on the other hand length-generalize on some non-star-free languages such as $(aa)^*$. The expressiveness of **C-RASP** correctly accounts for the observed behavior. (2) Within the star-free languages, a standard complexity metric is dot-depth, with increased dot-depth indicating increased complexity (non-star-free languages are plotted in gray color). Dot-depth does not predict length generalization, which succeeds on some languages at dot depths 1 and 12 and fails at some languages at intermediate depth. See Figure 3 for further discussion regarding another existing notion of complexity, circuit complexity, also much less successful than **C-RASP** expressiveness at predicting length generalization. Compare to **C-RASP** results in Figures 1 and 2.

*Given a transformer $T$ with $N(T) < \infty$, define:*

$$\alpha_{l,h,\mathcal{S}_1,\mathcal{S}_2,\boldsymbol{v},\boldsymbol{w}} := \boldsymbol{v}^T \left( \prod_{S \in \mathcal{S}_1} S \right)^T \boldsymbol{K}_{l,h}^T \boldsymbol{Q}_{l,h} \left( \prod_{S \in \mathcal{S}_2} S \right) \boldsymbol{w} \in \mathbb{R}$$

$$\text{for } 1 \leq l \leq L; \quad 1 \leq h \leq H; \quad \boldsymbol{v} \in \mathcal{VO}; \quad \boldsymbol{w} \in \mathcal{VO}; \quad \mathcal{S}_1, \mathcal{S}_2 \in \mathcal{P}$$

$$\beta_{\mathcal{S},\boldsymbol{v},\boldsymbol{w}} := \boldsymbol{v}^T \left( \prod_{S \in \mathcal{S}} S \right) \boldsymbol{w} \in \mathbb{R}$$

$$\text{for } \boldsymbol{v} \in \mathcal{VI}_{l_1}; \quad \boldsymbol{w} \in \mathcal{VO}_{l_2}; \quad \mathcal{S} \in \mathcal{P}$$

*where the matrix product over a set $\mathcal{S} \in \mathcal{P}$*

$$\prod_{S \in \mathcal{S}} S \tag{8}$$

*is computed in descending order of layers; with the $S$ associated with the lowest layer at the right. For instance,*

$$\prod_{S \in \{V_{1,h}, V_{3,h'}, V_{4,h''}\}} S = V_{4,h''} V_{3,h'} V_{1,h} \tag{9}$$

**Remark 44.** *Here, we exemplify the Product Parameterization (Definition 43).*

$$\alpha_{1,h,\emptyset,\emptyset,\boldsymbol{p}_i,\boldsymbol{E}_\sigma} = \boldsymbol{p}_i^T \boldsymbol{K}_{1,h}^T \boldsymbol{Q}_{1,h} \boldsymbol{E}_\sigma$$

$$\alpha_{2,h,\{\boldsymbol{V}_{1,h'}\},\emptyset,\boldsymbol{p}_i,\boldsymbol{p}_j} = \boldsymbol{p}_i^T \boldsymbol{V}_{1,h'}^T \boldsymbol{K}_{2,h}^T \boldsymbol{Q}_{2,h} \boldsymbol{p}_j$$

$$\alpha_{3,h,\{\boldsymbol{V}_{2,h'}\boldsymbol{V}_{1,h''}\},\{\boldsymbol{V}_{1,h'''}\},\boldsymbol{E}_\sigma,\boldsymbol{E}_\tau} = \boldsymbol{E}_\sigma^T \boldsymbol{V}_{1,h''}^T \boldsymbol{V}_{2,h'}^T \boldsymbol{K}_{2,h}^T \boldsymbol{Q}_{3,h} \boldsymbol{V}_{1,h'''} \boldsymbol{E}_\tau$$

$$\beta_{\emptyset,(\boldsymbol{A}_1)_{s,\cdot},\boldsymbol{p}_i} = (\boldsymbol{A}_1)_{s,\cdot}^T \boldsymbol{p}_i$$

$$\beta_{\{\boldsymbol{V}_{1,h}\},(\boldsymbol{A}_3)_{s,\cdot},\boldsymbol{E}_\sigma} = (\boldsymbol{A}_3)_{s,\cdot}^T \boldsymbol{V}_{1,h} \boldsymbol{E}_\sigma$$

$$\beta_{\{\boldsymbol{V}_{3,h'},\boldsymbol{V}_{1,h}\},\boldsymbol{U}_\tau,\boldsymbol{E}_\sigma} = \boldsymbol{U}_\tau^T \boldsymbol{V}_{3,h'} \boldsymbol{V}_{1,h} \boldsymbol{E}_\sigma$$

$$\beta_{\{\boldsymbol{V}_{3,h'},\boldsymbol{V}_{2,h}\},\boldsymbol{U}_\tau,(\boldsymbol{B}_1)_{\cdot,s}} = \boldsymbol{U}_\tau^T \boldsymbol{V}_{3,h'} \boldsymbol{V}_{2,h} (\boldsymbol{B}_1)_{\cdot,s}$$

**Remark 45.** *For ease of notation, we have not restricted the layers from which different vector parameters are taken in the definition of $\alpha$ and $\beta$; hence, they will also include products that are not relevant to actual computations, such as*

$$\boldsymbol{p}_i^T \boldsymbol{V}_{2,h}^T \boldsymbol{K}_{1,h'}^T \boldsymbol{Q}_{1,h''} \boldsymbol{V}_{3,h'''} \boldsymbol{p}_j \tag{10}$$

*where a vector of the form $V_{3,h'''} \boldsymbol{p}_j$ cannot actually feed into the computation of queries in the first layer. This is simply for simplicity of notation; such products will not impact results, though one could explicitly exclude them if one wants to obtain tighter quantitative bounds on the parameter count of Limit Transformers in Lemma 52.*

## F.2 FORMAL DEFINITION OF HYPOTHESIS CLASS

**Definition 46** (Hypothesis Class, corresponds to Definition 4). *Let $p \in \mathbb{N}$ be fixed. For each $n = 1, 2, 3, \ldots$, define the hypothesis class $\Theta_n$ as the set of transformers $T$ (as defined in Section 2) where*

1. $N(T_n) = n$

2. *each parameter vector and matrix of $T$ is represented at $p$ bits of precision*

3. *each product function (Definition 43) involving positional encodings is translation-invariant. That is, every product function involving exactly one positional encoding is constant across positions, and for every $1 \leq i, j, i + \Delta, j + \Delta \leq n$,*

$$\alpha_{l,h,\mathcal{S}_1,\mathcal{S}_2,\boldsymbol{p}_i,\boldsymbol{p}_j} = \alpha_{l,h,\mathcal{S}_1,\mathcal{S}_2,\boldsymbol{p}_{i+\Delta},\boldsymbol{p}_{j+\Delta}}$$

*for all $l, h, \mathcal{S}_1, \mathcal{S}_2$ making these objects well-defined.*

## F.3 FROM LIMIT TRANSFORMERS TO TRANSFORMERS

We give a translation from Limit Transformers to standard transformers for every finite context window $N$. The translation is uniform in the sense that $\mathcal{R}(T_N)$ from (8) is bounded across all $T_N$, though the width may need to increase. More intuition is provided further in the proof of the translation.

**Lemma 47.** *Let $T_\infty$ be a Limit Transformer satisfying* PERIODIC *and* LOCAL. *Then there are transformers $T_1, T_2, \ldots$ ($T_i \in \Theta_i$) such that, for all $i \in \mathbb{N}, x \in \mathfrak{S}, o \in \{0, \ldots, i - |x|\}$:*

$$T_i(x, o) \equiv T_\infty(x, 0) \quad when \quad o + |x| \leq i \tag{11}$$

*and*

$$\sup_i \mathcal{R}(T_i) < \infty \tag{12}$$

*Proof.* We use hats to indicate the parameters of the constructed transformers. Fix $N \in \mathbb{N}$; we construct $T_N$.

Let $\Delta$ be the periodicity of $\boldsymbol{p}_i$. The construction sets $\hat{L} = L + 2$ and $\hat{H} = \max\{1, H, \Delta\}$. Each activation has $\hat{d} := d + N + 3\Delta + 2$ dimensions, which can be partitioned into six regions:

$$\begin{pmatrix} d \text{ dimensions (Region I: main region)} \\ N \text{ dimensions (Region II: position region)} \\ \Delta \text{ dimensions (Region III: periodic region I)} \\ \Delta \text{ dimensions (Region IV: periodic Region II)} \\ 1 \text{ dimension (Region V: \texttt{SOS} Region)} \\ \Delta + 1 \text{ dimensions (Region VI: Copied \texttt{SOS} Region)} \end{pmatrix} \tag{13}$$

Region I directly emulates the computations of the Limit Transformer. Region II carries absolute positional information, used for simulating the positional functions $\phi_{l,h}$.

We define the token and positional encodings as

$$\hat{\boldsymbol{p}}_i = \begin{pmatrix} 0 \in \mathbb{R}^d \\ \boldsymbol{e}_i \in \mathbb{R}^N \\ \boldsymbol{e}_{(i\%\Delta)+1} \in \mathbb{R}^\Delta \\ 0 \in \mathbb{R}^\Delta \\ 0 \in \mathbb{R} \\ 0 \in \mathbb{R}^{\Delta+1} \end{pmatrix} \qquad \hat{\boldsymbol{E}}_\sigma = \begin{pmatrix} \boldsymbol{E}_\sigma \in \mathbb{R}^d \\ 0 \in \mathbb{R}^N \\ 0 \in \mathbb{R}^\Delta \\ 0 \in \mathbb{R}^\Delta \\ 1_{\sigma=\$} \in \mathbb{R} \\ 0 \in \mathbb{R}^{\Delta+1} \end{pmatrix}$$

where $\boldsymbol{e}_i$ is the $i$-th unit vector. Region I holds token information. Region II holds exact position information. Region III holds modular position information. Region V indicates whether the token is start-of-sequence (\texttt{SOS}) or not. Content will be written to Regions IV and VI by attention components.

**Intuition**  At first sight, a simple intuitive translation just uses Regions I and II, placing all parameters of $T_\infty$ into Region I, and taking one-hot vectors $\boldsymbol{e}_i$ for Region II to encode exact positional information, so that the functions $\phi_{l,h}$ can be implemented by products $\hat{\boldsymbol{p}}_i \hat{\boldsymbol{K}}_T \hat{\boldsymbol{Q}} \hat{\boldsymbol{p}}_j$. One would thus use the simpler encoding $\tilde{\boldsymbol{E}}_\sigma := [\boldsymbol{E}_\sigma, 0]$ and $\tilde{p}_i := [\boldsymbol{p}_i, \boldsymbol{e}_i]$. Such a translation would be able to reproduce the input-output behavior of $T_\infty$. However, it would fall short in two ways: First, the positional encodings $\boldsymbol{p}_i$ can give rise to patterns whereby $\boldsymbol{p}_i^T K^T Q \boldsymbol{p}_j$ is periodic in $j - i$ when $j - i$ is large, and thus bounded away from zero at unboundedly many distances $j - i$, making (8) unbounded. We will avoid this by routing modular positional information through a value matrix $\boldsymbol{V}_{1,1}$ before making it available to attention computations; intuitively, this is possible because the vectors $\boldsymbol{p}_i$ have a bounded-dimensional span. Modular positional information starts out in Region III of $\hat{\boldsymbol{p}}_i$, and is copied by $\boldsymbol{V}_{1,1}$ into Region IV; no $\boldsymbol{K}^T \boldsymbol{Q}$ matrix in the construction will directly address Region III. Second, transformers in $\Theta_N$ must satisfy the requirement that all product functions are translation-invariant; such a requirement need not be implemented by $T_\infty$ (e.g., MLPs could respond differently to different $\boldsymbol{p}_i$'s), and thus also not by the simple translation sketched. We overcome this by adding $\Delta$ different attention heads, each assigned to some $k \in \{0, \ldots, \Delta - 1\}$, each of which primarily attends to the \texttt{SOS} symbol (based on Region V), with a stronger weight in the $k$-th head

falling on SOS if the distance between the query position and the SOS position is congruent to $k$ modulo $\Delta$ (based on Region IV). These attention weights are written, via the value matrix, to Region VI. An MLP then compares each of these attention weights to the weights resulting from uniform attention, thereby determining the head giving rise to the highest attention weight, and places the matching encoding $\boldsymbol{p}_i$ into Region I. This construction maintains translation-invariance of all product functions; most importantly, it makes all $\beta$ functions involving positional encodings equal to zero: The MLP reads from Region VI, whose entries are linear combinations of entries from Region V, which is zero in all $\hat{p}_i$. Crucially, the dependence of the positional encoding $\boldsymbol{p}_i$ written to Region I on the original positional information in $\hat{p}_i$ is mediated entirely through attention weights, which do not enter the definition of the product functions. Once these computations are completed, Region I matches the activations in $T_\infty$ at offset 0, and a direct simulation of $T_\infty$ can proceed based on Regions I and II. Taken together, we expand the intuitive simple construction to make rigorous the intuition that bounded-rank positional information can be utilized by transformers even under the constraints that (8) be bounded and that product functions be translation-invariant.

**Layer 1: Copying Periodic Positional Information to Region IV**   In the lowest layer, each position attends to itself and moves the periodic positional information from Region III to Region IV. Formally:

$$
\boldsymbol{K}_{1,1} = \boldsymbol{Q}_{1,1} = \begin{pmatrix} 0 & 0 & 0 & 0 & 0 & 0 \\ 0 & \Omega \cdot \boldsymbol{I}_{N \times N} & 0 & 0 & 0 & 0 \\ 0 & 0 & 0 & 0 & 0 & 0 \\ 0 & 0 & 0 & 0 & 0 & 0 \\ 0 & 0 & 0 & 0 & 0 & 0 \\ 0 & 0 & 0 & 0 & 0 & 0 \end{pmatrix}
$$

$$
\boldsymbol{V}_{1,1} = \begin{pmatrix} 0 & 0 & 0 & 0 & 0 & 0 \\ 0 & 0 & 0 & 0 & 0 & 0 \\ 0 & 0 & 0 & 0 & 0 & 0 \\ 0 & 0 & \boldsymbol{I}_{\Delta \times \Delta} & 0 & 0 & 0 \\ 0 & 0 & 0 & 0 & 0 & 0 \\ 0 & 0 & 0 & 0 & 0 & 0 \end{pmatrix}
$$

for some $\Omega > 0$ to be chosen later. After attention (the MLP does nothing, $\boldsymbol{A}_1, \boldsymbol{B}_1, \boldsymbol{b}_1 \equiv 0$), the output will be

$$
\hat{\boldsymbol{y}}_i^{(1)} = \begin{pmatrix} \boldsymbol{E}_\sigma \\ \boldsymbol{e}_i \\ \boldsymbol{e}_{(i\%\Delta)+1} \\ (1-\epsilon)\boldsymbol{e}_{(i\%\Delta)+1} \\ 1_{x_i=\$} \\ 0 \end{pmatrix} \tag{14}
$$

where $\epsilon < 0.1$ when $\Omega$ is sufficiently large. Region III will not be addressed by any further downstream matrices or vectors. The idea behind this operation is to ensure no $\hat{\boldsymbol{K}}^T \hat{\boldsymbol{Q}}$ matrix will directly have to read from Region III – rather, any dependence of attention logits on modular positional information is mediated by the $\boldsymbol{V}_{1,1}$ matrix. This allows us to keep (8) bounded even in the presence of such dependence, as we detail below. Note that this strategy importantly relies on $\Delta \leq \mathcal{R}_\infty(T_\infty)$, so that $rank(\boldsymbol{V}_{1,1})$ is bounded independently of N. Intuitively, modular positional information (unlike the full positional information encoded in Region II) can be routed through bounded-rank components, which enables keeping (8) bounded.

**Layer 2: Determining Position Relative to SOS**   We now add a second layer, in which we attend with $\Delta + 1$ different heads, where the $s$-th head tests whether the distance to SOS is congruent to $s$ modulo $\Delta$, and the $\Delta + 1$-st head attends uniformly. By determining which head attends most strongly to SOS, we can read out the relative position with an MLP without breaking the translation invariance of product functions.

For $h = 1, \ldots, \Delta$, let $\boldsymbol{Q}_{2,h}$ be such that

$$\boldsymbol{Q}_{2,h} \begin{pmatrix} \cdots \\ \cdots \\ \cdots \\ \boldsymbol{e}_i \\ \cdots \\ \cdots \end{pmatrix} = \begin{pmatrix} 0 \\ 0 \\ 0 \\ \boldsymbol{e}_{i+h\%\Delta} \\ 0 \\ 0 \end{pmatrix} \tag{15}$$

Let $\boldsymbol{K}_{2,h}$ ($h = 1, \ldots, \Delta$) be the identity matrix restricted to Regions IV and V. Further, let

$$\boldsymbol{Q}_{2,\Delta+1} = \boldsymbol{K}_{2,\Delta+1} \equiv 0 \tag{16}$$

Then, for $h = 1, \ldots, \Delta$,

$$\hat{y}_i^{(1)} \hat{K}_{2,h}^T \hat{Q}_{2,h} \hat{y}_j^{(1)} = 1_{i-j \equiv h \pmod \Delta} \tag{17}$$

and

$$\hat{y}_i^{(1)} \boldsymbol{K}_{2,\Delta+1}^T \boldsymbol{Q}_{2,\Delta+1} \hat{y}_j^{(1)} = 0 \tag{18}$$

Intuitively, heads $1, \ldots, \Delta$ attend preferentially to positions at a given distance modulo $\Delta$; head $\Delta + 1$ attends everywhere. Define $\boldsymbol{V}_{2,h}$ for $h = 1, \ldots, \Delta + 1$ by

$$\boldsymbol{V}_{2,h} \begin{pmatrix} \cdots \\ \cdots \\ \cdots \\ \cdots \\ \boldsymbol{z} \\ \cdots \end{pmatrix} = \begin{pmatrix} 0 \\ 0 \\ 0 \\ 0 \\ 0 \\ \boldsymbol{z} \cdot \boldsymbol{e}_h \end{pmatrix} \tag{19}$$

As the only vector parameter with an entry in Region V is the token embedding for the SOS token, the outcome of this attention block effectively writes the attention falling on the SOS token for each of the $\Delta + 1$ attention heads. We then use a Heaviside MLP with number of hidden units bounded in $\Delta$ to determine for which $s = 1, \ldots \Delta$ it holds that entry $s$ in Region VI has a greater (as opposed to smaller) entry than entry $\Delta + 1$. The MLP, via the $B$ matrix, then writes $\boldsymbol{p}_s$ to Region I. A special case occurs at SOS, where Region V is 1 (it is 0 everywhere else); here, the $\Delta$ MLP units described above are disabled (a 1 in Region V causes a large negative number to be added to their inputs) and the MLP instead writes $\boldsymbol{p}_1$ to Region I. Overall, $\Delta + 1$ MLP units with Heaviside activation are sufficient for this construction. Overall, after the MLP, the $i$-th position in the string has $\boldsymbol{p}_{((i-1)\%\Delta)+1}$ added to Region I. Overall,

$$\hat{y}_i^{(2)} = \begin{pmatrix} \boldsymbol{E}_{x_i} + \boldsymbol{p}_{i-o+1} \\ \boldsymbol{e}_{i+o} \\ \cdots \\ \cdots \\ \cdots \\ \cdots \end{pmatrix} = \begin{pmatrix} \boldsymbol{y}_i^{(0)} \\ \boldsymbol{e}_{i+o} \\ \cdots \\ \cdots \\ \cdots \\ \cdots \end{pmatrix} \tag{20}$$

where $o$ is the offset, and the second equality holds if the Limit Transformer is run at $o = 0$. In layers $2, \ldots, \hat{L}$, only Regions I and II will receive any consideration.

**Higher Layers: Emulating $T_\infty$**   Next, for $l \geq 1$, define $\hat{K}_{l+2,h}^T \hat{Q}_{l+2,h} \in \mathbb{R}^{\hat{d} \times \hat{d}}$ as

$$\hat{K}_{l+2,h}^T \hat{Q}_{l+2,h} = \begin{pmatrix} \boldsymbol{K}_{l,h}^T \boldsymbol{Q}_{l,h} & 0 & 0 & 0 & 0 & 0 \\ 0 & W_{l,h} & 0 & 0 & 0 & 0 \\ 0 & 0 & 0 & 0 & 0 & 0 \\ 0 & 0 & 0 & 0 & 0 & 0 \\ 0 & 0 & 0 & 0 & 0 & 0 \\ 0 & 0 & 0 & 0 & 0 & 0 \end{pmatrix} \tag{21}$$

where $i \leq j$:

$$(W_{l,h})_{i,j} = \phi_{l,h}(i,j) \tag{22}$$

to satisfy

$$\boldsymbol{p}_i^T W_{l,h} \boldsymbol{p}_j = \phi_{l,h}(i,j) \tag{23}$$

Then $W$ is a sum of matrices each of which has the value $\phi_{l,h}(i-j,j)$ for some $i$ on an (off-)diagonal and zeros elsewhere; hence

$$\|W_{l,h}\|_2 \leq \sum_i |\phi_{l,h}(1,i)| \tag{24}$$

Overall, $\|\hat{K}_{l+1,h}^T \hat{Q}_{l+1,h}\|$ can be bounded, independently of $N$, in terms of $\|K_{l,h}^T Q_{l,h}\|$ and $\|\phi_{l,h}\|_1$. As $\phi_{l,h}$ is local, this 1-norm is finite.

We construct all other parameter matrices and vectors by placing the parameter from the Limit Transformer into the Region I and leaving all other regions zero. Now, by induction, the first $d$ dimensions of any activation will match those of the Limit Transformer, but shifted by two layers:

$$\hat{y}_i^{(l+2)} = \begin{pmatrix} \boldsymbol{y}_i^{(l)} \\ \boldsymbol{e}_{i+o} \\ \cdots \\ \cdots \\ \cdots \\ \cdots \end{pmatrix} \tag{25}$$

As the $U$ matrix only reads from Region I, the output will also be the same as for the Limit Transformer.

For the excess heads, the matrices are just set to $0$ – in the first or the higher layers – depending on whether $H$ or $\Delta$ is larger than the other.

**Bounding Norms and Ranks** At $l \geq 2$, now the ranks of $\boldsymbol{V}_{l,h}$ and the norms of $\boldsymbol{A}_l, \boldsymbol{B}_l, U$ and the $\ell_2$ norms of $e, b, c$ will be the same they were in the Limit Transformer. The increases from the first and second layer are bounded in terms of $\Delta$ and hence $\mathcal{R}_\infty(T_\infty)$

**Verifying Boundedness of (8)** By construction, $\hat{\boldsymbol{p}}_i^T \hat{K}_{1,h}^T \hat{Q}_{1,h} \hat{\boldsymbol{p}}_j = \delta_{ij}\delta_{h1}$ and $\hat{\boldsymbol{p}}_i^T \hat{K}_{2,h}^T \hat{Q}_{2,h} \hat{\boldsymbol{p}}_j \equiv 0$. For the higher layers, the boundedness follows because $T_\infty$ satisfies LOCAL.

**Verifying Translation Invariance** We need to verify that all product functions are translation-invariant. Each $\hat{\boldsymbol{p}}_i$ contains entries in Regions II and III. In the first layer, we have $\alpha_{1,1,\emptyset,\emptyset,\boldsymbol{p}_i,\boldsymbol{p}_j} = \delta_{ij}$, hence, these are translation-invariant. In the second layer, we have

$$\alpha_{2,s,\emptyset,\emptyset,\boldsymbol{p}_i,\boldsymbol{p}_j} = \begin{pmatrix} 0 \\ \boldsymbol{e}_i \\ \boldsymbol{e}_{(i\%\Delta)+1} \\ 0 \\ 0 \\ 0 \end{pmatrix}^T \boldsymbol{K}_{2,s}^T \boldsymbol{Q}_{2,s} \begin{pmatrix} 0 \\ \boldsymbol{e}_i \\ \boldsymbol{e}_{(j\%\Delta)+1} \\ 0 \\ 0 \\ 0 \end{pmatrix} = 0 \tag{26}$$

$$\alpha_{2,s,\{\boldsymbol{V}_1\},\emptyset,\boldsymbol{p}_i,\boldsymbol{p}_j} = 0 \tag{27}$$

$$\alpha_{2,s,\{\boldsymbol{V}_1\},\{\boldsymbol{V}_1\},\boldsymbol{p}_i,\boldsymbol{p}_j} = \begin{pmatrix} 0 \\ \boldsymbol{e}_i \\ \boldsymbol{e}_{(i\%\Delta)+1} \\ 0 \\ 0 \\ 0 \end{pmatrix}^T \boldsymbol{V}_{1,1}^T \boldsymbol{K}_{2,s}^T \boldsymbol{Q}_{2,s} \boldsymbol{V}_{1,1} \begin{pmatrix} 0 \\ \boldsymbol{e}_i \\ \boldsymbol{e}_{(j\%\Delta)+1} \\ 0 \\ 0 \\ 0 \end{pmatrix} \tag{28}$$

$$= \begin{pmatrix} 0 \\ 0 \\ 0 \\ \boldsymbol{e}_{(i\%\Delta)+1} \\ 0 \\ 0 \end{pmatrix}^T \boldsymbol{K}_{2,s}^T \boldsymbol{Q}_{2,s} \begin{pmatrix} 0 \\ 0 \\ 0 \\ \boldsymbol{e}_{(j\%\Delta)+1} \\ 0 \\ 0 \end{pmatrix} \tag{29}$$

$$= 1_{j-i\equiv s \pmod{\Delta}} \tag{30}$$

$$\tag{31}$$

These are all translation-invariant. All $\alpha_{2+l,h,\emptyset,\emptyset,\boldsymbol{p}_i,\boldsymbol{p}_j}$ equal a function $\phi_{l,h}(i,j)$ and hence are translation invariant. No $\boldsymbol{V}$ matrix ever reads from Region II. Overall, all $\alpha$ products are translation-invariant. In higher layers, $\boldsymbol{K}^T\boldsymbol{Q}$ matrices read from Regions I and II. The positional encodings $\hat{\boldsymbol{p}}_i$ write to Region II but not – not even when mediated directly through value matrices – Region I, and the products are translation-invariant because the functions $\phi_{l,h}$ are.

Consider

$$\beta_{\emptyset,(\boldsymbol{A}_1)_{s,\cdot},\boldsymbol{p}_i} = 0$$

because the first layer MLP does nothing. Second,

$$\beta_{\emptyset,(\boldsymbol{A}_2)_{s,\cdot},\boldsymbol{p}_i} = 0$$

because the second layer MLP only reads from Region VI, and none of $\boldsymbol{p}_i, \boldsymbol{V}_1\boldsymbol{p}_i$ contain any entries in Region VI. Also,

$$\beta_{\mathcal{S},\boldsymbol{U}_\sigma,\boldsymbol{p}_i} = 0$$
$$\beta_{\{\boldsymbol{V}_1\},\boldsymbol{U}_\sigma,\boldsymbol{p}_i} = 0$$
$$\beta_{\{\boldsymbol{V}_2\boldsymbol{V}_1\},\boldsymbol{U}_\sigma,\boldsymbol{p}_i} = 0$$
$$\beta_{\{\boldsymbol{V}_2\},\boldsymbol{U}_\sigma,\boldsymbol{p}_i} = 0$$

since none of $\boldsymbol{p}_i, \boldsymbol{V}_1\boldsymbol{p}_i, \boldsymbol{V}_2\boldsymbol{V}_1\boldsymbol{p}_i, \boldsymbol{V}_2\boldsymbol{p}_i$ contain any entries in Region I, where $\boldsymbol{U}_\sigma$ has its entries.

Since $\boldsymbol{V}_{2,s}$ all read only from Region V, whose entries have no connection to $\hat{p}_i$, we also have:

$$\beta_{\{\boldsymbol{V}_2,\dots\},(\boldsymbol{A}_l)_{s,\cdot},\boldsymbol{p}_i} = 0$$
$$\beta_{\{\boldsymbol{V}_2,\dots\},\boldsymbol{U}_\sigma,\boldsymbol{p}_i} = 0$$
$$\beta_{\{\boldsymbol{V}_2,\boldsymbol{V}_1,\dots\},(\boldsymbol{A}_l)_{s,\cdot},\boldsymbol{p}_i} = 0$$
$$\beta_{\{\boldsymbol{V}_2,\boldsymbol{V}_1,\dots\},\boldsymbol{U}_\sigma,\boldsymbol{p}_i} = 0$$

Overall, all $\beta$ products involving $\boldsymbol{p}_i$ are translation-invariant.

$\square$

### F.4 FROM TRANSFORMERS TO LIMIT TRANSFORMERS

We first establish various smaller lemmas. The first lemma informally says that, when $f(i,j) := \boldsymbol{p}_i^T\boldsymbol{A}\boldsymbol{p}_j$ is translation-invariant in $(i,j)$, and $\boldsymbol{A}$ has bounded rank, then $f(i,j)$ is periodic. This lemma is key to the prominent role of the PERIODIC property in Theorem 7.

**Lemma 48.** *Let $p \in \mathbb{N}$, and let $N \in \mathbb{N}$. Let $\boldsymbol{p}_1,\dots,\boldsymbol{p}_N \in \mathbb{R}^k$ such that $\|\boldsymbol{p}_i\|_2 < C$; let $\boldsymbol{A} \in \mathbb{R}^{k \times k}$. Let $f(i,j) := \boldsymbol{p}_i^T\boldsymbol{A}\boldsymbol{p}_j$ be translation invariant, in the sense that*

$$\forall 0 \le i \le j : \forall M \ge 0 : j + M \le N \Rightarrow f(i,j) = f(i+M, j+M) \tag{32}$$

*Further assume that, for $i \le j$, $f(i,j)$ can be expressed with $p$ fractional bits.[11] Define for $n \ge 0$*

$$G(n) := f(1, 1+n) \tag{33}$$

*Then there is $\Delta \in \mathbb{N}$ upper-bounded in terms of $\mathrm{rank}(\boldsymbol{A})$, $p$, $C$, and $\|\boldsymbol{A}\|$ (but not $N$) such that*

$$\forall n : n + \Delta < N \Rightarrow G(n) = G(n+\Delta) \tag{34}$$

*Proof.* Let $\rho := \mathrm{rank}(\boldsymbol{A})$; it is $> 0$ without loss of generality. We write the singular value decomposition of $\boldsymbol{A}$ as

$$\boldsymbol{A} = \boldsymbol{U}^T\Sigma\boldsymbol{V} \tag{35}$$

---

[11]In particular, this is satisfied if $\boldsymbol{p}_i, \boldsymbol{A}$ are each expressed at some fixed precision $q$, where $p \ge 3q$.

where $\Sigma \in \mathbb{R}^{\rho \times \rho}$, $\boldsymbol{U}, \boldsymbol{V} \in \mathbb{R}^{\rho \times d}$, where $\|\boldsymbol{U}\|, \|\boldsymbol{V}\| \leq 1$. Then we can write

$$\boldsymbol{p}_i^T \boldsymbol{A} \boldsymbol{p}_j = \boldsymbol{p}_i^T \boldsymbol{U}^T \Sigma \boldsymbol{V} \boldsymbol{p}_j = \begin{pmatrix} \boldsymbol{U}\boldsymbol{p}_i \\ \boldsymbol{V}\boldsymbol{p}_i \end{pmatrix}^T (\boldsymbol{I}_{\rho \times \rho} \quad \boldsymbol{0}_{\rho \times \rho})^T \Sigma (\boldsymbol{0}_{\rho \times \rho} \quad \boldsymbol{I}_{\rho \times \rho}) \begin{pmatrix} \boldsymbol{U}\boldsymbol{p}_j \\ \boldsymbol{V}\boldsymbol{p}_j \end{pmatrix} \quad (36)$$

We will henceforth replace $\boldsymbol{A}$ by $(\boldsymbol{I}_{\rho \times \rho} \quad \boldsymbol{0}_{\rho \times \rho})^T \Sigma (\boldsymbol{0}_{\rho \times \rho} \quad \boldsymbol{I}_{\rho \times \rho}) \in \mathbb{R}^{\rho \times \rho}$ and $\boldsymbol{p}_i$ by $\begin{pmatrix} \boldsymbol{U}\boldsymbol{p}_i \\ \boldsymbol{V}\boldsymbol{p}_i \end{pmatrix} \in$ $\mathbb{R}^{2 \operatorname{rank}(A)}$; note that this increases the norms of these objects at most by a multiplicative constant while preserving the products $\boldsymbol{p}_i^T \boldsymbol{A} \boldsymbol{p}_j$.

As $\|\boldsymbol{p}_i\|_2 < C$ for all $i$, we find that, when $N$ is sufficiently large, for each $\epsilon > 0$, there are $\boldsymbol{p}_i, \boldsymbol{p}_j$ $(i < j)$ such that:

$$\|\boldsymbol{p}_i - \boldsymbol{p}_j\|_2 \leq \epsilon \quad (37)$$

Take $\Delta := j - i$. The minimum required distance $\Delta$ can be upper bounded by considering the $\epsilon$-packing number of $\{v : \|v\| \leq C\}$ and applying the pigeonhole principle. Hence, overall, $\Delta$ can be upper-bounded in terms of $\epsilon$, $\operatorname{rank}(\boldsymbol{A})$, and $C$. Importantly, this bound is independent of $N$. Hence, $\forall k \in \{i + \Delta, \ldots, N\}$:

$$\begin{aligned} &= |G(k - i) - G(k - i - \Delta)| \\ &= |f(i, k) - f(i + \Delta, k)| \\ &= |f(i, k) - f(i + (j - i), k)| \\ &= |\boldsymbol{p}_i^T \boldsymbol{A} \boldsymbol{p}_k - \boldsymbol{p}_j^T \boldsymbol{A} \boldsymbol{p}_k| \\ &\leq \|\boldsymbol{p}_i - \boldsymbol{p}_j\|_2 \|\boldsymbol{A} \boldsymbol{p}_k\|_2 \\ &\leq \epsilon \cdot \|\boldsymbol{A} \boldsymbol{p}_k\|_2 \end{aligned}$$

Equivalently, using the substitution $l := k - i - \Delta$, we have, for any $l \in \{l, \ldots, N - \Delta\}$:

$$|G(l + \Delta) - G(l)| \leq \epsilon \cdot \|\boldsymbol{A} \boldsymbol{p}_k\|_2 \quad (38)$$

Take $\epsilon = \frac{2^{-p}}{4C\|A\|}$; then

$$G(l) = G(l + \Delta) \quad (39)$$

due to the assumption about fixed-precision outputs.

$\square$

**Lemma 49.** *Assume each parameter in a transformer is represented at $p$ bits of precision. Then each product function is exactly represented at $(4 + 2L)p$ bits of precision.*

*Proof.* Each product function consists at most of two vectors, a key and query matrix, and up to $2L$ value matrices. This results in a sum of numbers that each are a product of up to $4 + 2L$ numbers that each are individual parameters. As each number is represented at $p$ bits of precision, each product is represented at $(4 + 2L)p$ bits of precision. $\square$

It will be useful to define a complexity metric applicable to Limit Transformers:

**Definition 50.** *For a Limit Transformer $T$, define $\mathcal{R}_\infty(T)$ as the sum of*

1. $L + H + d$

2. *the precision $p$ used for expressing the parameters (Definition 2), and the precision $p$ used for rounding attention logits and the output of $\exp(\cdot)$ (Section 2).*

3. *the maximum $\ell^\infty$ norm of all parameter vectors and matrices (including positional encodings)*

4. *the minimal periodicity $\Delta$ of the positional encodings*[12]

5. $\max_{l,h} \sum_{i=1}^\infty |\phi_{l,h}(1, 1 + i)|^2$ *(short: $\max_{l,h} \|\phi_{l,h}\|_2^2$).*

---

[12]Formally, 1 plus the supremum of the set of $\Delta$'s for which $p_i \not\equiv \boldsymbol{p}_{i+\Delta}$.

**Proposition 51.** *Let $A \in \mathbb{R}_+$. Let $U$ be the set of Limit Transformers $T$ such that $\mathcal{R}_\infty(T) \leq A$. Then the set of parameter settings in $U$, other than the $\phi_{l,h}$ functions, is finite.*

*Proof.* Immediate. $\qquad\square$

We now state the key lemma translating ordinary transformers to Limit Transformers using the product parameterization:

**Lemma 52.** *Let $T \in \Theta_n$, at $p$ bits of precision. Let the alphabet $\Sigma$ be fixed.[13] Then there is a Limit Transformer $T_\infty$ such that $T \equiv T_\infty$ at length $\leq n$ and*

$$R_\infty(T_\infty) \leq F\left(\mathcal{R}(T)\right) \tag{40}$$

*for some universal function $F : \mathbb{R}_+ \to \mathbb{R}_+$ and*

$$\boldsymbol{p}_i^T \boldsymbol{K}_{l,h}^T \boldsymbol{Q}_{l,h} \boldsymbol{p}_j = \phi_{l,h}(i,j) \tag{41}$$

*for the $\boldsymbol{p}_i, \boldsymbol{K}_{l,h}, \boldsymbol{Q}_{l,h}$ parameters of $T$; for each $l, h$. In particular, $T_\infty$ satisfies* PERIODIC *and each $\phi_{l,h}$ is translation-invariant.*

We prove the lemma in the remainder of the section.

### F.4.1    PROVING LEMMA 52 (I): PRELIMINARIES

We discuss various preliminaries, before presenting the construction, explaining its intuition, and explaining how its soundness is formally proven.

**Basic Idea**    We will construct $T_\infty$ so that the entries in every parameter are 0, 1, or one of the product functions from Definition 43. This will automatically ensure that its parameters are represented at fixed precision $p$ bounded in terms of $\mathcal{R}(T)$, and with each entry bounded in terms of the spectral norms of parameter matrices and the $\ell_2$ of parameter vectors, hence, also bounded in terms of $\mathcal{R}(T)$. Importantly, we will use the definition of $\mathcal{R}(T)$ and the definition of $\Theta_n$ to restrict attention to a number of product functions that are bounded only in terms of $\mathcal{R}(T)$, independently of $n$.

**Bounding Active MLP Units**    First, given $\|\boldsymbol{A}_l\|_F < \infty$, the number of nonzero entries is bounded as $\leq \frac{\|\boldsymbol{A}_l\|_F^2}{2^{-2p}}$, which is bounded in terms of $\mathcal{R}(T)$. Similarly, the number of nonzero entries in $\boldsymbol{b}_l$ is bounded as $\leq \frac{\|b\|_2^2}{2^{-2p}}$, and similarly for $\boldsymbol{B}_l$. Let $d_{MLP} \leq d$ be, across $l = 1, \ldots, L$ the maximum of the maximum number of nonzero entries in $\boldsymbol{b}_l$, and of the maximum number of nonzero rows in $\boldsymbol{A}_l$. Without loss of generality, by reordering rows in $\boldsymbol{A}_l$ and columns in $\boldsymbol{B}_l$, we may assume that, in each layer, these entries are in the first $d_{MLP}$ dimensions. Then $d_{MLP}$ is upper-bounded in terms of these (or $d$, if the bounds exceed $d$); this is bounded in terms of $\mathcal{R}(T)$ and independent of $n$. In each layer, only $\leq d_{MLP}$ of the MLP units have nonzero input weights $\boldsymbol{A}_l$, output weights $\boldsymbol{B}_l$, or biases $\boldsymbol{b}_l$. Removing product functions belonging to the inactive units, we set:

$$\widehat{\mathcal{VI}}_l := \mathcal{VI}_l - \{(\boldsymbol{A}_l)_{s,\cdot} : s = d_{MLP} + 1, \ldots, d\}$$
$$\widehat{\mathcal{VI}} := \bigcup_{l=1}^L \widehat{\mathcal{VI}}_l$$

Then, the size of $\widehat{\mathcal{VI}}$ is bounded in terms of $\mathcal{R}(T)$.

**Periodicity of Bounded-Rank Positional Functions**    By Lemma 48 all products $\boldsymbol{p}_i^T \ldots \boldsymbol{p}_j$ where the intervening material has bounded rank are periodic in $j - i$ with period bounded in terms of the rank of the intervening material, and hence $\mathcal{R}(T)$. Let $\Delta$ be the least common multiple of the periods obtained from Lemma 48 across the (finitely many) different products of the form $\beta_{l,h,\mathcal{S}_1,\mathcal{S}_2,\boldsymbol{p}_i,\boldsymbol{p}_j}$

---

[13]The bound on $R_\infty(T_\infty)$ depends on it, but during the inference procedure, the alphabet is assumed fixed.

where $\mathcal{S}_1 \cup \mathcal{S}_2 \neq \emptyset$. Define

$$\widehat{\mathcal{VO}}_0 = \{p_i : i = 1, \ldots, \Delta\} \cup \{\boldsymbol{E}_\sigma : \sigma\}$$

$$\widehat{\mathcal{VO}}_l = \{(\boldsymbol{B}_l)_{\cdot,s} : s = 1, \ldots, d_{MLP}\}, \quad l = 1, \ldots, L$$

$$\widehat{\mathcal{VO}} = \widehat{\mathcal{VO}}_0 \cup \bigcup_{l=1}^{T} \widehat{\mathcal{VO}}_l$$

Then, the size of $\widehat{\mathcal{VO}}$ is bounded in terms of $\mathcal{R}(T)$.

### F.4.2 PROVING LEMMA 52 (II): CONSTRUCTION OF $T_\infty$

**Translation in Terms of Regions** We use hats (i.e., $\widehat{\cdot}$) to mark the parameters and activations of the Limit Transformer, distinguishing those from the parameters and activations of the original transformer $T$.

Each $d$-dimensional parameter vector and activation (residual streams and attention outputs) is translated to a vector consisting of three regions, each having a fixed number of dimensions bounded in terms of $\mathcal{R}(T)$, That is, each vector parameter or activation $\boldsymbol{v}$ (e.g., $\boldsymbol{p}_i$, $\boldsymbol{E}_\sigma$, $\boldsymbol{y}_i^{(l)}$) is translated to a parameter or activation $\widehat{\boldsymbol{v}}$ (e.g., $\widehat{\boldsymbol{p}_i}$, $\widehat{\boldsymbol{E}_\sigma}$, $\widehat{\boldsymbol{y}_i^{(l)}}$) vector consisting of the following three regions:

$$\widehat{\boldsymbol{v}} = \begin{pmatrix} \Gamma_{\mathcal{S},\boldsymbol{w}}(\widehat{\boldsymbol{v}}) : \mathcal{S} \in \mathcal{P}; \boldsymbol{w} \in \widehat{\mathcal{VI}_{l_2}} \\ \Lambda_{\mathcal{S}_1,\mathcal{T}_1,\mathcal{T}_2,l,h,\boldsymbol{w}_1,\boldsymbol{w}_2}(\boldsymbol{v}) : 1 \leq l \leq L; 1 \leq h \leq H; \mathcal{S}_1 \subseteq \mathcal{T}_1; \mathcal{T}_1, \mathcal{T}_2 \in \mathcal{P}, \boldsymbol{w}_1, \boldsymbol{w}_2 \in \widehat{\mathcal{VO}} \\ \Omega_{\mathcal{S}_2,\mathcal{T}_1,\mathcal{T}_2,l,h,\boldsymbol{w}_1,\boldsymbol{w}_2}(\boldsymbol{v}) : 1 \leq l \leq L; 1 \leq h \leq H; \mathcal{S}_2 \subseteq \mathcal{T}_2; \mathcal{T}_1, \mathcal{T}_2 \in \mathcal{P}, \boldsymbol{w}_1, \boldsymbol{w}_2 \in \widehat{\mathcal{VO}} \end{pmatrix} \tag{42}$$

**Intuition of the Construction** The first region, denoted $\Gamma_{\mathcal{S},\boldsymbol{w}}(\widehat{\boldsymbol{v}})$, has one entry for every choice of $\mathcal{S} \in \mathcal{P}; \boldsymbol{w} \in \widehat{\mathcal{VI}_{l_2}}$. Intuitively, the entry $\Gamma_{\mathcal{S},\boldsymbol{w}}(\widehat{\boldsymbol{v}})$ describes the outcome of applying all value matrices in $\mathcal{S}$ and then finally the vector $\boldsymbol{w}^T$:

$$\Gamma_{\mathcal{S},\boldsymbol{w}}(\widehat{\boldsymbol{v}}) = \boldsymbol{w}^T \left( \prod_{S \in \mathcal{S}} S \right) \boldsymbol{v} \in \mathbb{R} \tag{43}$$

(Recall Definition 43 for the notation $\prod_{S \in \mathcal{S}} S$.) The second and third regions each have one entry for every choice of $1 \leq l \leq L, \mathcal{S}_2 \subseteq \mathcal{T}_2; \mathcal{T}_1, \mathcal{T}_2 \in \mathcal{P}, \boldsymbol{w}_1, \boldsymbol{w}_2 \in \widehat{\mathcal{VO}}$. These regions contain the information necessary for computing attention logits. Intuitively, $\mathcal{T}_1, \mathcal{T}_2$ describe the value matrices through which $\boldsymbol{w}_1$ and $\boldsymbol{w}_2$, respectively, pass before the computation of attention logits in layer $l$. For parameter vectors $\boldsymbol{w}_1, \boldsymbol{w}_2$ (e.g., token embeddings or columns of a $\boldsymbol{B}_l$ matrix – positional encodings are somewhat special), we simply expect (note the duplicated arguments $\mathcal{T}_1, \mathcal{T}_2$ – these will be explained in the next paragraph):

$$\Lambda_{\mathcal{T}_1,\mathcal{T}_1,\mathcal{T}_2,l,h,\boldsymbol{w}_1,\boldsymbol{w}_2}(\boldsymbol{w}_1)\Omega_{\mathcal{T}_2,\mathcal{T}_1,\mathcal{T}_2,l,h,\boldsymbol{w}_1,\boldsymbol{w}_2}(\boldsymbol{w}_2) = \boldsymbol{w}_1^T \left( \prod_{S \in \mathcal{T}_1} S \right)^T \boldsymbol{K}_{l,h}^T \boldsymbol{Q}_{l,h} \left( \prod_{S \in \mathcal{T}_2} S \right) \boldsymbol{w}_2 \tag{44}$$

Thus, $\Lambda$ can be viewed as holding key parameters, whereas $\Omega$ can be viewed as holding query parameters, for the contribution that the pair of $\boldsymbol{w}_1, \boldsymbol{w}_2$ makes to attention logits in layer $l$, after passing through the value matrices in $\mathcal{T}_1, \mathcal{T}_2$, respectively. As a convention, at the level of parameter vectors, the $\Lambda$ component will hold the attention logit contribution (the RHS of this equation), whereas the $\Omega$ component will just hold zeros and ones. At the level of intermediate activations $\boldsymbol{v}$ ($\boldsymbol{y}_i^{(l)}$ or $\boldsymbol{Y}_i^{(l)}$), the situation is slightly more complex: here,

$$\Lambda_{\mathcal{S}_1,\mathcal{T}_1,\mathcal{T}_2,k,h,\boldsymbol{w}_1,\boldsymbol{w}_2}(\widehat{\boldsymbol{y}_i^{(l)}}) \tag{45}$$

denotes the contribution to attention logits for head $h$ at layer $k$ arising from multiples of $\left( \prod_{S \in \mathcal{T}_1} S \right) \widehat{\boldsymbol{w}_1}$ in an activation $\widehat{\boldsymbol{y}_i^{(k-1)}}$ interacting with multiples of $\left( \prod_{S \in \mathcal{T}_2} S \right) \widehat{\boldsymbol{w}_2}$ in an activation $\widehat{\boldsymbol{y}_j^{(k-1)}}$; a similar idea applies to $\Omega_{\ldots}$. However, additional care is needed to ensure that only

contributions from value matrices are counted that were actually passed through. The additional argument $\mathcal{S}_1$ serves as a "to-do-list": it records which of the value matrices in $\mathcal{T}_1$ still have to be traversed; whenever an activation passes through a value matrix $V_{l,h'}$, the value matrix $\widehat{V_{l,h'}}$ of the Limit Transformer moves entries from $\Lambda_{\mathcal{S}_1,\mathcal{T}_1,\mathcal{T}_2,k,h,w_1,w_2}$ to $\Lambda_{\mathcal{S}_1-\{V_{l,h'}\},\mathcal{T}_1,\mathcal{T}_2,k,h,w_1,w_2}$ – effectively removing itself from the "to-do-list". The same princple applies to $\Omega$, which maintains a to-do-list $\mathcal{S}_2$ for $\mathcal{T}_2$. In the end, only those components where the to-do-lists are empty (formally, $\mathcal{S}_1 = \mathcal{S}_2 = \emptyset$) will enter attention logit computations of the Limit Transformer:

$$\phi_{k,h}(i,j) + \sum_{\boldsymbol{v},\boldsymbol{w},\mathcal{T}_1,\mathcal{T}_2} \Lambda_{\emptyset,\mathcal{T}_1,\mathcal{T}_2,k,h,\boldsymbol{v},\boldsymbol{w}}(\widehat{\boldsymbol{y}_i^{(l)}}) \cdot \Omega_{\emptyset,\mathcal{T}_1,\mathcal{T}_2,l,h,\boldsymbol{v},\boldsymbol{w}}(\widehat{\boldsymbol{y}_j^{(l)}}) = (\boldsymbol{y}_i^{(l)})^T \boldsymbol{K}_{k,h}^T \boldsymbol{Q}_{k,h} \boldsymbol{y}_j^{(l)} \quad (46)$$

where the sum runs over all $\boldsymbol{v}, \boldsymbol{w} \in \widehat{\mathcal{VO}}$, and $\mathcal{T}_1, \mathcal{T}_2$ runs over all sets of value matrices from layers $\leq l$.

In the remainder of the proof, we present a detailed formal construction implementing this intuition. We first define, for each vector $\boldsymbol{v} \in \{\boldsymbol{p}_i, \boldsymbol{E}_\sigma, \boldsymbol{b}_l : i, l, \sigma\}$ its translation $\widehat{\boldsymbol{v}}$; throughout, we will define each of the three regions.

**Vector Parameters $\boldsymbol{b}_l \in \mathcal{VO}$**   We take $(\widehat{\boldsymbol{b}_l})_s := (\boldsymbol{b}_l)_s$ for $s = 1, \ldots, d_{MLP}$.

**Vector Parameters $\boldsymbol{E}_\sigma \in \mathcal{VO}$**   The first region provides products with other vectors appearing at higher layers (rows/columns of the $\boldsymbol{A}_l$ matrices and the unembedding matrix). The second and third regions provide products leading up to keys and values.

$$\Gamma_{\mathcal{S},\boldsymbol{w}}(\widehat{\boldsymbol{E}_\sigma}) := \begin{cases} \beta_{\mathcal{S},\boldsymbol{w},\boldsymbol{E}_\sigma} & \text{if } \mathcal{S} \in \mathcal{P}, \boldsymbol{w} \in \widehat{\mathcal{VI}} \\ 0 & \text{else} \end{cases}$$

$$\Lambda_{\mathcal{S}_1,\mathcal{T}_1,\mathcal{T}_2,l,h,\boldsymbol{w}_1,\boldsymbol{w}_2}(\widehat{\boldsymbol{E}_\sigma}) := \begin{cases} \alpha_{l,h,\mathcal{T}_1,\mathcal{T}_2,\boldsymbol{w}_1,\boldsymbol{w}_2} & \text{if } \boldsymbol{E}_\sigma = \boldsymbol{w}_1, \mathcal{S}_1 = \mathcal{T}_1 \\ 0 & \text{else} \end{cases}$$

$$\Omega_{\mathcal{S}_2,\mathcal{T}_1,\mathcal{T}_2,l,h,\boldsymbol{w}_1,\boldsymbol{w}_2}(\widehat{\boldsymbol{E}_\sigma}) := \begin{cases} 1 & \text{if } \boldsymbol{E}_\sigma = \boldsymbol{w}_2, \mathcal{S}_2 = \mathcal{T}_2 \\ 0 & \text{else} \end{cases}$$

**Vector Parameters: $\widehat{\boldsymbol{p}_i}$**   For $\boldsymbol{v} = \boldsymbol{p}_i$, the construction is analogous, however, we zero out the entries for $\Lambda_{\emptyset,\emptyset,l,h,\boldsymbol{p}_i,\boldsymbol{p}_j}$ and $\Omega_{\emptyset,\emptyset,l,h,\boldsymbol{p}_i,\boldsymbol{p}_j}$, as these will be taken care of by the $\phi_{l,h}$ functions. Formally:

$$\Gamma_{\mathcal{S},\boldsymbol{w}}(\widehat{\boldsymbol{p}_i}) := \begin{cases} \beta_{\mathcal{S},\boldsymbol{w},\boldsymbol{p}_i} & \text{if } \mathcal{S} \in \mathcal{P}, \boldsymbol{w} \in \widehat{\mathcal{VI}} \\ 0 & \text{else} \end{cases}$$

$$\Lambda_{\mathcal{S}_1,\mathcal{T}_1,\mathcal{T}_2,l,h,\boldsymbol{w}_1,\boldsymbol{w}_2}(\widehat{\boldsymbol{p}_i}) := \begin{cases} \alpha_{l,h,\mathcal{T}_1,\mathcal{T}_2,\boldsymbol{w}_1,\boldsymbol{w}_2} & \text{if } \boldsymbol{p}_{i\%\Delta} = \boldsymbol{w}_1; (\boldsymbol{w}_2 \notin \{\boldsymbol{p}_j : j\} \vee \mathcal{T}_1 \cup \mathcal{T}_2 \neq \emptyset), \\ & \quad\quad\quad \mathcal{S}_1 = \mathcal{T}_1 \\ 0 & \text{else} \end{cases}$$

$$\Omega_{\mathcal{S}_2,\mathcal{T}_1,\mathcal{T}_2,l,h,\boldsymbol{w}_1,\boldsymbol{w}_2}(\widehat{\boldsymbol{p}_i}) := \begin{cases} 1 & \text{if } \boldsymbol{p}_{i\%\Delta} = \boldsymbol{w}_2; (\boldsymbol{w}_1 \notin \{\boldsymbol{p}_j : j\} \vee \mathcal{T}_1 \cup \mathcal{T}_2 \neq \emptyset), \\ & \quad\quad\quad \mathcal{S}_2 = \mathcal{T}_2 \\ 0 & \text{else} \end{cases}$$

We need to establish that $T_\infty$ satisfies PERIODIC with the period $\Delta$ given above. First, $\beta_{\mathcal{S},\boldsymbol{v},\boldsymbol{p}_i}$ is independent of $i$ by translation-invariance, thus trivially periodic in $i$. Second, the $\Lambda$ and $\Omega$ entries are periodic in $i$ with period $\Delta$ by construction.

**Matrix Parameters:** $\widehat{V_{l,h}}$    Each entry in the $\widehat{V_{l,h}}$ matrix is zero or one. We define it implicitly, in terms of its action on the three different regions:

$$\Gamma_{\mathcal{S},\boldsymbol{w}}(\widehat{V_{l,h}}(\widehat{\boldsymbol{y}_l^{(l)}})) = \begin{cases} \Gamma_{\mathcal{S}\cup\{\boldsymbol{V}_{l,h}\},\boldsymbol{w}}(\widehat{\boldsymbol{y}_l^{(l)}}) & \boldsymbol{V}_{l,h} \notin \mathcal{S} \\ 0 & \text{else} \end{cases}$$

$$\Lambda_{\mathcal{S}_1,\mathcal{T}_1,\mathcal{T}_2,l,h,\boldsymbol{w}_1,\boldsymbol{w}_2}(\widehat{V_{l,h}}(\widehat{\boldsymbol{y}_l^{(l)}})) = \begin{cases} \Lambda_{\mathcal{S}_1\cup\{\boldsymbol{V}_{l,h}\},\mathcal{T}_1,\mathcal{T}_2,l,h,\boldsymbol{w}_1,\boldsymbol{w}_2}(\widehat{\boldsymbol{y}_l^{(l)}}) & \boldsymbol{V}_{l,h} \notin \mathcal{S}_1, \boldsymbol{V}_{l,h} \in \mathcal{T}_1 \\ 0 & \text{else} \end{cases}$$

$$\Omega_{\mathcal{S}_2,\mathcal{T}_1,\mathcal{T}_2,l,h,\boldsymbol{w}_1,\boldsymbol{w}_2}(\widehat{V_{l,h}}(\widehat{\boldsymbol{y}_l^{(l)}})) = \begin{cases} \Omega_{\mathcal{S}_2\cup\{\boldsymbol{V}_{l,h}\},\mathcal{T}_1,\mathcal{T}_2,l,h,\boldsymbol{w}_1,\boldsymbol{w}_2}(\widehat{\boldsymbol{y}_l^{(l)}}) & \boldsymbol{V}_{l,h} \notin \mathcal{S}_2, \boldsymbol{V}_{l,h} \in \mathcal{T}_2 \\ 0 & \text{else} \end{cases}$$

**Matrix Parameters:** $\widehat{A_l}, \widehat{B_l}$    Let $s \in \{1, \dots, d_{MLP}\}$. For the $s$-th unit in the MLP at layer $l$, we first define the $s$-th row of $\widehat{A_l}$ by setting

$$(\widehat{A_l})_{s,\cdot} \cdot \widehat{Y_i^{(l)}} = \Gamma_{\emptyset,(A_l)_{s,\cdot}}\left(\widehat{Y_i^{(l)}}\right) \in \mathbb{R}$$

and define the $s$-th column of $\widehat{B_l}$ as follows – writing $\hat{X} \in \mathbb{R}$ for the $s$-th hidden unit activation:

$$\Gamma_{\mathcal{S},\boldsymbol{w}}(\widehat{B}_{s,\cdot}\hat{X}) := \hat{X} \cdot \beta_{\mathcal{S},\boldsymbol{w},(B_l)_{\cdot,s}}$$

$$\Lambda_{\mathcal{S}_1,\mathcal{T}_1,\mathcal{T}_2,l,h,\boldsymbol{w}_1,\boldsymbol{w}_2}(\widehat{B}_{l\cdot,s}\hat{X}) := \hat{X} \cdot \begin{cases} \alpha_{l_1,h,l_2,\mathcal{T}_1,\mathcal{T}_2,(B_l)_{\cdot,s},\boldsymbol{w}_2} & \text{if } \boldsymbol{w}_1 = (B_l)_{\cdot,s}, \mathcal{S}_1 = \mathcal{T}_1 \\ 0 & \text{else} \end{cases}$$

$$\Omega_{\mathcal{S}_2,\mathcal{T}_1,\mathcal{T}_2,l,h,\boldsymbol{w}_1,\boldsymbol{w}_2}(\widehat{B}_{l\cdot,s}\hat{X}) := \hat{X} \cdot \begin{cases} 1 & \text{if } \boldsymbol{w}_2 = (B_l)_{\cdot,s}, \mathcal{S}_2 = \mathcal{T}_2 \\ 0 & \text{else} \end{cases}$$

This defines the $\widehat{A_l}$ and $\widehat{B_l}$ matrix parameters, and we can write, letting $\psi_{l,s}$ denote the activation function (ReLU or Heaviside) applying to the $s$-th hidden MLP unit:

$$\widehat{\boldsymbol{y}_i^{(l)}} = \widehat{Y_i^{(l)}} + \sum_{s=1}^{d_{MLP}} (\widehat{B_l})_{\cdot,s} \cdot \psi_{l,s}\left((\widehat{A_l})_{s,\cdot}(\widehat{Y_i^{(l)}}) + (\widehat{b_l})_s\right) \tag{47}$$

or equivalently

$$\widehat{\boldsymbol{y}_i^{(l)}} = \widehat{Y_i^{(l)}} + \widehat{B_l} \cdot \phi_l(\widehat{A_l} \cdot \widehat{Y_i^{(l)}} + \widehat{b_l}) \tag{48}$$

matching the formulation of MLPs for our model of transformers (Equation 4).

Note that the hidden dimension of the MLP in the Limit Transformer is now $d_{MLP}$, which will be smaller than $\widehat{d}$. We thus pad the remaining rows/columns of $\widehat{A_l}, \widehat{B_l}$, and the remaining entries of $\widehat{b_l}$ with zeros.

**A partial order on sets of value matrices**    For $\mathcal{S}, \mathcal{T} \in \mathcal{P}$, we write $\mathcal{T} \geq_l \mathcal{S}$ to denote that

1. $\mathcal{T} \supseteq \mathcal{S}$
2. $\forall l' \in \{1, \dots, L\} : [(\boldsymbol{V}_{l',h'} \in \mathcal{S}) \Rightarrow l' > l]$
3. $\forall l' \in \{1, \dots, L\} : [(\boldsymbol{V}_{l',h'} \in \mathcal{T} - \mathcal{S}) \Rightarrow l \geq l']$

**Intuitively**, "$\mathcal{T} \geq_l \mathcal{S}$" says that "*among the value matrices in $\mathcal{T}$, the activation has already passed through all value matrices at layer $l$ and below*". For example:

$$\{V_{1,h}, V_{2,h'}, V_{3,h''}, V_{5,'''}\} \geq_2 \{V_{3,h''}, V_{5,'''}\}$$
$$\{V_{1,h}, V_{2,h'}, V_{3,h''}, V_{5,'''}\} \not\geq_2 \{V_{4,h'''}\}$$
$$\{V_{1,h}, V_{2,h'}, V_{3,h''}, V_{5,'''}\} \not\geq_1 \{V_{3,h''}, V_{5,'''}\}$$
$$\{V_{1,h}, V_{2,h'}, V_{3,h''}, V_{5,'''}\} \not\geq_3 \{V_{3,h''}, V_{5,'''}\}$$

**Matrix Parameters:** $\widehat{K^T_{l,h}Q_{l,h}}$   We again define them implicitly in terms of regions; this can be realized using matrices $\widehat{K^T_{l,h}}, \widehat{Q_{l,h}}$ where all entries are 0 or 1. Importantly, we sum only those entries where the "to-do-lists" $\mathcal{S}_1, \mathcal{S}_2$ are empty, and the sets $\mathcal{T}_1, \mathcal{T}_2$ only contain value matrices at layers $\leq l$:

$$(\widehat{y_i^{(l)}})^T \widehat{K^T_{l,h}Q_{l,h}} \widehat{y_j^{(l)}} = \sum_{\boldsymbol{v},\boldsymbol{w},\mathcal{T}_1 \geq_l \emptyset, \mathcal{T}_2 \geq_l \emptyset} \Lambda_{\emptyset,\mathcal{T}_1,\mathcal{T}_2,l,h,\boldsymbol{v},\boldsymbol{w}}(\widehat{y_i^{(l)}}) \cdot \Omega_{\emptyset,\mathcal{T}_1,\mathcal{T}_2,l,h,\boldsymbol{v},\boldsymbol{w}}(\widehat{y_j^{(l)}}) \tag{49}$$

**Matrix Parameters:** $U$   The $U$ matrix is translated as follows:

$$\widehat{U}_\sigma^T \widehat{y_i^{(L)}} = \Gamma_{\emptyset,U_\sigma}(\widehat{y_i^{(L)}}) \tag{50}$$

**Positional Function**   Define for $l = 1, \ldots, L$ and $h = 1, \ldots, H$, when $1 \leq i \leq j \leq N(T)$:

$$\phi_{l,h}(i,j) = \boldsymbol{p}_i^T \boldsymbol{K}_{l,h}^T \boldsymbol{Q}_{l,h} \boldsymbol{p}_j \tag{51}$$

As $T \in \Theta_n$, $\phi_{l,h}(i,j)$ only depends on $j - i$.

**Bounding** $\mathcal{R}_\infty(T_\infty)$   First, we showed above that $\widehat{d}$ is upper-bounded in $\mathcal{R}(T)$. Second, all parameters are represented at precision bounded in terms of $\mathcal{R}(T)$: those parameters that are taken from product functions have precision $\leq 4Lp$ bits; those involving the SVDs of $K^TQ$ matrices also have bounded precision. Third, the $\ell^\infty$ norm of all parameter vectors is bounded in terms of $\mathcal{R}(T)$ by construction. Fourth, $\Delta$ is bounded in terms of $\mathcal{R}(T)$ as discussed above. The boundedness of the fifth term is immediate.

**Summary**   We have constructed a Limit Transformer $T_\infty$ such that

$$R_\infty(T_\infty) \leq F(\mathcal{R}(T)) \tag{52}$$

for some universal function $F : \mathbb{R}_+ \to \mathbb{R}_+$ and

$$\boldsymbol{p}_i^T \boldsymbol{K}_{l,h}^T \boldsymbol{Q}_{l,h} \boldsymbol{p}_j = \phi_{l,h}(i,j) \tag{53}$$

for the $\boldsymbol{p}_i, \boldsymbol{K}_{l,h}, \boldsymbol{Q}_{l,h}$ parameters of $T$; for each $l, h$. By assumption on $T$, each $\phi_{l,h}$ is translation-invariant. We have also constructed $\widehat{p}_i$ with period $\Delta$, so that $T_\infty$ satisfies PERIODIC.

### F.4.3   PROVING LEMMA 52 (III): PROVING CORRECTNESS

In order to conclude Lemma 52, it remains to establish the correctness of the translation; that is, $T \equiv T_\infty$ at length $\leq N(T)$. To do this, it suffices to show that both transformers provide the same next-token predictions for each $i = 1, \ldots, N(T)$:

$$\boxed{\widehat{U}_\sigma^T \widehat{y_i^{(L)}} = \Gamma_{\emptyset,U_\sigma}(\widehat{y_i^{(L)}}) = U_\sigma^T y_i^{(L)}} \tag{54}$$

**Informally**, proving this requires showing that the attention logits and MLP activations in $T_\infty$ match those in $T$; the result then follows from the linearity of $\Gamma_{\emptyset,U_\sigma}$ and the way $\Gamma_{\ldots}$ is defined for the vector parameters and how value matrices $\widehat{V_{l,h}}$ move information. **Formally**, we prove the correctness of the translation inductively, by showing the following equalities. Recall (from Definition 43) that, when $\mathcal{S} \in \mathcal{P}$ is a set of value matrices, we write $\prod_{S \in \mathcal{S}} S$ for the product of these matrices, ordered by layers, with the matrix associated with the lowest layer at the right. Then

**Lemma 53** (Preservation of Products by Translation). *For layer $l \in \{0, 1, \ldots, L\}$, for any $\mathcal{S}, \mathcal{S}_1, \mathcal{S}_2 \in \mathcal{P}$ for any $k > l$, for any $m \geq l$, and for any $\boldsymbol{w} \in \widehat{\mathcal{VI}}$, by induction over the layers $l$:*

$(A)$   *Preservation of Products with Vector Parameters*

$$\Gamma_{\mathcal{S},\boldsymbol{w}}(\widehat{\boldsymbol{y}_i^{(l)}}) = \boldsymbol{w}^T \left( \prod_{S \in \mathcal{S}} S \right) \boldsymbol{y}_i^{(l)}$$

$if \, \forall l' \in \{1, \ldots, L\} : [(\boldsymbol{V}_{l',h'} \in \mathcal{S}) \Rightarrow l' > l]$

$(B)$   *Preservation of Attention Logits (I)*

$$\sum_{\boldsymbol{v},\boldsymbol{w},\mathcal{T}_1 \geq_l \mathcal{S}_1, \mathcal{T}_2 \geq_l \mathcal{S}_2} \Lambda_{\mathcal{S}_1,\mathcal{T}_1,\mathcal{T}_2,k,h,\boldsymbol{v},\boldsymbol{w}}(\widehat{\boldsymbol{y}_i^{(l)}}) \cdot \Omega_{\mathcal{S}_2,\mathcal{T}_1,\mathcal{T}_2,k,h,\boldsymbol{v},\boldsymbol{w}}(\widehat{\boldsymbol{y}_j^{(l)}})$$

$$= (\boldsymbol{y}_i^{(l)})^T \left( \prod_{S \in \mathcal{S}_1} S \right)^T \boldsymbol{K}_{k,h}^T \boldsymbol{Q}_{k,h} \left( \prod_{S \in \mathcal{S}_2} S \right) \boldsymbol{y}_j^{(l)}$$

$if \, \mathcal{S}_1 \cup \mathcal{S}_2 \neq \emptyset$

$(C)$   *Preservation of Attention Logits (II)*

$$\phi_{k,h}(i,j) + \sum_{\boldsymbol{v},\boldsymbol{w},\mathcal{T}_1 \geq_l \emptyset, \mathcal{T}_2 \geq_l \emptyset} \Lambda_{\emptyset,\mathcal{T}_1,\mathcal{T}_2,k,h,\boldsymbol{v},\boldsymbol{w}}(\widehat{\boldsymbol{y}_i^{(l)}}) \cdot \Omega_{\emptyset,\mathcal{T}_1,\mathcal{T}_2,k,h,\boldsymbol{v},\boldsymbol{w}}(\widehat{\boldsymbol{y}_j^{(l)}})$$

$$= (\boldsymbol{y}_i^{(l)})^T \boldsymbol{K}_{k,h}^T \boldsymbol{Q}_{k,h} \boldsymbol{y}_j^{(l)}$$

(55)

$(D)$   *Preservation of Attention Logits (III)*

$$\sum_{\boldsymbol{v},\boldsymbol{w},\mathcal{T}_1 \geq_l \mathcal{S}_1, \mathcal{T}_2 \geq_l \mathcal{S}_2} \Lambda_{\mathcal{S}_1,\mathcal{T}_1,\mathcal{T}_2,k,h,\boldsymbol{v},\boldsymbol{w}}((\widehat{\boldsymbol{B}_m})_{\cdot,s}) \cdot \Omega_{\mathcal{S}_2,\mathcal{T}_1,\mathcal{T}_2,k,h,\boldsymbol{v},\boldsymbol{w}}(\widehat{\boldsymbol{y}_j^{(l)}})$$

$$= (\boldsymbol{B}_m)_{\cdot,s}^T \left( \prod_{S \in \mathcal{S}_1} S \right)^T \boldsymbol{K}_{k,h}^T \boldsymbol{Q}_{k,h} \left( \prod_{S \in \mathcal{S}_2} S \right) \boldsymbol{y}_j^{(l)}$$

$(E)$   *Preservation of Attention Logits (IV)*

$$\sum_{\boldsymbol{v},\boldsymbol{w},\mathcal{T}_1 \geq_l \mathcal{S}_1, \mathcal{T}_2 \geq_l \mathcal{S}_2} \Lambda_{\mathcal{S}_1,\mathcal{T}_1,\mathcal{T}_2,k,h,\boldsymbol{v},\boldsymbol{w}}(\widehat{\boldsymbol{y}_i^{(l)}}) \cdot \Omega_{\mathcal{S}_2,\mathcal{T}_1,\mathcal{T}_2,k,h,\boldsymbol{v},\boldsymbol{w}}((\widehat{\boldsymbol{B}_m})_{\cdot,s})$$

$$= (\boldsymbol{y}_i^{(l)})^T \left( \prod_{S \in \mathcal{S}_1} S \right)^T \boldsymbol{K}_{k,h}^T \boldsymbol{Q}_{k,h} \left( \prod_{S \in \mathcal{S}_2} S \right) (\boldsymbol{B}_m)_{\cdot,s}$$

*and analogous statements with the post-MLP activations* $\boldsymbol{y}_i^{(l)}, \widehat{\boldsymbol{y}_i^{(l)}}$ *replaced by the pre-MLP activations* $\boldsymbol{Y}_i^{(l)}, \widehat{\boldsymbol{Y}}_i^{(l)}$.

From (A), we in particular obtain the correctness of the translation by noting that next-token predictions are replicated:

$$\widehat{\boldsymbol{U}}_\sigma^T \widehat{\boldsymbol{y}_i^{(L)}} = \Gamma_{\emptyset,\boldsymbol{U}_\sigma}(\widehat{\boldsymbol{y}_i^{(L)}}) = \boldsymbol{U}_\sigma^T \boldsymbol{y}_i^{(L)}$$

(56)

*Proof of Lemma 53.* The formal proof proceeds by induction over $l$. It is conceptually straightforward, consisting of expanding definitions and taking care of the special treatment of positional encodings. We show it in considerable detail to build intuition. For the **inductive base**, at $l = 0$, where $\boldsymbol{y}_i^{(0)}$ is a sum of a word embedding and a positional encoding, the claims are immediate from the definitions. For expository purposes, and for building intuition for the more complex inductive step, we show them in more detail. Starting from (for simplicity, we are taking the offset to be zero here):

$$\boldsymbol{y}_i^{(0)} = \boldsymbol{E}_{x_i} + \boldsymbol{p}_i$$
$$\widehat{\boldsymbol{y}_i^{(0)}} = \widehat{\boldsymbol{E}_{x_i}} + \widehat{\boldsymbol{p}}_i$$

(57)

we first, for (A), write

$$\Gamma_{\mathcal{S},\boldsymbol{w}}(\widehat{\boldsymbol{E}_{x_i}} + \widehat{\boldsymbol{p}_i}) = \Gamma_{\mathcal{S},\boldsymbol{w}}(\widehat{\boldsymbol{E}_{x_i}}) + \Gamma_{\mathcal{S},\boldsymbol{w}}(\widehat{\boldsymbol{p}_i})$$
$$= \beta_{\mathcal{S},\boldsymbol{w},\boldsymbol{E}_{x_i}} + \beta_{\mathcal{S},\boldsymbol{w},\boldsymbol{p}_{i\%\Delta}}$$
$$= \boldsymbol{w}^T \left(\prod_{S\in\mathcal{S}} S\right) \boldsymbol{y}_i^{(0)}$$

proving case (A) of the inductive base. Second, for (B) and (C), write using the linearity of $\Lambda_{\ldots}, \Omega_{\ldots}$:

$$\sum_{\boldsymbol{v},\boldsymbol{w},\mathcal{T}_1 \geq_0 \mathcal{S}_1,\mathcal{T}_2 \geq_0 \mathcal{S}_2} \Lambda_{\mathcal{S}_1,\mathcal{T}_1,\mathcal{T}_2,k,h,\boldsymbol{v},\boldsymbol{w}}(\widehat{\boldsymbol{E}_{x_i}} + \widehat{\boldsymbol{p}_i}) \cdot \Omega_{\mathcal{S}_2,\mathcal{T}_1,\mathcal{T}_2,k,h,\boldsymbol{v},\boldsymbol{w}}(\widehat{\boldsymbol{E}_{x_i}} + \widehat{\boldsymbol{p}_i})$$

$$= \sum_{\boldsymbol{v},\boldsymbol{w},\mathcal{T}_1 \geq_0 \mathcal{S}_1,\mathcal{T}_2 \geq_0 \mathcal{S}_2} \Lambda_{\mathcal{S}_1,\mathcal{T}_1,\mathcal{T}_2,k,h,\boldsymbol{v},\boldsymbol{w}}(\widehat{\boldsymbol{E}_{x_i}}) \cdot \Omega_{\mathcal{S}_2,\mathcal{T}_1,\mathcal{T}_2,k,h,\boldsymbol{v},\boldsymbol{w}}(\widehat{\boldsymbol{E}_{x_i}})$$

$$+ \sum_{\boldsymbol{v},\boldsymbol{w},\mathcal{T}_1 \geq_0 \mathcal{S}_1,\mathcal{T}_2 \geq_0 \mathcal{S}_2} \Lambda_{\mathcal{S}_1,\mathcal{T}_1,\mathcal{T}_2,k,h,\boldsymbol{v},\boldsymbol{w}}(\widehat{\boldsymbol{E}_{x_i}}) \cdot \Omega_{\mathcal{S}_2,\mathcal{T}_1,\mathcal{T}_2,k,h,\boldsymbol{v},\boldsymbol{w}}(\widehat{\boldsymbol{p}_i})$$

$$+ \sum_{\boldsymbol{v},\boldsymbol{w},\mathcal{T}_1 \geq_0 \mathcal{S}_1,\mathcal{T}_2 \geq_0 \mathcal{S}_2} \Lambda_{\mathcal{S}_1,\mathcal{T}_1,\mathcal{T}_2,k,h,\boldsymbol{v},\boldsymbol{w}}(\widehat{\boldsymbol{p}_i}) \cdot \Omega_{\mathcal{S}_2,\mathcal{T}_1,\mathcal{T}_2,k,h,\boldsymbol{v},\boldsymbol{w}}(\widehat{\boldsymbol{E}_{x_i}})$$

$$+ \sum_{\boldsymbol{v},\boldsymbol{w},\mathcal{T}_1 \geq_0 \mathcal{S}_1,\mathcal{T}_2 \geq_0 \mathcal{S}_2} \Lambda_{\mathcal{S}_1,\mathcal{T}_1,\mathcal{T}_2,k,h,\boldsymbol{v},\boldsymbol{w}}(\widehat{\boldsymbol{p}_i}) \cdot \Omega_{\mathcal{S}_2,\mathcal{T}_1,\mathcal{T}_2,k,h,\boldsymbol{v},\boldsymbol{w}}(\widehat{\boldsymbol{p}_i})$$

The only way of satisfying $\mathcal{T} \geq_0 \mathcal{S}$ is for $\mathcal{T}$ to equal $\mathcal{S}$. After plugging in the definitions, the sums collapse due to the indicator terms in the definition of the token and positional encodings, and we obtain after simplifying:

$$= \alpha_{k,h,\mathcal{S}_1,\mathcal{S}_2,\boldsymbol{E}_{x_i},\boldsymbol{E}_{x_j}}$$
$$+ \alpha_{\mathcal{S}_1,\mathcal{S}_1,\mathcal{S}_2,k,h,\boldsymbol{E}_{x_i},\boldsymbol{p}_{j\%\Delta}}$$
$$+ \alpha_{\mathcal{S}_1,\mathcal{S}_1,\mathcal{S}_2,k,h,\boldsymbol{p}_{i\%\Delta},\boldsymbol{E}_{x_j}}$$
$$+ \alpha_{\mathcal{S}_1,\mathcal{S}_1,\mathcal{S}_2,k,h,\boldsymbol{p}_{i\%\Delta},\boldsymbol{p}_{j\%\Delta}} \cdot 1_{\mathcal{S}_1\cup\mathcal{S}_2\neq\emptyset}$$

By translation-invariance, the second and third term are independent of the positional encoding arguments. By our choice of $\Delta$ at the beginning of the proof, the fourth term equals $\alpha_{\mathcal{S}_1,\mathcal{S}_1,\mathcal{S}_2,k,h,\boldsymbol{p}_i,\boldsymbol{p}_j} \cdot 1_{\mathcal{S}_1\cup\mathcal{S}_2\neq\emptyset}$, as this is periodic in $(i,j)$ with periodicity $\Delta$. We can thus rewrite as

$$= \alpha_{k,h,\mathcal{S}_1,\mathcal{S}_2,\boldsymbol{E}_{x_i},\boldsymbol{E}_{x_j}}$$
$$+ \alpha_{\mathcal{S}_1,\mathcal{S}_1,\mathcal{S}_2,k,h,\boldsymbol{E}_{x_i},\boldsymbol{p}_j}$$
$$+ \alpha_{\mathcal{S}_1,\mathcal{S}_1,\mathcal{S}_2,k,h,\boldsymbol{p}_i,\boldsymbol{E}_{x_j}}$$
$$+ \alpha_{\mathcal{S}_1,\mathcal{S}_1,\mathcal{S}_2,k,h,\boldsymbol{p}_i,\boldsymbol{p}_j} \cdot 1_{\mathcal{S}_1\cup\mathcal{S}_2\neq\emptyset}$$

Applying the definition of $\alpha_{\ldots}$, the above equals

$$= \begin{cases} (\boldsymbol{y}_i^{(0)})^T \boldsymbol{K}_{k,h}^T \boldsymbol{Q}_{k,h}(\boldsymbol{y}_j^{(0)}) - \phi_{k,h}(i,j) & \mathcal{S}_1 \cup \mathcal{S}_2 = \emptyset \\ (\boldsymbol{y}_i^{(0)})^T \left(\prod_{S\in\mathcal{S}_1} S\right)^T \boldsymbol{K}_{k,h}^T \boldsymbol{Q}_{k,h} \left(\prod_{S\in\mathcal{S}_2} S\right) (\boldsymbol{y}_j^{(0)}) & \mathcal{S}_1 \cup \mathcal{S}_2 \neq \emptyset \end{cases}$$

This establishes cases (B) and (C) of the inductive base. The proof of cases (D) and (E) in the inductive base is analogous.

For the **inductive step**, the **intuition** is that each activation $\boldsymbol{y}_i^{(l)}$ is a linear combination of vector parameters, with different sets of value matrices acting on those:

$$\boldsymbol{y}_i^{(l)} = \sum_{\boldsymbol{v}\in\mathcal{VO}} \sum_{\mathcal{S}\in\mathcal{P}} \lambda_{\boldsymbol{v},i,l,\mathcal{S}} \left(\prod_{S\in\mathcal{S}} S\right) \boldsymbol{v} \tag{58}$$

where the coefficients are determined by attention weights and the activations of MLP hidden units. Importantly, the attention weights and MLP activations turn out to be the same in the Limit Transformer as in the original transformer, provided we can prove that attention and MLPs are faithfully

simulated (which indeed follows from cases A and C of the inductive claim). Hence, the same decomposition is valid in the Limit Transformer:

$$\widehat{\boldsymbol{y}_i^{(l)}} = \sum_{\boldsymbol{v} \in \mathcal{VO}} \sum_{\mathcal{S} \in \mathcal{P}} \lambda_{\boldsymbol{v},i,l,\mathcal{S}} \left( \prod_{S \in \mathcal{S}} \widehat{S} \right) \widehat{\boldsymbol{v}} \tag{59}$$

with the same $\lambda_{\boldsymbol{v},i,l,\mathcal{S}}$ coefficients as in the original transformer. Then, case (A) of the inductive claim follows intuitively by the calculation:

$$\boldsymbol{w}^T \boldsymbol{y}_i^{(l)} = \sum_{\boldsymbol{v} \in \mathcal{VO}} \sum_{\mathcal{S} \in \mathcal{P}} \lambda_{\boldsymbol{v},i,l,\mathcal{S}} \boldsymbol{w}^T \left( \prod_{S \in \mathcal{S}} S \right) \boldsymbol{v}$$

$$= \sum_{\boldsymbol{v} \in \mathcal{VO}} \sum_{\mathcal{S} \in \mathcal{P}} \lambda_{\boldsymbol{v},i,l,\mathcal{S}} \widehat{\boldsymbol{w}}^T \left( \prod_{S \in \mathcal{S}} \widehat{S} \right) \widehat{\boldsymbol{v}}$$

$$= \widehat{\boldsymbol{w}}^T \sum_{\boldsymbol{v} \in \mathcal{VO}} \sum_{\mathcal{S} \in \mathcal{P}} \lambda_{\boldsymbol{v},i,l,\mathcal{S}} \left( \prod_{S \in \mathcal{S}} \widehat{S} \right) \widehat{\boldsymbol{v}}$$

$$= \widehat{\boldsymbol{w}}^T \widehat{\boldsymbol{y}_i^{(l)}}$$

which is warranted provided that, when $\boldsymbol{v} \in \mathcal{VO}$ and $\boldsymbol{w} \in \mathcal{VI}$, we have that $\widehat{\boldsymbol{w}}^T \left( \prod_{S \in \mathcal{S}} \widehat{S} \right) \widehat{\boldsymbol{v}}$ equals $\boldsymbol{w}^T \left( \prod_{S \in \mathcal{S}} S \right) \boldsymbol{v}$ – this is ensured because of the way the vector parameters $\widehat{\boldsymbol{v}}$ and the value matrices $\widehat{\boldsymbol{V}_{l,h}}$ are defined. The same idea establishes cases (D–E). A similar, though somewhat more complex (due to the *bilinear* nature of attention) calculation establishes cases (B–C). Formalizing this reasoning essentially amounts to inductively proving cases (A–E); it will not be necessary to keep track of an explicit decomposition using $\lambda_{...}$ coefficients; rather, one can mechanically verify these conditions inductively by plugging in definitions and applying the inductive hypothesis.

**Formally proving the inductive step** consists in mechanically expanding definitions and applying the inductive hypothesis. First, (C) applied to layer $l - 1$ entails that the attention logits for attention heads operating in layer $l$ match those of the original transformer. We start by establishing the inductive step for the pre-MLP activations $\boldsymbol{Y}_i^{(l)}$. Starting from:

$$\boldsymbol{Y}_i^{(l)} = \boldsymbol{y}_i^{(l-1)} + \sum_{h=1}^{H} \sum_{j=1}^{i} \tilde{a}_{i,j}^{(l,h)} \boldsymbol{V}_{l,h} \boldsymbol{y}_j^{(l-1)}$$

$$\widehat{\boldsymbol{Y}_i^{(l)}} = \widehat{\boldsymbol{y}_i^{(l-1)}} + \sum_{h=1}^{H} \sum_{j=1}^{i} \tilde{a}_{i,j}^{(l,h)} \widehat{\boldsymbol{V}_{l,h}} \widehat{\boldsymbol{y}_j^{(l-1)}} \tag{60}$$

we show the inductive step first for (A) in the case of the pre-MLP activation $\boldsymbol{Y}_i^{(l)}$. For $\mathcal{S}$ satisfying

$$\forall l' \in \{1, \dots, L\} : [(\boldsymbol{V}_{l',h'} \in \mathcal{S}) \Rightarrow l' > l] \tag{61}$$

we consider

$$\Gamma_{\mathcal{S},\boldsymbol{w}} \left( \widehat{\boldsymbol{Y}_i^{(l)}} \right) = \Gamma_{\mathcal{S},\boldsymbol{w}} \left( \widehat{\boldsymbol{y}_i^{(l-1)}} \right) + \sum_{h=1}^{H} \sum_{j=1}^{i} \tilde{a}_{i,j}^{(l,h)} \Gamma_{\mathcal{S},\boldsymbol{w}} \left( \widehat{\boldsymbol{V}_{l,h} \boldsymbol{y}_j^{(l-1)}} \right)$$

$$= \Gamma_{\mathcal{S},\boldsymbol{w}} \left( \widehat{\boldsymbol{y}_i^{(l-1)}} \right) + \sum_{h=1}^{H} \sum_{j=1}^{i} \tilde{a}_{i,j}^{(l,h)} \Gamma_{\mathcal{S} \cup \{\boldsymbol{V}_{l,h}\},\boldsymbol{w}} \left( \widehat{\boldsymbol{y}_j^{(l-1)}} \right)$$

where $\tilde{a}_{i,j}$ denotes attention weights. The claim here now follows from the inductive hypothesis for (A):

$$= \boldsymbol{w}^T \left( \prod_{S \in \mathcal{S}} S \right) \boldsymbol{y}_i^{(l-1)} + \sum_{h=1}^{H} \sum_{j=1}^{i} \tilde{a}_{i,j}^{(l,h)} \boldsymbol{w}^T \left( \prod_{S \in \mathcal{S}} S \right) \boldsymbol{V}_{l,h} \boldsymbol{y}_j^{(l-1)}$$

$$= \boldsymbol{w}^T \left( \prod_{S \in \mathcal{S}} S \right) \boldsymbol{Y}_i^{(l)}$$

where the last step used (60). Next, we consider (B) and (C). First, for (B), assuming $\mathcal{T}_1 \cup \mathcal{T}_2 \neq \emptyset$, we first find using (60) and the linearity of $\Lambda_{\cdots}$ and $\Omega_{\cdots}$ (portions changed marked in blue):

$$
\sum_{\boldsymbol{v},\boldsymbol{w},\mathcal{T}_1,\mathcal{T}_2} \Lambda_{\mathcal{S}_1,\mathcal{T}_1,\mathcal{T}_2,k,h,\boldsymbol{v},\boldsymbol{w}}(\widehat{\boldsymbol{Y}_i^{(l)}}) \Omega_{\mathcal{S}_2,\mathcal{T}_1,\mathcal{T}_2,k,h,\boldsymbol{v},\boldsymbol{w}}(\widehat{\boldsymbol{Y}_j^{(l)}})
$$

$$
= \sum_{\boldsymbol{v},\boldsymbol{w},\mathcal{T}_1,\mathcal{T}_2} \Lambda_{\mathcal{S}_1,\mathcal{T}_1,\mathcal{T}_2,k,h,\boldsymbol{v},\boldsymbol{w}} \left( \widehat{\boldsymbol{y}_i^{(l-1)}} + \sum_{h'=1}^{H} \sum_{w=1}^{i} \tilde{a}_{iw}^{(l,h')} \widehat{\boldsymbol{V}_{l,h'}} \widehat{\boldsymbol{y}_w^{(l-1)}} \right)
$$

$$
\cdot \, \Omega_{\mathcal{S}_2,\mathcal{T}_1,\mathcal{T}_2,k,h,\boldsymbol{v},\boldsymbol{w}} \left( \widehat{\boldsymbol{y}_j^{(l-1)}} + \sum_{h''=1}^{H} \sum_{w'=1}^{j} \tilde{a}_{jw'}^{(l,h'')} \widehat{\boldsymbol{V}_{l,h''}} \widehat{\boldsymbol{y}_{w'}^{(l-1)}} \right)
$$

$$
= \sum_{\boldsymbol{v},\boldsymbol{w},\mathcal{T}_1,\mathcal{T}_2} \Lambda_{\mathcal{S}_1,\mathcal{T}_1,\mathcal{T}_2,k,h,\boldsymbol{v},\boldsymbol{w}} \left( \widehat{\boldsymbol{y}_i^{(l-1)}} \right) \Omega_{\mathcal{S}_2,\mathcal{T}_1,\mathcal{T}_2,k,h,\boldsymbol{v},\boldsymbol{w}} \left( \widehat{\boldsymbol{y}_j^{(l-1)}} \right)
$$

$$
+ \sum_{\boldsymbol{v},\boldsymbol{w},\mathcal{T}_1,\mathcal{T}_2} \Lambda_{\mathcal{S}_1,\mathcal{T}_1,\mathcal{T}_2,k,h,\boldsymbol{v},\boldsymbol{w}} \left( \widehat{\boldsymbol{y}_i^{(l-1)}} \right) \sum_{h'=1}^{H} \sum_{w=1}^{j} \tilde{a}_{jw}^{(l,h')} \Omega_{\mathcal{S}_2,\mathcal{T}_1,\mathcal{T}_2,l,h,\boldsymbol{v},\boldsymbol{w}} \left( \widehat{\boldsymbol{V}_{l,h'}} \widehat{\boldsymbol{y}_w^{(l-1)}} \right)
$$

$$
+ \sum_{\boldsymbol{v},\boldsymbol{w},\mathcal{T}_1,\mathcal{T}_2} \sum_{h''=1}^{H} \sum_{w=1}^{i} \tilde{a}_{iw}^{(l,h'')} \Lambda_{\mathcal{S}_1,\mathcal{T}_1,\mathcal{T}_2,l,h,\boldsymbol{v},\boldsymbol{w}} \left( \widehat{\boldsymbol{V}_{l,h''}} \widehat{\boldsymbol{y}_w^{(l-1)}} \right) \Omega_{\mathcal{S}_2,\mathcal{T}_1,\mathcal{T}_2,k,h,\boldsymbol{v},\boldsymbol{w}} \left( \widehat{\boldsymbol{y}_j^{(l-1)}} \right)
$$

$$
+ \sum_{\boldsymbol{v},\boldsymbol{w},\mathcal{T}_1,\mathcal{T}_2} \sum_{h'=1}^{H} \sum_{w=1}^{i} \tilde{a}_{iw}^{(l,h')} \Lambda_{\mathcal{S}_1,\mathcal{T}_1,\mathcal{T}_2,k,h,\boldsymbol{v},\boldsymbol{w}} \left( \widehat{\boldsymbol{V}_{l,h'}} \widehat{\boldsymbol{y}_w^{(l-1)}} \right)
$$

$$
\cdot \sum_{h''=1}^{H} \sum_{w'=1}^{j} \tilde{a}_{jw'}^{(l,h'')} \Omega_{\mathcal{S}_2,\mathcal{T}_1,\mathcal{T}_2,k,h,\boldsymbol{v},\boldsymbol{w}} \left( \widehat{\boldsymbol{V}_{l,h''}} \widehat{\boldsymbol{y}_{w'}^{(l-1)}} \right)
$$

Now the definition of $\widehat{\boldsymbol{V}}_{l,h}$ allows us to rewrite this as:

$$
= \sum_{\boldsymbol{v},\boldsymbol{w},\mathcal{T}_1 \geq_l \mathcal{S}_1,\mathcal{T}_2 \geq_l \mathcal{S}_2} \Lambda_{\mathcal{S}_1,\mathcal{T}_1,\mathcal{T}_2,l,h,\boldsymbol{v},\boldsymbol{w}} \left( \widehat{\boldsymbol{y}_i^{(l-1)}} \right) \Omega_{\mathcal{S}_2,\mathcal{T}_1,\mathcal{T}_2,l,h,\boldsymbol{v},\boldsymbol{w}} \left( \widehat{\boldsymbol{y}_j^{(l-1)}} \right)
$$

$$
+ \sum_{\boldsymbol{v},\boldsymbol{w},\mathcal{T}_1 \geq_l \mathcal{S}_1,\mathcal{T}_2 \geq_l \mathcal{S}_2} \Lambda_{\mathcal{S}_1,\mathcal{T}_1,\mathcal{T}_2,l,h,\boldsymbol{v},\boldsymbol{w}} \left( \widehat{\boldsymbol{y}_i^{(l-1)}} \right) \sum_{h'=1}^{H} \sum_{w=1}^{j} \tilde{a}_{jw}^{(l,h')} \Omega_{\mathcal{S}_2 \cup \{\boldsymbol{V}_{l,h'}\},\mathcal{T}_1,\mathcal{T}_2,l,h,\boldsymbol{v},\boldsymbol{w}} \left( \widehat{\boldsymbol{y}_w^{(l-1)}} \right)
$$

$$
+ \sum_{\boldsymbol{v},\boldsymbol{w},\mathcal{T}_1 \geq_l \mathcal{S}_1,\mathcal{T}_2 \geq_l \mathcal{S}_2} \sum_{h'=1}^{H} \sum_{w=1}^{i} \tilde{a}_{iw}^{(l,h')} \Lambda_{\mathcal{S}_1 \cup \{\boldsymbol{V}_{l,h'}\},\mathcal{T}_1,\mathcal{T}_2,l,h,\boldsymbol{v},\boldsymbol{w}} \left( \widehat{\boldsymbol{y}_w^{(l-1)}} \right) \Omega_{\mathcal{S}_2,\mathcal{T}_1,\mathcal{T}_2,l,h,\boldsymbol{v},\boldsymbol{w}} \left( \widehat{\boldsymbol{y}_j^{(l-1)}} \right)
$$

$$
+ \sum_{\boldsymbol{v},\boldsymbol{w},\mathcal{T}_1 \geq_l \mathcal{S}_1,\mathcal{T}_2 \geq_l \mathcal{S}_2} \sum_{h'=1}^{H} \sum_{j=1}^{i} \tilde{a}_{i,j}^{(l,h)} \Lambda_{\mathcal{S}_1 \cup \{\boldsymbol{V}_{l,h'}\},\mathcal{T}_1,\mathcal{T}_2,l,h',\boldsymbol{v},\boldsymbol{w}} \left( \widehat{\boldsymbol{y}_j^{(l-1)}} \right)
$$

$$
\cdot \sum_{h''=1}^{H} \sum_{w'=1}^{j} \tilde{a}_{jw'}^{(l,h'')} \Omega_{\mathcal{S}_2 \cup \{\boldsymbol{V}_{l,h''}\},\mathcal{T}_1,\mathcal{T}_2,l,h,\boldsymbol{v},\boldsymbol{w}} \left( \widehat{\boldsymbol{y}_{w'}^{(l-1)}} \right)
$$

In order to directly apply the inductive hypothesis, we rearrange the summations:

$$
= \sum_{\boldsymbol{v},\boldsymbol{w},\mathcal{T}_1 \geq_l \mathcal{S}_1, \mathcal{T}_2 \geq_l \mathcal{S}_2} \Lambda_{\mathcal{S}_1,\mathcal{T}_1,\mathcal{T}_2,l,h,\boldsymbol{v},\boldsymbol{w}} \left( \widehat{\boldsymbol{y}_i^{(l-1)}} \right) \Omega_{\mathcal{S}_2,\mathcal{T}_1,\mathcal{T}_2,l,h,\boldsymbol{v},\boldsymbol{w}} \left( \widehat{\boldsymbol{y}_j^{(l-1)}} \right)
$$

$$
+ \sum_{h'=1}^{H} \sum_{w=1}^{j} \tilde{a}_{jw}^{(l,h')} \sum_{\boldsymbol{v},\boldsymbol{w},\mathcal{T}_1 \geq_l \mathcal{S}_1, \mathcal{T}_2 \geq_l \mathcal{S}_2} \Lambda_{\mathcal{S}_1,\mathcal{T}_1,\mathcal{T}_2,l,h,\boldsymbol{v},\boldsymbol{w}} \left( \widehat{\boldsymbol{y}_i^{(l-1)}} \right) \Omega_{\mathcal{S}_2 \cup \{\boldsymbol{V}_{l,h'}\},\mathcal{T}_1,\mathcal{T}_2,l,h,\boldsymbol{v},\boldsymbol{w}} \left( \widehat{\boldsymbol{y}_w^{(l-1)}} \right)
$$

$$
+ \sum_{h'=1}^{H} \sum_{w=1}^{i} \tilde{a}_{iw}^{(l,h')} \sum_{\boldsymbol{v},\boldsymbol{w},\mathcal{T}_1 \geq_l \mathcal{S}_1, \mathcal{T}_2 \geq_l \mathcal{S}_2} \Lambda_{\mathcal{S}_1 \cup \{\boldsymbol{V}_{l,h'}\},\mathcal{T}_1,\mathcal{T}_2,l,h,\boldsymbol{v},\boldsymbol{w}} \left( \widehat{\boldsymbol{y}_w^{(l-1)}} \right) \Omega_{\mathcal{S}_2,\mathcal{T}_1,\mathcal{T}_2,l,h,\boldsymbol{v},\boldsymbol{w}} \left( \widehat{\boldsymbol{y}_j^{(l-1)}} \right)
$$

$$
+ \sum_{h'=1}^{H} \sum_{w=1}^{i} \tilde{a}_{iw}^{(l,h')} \sum_{h''=1}^{H} \sum_{w'=1}^{j} \tilde{a}_{jw'}^{(l,h'')} \sum_{\boldsymbol{v},\boldsymbol{w},\mathcal{T}_1 \geq_l \mathcal{S}_1, \mathcal{T}_2 \geq_l \mathcal{S}_2} \Lambda_{\mathcal{S}_1 \cup \{\boldsymbol{V}_{l,h'}\},\mathcal{T}_1,\mathcal{T}_2,l,h,\boldsymbol{v},\boldsymbol{w}} \left( \widehat{\boldsymbol{y}_w^{(l-1)}} \right)
$$

$$
\cdot \Omega_{\mathcal{S}_2 \cup \{\boldsymbol{V}_{l,h''}\},\mathcal{T}_1,\mathcal{T}_2,l,h,\boldsymbol{v},\boldsymbol{w}} \left( \widehat{\boldsymbol{y}_{w'}^{(l-1)}} \right)
$$

Note that, as above, $\mathcal{S}_1, \mathcal{S}_2$ are fixed and the sums run over $\mathcal{T}_1, \mathcal{T}_2$. We can rewrite the above as:

$$
= \sum_{\boldsymbol{v},\boldsymbol{w},\mathcal{T}_1 \geq_{l-1} \mathcal{S}_1, \mathcal{T}_2 \geq_{l-1} \mathcal{S}_2} \Lambda_{\mathcal{S}_1,\mathcal{T}_1,\mathcal{T}_2,l,h,\boldsymbol{v},\boldsymbol{w}} \left( \widehat{\boldsymbol{y}_i^{(l-1)}} \right) \Omega_{\mathcal{S}_2,\mathcal{T}_1,\mathcal{T}_2,l,h,\boldsymbol{v},\boldsymbol{w}} \left( \widehat{\boldsymbol{y}_j^{(l-1)}} \right)
$$

$$
+ \sum_{h'=1}^{H} \sum_{w=1}^{j} \tilde{a}_{jw}^{(l,h')} \sum_{\boldsymbol{v},\boldsymbol{w},\mathcal{T}_1 \geq_{l-1} \mathcal{S}_1, \mathcal{T}_2 \geq_{l-1} \mathcal{S}_2 \cup \{\boldsymbol{V}_{l,h'}\}} \Lambda_{\mathcal{S}_1,\mathcal{T}_1,\mathcal{T}_2,l,h,\boldsymbol{v},\boldsymbol{w}} \left( \widehat{\boldsymbol{y}_i^{(l-1)}} \right)
$$

$$
\cdot \Omega_{\mathcal{S}_2 \cup \{\boldsymbol{V}_{l,h}\},\mathcal{T}_1,\mathcal{T}_2,l,h,\boldsymbol{v},\boldsymbol{w}} \left( \widehat{\boldsymbol{y}_w^{(l-1)}} \right)
$$

$$
+ \sum_{h'=1}^{H} \sum_{w=1}^{i} \tilde{a}_{i,j}^{(l,h)} \sum_{\boldsymbol{v},\boldsymbol{w},\mathcal{T}_1 \geq_{l-1} \mathcal{S}_1 \cup \{\boldsymbol{V}_{l,h'}\}, \mathcal{T}_2 \geq_{l-1} \mathcal{S}_2} \Lambda_{\mathcal{S}_1 \cup \{\boldsymbol{V}_{l,h'}\},\mathcal{T}_1,\mathcal{T}_2,l,h,\boldsymbol{v},\boldsymbol{w}} \left( \widehat{\boldsymbol{y}_j^{(l-1)}} \right)
$$

$$
\cdot \Omega_{\mathcal{S}_2,\mathcal{T}_1,\mathcal{T}_2,l,h,\boldsymbol{v},\boldsymbol{w}} \left( \widehat{\boldsymbol{y}_j^{(l-1)}} \right)
$$

$$
+ \sum_{h'=1}^{H} \sum_{w=1}^{i} \tilde{a}_{iw}^{(l,h')} \sum_{h''=1}^{H} \sum_{w'=1}^{j} \tilde{a}_{jw'}^{(l,h'')} \sum_{\substack{\boldsymbol{v},\boldsymbol{w},\mathcal{T}_1 \geq_{l-1} \mathcal{S}_1 \cup \{\boldsymbol{V}_{l,h''}\}, \\ \mathcal{T}_2 \geq_{l-1} \mathcal{S}_2 \cup \{\boldsymbol{V}_{l,h'}\}}} \Lambda_{\mathcal{S}_1 \cup \{\boldsymbol{V}_{l,h'}\},\mathcal{T}_1,\mathcal{T}_2,l,h,\boldsymbol{v},\boldsymbol{w}} \left( \widehat{\boldsymbol{y}_w^{(l-1)}} \right)
$$

$$
\cdot \Omega_{\mathcal{S}_2 \cup \{\boldsymbol{V}_{l,h''}\},\mathcal{T}_1,\mathcal{T}_2,l,h,\boldsymbol{v},\boldsymbol{w}} \left( \widehat{\boldsymbol{y}_{w'}^{(l-1)}} \right)
$$

We are now ready to apply the inductive hypothesis: Directly plugging the inductive hypothesis for (B) into the second through fourth terms gives us:

$$
= \sum_{\boldsymbol{v},\boldsymbol{w},\mathcal{T}_1 \geq_{l-1} \mathcal{S}_1, \mathcal{T}_2 \geq_{l-1} \mathcal{S}_2} \Lambda_{\mathcal{S}_1,\mathcal{T}_1,\mathcal{T}_2,l,h,\boldsymbol{v},\boldsymbol{w}} \left( \widehat{\boldsymbol{y}_i^{(l-1)}} \right) \Omega_{\mathcal{S}_2,\mathcal{T}_1,\mathcal{T}_2,l,h,\boldsymbol{v},\boldsymbol{w}} \left( \widehat{\boldsymbol{y}_j^{(l-1)}} \right)
$$

$$
+ \sum_{h'=1}^{H} \sum_{w=1}^{j} \tilde{a}_{jw}^{(l,h')} (\boldsymbol{y}_i^{(l-1)})^T \left( \prod_{S \in \mathcal{S}_1} S \right)^T \boldsymbol{K}_{k,h}^T \boldsymbol{Q}_{k,h} \left( \prod_{S \in \mathcal{S}_2} S \right)^T \boldsymbol{V}_{l,h} \boldsymbol{y}_w^{(l-1)}
$$

$$
+ \sum_{h'=1}^{H} \sum_{w=1}^{i} \tilde{a}_{iw}^{(l,h')} \boldsymbol{y}_j^{(l-1)} \boldsymbol{V}_{l,h'}^T \left( \prod_{S \in \mathcal{S}_1} S \right)^T \boldsymbol{K}_{l,h}^T \boldsymbol{Q}_{l,h} \left( \prod_{S \in \mathcal{S}_2} S \right) \boldsymbol{y}_j^{(l-1)}
$$

$$
+ \sum_{h'=1}^{H} \sum_{w=1}^{i} \tilde{a}_{iw}^{(l,h')} \sum_{h''=1}^{H} \sum_{w'=1}^{j} \tilde{a}_{jw'}^{(l,h'')} (\boldsymbol{y}_w^{(l-1)})^T \boldsymbol{V}_{l,h'}^T \left( \prod_{S \in \mathcal{S}_1} S \right)^T \boldsymbol{K}_{l,h}^T \boldsymbol{Q}_{l,h} \left( \prod_{S \in \mathcal{S}_2} S \right) \boldsymbol{V}_{l,h''} \boldsymbol{y}_{w'}^{(l-1)}
$$

We now distinguish two cases, for proving (B) and (C). The first one is that $\mathcal{S}_1 = \mathcal{S}_2 = \emptyset$. In this case, by case (C) of the inductive hypothesis:

$$= (\boldsymbol{y}_i^{(l-1)})^T \boldsymbol{K}_{k,h}^T \boldsymbol{Q}_{k,h} \boldsymbol{y}_j^{(l-1)} - \phi_{k,h}(i,j)$$

$$+ \sum_{h'=1}^{H} \sum_{w=1}^{j} \tilde{a}_{jw}^{(l,h')} (\boldsymbol{y}_i^{(l-1)})^T \boldsymbol{K}_{k,h}^T \boldsymbol{Q}_{k,h} \boldsymbol{V}_{l,h} \boldsymbol{y}_w^{(l-1)}$$

$$+ \sum_{h'=1}^{H} \sum_{w=1}^{i} \tilde{a}_{iw}^{(l,h')} \boldsymbol{y}_j^{(l-1)} \boldsymbol{V}_{l,h'}^T \boldsymbol{K}_{l,h}^T \boldsymbol{Q}_{l,h} \boldsymbol{y}_j^{(l-1)}$$

$$+ \sum_{h'=1}^{H} \sum_{w=1}^{i} \tilde{a}_{iw}^{(l,h')} \sum_{h''=1}^{H} \sum_{w'=1}^{j} \tilde{a}_{jw'}^{(l,h'')} (\boldsymbol{y}_w^{(l-1)})^T \boldsymbol{V}_{l,h'}^T \boldsymbol{K}_{l,h}^T \boldsymbol{Q}_{l,h} \boldsymbol{V}_{l,h''} \boldsymbol{y}_{w'}^{(l-1)}$$

Now, applying (60) again, we rearrange to sums to obtain the conclusion

$$= (\boldsymbol{Y}_i^{(l)})^T \boldsymbol{K}_{k,h}^T \boldsymbol{Q}_{k,h} \boldsymbol{Y}_j^{(l)} - \phi_{k,h}(i,j) \tag{62}$$

proving (upon rearranging) the inductive step for (C) in the case of $\boldsymbol{Y}_i^{(l)}$. In the second case, $\mathcal{S}_1 \cup \mathcal{S}_2 \neq \emptyset$; here, we use case (B) of the inductive hypothesis to instead rewrite as

$$= (\boldsymbol{y}_i^{(l-1)})^T \left( \prod_{S \in \mathcal{S}_1} S \right)^T \boldsymbol{K}_{k,h}^T \boldsymbol{Q}_{k,h} \left( \prod_{S \in \mathcal{S}_2} S \right) (\boldsymbol{y}_j^{(l-1)})$$

$$+ \sum_{h'=1}^{H} \sum_{w=1}^{j} \tilde{a}_{jw}^{(l,h')} (\boldsymbol{y}_i^{(l-1)})^T \left( \prod_{S \in \mathcal{S}_1} S \right)^T \boldsymbol{K}_{k,h}^T \boldsymbol{Q}_{k,h} \left( \prod_{S \in \mathcal{S}_2} S \right) \boldsymbol{V}_{l,h'} \boldsymbol{y}_w^{(l-1)}$$

$$+ \sum_{h'=1}^{H} \sum_{w=1}^{i} \tilde{a}_{iw}^{(l,h')} \boldsymbol{y}_j^{(l-1)} \boldsymbol{V}_{l,h'}^T \left( \prod_{S \in \mathcal{S}_1} S \right)^T \boldsymbol{K}_{l,h}^T \boldsymbol{Q}_{l,h} \left( \prod_{S \in \mathcal{S}_2} S \right) \boldsymbol{y}_j^{(l-1)}$$

$$+ \sum_{h'=1}^{H} \sum_{w=1}^{i} \tilde{a}_{iw}^{(l,h')} \sum_{h''=1}^{H} \sum_{w'=1}^{j} \tilde{a}_{jw'}^{(l,h'')} (\boldsymbol{y}_w^{(l-1)})^T \boldsymbol{V}_{l,h'}^T \left( \prod_{S \in \mathcal{S}_1} S \right)^T \boldsymbol{K}_{l,h}^T \boldsymbol{Q}_{l,h} \left( \prod_{S \in \mathcal{S}_2} S \right) \boldsymbol{V}_{l,h''} \boldsymbol{y}_{w'}^{(l-1)}$$

Now, applying (60) again, we rearrange to sums to obtain the conclusion

$$= (\boldsymbol{Y}_i^{(l)})^T \left( \prod_{S \in \mathcal{S}_1} S \right)^T \boldsymbol{K}_{k,h}^T \boldsymbol{Q}_{k,h} \left( \prod_{S \in \mathcal{S}_2} S \right) \boldsymbol{Y}_j^{(l)} \tag{63}$$

This proves the inductive step for (B) in the case of $\boldsymbol{Y}_i^{(l)}$. We next address the inductive step for (D) in the case of the pre-MLP activation:

$$\sum_{\boldsymbol{v},\boldsymbol{w},\mathcal{T}_1 \geq_l \mathcal{S}_1, \mathcal{T}_2 \geq_l \mathcal{S}_2} \Lambda_{\mathcal{S}_1,\mathcal{T}_1,\mathcal{T}_2,k,h,\boldsymbol{v},\boldsymbol{w}}(\widehat{\boldsymbol{B}_m}) \cdot \Omega_{\mathcal{S}_2,\mathcal{T}_1,\mathcal{T}_2,k,h,\boldsymbol{v},\boldsymbol{w}}(\widehat{\boldsymbol{Y}_j^{(l)}})$$

$$= (\boldsymbol{B}_m)^T \left( \prod_{S \in \mathcal{S}_1} S \right)^T \boldsymbol{K}_{k,h}^T \boldsymbol{Q}_{k,h} \left( \prod_{S \in \mathcal{S}_2} S \right) \boldsymbol{Y}_j^{(l)}$$

By unfolding $\widehat{\boldsymbol{Y}_j^{(l)}}$ using (60) and using the linearity of $\Omega_{\mathcal{S}_2,\mathcal{T}_1,\mathcal{T}_2,k,h,\boldsymbol{v},\boldsymbol{w}}$, the claim follows directly from the inductive hypothesis for (D). The same reasoning applies to (E). Overall, we have proven the inductive step (A–E) for the pre-MLP activations $\boldsymbol{Y}_i^{(l)}$.

We now need to show that the inductive step also holds for the post-MLP activations. Recall that the MLP acts as

$$\boldsymbol{y}_i^{(l)} = \boldsymbol{Y}_i^{(l)} + \sum_{s=1}^{d_{MLP}} (\boldsymbol{B}_l)_{\cdot,s} \cdot \psi_{l,s} \left( (\boldsymbol{A}_l)_{s,\cdot} (\boldsymbol{Y}_i^{(l)}) + (\boldsymbol{b}_l)_s \right)$$

$$\widehat{\boldsymbol{y}_i^{(l)}} = \widehat{\boldsymbol{Y}_i^{(l)}} + \sum_{s=1}^{d_{MLP}} (\widehat{\boldsymbol{B}}_l)_{s,\cdot} \cdot \psi_{l,s} \left( (\widehat{\boldsymbol{A}}_l)_{s,\cdot} (\widehat{\boldsymbol{Y}_i^{(l)}}) + (\widehat{\boldsymbol{b}}_l)_s \right) \tag{64}$$

The proof proceeds by expanding this equation and reducing the claim to the already-proven inductive step for pre-MLP activations (for handling the direct contribution from the pre-MLP activation), and cases (D) and (E) (for handling the contributions of the MLP units). First, we note that, for each $l, s$, by the case (A) of the inductive hypothesis and by the definition of $\widehat{\boldsymbol{b}}_l$,

$$\psi_{l,s}\left((\boldsymbol{A}_l)_{s,\cdot}(\boldsymbol{Y}_i^{(l)}) + (\boldsymbol{b}_l)_s\right) = \psi_{l,s}\left((\widehat{\boldsymbol{A}}_l)_{s,\cdot}(\widehat{\boldsymbol{Y}_i^{(l)}}) + (\widehat{\boldsymbol{b}}_l)_s\right) \tag{65}$$

We will abbreviate this number as $\Xi_{l,s,i} \in \mathbb{R}$. We now prove the case (A) of the inductive step for the post-MLP activation:

$$
\begin{aligned}
\Gamma_{\mathcal{S},\boldsymbol{w}}(\widehat{\boldsymbol{y}_i^{(l)}}) &= \Gamma_{\mathcal{S},\boldsymbol{w}}\left(\widehat{\boldsymbol{Y}_i^{(l)}} + \sum_{s=1}^{d_{MLP}} (\widehat{\boldsymbol{B}}_l)_{s,\cdot} \cdot \Xi_{l,s,i}\right)\\
&= \Gamma_{\mathcal{S},\boldsymbol{w}}(\widehat{\boldsymbol{Y}_i^{(l)}}) + \sum_{s=1}^{d_{MLP}} \Gamma_{\mathcal{S},\boldsymbol{w}}((\widehat{\boldsymbol{B}}_l)_{s,\cdot}) \cdot \Xi_{l,s,i}\\
&= \boldsymbol{w}^T \left(\prod_{S\in\mathcal{S}} S\right) \boldsymbol{Y}_i^{(l)} + \sum_{s=1}^{d_{MLP}} \boldsymbol{w}^T \left(\prod_{S\in\mathcal{S}} S\right) (\boldsymbol{B}_l)_{\cdot,s} \cdot \Xi_{l,s,i}\\
&= \boldsymbol{w}^T \left(\prod_{S\in\mathcal{S}} S\right) \boldsymbol{y}_i^{(l)}
\end{aligned}
$$

To prove cases (B) and (C) for the post-MLP activations $\boldsymbol{y}_i^{(l)}$, we now consider

$$
\begin{aligned}
&\sum_{\boldsymbol{v},\boldsymbol{w},\mathcal{T}_1 \geq_l \mathcal{S}_1, \mathcal{T}_2 \geq_l \mathcal{S}_2} \Lambda_{\mathcal{S}_1,\mathcal{T}_1,\mathcal{T}_2,k,h,\boldsymbol{v},\boldsymbol{w}}\left(\widehat{\boldsymbol{y}_i^{(l)}}\right) \cdot \Omega_{\mathcal{S}_2,\mathcal{T}_1,\mathcal{T}_2,k,h,\boldsymbol{v},\boldsymbol{w}}\left(\widehat{\boldsymbol{y}_j^{(l)}}\right)\\
&= \sum_{\boldsymbol{v},\boldsymbol{w},\mathcal{T}_1 \geq_l \mathcal{S}_1, \mathcal{T}_2 \geq_l \mathcal{S}_2} \Lambda_{\mathcal{S}_1,\mathcal{T}_1,\mathcal{T}_2,k,h,\boldsymbol{v},\boldsymbol{w}}\left(\widehat{\boldsymbol{Y}_i^{(l)}} + \sum_{s=1}^{d_{MLP}} \Xi_{l,s,i} \cdot (\widehat{\boldsymbol{B}}_l)_{\cdot,s}\right)\\
&\qquad\qquad \cdot \Omega_{\mathcal{S}_2,\mathcal{T}_1,\mathcal{T}_2,k,h,\boldsymbol{v},\boldsymbol{w}}\left(\widehat{\boldsymbol{Y}_j^{(l)}} + \sum_{t=1}^{d_{MLP}} \Xi_{l,t,j} \cdot (\widehat{\boldsymbol{B}}_l)_{\cdot,t}\right)\\
&= \sum_{\boldsymbol{v},\boldsymbol{w},\mathcal{T}_1 \geq_l \mathcal{S}_1, \mathcal{T}_2 \geq_l \mathcal{S}_2} \Lambda_{\mathcal{S}_1,\mathcal{T}_1,\mathcal{T}_2,k,h,\boldsymbol{v},\boldsymbol{w}}\left(\widehat{\boldsymbol{Y}_i^{(l)}}\right) \Omega_{\mathcal{S}_2,\mathcal{T}_1,\mathcal{T}_2,k,h,\boldsymbol{v},\boldsymbol{w}}\left(\widehat{\boldsymbol{Y}_j^{(l)}}\right)\\
&+ \sum_{s=1}^{d_{MLP}} \Xi_{l,s,j} \cdot \sum_{\boldsymbol{v},\boldsymbol{w},\mathcal{T}_1 \geq_l \mathcal{S}_1, \mathcal{T}_2 \geq_l \mathcal{S}_2} \Lambda_{\mathcal{S}_1,\mathcal{T}_1,\mathcal{T}_2,k,h,\boldsymbol{v},\boldsymbol{w}}\left(\widehat{\boldsymbol{Y}_i^{(l)}}\right) \Omega_{\mathcal{S}_2,\mathcal{T}_1,\mathcal{T}_2,k,h,\boldsymbol{v},\boldsymbol{w}}\left((\widehat{\boldsymbol{B}}_l)_{\cdot,s}\right)\\
&+ \sum_{s=1}^{d_{MLP}} \Xi_{l,s,i} \cdot \sum_{\boldsymbol{v},\boldsymbol{w},\mathcal{T}_1 \geq_l \mathcal{S}_1, \mathcal{T}_2 \geq_l \mathcal{S}_2} \Lambda_{\mathcal{S}_1,\mathcal{T}_1,\mathcal{T}_2,k,h,\boldsymbol{v},\boldsymbol{w}}\left((\widehat{\boldsymbol{B}}_l)_{\cdot,s}\right) \Omega_{\mathcal{S}_2,\mathcal{T}_1,\mathcal{T}_2,k,h,\boldsymbol{v},\boldsymbol{w}}\left(\widehat{\boldsymbol{Y}_j^{(l)}}\right)\\
&+ \sum_{s=1}^{d_{MLP}} \Xi_{l,s,i} \cdot \sum_{t=1}^{d_{MLP}} \Xi_{l,t,j} \cdot \sum_{\boldsymbol{v},\boldsymbol{w},\mathcal{T}_1 \geq_l \mathcal{S}_1, \mathcal{T}_2 \geq_l \mathcal{S}_2} \Lambda_{\mathcal{S}_1,\mathcal{T}_1,\mathcal{T}_2,k,h,\boldsymbol{v},\boldsymbol{w}}\left((\widehat{\boldsymbol{B}}_l)_{\cdot,s}\right)\\
&\qquad\qquad \cdot \Omega_{\mathcal{S}_2,\mathcal{T}_1,\mathcal{T}_2,k,h,\boldsymbol{v},\boldsymbol{w}}\left((\widehat{\boldsymbol{B}}_l)_{\cdot,t}\right)
\end{aligned}
$$

We apply the inductive step for the pre-MLP activation in cases (D), and (E) to rewrite the second and third term, and apply the definition of $\widehat{B}_l$ to rewrite the fourth term:

$$
\begin{aligned}
= &\sum_{\boldsymbol{v},\boldsymbol{w},\mathcal{T}_1 \geq_l \mathcal{S}_1, \mathcal{T}_2 \geq_l \mathcal{S}_2} \Lambda_{\mathcal{S}_1,\mathcal{T}_1,\mathcal{T}_2,k,h,\boldsymbol{v},\boldsymbol{w}} \left( \widehat{\boldsymbol{Y}_i^{(l)}} \right) \Omega_{\mathcal{S}_2,\mathcal{T}_1,\mathcal{T}_2,k,h,\boldsymbol{v},\boldsymbol{w}} \left( \widehat{\boldsymbol{Y}_j^{(l)}} \right) \\
&+ \sum_{s=1}^{d_{MLP}} \Xi_{l,s,j} \cdot (\boldsymbol{Y}_i^{(l)})^T \left( \prod_{S \in \mathcal{S}_1} S \right)^T \boldsymbol{K}_{k,h}^T \boldsymbol{Q}_{k,h} \left( \prod_{S \in \mathcal{S}_2} S \right) (\boldsymbol{B}_l)_{s,\cdot} \\
&+ \sum_{s=1}^{d_{MLP}} \Xi_{l,s,i} \cdot (\boldsymbol{B}_l)_{\cdot,s}^T \left( \prod_{S \in \mathcal{S}_1} S \right)^T \boldsymbol{K}_{k,h}^T \boldsymbol{Q}_{k,h} \left( \prod_{S \in \mathcal{S}_2} S \right) \boldsymbol{Y}_j^{(l)} \\
&+ \sum_{s=1}^{d_{MLP}} \Xi_{l,s,i} \cdot \sum_{t=1}^{d_{MLP}} \Xi_{l,t,j} \cdot (\boldsymbol{B}_l)_{\cdot,s}^T (\prod_{S \in \mathcal{S}_1} S)^T \boldsymbol{K}_{k,h}^T \boldsymbol{Q}_{k,h} (\prod_{S \in \mathcal{S}_2} S)(\boldsymbol{B}_l)_{\cdot,t}
\end{aligned}
$$

In the case where $\mathcal{S}_1 = \mathcal{S}_2 = \emptyset$, we obtain using case (C) of the inductive hypothesis for the pre-MLP activation:

$$
\begin{aligned}
= &(\boldsymbol{Y}_i^{(l)})^T \boldsymbol{K}_{k,h}^T \boldsymbol{Q}_{k,h} \boldsymbol{Y}_j^{(l)} - \phi_{k,h}(i,j) \\
&+ \sum_{s=1}^{d_{MLP}} \Xi_{l,s,j} \cdot (\boldsymbol{Y}_i^{(l)})^T \boldsymbol{K}_{k,h}^T \boldsymbol{Q}_{k,h} (\boldsymbol{B}_l)_{s,\cdot} \\
&+ \sum_{s=1}^{d_{MLP}} \Xi_{l,s,i} \cdot (\boldsymbol{B}_l)_{\cdot,s}^T \boldsymbol{K}_{k,h}^T \boldsymbol{Q}_{k,h} \boldsymbol{Y}_j^{(l)} \\
&+ \sum_{s=1}^{d_{MLP}} \Xi_{l,s,i} \cdot \sum_{t=1}^{d_{MLP}} \Xi_{l,t,j} \cdot (\boldsymbol{B}_l)_{\cdot,s}^T \boldsymbol{K}_{k,h}^T \boldsymbol{Q}_{k,h} (\boldsymbol{B}_l)_{\cdot,t}
\end{aligned}
$$

Using (64) and the definition of $\Xi_{l,s,i}$, this rewrites to

$$
= (\boldsymbol{y}_i^{(l)})^T \boldsymbol{K}_{k,h}^T \boldsymbol{Q}_{k,h} \boldsymbol{y}_j^{(l)} - \phi_{k,h}(i,j)
$$

from which case (B) of the inductive hypothesis follows by rearranging. If instead $\mathcal{S}_1 \cup \mathcal{S}_2 \neq \emptyset$, the same reasoning leads to case (C) of the inductive hypothesis. Analogous reasoning establishes cases (D) and (E) for the post-MLP activations. Overall, we have established the inductive step for cases (A–E) for the post-MLP activations. □

# G  ADDITIONAL SUPPORTING RESULTS

## G.1  REGULARIZER AT INITIALIZATION

Here, we provide evidence that the additional regularizer (8) is bounded independently of $N$ under plausible initializations of parameters. Recall

$$
(8) = \sum_{l=1}^{L} \sum_{h=1}^{H} \sum_{j=1}^{N(T)} \left| \boldsymbol{p}_1^T \boldsymbol{K}_{l,h}^T \boldsymbol{Q}_{l,h} \boldsymbol{p}_j \right|^2 \tag{66}
$$

Intuitively, and as formalized in Proposition 54, when independently initializing the positional encodings $\boldsymbol{p}_i$, their inner products as mediated through $\boldsymbol{K}_{l,h}^T \boldsymbol{Q}_{l,h}$ will tend to be small. As long as the width grows linearly with $N$, the aggregate value of (8) will tend to be bounded independently of $N$. Note that (8) only includes products involving position 1, which due to translation invariance for $T \in \Theta_n$ places a bound on all products. As standard training does not enforce translation invariance of the products $\boldsymbol{p}_i^T \boldsymbol{K}_{l,h}^T \boldsymbol{Q}_{l,h} \boldsymbol{p}_j$, one may also be interested in a variant that takes all pairs of

positions into account, to the extent that they can enter causal attention:

$$\frac{1}{N(T)} \sum_{l=1}^{L} \sum_{h=1}^{H} \sum_{j=1}^{N(T)} \sum_{i=1}^{j} \left| \boldsymbol{p}_i^T \boldsymbol{K}_{l,h}^T \boldsymbol{Q}_{l,h} \boldsymbol{p}_j \right|^2 \tag{67}$$

Here, the same conclusion holds. We describe it formally, at the example of the second variant, in Proposition 54.

**Proposition 54.** *Assume $d = \Theta(N)$. Assume the entries of each $\boldsymbol{p}_1, \ldots, \boldsymbol{p}_N \in \mathbb{R}^d$ and $\boldsymbol{K}_{l,h}^T \boldsymbol{Q}_{l,h} \in \mathbb{R}^{d \times d}$ ($l = 1, \ldots, L$; $h = 1, \ldots, H$) are initialized i.i.d. from $\mathcal{N}(0, \frac{1}{d})$. The number of layers $L$ and heads $H$ are constant with respect to $N$. Then*

$$\mathbb{E}\left( \frac{1}{N} \sum_{l=1}^{L} \sum_{h=1}^{H} \sum_{1 \leq i \leq j \leq N} \left| \boldsymbol{p}_i^T \boldsymbol{K}_{l,h}^T \boldsymbol{Q}_{l,h} \boldsymbol{p}_j \right|^2 \right) = O(1) \tag{68}$$

*Proof.* We begin by showing that the expectation of each term in the sum is $O(1/d)$ and hence the sum is bounded by a constant. There are two cases for the expectation of terms: (i) The first is $i \neq j$ when the vectors $\boldsymbol{p}_i$ and $\boldsymbol{p}_j$ are independent and the second is $i = j$ when $\boldsymbol{p}_i$ and $\boldsymbol{p}_j$ are dependent.

For this section, let $\boldsymbol{K}, \boldsymbol{Q}$ denote the matrices $\boldsymbol{K}_{l,h} \boldsymbol{Q}_{l,h}$ for any fixed $l, h$. Let $\boldsymbol{K}_{ij}$ and $\boldsymbol{Q}_{ij}$ denote the entry of the corresponding matrices $i$th column and $j$th row. Let $A = \boldsymbol{K}^T \boldsymbol{Q} \in \mathbb{R}^{d \times d}$. Note that the expectation of any entry of $A$,

$$\mathbb{E}[\boldsymbol{A}_{ij}] = \mathbb{E}[\boldsymbol{K}_i^T \boldsymbol{Q}_j] = \mathbb{E}[\sum_{k=1}^{d} \boldsymbol{K}_{i,k} \boldsymbol{Q}_{j,k}] = 0.$$

Further,

$$\mathbb{E}[\boldsymbol{A}_{ij}^2] = \mathbb{E}[(\sum_{k=1}^{d} \boldsymbol{K}_{i,k} \boldsymbol{Q}_{j,k})^2] = \mathbb{E}[\sum_{k=1}^{d} \boldsymbol{K}_{i,k}^2 \boldsymbol{Q}_{j,k}^2] + 2\mathbb{E}[\sum_{m=1}^{d-1} \sum_{n=m+1}^{d} \boldsymbol{K}_{i,m} \boldsymbol{Q}_{j,m} \boldsymbol{K}_{i,n} \boldsymbol{Q}_{j,n}]$$

$$= \mathbb{E}[\sum_{k=1}^{d} \boldsymbol{K}_{i,k}^2 \boldsymbol{Q}_{j,k}^2] = d\sigma^4 = \sigma^2$$

For products of two different entries $\boldsymbol{A}_{i,j} \boldsymbol{A}_{mn}$, note that $\mathbb{E}[\boldsymbol{A}_{i,j} \boldsymbol{A}_{m,n}] = \mathbb{E}[\sum_{u=1}^{d} \sum_{v=1}^{d} \boldsymbol{K}_{i,u} \boldsymbol{Q}_{j,u} \boldsymbol{K}_{m,v} \boldsymbol{Q}_{n,v}]$ which is 0 when $i \neq m$ or $j \neq n$.

For each term $|\boldsymbol{p}_i^T A \boldsymbol{p}_j|^2$, we have

$$\mathbb{E}[|\boldsymbol{p}_i^T A Q \boldsymbol{p}_j|^2] = \mathbb{E}[(\sum_{u=1}^{d} \sum_{v=1}^{d} \boldsymbol{p}_{i,u} \boldsymbol{A}_{u,v} \boldsymbol{p}_{j,v})^2]$$

$$= \mathbb{E}[\sum_{u=1}^{d} \sum_{v=1}^{d} (\boldsymbol{p}_{i,u} \boldsymbol{A}_{u,v} \boldsymbol{p}_{j,v})^2] + 2\mathbb{E}[\sum_{u,v \neq m,n} \boldsymbol{p}_{i,u} \boldsymbol{p}_{i,v} \boldsymbol{A}_{u,v} \boldsymbol{A}_{m,n} \boldsymbol{p}_{j,v} \boldsymbol{p}_{j,n}]$$

$$= \mathbb{E}[\sum_{u=1}^{d} \sum_{v=1}^{d} \boldsymbol{p}_{i,u}^2 \boldsymbol{A}_{u,v}^2 \boldsymbol{p}_{j,v}^2]$$

since all terms of the form $\mathbb{E}[\boldsymbol{p}_{i,u} \boldsymbol{p}_{i,v} \boldsymbol{A}_{u,v} \boldsymbol{A}_{m,n} \boldsymbol{p}_{j,v} \boldsymbol{p}_{j,n}]$ are 0 due to independence of $\boldsymbol{A}$ and $\boldsymbol{p}$.

For $i \neq j$, we have

$$\mathbb{E}[\sum_{u=1}^{d} \sum_{v=1}^{d} \boldsymbol{p}_{i,u}^2 \boldsymbol{A}_{u,v}^2 \boldsymbol{p}_{j,v}^2] = \sum_{u=1}^{d} \sum_{v=1}^{d} \mathbb{E}[\boldsymbol{p}_{i,u}^2] \mathbb{E}[\boldsymbol{A}_{u,v}^2] \mathbb{E}[\boldsymbol{p}_{j,v}^2] = d^2 \sigma^6 = \frac{1}{d}.$$

For $i = j$, we have

$$\mathbb{E}[\sum_{u=1}^{d}\sum_{v=1}^{d}\boldsymbol{p}_{i,u}^2\boldsymbol{A}_{u,v}^2\boldsymbol{p}_{i,v}^2] = \mathbb{E}[\sum_{u=1}^{d}\boldsymbol{p}_{i,u}^4\boldsymbol{A}_{u,v}^2] + \mathbb{E}[\sum_{u=1}^{d}\sum_{v\neq u}\boldsymbol{p}_{i,u}^2\boldsymbol{A}_{u,v}^2\boldsymbol{p}_{i,v}^2]$$

$$= d(3\sigma^6) + (d^2 - d)\sigma^6 < \frac{3}{d}.$$

Since $d = \Theta(N)$ and each term is less than $\frac{3}{d}$, we have that the sum in Eq.68 is $O(1)$.

$\square$

## G.2 EMPIRICAL LENGTH GENERALIZATION OF POSITIONAL FUNCTIONS

Here, we show empirically that, when directly fitting parameters so that a product $\boldsymbol{p}_i \boldsymbol{K}^T \boldsymbol{Q} \boldsymbol{p}_j$ reproduces some function $\phi(\cdot, \cdot)$ at smaller distances, it will length generalize when these are local or periodic, but under different conditions matching the role of local and periodic functions in our theory. Specifically, we show that they length-generalize well at large $d$ when they are LOCAL; whereas, when $d$ is smaller, length generalization works well when they are PERIODIC. Length generalization is poor on functions that are neither local nor periodic.

**Experimental Setup** We randomly initialize 200 position embeddings $\{\boldsymbol{p}_i \in \mathbb{R}^d : 1 \leq i < 201\}$, as well as query and key matrices, $\boldsymbol{Q}, \boldsymbol{K} \in \mathbb{R}^{d\times d}$. We experiment with $d = \{32, 256\}$. We optimize the mean square error (MSE) between $\boldsymbol{p}_i^T \boldsymbol{K}^T \boldsymbol{Q} \boldsymbol{p}_j$ and $\phi(\cdot, \cdot)$ on length of 50, and test on length $\{50, 100, 150\}$. Concretely, during training, we sample random offsets $o$ from [0, 150] and take the sequence of $\boldsymbol{p}_{1+o}, \ldots, \boldsymbol{p}_{50+o}$ to compute the loss. When testing on length $n$, we compute the average loss over all offsets in [0, 200-n]. We ignore the loss on those entries where $j < i$ to mimic causal masking. The $\phi(\cdot, \cdot)$ we use in experiments (except for the one combined from two $\phi(\cdot, \cdot)$) only takes two values, 0 when condition is false and $2\log 50$ when condition is true. We thus use different condition to describe different $\phi(\cdot, \cdot)$. For example, we use $\phi$ : j=i-c to denote the following function:

$$\phi(j, i) = \begin{cases} 2\log 50 & j = i - c \\ 0 & else \end{cases} \tag{69}$$

where $c$ is a constant number.

The embeddings and weight matrices are trained with Adam optimizer, using batch size of 64, learning rate of 1e-3, for 15k steps. Additionally, we add mean squared weights (i.e., squared Frobenius norm divided by number of elements) to the loss to mimic training regularizer, with coefficient of 0.01.

Results are in Figure 5 and 6. Note that in both figures, y axis is using logarithmic scale above 1.0 and stays linear scale below 1.0. In the last column of 6, "combined" denotes functions that combine two functions as follows: $\phi = \phi_1 + \phi_2$, where $\phi_1$ : j=i-c and $\phi_2$ : (i-j)=$c_2$ mod $c_1$.

## G.3 BOUND FOR ENCODINGS NORM IN TERMS OF FUNCTION COMPLEXITY

Recall that our regularizer includes a penalty (8) on attention dot products. Here, we discuss a conjecture:

**Conjecture 55.** *The term (8) can be removed from the regularizer while maintaining a (potentially weaker) length generalization guarantee for Limit Transformers.*

To provide a heuristic argument for this, assume that for each upper bound N on the input length, we have a configuration of positional encodings and the matrix $\boldsymbol{A}$, such that the following property holds: For any indices $N \geq j > i > 0$, let

$$\boldsymbol{p}_i^T \boldsymbol{A} \boldsymbol{p}_j = F(j - i) \tag{70}$$

where $F : \mathbb{N} \to \mathbb{R}$ is a function that maps to numbers representable in $p$-bit precision. The function $F$ and the precision $p$ are chosen globally, across the different N's. Boundedness of $\mathcal{R}(T_n)$ across

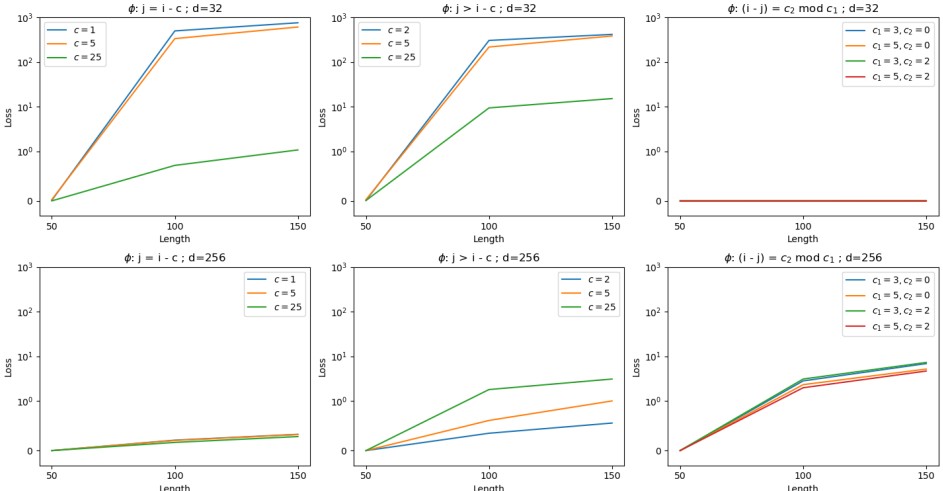

Figure 5: Appendix G.2: MSE loss in fitting (length $= 50$) and generalizing (higher lengths) functions $\phi(\cdot, \cdot)$ with products $\boldsymbol{p}_j^T \boldsymbol{K}^T \boldsymbol{Q} \boldsymbol{p}_i$. We show local functions testing if $j = i - c$ (left), if $j > i - c$ (center), periodic functions testing whether $i - j \equiv c_2 \pmod{c_1}$ (right). We show result at small (top, $d = 32$) and high (bottom, $d = 256$) dimensionality. Local functions length-generalize well when dimensionality is high (bottom left, bottom center); generalization is more successful with functions concentrated on few pairs (bottom left is nonzero at only one value of $j - i$; bottom center is nonzero at $c$ different values of $j - i$). Periodic functions length-generalize well when dimensionality is low (top right). The results match the distinct roles played by local and periodic functions in our theoretical constructions: Periodic functions are mediated by bounded-rank products (Lemma 48), local functions are mediated by the products $\boldsymbol{p}^T \boldsymbol{K}_{l,h}^T \boldsymbol{Q} \boldsymbol{p}$.

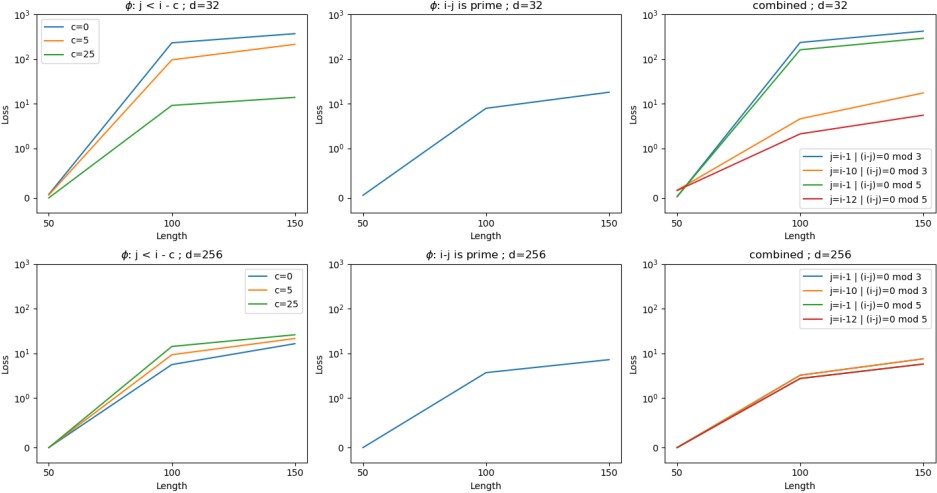

Figure 6: Appendix G.2: MSE loss in fitting (length $= 50$) and generalizing (higher lengths) functions $\phi(\cdot, \cdot)$ with products $\boldsymbol{p}_j^T \boldsymbol{K}^T \boldsymbol{Q} \boldsymbol{p}_i$. We show functions that are neither local nor periodic, which tests if $j < i - c$ (left), if $i - j$ is prime number (center), and a function created by adding a local and a periodic function (right). We show result at small (top, $d = 32$) and high (bottom, $d = 256$) dimensionality. Compared with results in Figure 5, we can see that such functions, neither local nor periodic, length-generalize poorly.

$n$ entails $\|\boldsymbol{p}_i\|_2, \|\boldsymbol{A}\| < C$, for $C$ a global constant. We also know that $\sup_{x \in \mathbb{N}} |F(x)| < \infty$. We conjecture that one can use these assumptions, and Lemma 56, to prove that F cannot be "too complicated". Specifically, we conjecture that $F$ will be **ultimately periodic**: when $x$ exceeds some threshold, $F(x + \Delta) = F(x)$ for some period $\Delta$. For, if $F$ is not ultimately periodic, we hope to construct a matrix $G$ whose nuclear norm can be made arbitrarily large, so large as to give a superconstant lower bound on $\|\boldsymbol{A}\|$ – which is a contradiction. First note that Lemma 56 even holds if $\boldsymbol{G} = \boldsymbol{Y}^T \boldsymbol{A} \boldsymbol{X}$ where $\boldsymbol{X}, \boldsymbol{Y}$ are two different matrices with $n$ unit-norm columns. That is, we can consider a matrix $\boldsymbol{G}_{ij} = \boldsymbol{p}_{x_i}^T \boldsymbol{A} \boldsymbol{p}_{y_j}$ where we conjecture that one can choose $\boldsymbol{x}_1, ..., \boldsymbol{x}_n$ and $\boldsymbol{y}_1, ..., \boldsymbol{y}_n$ to give an arbitrarily large lower bound on $\|\boldsymbol{A}\|$, under the assumption that $F$ is not ultimately periodic. Here, it is important that $F$ maps to fixed-precision output; otherwise, one could get functions that have irrational periods and thus are not periodic when restricted to $\mathbb{N}$.

**Lemma 56.** *Let $x_1, \ldots, x_n \in \mathbb{R}^d$ be vectors with $\|x_i\|_2 = 1$, and let $A \in \mathbb{R}^{d \times d}$ arbitrary. Let $G \in \mathbb{R}^{n \times n}$ such that $G_{ij} = x_i^T A x_j$. Then*

$$\|A\| \geq \frac{\|G\|_*}{n} \tag{71}$$

*where $\|A\|$ denotes the spectral norm.*

*Proof.* First, note that for any matrix $B = U\Sigma V$, we have

$$\|B\|_* = tr(\Sigma) = tr(\Sigma V V^T U^T U) = tr(U \Sigma V V^T U^T) = tr(B V^T U^T) \tag{72}$$

where $V^T U^T$ has each singular value bounded by 1 (in fact, it's orthogonal). In other words, $\|B\|_* = tr(BM)$ where $M$ is an orthogonal matrix.

For any two real-valued and possibly non-square matrices, we have

$$tr(A^T B) \leq \|A\|_F \|B\|_F \tag{73}$$

and by submultiplicativity, we have

$$\|AB\|_F \leq \|A\| \|B\|_F \tag{74}$$

Using Eq. 72 and the above two properties, it follows that,

$$tr(X^T A X M) = tr(A X M X^T) \leq \|X^T A^T\|_F \|M X^T\|_F$$

$$\leq \|AX\|_F \|M X^T\|_F \leq \|A\| \|X\|_F \|M\| \|X\|_F \leq n \|A\|.$$

$\square$

### G.4 Positional Abilities in NoPE vs APE

A recent study by Kazemnejad et al. (2024) found NoPE to work better than APE at length generalization and to theoretically simulate positional information – an apparent contrast to our findings, which find APE to be more powerful than NoPE both theoretically and empirically (Figure 1). There are a few important aspects that resolve this apparent contradiction.

1. The APE setting in Kazemnejad et al. (2024) is different from ours, as they used fixed sinusoidal functions (Vaswani et al., 2017), whereas we follow Zhou et al. (2024a) in examining the potentially more expressive case of arbitrary learnable positional encodings.

2. Kazemnejad et al. (2024) trained APE transformers only on an initial segment of the full set of positions, so that the positional encodings at higher positions did not appear during training. This inherently puts APE at a potential disadvantage compared to our setting, where (in line both with Zhou et al. (2024a) and standard LLM training) all positional encodings appear during training.

3. In support of their empirical findings, Kazemnejad et al. (2024) also showed theoretically that NoPE transformers can simulate positional information. Specifically, they found (their Theorem 1) that NoPE transformers can compute an activation with value $1/t$ at position $t$, e.g. by attending uniformly and reading out the contribution coming from a BOS token. Importantly, this is weaker than the ways in which APE transformers can use positional information. For instance, distinguishing close-by positions, say $t - 1$ and $t - 2$, in such an encoding requires rapidly increasing parameter values as $t \to \infty$, as the values get arbitrarily close to 0. In this sense, even a simple task such as an induction head, which requires attention from $t$ to $t-1$, requires parameter norms rapidly increasing with the input length[14], not allowed under our inference procedure. In our theoretical framework, NoPE is not predicted to length-generalize as well as APE on such a task. Indeed, empirically, we find NoPE transformers not to perform well on the induction head task, and a variety of other tasks not expressible in **C-RASP**[$\emptyset$] (dotted lines in Figure 1) despite extensive hyperparameter search (Figure 1 and Section G.7).

We further note that Kazemnejad et al. (2024) also report strong performance for NoPE on a copying task (their Figure 3), in apparent contradiction to our finding that COPY is hard even for APE. However, note COPY in Kazemnejad et al. (2024) had three variants: The first two just asked the model to replicate the number of tokens but not their identities, a task well in **C-RASP**[$\emptyset$], and indeed NoPE worked well (their Figure F.4). Their third variant matched our COPY task and is not in **C-RASP**[periodic,local]; neither NoPE nor APE generalize to 2x training length (their Figure F.4), in line with both our empirical and theoretical findings.

### G.5 Conceptual Relation to Solomonoff Induction

A standard approach to studying learning in algorithmic information theory, specifically Solomonoff Induction (Li & Vitányi, 2008), does not specifically ask for identifying the correct function at some point, but rather aims to bound the overall number of mistakes. In the case of Solomonoff Induction, computability of the underlying function guarantees that this number is finite. Here, we show that our length generalization guarantee can also be viewed as bounding the number of mistakes made by the inference procedure:

**Definition 57.** *When applying the Inference Procedure from Definition 6 on a target function $f$, we define the* Number of Errors *as the summed numbers of inputs on which $T_n$ makes a mistake:*

$$\text{ErrorCount}(T_1, T_2, \dots) := \sum_{n=1}^{\infty} \sum_{x \in \mathfrak{S} : |x| \leq n} 1_{T_n(x,o) \neq f(x)} \tag{75}$$

Then we can show the following variant of Theorem 7:

**Theorem 58.** *Let $f \in \mathcal{F}(\Sigma)$. Then the following are equivalent:*

1. *$f$ is expressible by a Limit Transformer satisfying* PERIODIC *and* LOCAL.

2. *(Guaranteed Length Generalization) Applying the Inference Procedure from Definition 6 to $f$ generates a sequence $T_1, T_2, \dots$ with $\sup_{n=1,2,3,\dots} \mathcal{R}(T_n) < \infty$ and $\text{ErrorCount}(T_1, T_2, \dots) < \infty$.*

*Proof.* In view of Theorem 7, it suffices to note that the following two conditions are equivalent:

1. $\text{ErrorCount}(T_1, T_2, \dots) < \infty$

2. There is some $N_0$ such that, for all $m > N_0$, $T_m$ matches $f$ on all inputs of any length $k \leq m$.

$\square$

---

[14]Kazemnejad et al. (2024) use an MLP to convert $1/t$ to $t$. Such an MLP would need an increasing number of hidden units as the maximum context size $n$ grows, hence, $\|\boldsymbol{A}\|_F$, $\|\boldsymbol{B}\|_F$ growing with $n$. Hence, $\mathcal{R}(T_n) \to \infty$ by Definition 5. Such a construction is thus not predicted to length-generalize under our theoretical framework.

## G.6 THE ROLE OF TRANSLATION INVARIANCE

The hypothesis class (Definition 4) assumes that product functions involving positional encodings are translation invariant: That is, every product function involving exactly one positional encoding is constant across positions, and for every $1 \leq i, j, i + \Delta, j + \Delta \leq n$,

$$\boldsymbol{p}_i^T \boldsymbol{M}_1 \ldots \boldsymbol{M}_k \boldsymbol{p}_j = \boldsymbol{p}_{i+\Delta}^T \boldsymbol{M}_1 \ldots \boldsymbol{M}_k \boldsymbol{p}_{j+\Delta} \qquad (76)$$

whenever $\boldsymbol{M}_1 \ldots \boldsymbol{M}_k$ is a product of parameter matrices linking the input layer, such as $\boldsymbol{K}_{1,h}^T \boldsymbol{Q}_{1,h}$, $\boldsymbol{V}_{2,h}^T \boldsymbol{K}_{3,h'}^T \boldsymbol{Q}_{3,h'} \boldsymbol{V}_{1,h''}$, and similar.

Translation invariance formalizes the idea that the algorithm employed by the transformer is the same independently of the offset. However, it is not explicitly enforced in standard training. Here, we justify this assumption as follows:

1. We theoretically show that translation invariance is beneficial for length generalization, by describing a sequence of transformers $T_n$ that violate it and that fail to length generalize on a simple induction head task (Section G.6.1).

2. We empirically show that trained transformers often exhibit translation invariance, both in transformers that we trained from scratch on algorithmic problems, and in GPT-2 (Section G.6.2).

We conjecture that translation-invariance happens when the training signal itself is offset-invariant: On the algorithmic problems, our theoretical and experimental setups assume that each training example is presented with a random offset; hence, the target function is translation-invariant. In the case of language modeling, the task is also approximately translation-invariant, because – except for the beginning of a document – the structure of language (e.g., its grammar) is likely to be largely independent of absolute position. Hence, we expect that the behavior of the model will be approximately translation-invariant. Thus, with an offset-invariant target function, we conjecture that standard training implicitly favors translation-invariance of the transformer's algorithm. Understanding this theoretically, in a theoretical account of SGD dynamics and generalization on multi-layer transformers, is an interesting problem for future research.

### G.6.1 NON-OFFSET-INVARIANT TRANSFORMERS FAILING TO LENGTH-GENERALIZE

Here, we theoretically show on a simple induction head task how offset invariance is beneficial for length generalization. We construct a sequence $(T_n)_n$ of transformers violating offset invariance that each compute the induction head task at lengths $\leq n/2$, but fail to compute it at length $n$:

**Proposition 59.** *There is a function $f \in \mathcal{F}$ and a sequence $T_n$ of transformers violating offset-invariance, where $n$ operates at lengths $\leq n$, each product function $\boldsymbol{p}_i \boldsymbol{K}_{l,h}^T \boldsymbol{Q}_{l,h} \boldsymbol{p}_j$ is local for $\tau = 2$ where $\sup_n \mathcal{R}(T_n) < \infty$ and each $T_n$ matches $f$ at lengths $\leq n/2$, but no $T_n$ matches $f$ at length $n$.*

This is an important contrast to the translation-invariant setup, where such a sequence $T_n$ where (i) $\sup_n \mathcal{R}(T_n) < \infty$ and (ii) all $\boldsymbol{p}_i^T \boldsymbol{K}^T \boldsymbol{Q} \boldsymbol{p}_j$ are local for a single $\tau < \infty$, will necessarily match $f$ at length $n$ when $n$ is large. This property is used in showing our length generalization guarantee. This property is not available when translation invariance is violated, exemplifying that offset invariance is theoretically beneficial for length generalization.

*Proof.* We take $f$ to compute a simple induction head task (Section 4.1). There is a simple translation-invariant construction: In Layer 1, each position collects the token appearing at the previous condition; in Layer 2, each position $j$ attends to previous positions $i$ that had followed the token also appearing at $j$. This construction performs correctly across input lengths, reflecting length generalization.

Here, we describe a transformer performing this task while violating translation invariance, and *failing to length-generalize*. We use two heads, one performing "correctly" at smaller indices; the other one performing "correctly" at large indices. We note that this version is more complex and not minimal; we conjecture that translation-invariant solutions will generally tend to be simpler for

**C-RASP**-expressible problems. We consider an alphabet $\Sigma = \{SOS, \sigma_1, \ldots, \sigma_k\}$. For each $n$, we define a transformer $T_n$ as follows. We define the encodings:

$$
\boldsymbol{e}_{SOS} = \begin{pmatrix} 1 \\ 0 \in \mathbb{R}^k \\ 0 \in \mathbb{R}^k \\ 0 \in \mathbb{R}^k \\ 0 \in \mathbb{R}^k \\ 0 \in \mathbb{R}^n \end{pmatrix}
\qquad
\boldsymbol{e}_{\sigma_i} = \begin{pmatrix} 0 \\ e_i \in \mathbb{R}^k \\ 0 \in \mathbb{R}^k \\ 0 \in \mathbb{R}^k \\ 0 \in \mathbb{R}^k \\ 0 \in \mathbb{R}^n \end{pmatrix}
\qquad
\boldsymbol{p}_i = \begin{pmatrix} 0 \\ 0 \\ 0 \\ 0 \\ 0 \in \mathbb{R}^k \\ e_i \in \mathbb{R}^n \end{pmatrix}
$$

where $e_i \in \mathbb{R}^n$ is the $i$-th one-hot vector. We define $\boldsymbol{p}_i$ and $\boldsymbol{K}_{1,h}$, $\boldsymbol{Q}_{1,h}$ by:

$$
\boldsymbol{p}_i^T K_{1,1}^T Q_{1,1} \boldsymbol{p}_j = \begin{cases} 2\delta_{i,j-1} & 1 \le j \le \frac{3n}{4} \\ 2\delta_{i,j-2} & else \end{cases}
$$

$$
\boldsymbol{p}_i^T K_{1,2}^T Q_{1,2} \boldsymbol{p}_j = \begin{cases} 2\delta_{i,j-2} & 1 \le j \le \frac{n}{4} \\ 2\delta_{i,j-1} & else \end{cases}
$$

This is **not translation-invariant**: Informally, one of the two heads takes the role of copying from the preceding position for smaller indices; the other one takes it for large indices. We set

$$
\boldsymbol{V}_{1,1} \cdot \begin{pmatrix} \ldots \\ e_i \\ \ldots \\ \ldots \\ \ldots \\ \ldots \end{pmatrix} = \begin{pmatrix} 0 \\ 0 \\ e_i \\ 0 \\ 0 \\ 0 \end{pmatrix}
$$

$$
\boldsymbol{V}_{1,2} \cdot \begin{pmatrix} \ldots \\ e_i \\ \ldots \\ \ldots \\ \ldots \\ \ldots \end{pmatrix} = \begin{pmatrix} 0 \\ 0 \\ 0 \\ e_i \\ 0 \\ 0 \end{pmatrix}
$$

Now, given an input of the form $x_1 x_2 \cdots \in \Sigma^*$ (where $x_1 = \$$), and an offset $o$, we have

$$
\boldsymbol{y}_i^{(1)} = \begin{pmatrix} 0 \\ e_{x_i} \\ e_{x_{i-1}} \\ e_{x_{i-2}} \\ 0 \\ e_{i+o} \end{pmatrix} \text{ if } i + o < \frac{n}{4}
\qquad
\begin{pmatrix} 0 \\ e_{x_i} \\ e_{x_{i-1}} \\ e_{x_{i-1}} \\ 0 \\ e_{i+o} \end{pmatrix} \text{ if } \frac{n}{4} \le i + o \le \frac{3n}{4}
\qquad
\begin{pmatrix} 0 \\ e_{x_i} \\ e_{x_{i-2}} \\ e_{x_{i-1}} \\ 0 \\ e_{i+o} \end{pmatrix} \text{ if } \frac{3n}{4} < i + o \le n
$$

Then, we set $\boldsymbol{K}_{2,1}^T \boldsymbol{Q}_{2,1}$ and $\boldsymbol{K}_{2,2}^T \boldsymbol{Q}_{2,2}$ to satisfy:

$$
(\boldsymbol{y}_i^{(1)})^T \boldsymbol{K}_{2,1}^T \boldsymbol{Q}_{2,1} \boldsymbol{y}_j^{(1)} = \begin{cases} 10 & x_i = \$; j + o < \frac{3n}{4} \\ 0 & else \end{cases} + \begin{cases} 1 & (\boldsymbol{y}_i^{(1)})_3 = (\boldsymbol{y}_j^{(1)})_2 \\ 0 & else \end{cases}
$$

$$
(\boldsymbol{y}_i^{(1)})^T \boldsymbol{K}_{2,2}^T \boldsymbol{Q}_{2,2} \boldsymbol{y}_j^{(1)} = \begin{cases} 10 & x_i = \$; j + o \ge \frac{3n}{4} \\ 0 & else \end{cases} + \begin{cases} 1 & (\boldsymbol{y}_i^{(1)})_4 = (\boldsymbol{y}_j^{(1)})_2 \\ 0 & else \end{cases}
$$

The value matrices are given as ($h = 1, 2$):

$$
\boldsymbol{V}_{2,h} \begin{pmatrix} \ldots \\ e_i \\ \ldots \\ \ldots \\ \ldots \\ \ldots \end{pmatrix} = \begin{pmatrix} 0 \\ 0 \\ 0 \\ 0 \\ e_i \\ 0 \end{pmatrix}
\tag{77}
$$

By design, $\sup_n \mathcal{R}(T_n) < \infty$.

Due to the design of the token embeddings, attention to the SOS token contributes a zero output. Hence, at query position $j$, the first head provides the output

$$\sum_{i=1+o,\dots,j} \frac{\exp\left(\begin{cases} 10 & x_i = \$; j+o < \frac{3n}{4} \\ 0 & else \end{cases} + \begin{cases} 10 & (\boldsymbol{y}_i^{(1)})_3 = (\boldsymbol{y}_j^{(1)})_2 \\ 0 & else \end{cases}\right)}{\text{Normalization}} \boldsymbol{V}_{2,h} \boldsymbol{y}_j^{(2)}$$

$$= \sum_{i=1+o,\dots,j} \frac{\exp\left(\begin{cases} 20 & i = 1; j+o < \frac{3n}{4} \\ 10 & \boldsymbol{e}_{x_{i-1}} = \boldsymbol{e}_{x_j}, i+o < \frac{3n}{4} \\ 10 & \boldsymbol{e}_{x_{i-2}} = \boldsymbol{e}_{x_j}, i+o \geq \frac{3n}{4} \\ 0 & else \end{cases}\right)}{\text{Normalization}} \boldsymbol{V}_{2,h} \boldsymbol{y}_j^{(2)}$$

whereas the second head provides the output:

$$\sum_{i=1,\dots,j} \frac{\exp\left(\begin{cases} 10 & x_i = \$; j+o \geq \frac{3n}{4} \\ 0 & else \end{cases} + \begin{cases} 1 & (\boldsymbol{y}_i^{(1)})_4 = (\boldsymbol{y}_j^{(1)})_2 \\ 0 & else \end{cases}\right)}{\text{Normalization}} \boldsymbol{V}_{2,h} \boldsymbol{y}_j^{(2)}$$

$$= \sum_{i=1+o,\dots,j} \frac{\exp\left(\begin{cases} 20 & i = 1; j+o \geq \frac{3n}{4} \\ 10 & \boldsymbol{e}_{x_{i-2}} = \boldsymbol{e}_{x_j}, i+o < \frac{n}{4} \\ 10 & \boldsymbol{e}_{x_{i-1}} = \boldsymbol{e}_{x_j}, i+o \geq \frac{n}{4} \\ 0 & else \end{cases}\right)}{\text{Normalization}} \boldsymbol{V}_{2,h} \boldsymbol{y}_j^{(2)}$$

If the input length is $\leq \frac{n}{2}$, this simplifies as follows: If $j + o < \frac{3n}{2}$, the first head's contribution is dominated by SOS (which leads to zero value vector), and an overall near-zero contribution. The second head contributes the frequencies of symbols appearing immediately after prior occurrences of $x_j$ (10). On the other hand, if $j + o \geq \frac{3n}{2}$, the reverse situation holds; the second head makes a near-zero contribution, and the first head contributes (10). Hence, $T_n$ computes (10) when the input length is $\leq \frac{n}{2}$.

However, for longer inputs, this does not hold any more: if, say, $o = 0$ and $j = \frac{3n}{4}$, the second head additionally has contributions counting, at positions where $i < \frac{n}{4}$, how often a symbol appeared *two positions* after a symbol matching $x_j$. Hence, $T_n$ performs correctly at lengths $\leq \frac{n}{2}$, but not at longer lengths. $\qquad\square$

### G.6.2 TRANSLATION-INVARIANCE IN TRAINED TRANSFORMERS

Here, we provide empirical evidence that translation invariance, even though it is not explicitly enforced in standard training, may be *approximately satisfied in trained transformers*. In this section, we aim to check, empirically, how often or how much the product

$$\boldsymbol{p}_i^T \boldsymbol{K}_{l,h}^T \boldsymbol{Q}_{l,h} \boldsymbol{p}_j \tag{78}$$

is translation-invariant as a function of $(i, j)$. We conduct experiments for both models trained on algorithmic problems and a transformer language model trained on real-world data.

**Transformers trained on algorithmic problems** We select five small transformers which generalize well on algorithmic tasks. They are trained on BINARY MAJORITY, BINARY MAJORITY INTERLEAVE, MAJORITY, SORT, and COPY UNIQUE (defined in Section E.2.1). For a given head $(l, h)$, we assemble each product $\boldsymbol{p}_i^T \boldsymbol{K}_{l,h}^T \boldsymbol{Q}_{l,h} \boldsymbol{p}_j$ into a matrix $\boldsymbol{M} := \boldsymbol{P}^T \boldsymbol{Q}_{l,h}^T \boldsymbol{K}_{l,h} \boldsymbol{P} \in \mathbb{R}^{n \times n}$.[15] Then such a product is **translation-invariant** if and only if the entries are *constant along*

---

[15]Our experiments use a standard transformer implementation including layer norm, whereas our theoretical analysis disregards layer norm (see Appendix D.3). Layer norm can in principle have a nontrivial impact on

*each sub-diagonal*[16]:

$$\boldsymbol{p}_i^T \boldsymbol{K}_{l,h}^T \boldsymbol{Q}_{l,h} \boldsymbol{p}_j = \boldsymbol{p}_{i+\Delta}^T \boldsymbol{K}_{l,h}^T \boldsymbol{Q}_{l,h} \boldsymbol{p}_{j+\Delta} \quad \forall 1 \le i \le j \le j + \Delta \le n \tag{79}$$

Hence, we quantified the deviance from translation-invariance by calculating the **offset variance** $\text{Var}_{offset}$, the average variance along each sub-diagonal, as follows:

$$\overline{\text{Diag}_c} = \frac{1}{N-c} \sum_{i=c}^{N-1} \boldsymbol{M}_{i(i-c)} \tag{80}$$

$$\text{Var}_{offset} = \frac{2}{N(N+1)} \sum_{i=0}^{N-1} \sum_{j=0}^{i} (\boldsymbol{M}_{ij} - \overline{\text{Diag}_{i-j}})^2 \tag{81}$$

We calculate the $\text{Var}_{offset}$ for each head in these models, and show a histogram of the values across heads in Figure 7. Most of the heads have very small variance, and a single head across these models has variance around 8.

We next investigated the behavior of these heads in greater detail (Figure 8–9). Figure 8 provides examples for the first two bins in Figure 7. The head in (a) shows an entirely uniform pattern, translation-invariant by definition. More interestingly, the head in (b), from the transformer trained on BINARY MAJORITY INTER-LEAVE, shows a perfectly periodic pattern reflecting the parity of $j - i$:

$$\boldsymbol{p}_i \boldsymbol{K}^T \boldsymbol{Q} \boldsymbol{p}_j = \begin{cases} \approx 2 & j - i \text{ even} \\ \approx -1.5 & j - i \text{ odd} \end{cases} \tag{82}$$

Intuitively, this pattern allows the head to specifically attend to positions at an *even* distance from the query position, which is important for solving BINARY MAJORITY INTERLEAVE. As it only depends on $j - i$, this satisfies (79) and the pattern is translation-invariant, in accordance with our hypothesis class in Definition 4.[17] Finally, in Figure 9, we study the single head which has $\text{Var}_{offset} \approx 8$. Subfigure (a) shows the overall pattern of $\boldsymbol{P}^T \boldsymbol{Q}_{l,h}^T \boldsymbol{K}_{l,h} \boldsymbol{P}$. It is not translation-invariant, but still shows similarity to a comparable translation-invariant pattern in (b): $\boldsymbol{p}_i \boldsymbol{K}^T \boldsymbol{Q} \boldsymbol{p}_j$ is large when $j - i$ is small or very large, and small in between. In order to understand the variability of the contributions that inner products $\boldsymbol{p}_i^T \boldsymbol{K}^T \boldsymbol{Q} \boldsymbol{p}_j$ make to attention weights across different offsets, we measure the variance of post-softmax weights across context windows. Specifically, given a window size $w < N$, we apply softmax to $\boldsymbol{P}_{[:,o:o+w]}^T \boldsymbol{Q}_{l,h}^T \boldsymbol{K}_{l,h} \boldsymbol{P}_{[:,o:o+w]}$ together with causal masking, where $o \le N - w$ is the offset. We denote the post-softmax matrix when offset is $o$ as $\boldsymbol{M}^{(o)}$. Subfigure (c) shows this matrix at different offsets $o$. Throughout, the softmax output is focused on the immediately preceding position:

$$(\boldsymbol{M}^{(o)})_{ij} = \begin{cases} \approx 0.8 & i - 1 = j \\ < 0.2 & \text{else} \end{cases} \tag{83}$$

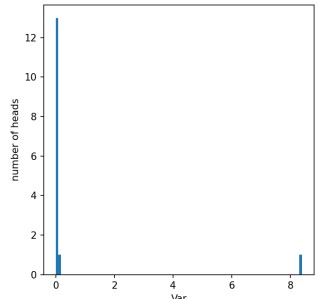

Figure 7: Product functions across five small transformers trained on algorithmic tasks. Histogram of heads by their offset variance measured by $\text{Var}_{offset}$ (Eq. 81), for all heads from five small transformers trained on algorithmic problems. Most heads have very low Offset Variance; we investigate these in Figure 8. One head across the five models has a higher variance; we investigate it in Figure 9.

---

the attention logit dot products. We accounted for this, by fixing the variance term in layer norm as its average value across a big amount of input data (around 240k tokens for small transformers and 400k tokens for GPT-2), thus making it a linear operation. Effectively, we are thus able to account for layer norm by applying it as a linear operation to the columns of $\boldsymbol{P}$.

[16]Due to causal masking, super-diagonal entries are irrelevant, see Definition 4.

[17]In the asymptotic regime of $n \to \infty$, our Inference Procedure prefers transformers where product functions $\boldsymbol{p}_i^T \boldsymbol{K}^T \boldsymbol{Q} \boldsymbol{p}_j$ are local, and thus cannot be periodic over unbounded distances. Such periodic relations are instead, asymptotically, predicted to be covered by product functions additionally involving components whose rank is penalized by $\mathcal{R}(T_n)$, such as $\boldsymbol{p}_i^T \boldsymbol{K}^T \boldsymbol{Q} \boldsymbol{V} \boldsymbol{p}_j$. Our theory applies in the asymptotic limit, and thus predicts that as $n \to \infty$ (and thus $d \to \infty$), this periodic pattern might instead be taken over by such products. An extension of our theory that accounts for training dynamics, beyond an idealized inference procedure, may be able to predict in which settings which kind of product will represent periodic relations.

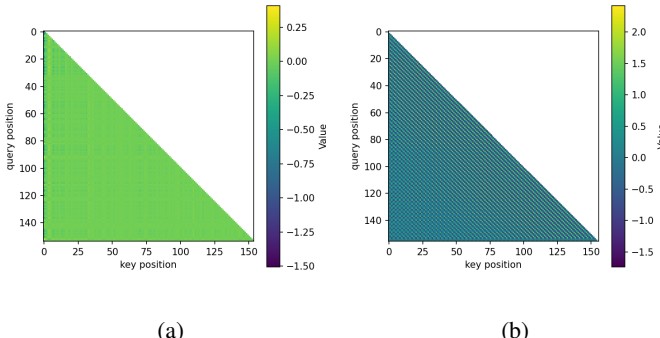

(a)                                           (b)

Figure 8: Product functions across five small transformers trained on algorithmic tasks. (a) $\boldsymbol{P}^T\boldsymbol{Q}_{l,h}^T\boldsymbol{K}_{l,h}\boldsymbol{P}$ of a head whose variance is 0.001 (Figure 7 a). (b) $\boldsymbol{P}^T\boldsymbol{Q}_{l,h}^T\boldsymbol{K}_{l,h}\boldsymbol{P}$ of a head whose variance is 0.098 (second bin in Figure 7 (a)). This head represents an almost perfectly periodic pattern, depending only on the parity of $j - i$. All heads with Offset Variance close to 0 look qualitatively like (a) or (b). We investigate the higher-variance head in Figure 9.

Thus, the product function makes an approximately translation-invariant contribution to attention weights. Overall, the transformers trained on algorithmic problems show approximate translation-invariance even though this is not enforced as part of standard initialization or training.

**Results on an LM: GPT-2 Small** We conduct the same calculation on GPT-2 Small (Radford et al., 2019), a language model with 12 layers, and 12 heads per layer, which has served as a common object of study in the mechanistic analysis of language models (Conmy et al., 2023). The results are shown in Figure 10 and 11. Again, there are heads producing near-constant products, and heads producing a focus on positions $i$ with small distance $j - i$. In addition to products of the form $\boldsymbol{p}_i\boldsymbol{K}^T\boldsymbol{Q}\boldsymbol{p}_j$, we also inspected more complex products including value matrices[18], finding these to generally be close to zero and thus also translation-invariant. Overall, we have found product functions in GPT-2 Small to display approximately offset-invariant behavior, again even though this is not enforced in training or initialization.

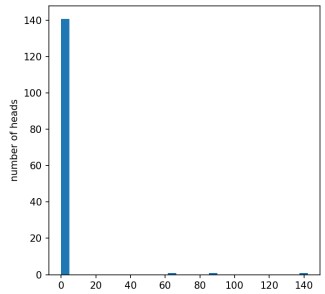

Figure 10: Product Functions in GPT-2 Small: Histograms of Offset Variance measured. Heads with variance close to 0 have approximately uniform values (Figure 11a). Heads with higher variances are shown in Figure 11 (b–d).

### G.7 COMPARING APE AND NoPE ON INDUCTION HEAD TASKS

We further conduct more experiments comparing the generalization ability of transformers with APE and NoPE. We have already found that APE performs better at NoPE at a variety of task that are not expressible in **C-RASP**[∅], such as COPY UNIQUE (Figure 1). Here, we further bolster these results by (i) considering a parametric family of further such tasks, and (ii) conducting a much broader hyperparameter search for NoPE.

We introduce 3 problems, and refer them as COPY UNIQUE $t - 1$, $t - 2$, $t - 3$ respectively. Here, "$t - c$" means the problem is naturally solved with an attention pattern where each token attends to the $c$-th last previous token. The $t - 1$ problem is identical to the COPY UNIQUE task we described before; it asks the model to copy a string in which each token appears at most once (hence, "UNIQUE"), and is naturally solved using an induction head circuit (Section 4.1). The $t - 2$

---

[18]GPT-2 Small uses additional output matrices, e.g. $\boldsymbol{O}_{l,h}\boldsymbol{V}_{l,h}$, corresponding to the value matrices in our theoretical analysis.

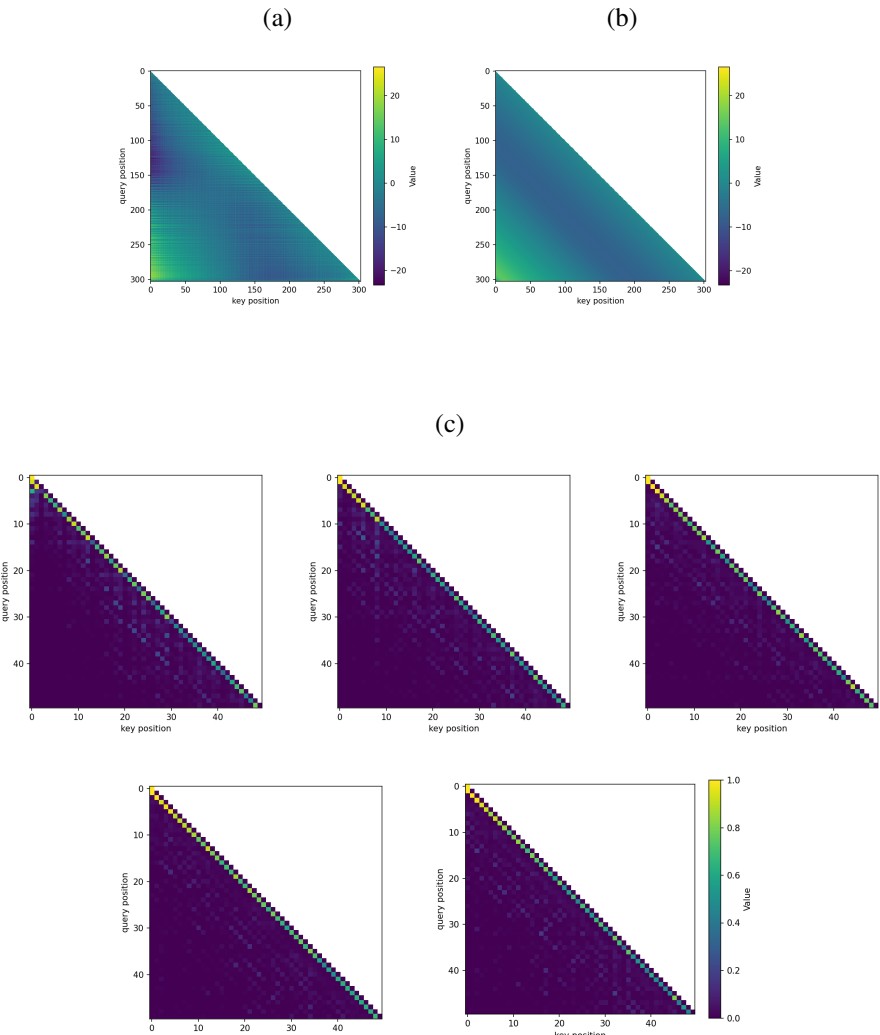

Figure 9: Head 0,0 in a small transformer trained for COPY UNIQUE. This is the head whose variance is 8.39 (the single head in the largest bin in Figure 7). (a) $\boldsymbol{P}^T \boldsymbol{Q}_{l,h}^T \boldsymbol{K}_{l,h} \boldsymbol{P}$. Values are high when $j - i$ is small or very large, and low in between. (b) For comparison, we show a similar but perfectly translation-invariant matrix. We replace values $\boldsymbol{P}^T \boldsymbol{Q}_{l,h}^T \boldsymbol{K}_{l,h} \boldsymbol{P}$ with their average value along the diagonal. The pattern is qualitatively similar to (a), highlighting how (a) is similar to though does not fully satisfy translation-invariance. (c) Heatmaps showing the post-softmax matrices $M^{(o)}$, where $o = 0, 40, 80, 120, 160$ respectively, and at a context window $w = 50$. Across offsets, the largest contribution at position $t$ is to the immediately preceding position $t-1$; that is, the entries of the post-softmax matrix $\boldsymbol{M}^{(o)}$ are largely determined by the relative distance $j - i$. Hence, while the overall function is not exactly translation-invariant, the resulting attention behavior is approximately offset-independent, and implements attention to the immediately preceding position.

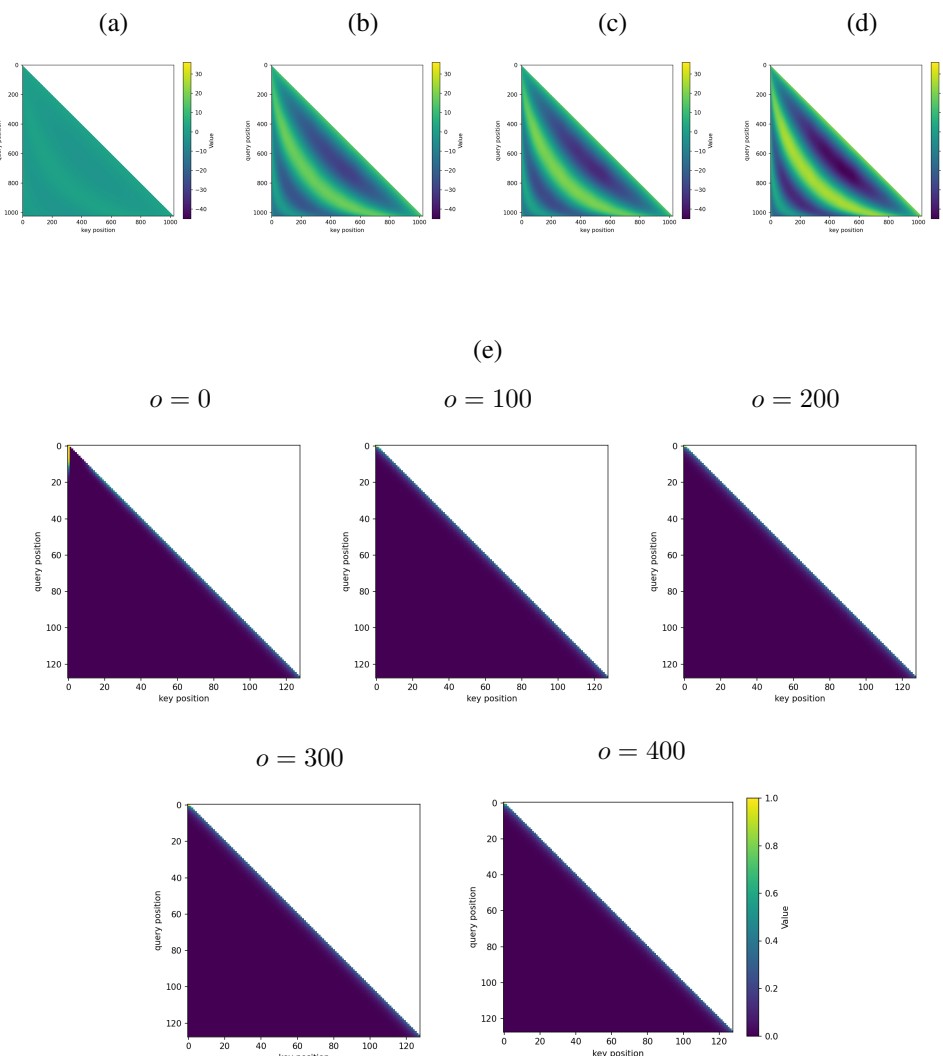

Figure 11: **Product functions in GPT-2 small.** Heatmaps showing the product $\boldsymbol{P}^T\boldsymbol{Q}_{l,h}^T\boldsymbol{K}_{l,h}\boldsymbol{P}$ of heads in different bins in Figure 10. (a) A head whose Offset Variance is 0.6. The pattern is close to uniform, making it trivially translation-invariant. (b–d) Heads with Offset Variances 66.0 (b), 87.9 (c), 142.1 (d). These heads (b–d) produce high products at small distances, and at distances $\approx 300$. The first aspect is exactly translation-invariant; the second one is approximately translation-invariant in the middle of the range, though not when closer to the boundaries. (e) Heatmaps showing the post-softmax matrices $M^{(o)}$ for Head 0,3, the head with largest variance in Figure 10, where $o = 0, 100,$ 200, 300, 400 respectively, and window size $w = 128$. With the exception of positions very close to the beginning ($o = 0$), the weights from query position $j$ are focused on key positions $i$ with small $j - i$ (note the green/yellow diagonal). Note that the green/yellow diagonal is faint as it is spread over multiple small distances $j - i$, but it is consistently present and of consistent width. Hence, the contribution of $\boldsymbol{p}_i\boldsymbol{K}^T\boldsymbol{Q}\boldsymbol{p}_j$ to attention patterns is largely independent of the offset here, even in the head with the numerically largest Offset Variance.

|  | APE | | | NoPE | | |
|---|---|---|---|---|---|---|
|  | $\le 50$ | [51, 100] | [101, 150] | $\le 50$ | [51, 100] | [101, 150] |
| t-1 | 1.000 | 1.000 | 0.986 | 1.000 | 0.770 | 0.044 |
| t-2 | 1.000 | 1.000 | 0.977 | 0.999 | 0.232 | 0.000 |
| t-3 | 1.000 | 1.000 | 0.990 | 1.000 | 0.116 | 0.000 |

Table 8: Experimental results for transformer with APE and NoPE, on the three problems we defined. The models are trained on data where LEN$\le 50$, and tested on all three length ranges. Accuracy is averaged over 5 successful runs (accuracy on LEN $\le 50$ is greater than 0.99). We find the set of hyperparameters that generalizes best in each case, shown in Table 10.

|  | APE | | | NoPE | | |
|---|---|---|---|---|---|---|
|  | $\le 128$ | [129, 256] | [257, 384] | $\le 128$ | [129, 256] | [257, 384] |
| t-1 | 1.000 | 1.000 | 0.989 | 1.000 | 0.602 | 0.000 |
| t-2 | 1.000 | 1.000 | 0.998 | 1.000 | 0.241 | 0.000 |
| t-3 | 1.000 | 1.000 | 0.997 | 0.997 | 0.058 | 0.000 |

Table 9: Experimental results for transformer with APE and NoPE, on the three problems we defined. The models are trained on data where LEN$\le 128$, and tested on all three length ranges. Accuracy is averaged over 5 successful runs (accuracy on LEN $\le 128$ is greater than 0.99). We find the set of hyperparameters that generalizes best in each case, shown in Table 11.

problem consists of inputs such as `SOS # 14 23 6 9 1 18 SEP 14 6 1 EOS` That is, the problem is tasked with copying every second token from the input, which can be naturally solved using attention from position $t$ to position $t - 2$. Finally, the $t - 3$ problem consists of inputs such as `SOS # # 14 23 6 9 1 18 SEP 14 9 EOS`

In general, for a $t - c$ problem, each input sequence starts with `SOS` and $c - 1$ special tokens #, and then continues with $k \cdot c$ unique tokens, where $k \in \mathbb{Z}^+$ is a positive integer, finally the output sequence after SEP, which consists of every $c$th token in the input sequence. We introduce the extra special token # so that the task is easily solved using an induction head-like construction attending from position $t$ to position $t - c$. The model is trained to output the tokens after SEP. In these problems, LEN (recall Section E.2.1) refers to the length of the portion between `SOS` and SEP. The minimum LEN is $l_{min} = 2c - 1$. The total vocabulary size of tokens except for special tokens is equal to the maximum number of unique tokens needed, i.e., 150.

As before, the models are trained on sequences of LEN $\le 50$, and we optimize hyperparameters for accuracy on lengths [51, 100]. For APE transformers, we fixed the architecture to have 2 layers and 1 head per layer, and the searching space is: learning rate = {0.001, 0.0001}, model dimension = {64, 256, 512}, dropout rate = {0, 0.1}. For NoPE transformers, we search in a *much larger space*: number of layers = {2, 3, 4, 5}, number of heads = {1, 2, 4}, learning rate = {0.001, 0.0001}, model dimension = {64, 256, 512}, dropout rate = {0, 0.1}. Other hyperparameters are fixed and same as before; the attention dropout is still 0. We sweep all hyperparameter combinations and select the combination with highest accuracy on on [51, 100], shown in Table 10. The selected combination is then used to train 5 models with different random seeds. The average accuracy on the test set (different from before, each now 5,000 examples, more than before) is shown in Table 8. We can see that, even though we search in a much larger space for NoPE and select the most generalizable model according to accuracy on on [51, 100], APE's performance is still far stronger than NoPE across the board. We also change the length range, that is, we train model on sequences of LEN $\le 128$, and we optimize hyperparameters for accuracy on lengths [129, 256], and test the model's performance on [257, 384] as well as the previous two ranges. Results are shown in Table 9. We see that, again, APE's performance is superior than NoPE. These problems are easily expressed in **C-RASP**[periodic, local] but not **C-RASP**[∅], and our theory thus predicts length generalization for APE but not NoPE, in line with the empirical results.

| | APE | | | NoPE | | |
|---|---|---|---|---|---|---|
| | Model Arch | LR | Dropout | Model Arch | LR | Dropout |
| t-1 | 2 layer; 1 head; 64 dim | 1e-3 | 0.1 | 5 layer; 2 head; 512 dim | 1e-3 | 0.1 |
| t-2 | 2 layer; 1 head; 256 dim | 1e-4 | 0.1 | 5 layer; 4 head; 256 dim | 1e-3 | 0.1 |
| t-3 | 2 layer; 1 head; 256 dim | 1e-3 | 0.1 | 5 layer; 4 head; 256 dim | 1e-3 | 0 |

Table 10: The best configuration of hyperparameters we found for each case in Table 8. Gray text stands for fixed hyperparameter. Because we know 2 layers with 1 head can solve these problems when using APE, we do not perform further search there. On the other hand, for NoPE, we perform hyperparameter search over a larger space.

| | APE | | | NoPE | | |
|---|---|---|---|---|---|---|
| | Model Arch | LR | Dropout | Model Arch | LR | Dropout |
| t-1 | 2 layer; 1 head; 64 dim | 1e-3 | 0.1 | 5 layer; 1 head; 512 dim | 1e-3 | 0.1 |
| t-2 | 2 layer; 1 head; 256 dim | 1e-3 | 0.1 | 5 layer; 2 head; 512 dim | 1e-3 | 0.1 |
| t-3 | 2 layer; 1 head; 256 dim | 1e-3 | 0.1 | 5 layer; 2 head; 512 dim | 1e-3 | 0.1 |

Table 11: The best configuration of hyperparameters we found for each case in 9.

