# OpenReview forum: "A Formal Framework for Understanding Length Generalization in Transformers"
_ICLR.cc/2025/Conference — ICLR 2025 Poster_

### Official Review · Reviewer_H3pw · 2024-10-30

**Soundness:** 2
**Presentation:** 3
**Contribution:** 2
**Rating:** 6
**Confidence:** 3

**Summary:**

In the recent years, with the surge of LLMs and transformers, there has been an increasing interest to formally understand the generalization failures of transformer. A notorious and common failure mode that is often mentioned in the literature pertains to length generalization, where the model is trained on sequences of certain length, and then is tested to longer sequences. Many works have shown the failures of modern architectures on a range of tasks (most popular ones being parity) and others have come up with strategies to fix the failures (e.g., changes in positional embeddings). In recent work, Zhou et al., authors proposed an insightful conjecture, the RASP-L conjecture, that aims to dilineates the tasks that are easy for transformers to learn and length generalize on, and then tasks that continue to be hard for transformers. While the conjecture was empirically backed, theoretical justifications for the same have been lacking. In this work, the authors aim to formalize and provide a theoretical justification to the RASP-L conjecture.

The authors study two types of positional encodings -- no positional encodings, and absolute positional encodings. When dealing with absolute positional encodings the number of parameters grow with length of the input. To address this challenge, the authors introduce a new object, the limit transformer. This transformer encapsulates the behavior of transformers on longer and longer sequences into a single object. The authors define an idealized inference procedure, which searches for the transformer that minimizes the risk along with a regularization constraint. The regularization constraint is a special one, i.e., it is not the standard constraint based on purely say the l2 norm of the weights. With these constraints in place, the authors show that if the ideal function f is expressable in a limit transformer that satisfies two properties -- Local and Periodic, then the transformer is able to learn the function f and length generalize on it. In the second half of the paper, the authors show that for every program C-RASP[phi, Psi) with local and periodic phi and Psi respectively, there exists a limit transformer that accepts the same set of strings that P accepts. The authors also provide communication complexity based arguments to explain the limits of limit transformers. Finally, the authors conduct experiments to match the predictions of the theory.

**Strengths:**

1. The authors tackle a hard problem, i.e., providing a formal justification of RASP conjecture for multi-block transformer models. This is both a hard problem and an important one.
2. The authors have been creative in several aspects of the paper -- i) the regularization constraints that have been imposed seem particularly important to the inference procedure's success, ii) the construction of the limit transformer, iii) the tight connection between the limit transformer and the C-RASP language from Yang and Chiang.
3. The main body of the paper is nicely written and does a nice job of getting to the main results quite fast.

**Weaknesses:**

There are quite a few concerns that I have for various parts of the paper. My current score is a reflection of these weaknesses. I would be happy to change my score if the authors can provide satisfactory explanations and no other major concerns appear in the course of discussion.

1. **No Positional Encoding vs. Absolute Positional Encoding**:  In the current work, the authors went through great detail to construct limit transformer with the idea that limit transformer can encapsulate longer and longer transformers into one limiting object. The first thing that bothers me is that if we take no positional encodings then we do not need this object. The authors do not explain the need of limiting transformer in the context of NoPE as there is no growth in number of parameters anymore that limit transformer needs to cater to.
The second thing that bothers me is that from an expressivity point of view, the NoPE based transformer can express APE up to a large length. So why can't we use
Theorem 1  (https://proceedings.neurips.cc/paper_files/paper/2023/file/4e85362c02172c0c6567ce593122d31c-Paper-Conference.pdf) from this work. This work essentially is arguing that NoPE can approximate APE (or RoPE).

2. **About invariance to offsets and regularization**
     a) The authors introduce the constraint that the transformer should be invariant to offsets, i.e., if the problem appears at different locations in the context window then the solution should not change. This constraint is not explicitly enforced.
     b)  The authors also have an idealized inference procedure, where the regularizer has been introduced for the purpose of theory. The authors argue that there is an implicit bias towards small values of it at initialization. I don't quite see how. Also, this regularizer is also not explicitly enforced.
  Since there is quite some gap between the theory and expmts, what do you think is the explanation for this gap?

3. **Regarding the definition of limit transformers**
     a) The limit transformer is introduced in Definition 2. The term y_i(l) in equation (6), is it the same as how it was defined in equation 4. If so, then does it already have positional encoding in it like in equation (1). If so, then how does the function phi that is introduced additionally absorb the terms that involve inner product of two different positional encodings. It feels if y_i^(0) already had positional encodings in it then won't the first term in equation 6 already take care of the stuff.

   b) Is the rest of the construction of limit transformer same as standard transformer and the only difference is equation (6)? I ask this because following equation (6), we are not told what happens to attention logits.

   c) In point 2 in the definition u say that the positional encodings p_i are encoded in finite precision. If that is the case, then when we increase the length to arbitrary large values, the positional encodings start overlapping and we only have finitely many positional encodings.    This does not address the increase in the parameter count issue that the author state was the very reason to define the limit transformer. If we are happy with finitely many positional encodings, then why not just do some periodic encodings in the standard transformer? Also, if we are happy to do everything with finitely many positonal encodings then this goes back to my first concern on NoPE, we can operate with NoPE, express finitely many positional encodings, and simplify the whole story right?


4. **Concerning periodic and local in definition 3.** In Definition 3, you state that phi_l,h is translation-invariant and local. You also stated that phi_l,h expresses the inner product involving positional encodings. This translates into a constraint on the positional encodings. This creates confusions. The authors should be more clear on this whole connection in the main body.

5. **About the hypothesis class definition 4** In definition 4, you say that each product function involving position encoding is translation invariant. You also say that each function involving exactly one product function is translation invariant. These constraints seem very restrictive. Is there a reason to believe that imposition of these constraints is not over simplifying the problem somehow? Are these constraints implying offset invariant condition? I think offset invariance on its own was reasonable but this seems not very digestable. I would appreciate if authors gave more insights into why these constraints are not unreasonable? Also, some numerical insights into what it means to enforce these constraints? This goes back to my point 2. If you see point 2, I state that there is gap between theory and expmts. In this case, the theory would require some of these constraints, which how do u really enforce? Perhaps these constraints are strong sufficient conditions for length generalization? and far from necessary? Is offset invariance a necessary condition btw for length generalization?

6. **On the regularizer in definition 5**
    a) After definition 5, you state that the idea of this regularizer is to discourage attention between far-away positions that do not appear together during training, which could hamper length generalization. At what point do you use this insight in the proofs. For instance in Lemma  17 how does it come up? I don't quite see it. Also, since all positions are equally penalized in this regularizer, why would things far off be more penalized? Also, the justification based on initialization making the regularizer small is not fully clear.

   b) If we use NoPE positional encoding, then p_i is set to zero for all i. As a result, I don't quite understand the role of the regularizer anymore. Since the regularizer is supposed to penalize far away positions, those positions don't seem distinguisable under the regularizer as p_i is set to zero. Further, if p_i is set to zero, then what is the role of the phi function in equation (6), and eventually why do we need the limit transformer?

7. **On phi function** In the line 260, you indicate that for phi function, we need to only care abt it its values such as phi(1,1), phi(1,2),..phi(1,tau). The values above the diagonal are taken care of by translation invariance, but what abt the values below the diagonal, i.e., phi(2,1)..I don't think you assume symmetry, do you?

8. **Minor remark on line 326/327** Shouldn't the Q_a in the RHS be Q_a(j) and not Q_a(i)?

9. **Notation remark on Theorem 9**. In an unfortunate use of notation, u call Psi function local and Phi function periodic. Earlier u had used phi for local in the definition 3.

10. Since the results from the work hold for NoPE positional embeddings. From Theorem 2 in https://proceedings.neurips.cc/paper_files/paper/2023/file/4e85362c02172c0c6567ce593122d31c-Paper-Conference.pdf, the results should extend to relative positional encodings too?

11. Currently the results require a large N_0, which practically speaking can be very large. The authors do mention this limitation. While I am not expecting a bound of any sort in this work, I want to understand the consequences of the results better. The current machinery in this work (if correct), seems to indicate that allowing for a very large N_0 and some strong constraints (periodic, local on hypothesis class), length generalization is achievable for a large N_0 seen during training. If this N_0 from the theory is quite large, then would you say that research in this should try to explain why can transformers do it with a much smaller N_0 than theory predicts? I want to understand the hunch of the authors here. If one tries to bound N_0, then would we run the risk of vaccuous bounds?


12. **Concerns on the proof of Lemma 17**:
    a) In line 976 you say there are only a finitely many settings traversed by the limit transformer \tilde{T}_i. Why is this the case? Can you properly justify?

      b) In equation 9, you say that we select an R(Tn) that is less than 1/n + inf (R(T)). In line 981, you argue that R(Tn) should converge because inf R(Tn) is bounded and monotonically increasing. This argument only tells that the RHS in equation (9) converges, which is the upper bound. Why does it imply that the LHS converges? You crucially use the existence of this limit in equation (14).

     c) In line 995, you say that lim D_v_i(v_i) is D_0. Why does this limit exist?

     d) Below equation 17, you state "As this function is monotonically increasing, and as phi_{l,h} has bounded precision, there must be t_infty..." Why does this hold true? I don't quite follow.

      e) In line 1010-1017, you construct a sequence of T'n satisfying certain properties. Why does this have to exist? For instance why is  D_{vi(n)}(n) = D_{infty}(n) true?

      f) The phi_l,h(i,j) in equation (10) should also bear the index n as it would be different for each limit transformer. This makes the rest of the stuff bit confusing.  For instance, you define D_n(tau) in line 984. Why would it be the case that tau can be larger than n in the summation? Since positional embeddings for that transformer would only be defined up to n right?

      g) I do not follow the inequality in line 1044?

      h) In line 1049-1052, you say that set of functions traversed becomes stationary, what do u precisely mean here?

13. In line 1121 to 1127, how does the C logN fall out. Do you mean that since you partition stuff into N positions and that takes log N bits to compute, we get C log N?


14. I realize that there was one question I had forgotten to add in the above list. In the Definition 4 of your hypothesis class, I do not see any constraint on positional encodings being periodic. Does periodicity fall out as a consequence of the offset independent constraint you add later in the definition? What confuses me quite a bit is that while your transformer is not periodic but your limit transformer is periodic. How is it that a transformer with non-periodic positional encodings with an ever growing context window is captured by another transformer like object with periodic positional encodings? And relatedly why not just have periodic positional encodings on the original transformer to begin with instead of the very strong offset independence conditions in definition 4.

**Questions:**

Please see weakness section, where I list both the weaknesses and questions.

---

> ### Author Response · Authors · 2024-11-21
> **Response (Part 1)**
>
> > There are quite a few concerns that I have for various parts of the paper. My current score is a reflection of these weaknesses. I would be happy to change my score if the authors can provide satisfactory explanations and no other major concerns appear in the course of discussion.
>
> We thank the reviewer for the detailed reading and their feedback.
>
> We believe the most important points to revolve around
>
> * Detailed questions about the proof of Lemma 17, which we address in detail
> * The role of APE vs NoPE, which we now discuss in Appendix G.4. While NoPE can partially simulate positional information, it is not as powerful as APE. We find in both theory and experiment that NoPE does not perform well on tasks not expressible in C-RASP$[\emptyset]$ (Figure 1 and Appendix G.4).
> * The role of offset invariance and translation invariance, which we now discuss in Appendix G.6. We empirically show that translation invariance is often approximately satisfied in trained transformers, both small transformers trained on algorithmic problems, and a real-world LM with APE (GPT-2), suggesting that it is favored by standard training.
> We also theoretically show that translation invariance is beneficial for ensuring length generalization on a simple induction head task.
>
> ### Point-by-Point Reponse
>
> We address each question point-by-point. We paraphrase the questions for the sake of brevity.
>
> >  (1) Can't NoPE simulate APE? Hence, isn't it sufficient to study NoPE, making the analysis much simpler?
>
> It is true that up to a fixed input length $T$, a NoPE transformer can in principle compute positional information, in the sense of computing an activation with value $1/t$ at position $t$ (as in the proof of Theorem 1 in the Kazemnejad et al paper referenced by the reviewer). Importantly, this is weaker than the ways in which APE transformers can use positional information.
> In order to perform this simulation, the transformer would need either rapidly increasing MLP weight values or rapidly increasing width for larger values of $T$. For instance, distinguishing close-by positions, say $t-1$ and $t-2$, in such an encoding requires rapidly increasing parameter values as $t \rightarrow \infty$, as the values get arbitrarily close to $0$. In this sense, even a simple task such as an induction head, which requires attention from $t$ to $t-1$, requires parameters rapidly increasing with the input length Appendix G.4). In our theoretical framework, NoPE is not predicted to length-generalize as well as APE on such a task.
> Indeed, empirically, we find NoPE transformers not to perform well on a variety of tasks not expressible in $C-RASP[\emptyset]$ (Figure 1).
>
> In our theory, NoPE is a simple special case of the more powerful APE setup (lines 265), in which special case one can indeed prove Theorem 7 without limit transformers. This is explained in Appendix B.2.
>
> > (2) Regarding assumptions made by the theory
>
> > (2a) The theory uses the constraint that all transformers are offset-invariant
>
> We believe that the reviewer refers to ''*In line with the assumed setup, we focus on transformers whose input-output behavior is invariant across offsets: $T(x,o) = T(x,o')$ for any $0 \leq o, o' \leq N(T) - |x|$*''.
> We have removed this statement, as it is not needed at this point.
> However, we do assume that the Hypothesis Class requires translation invariance, which we discuss under (5).
>
> > (2b) The inference procedure uses an additional regularizer that is not explicitly enforced in standard training
>
> It is true that the additional regularizer (Equation 8) is not explicitly enforced in standard training.
> We argue that, nonetheless, it is likely to reflect an inductive bias of standard initialization: We know from Proposition 54 that, if one randomly initializes transformers at context length $N$, and sufficient width, the regularizer will in expectation remain bounded even as $N \rightarrow \infty$. Boundedness of the additional regularizer as $N \rightarrow \infty$ is thus likely to be favored by standard initialization.
>
> More broadly, we argue that our results are highly interesting even if the theoretical setup makes idealizing assumptions. Despite a lot of empirical research, theoretical understanding of length generalization of transformers is in its infancy. The RASP-L Conjecture has not previously been formalized; there is no existing evidence beyond experiments. Our work already enables a proper formalization of the conjecture, in terms of limit transformers and C-RASP. As a step towards theoretically understanding  length generalization, we study length generalization in an idealized setup abstracting away from training dynamics, as we clearly acknowledge in Sections 1 and 6. We lay groundwork for future work, both through theory (we provided a formal class of languages whose expressivity can be rigorously understood) and experiments (we provide evidence that this language class tracks empirical length generalization behavior).

---

> ### Author Response · Authors · 2024-11-21
> **Part 2**
>
> > (3)   Regarding the definition of limit transformers
>
> > (3a) Why do limit transfomers need both positional encodings and $\phi_{l,h}$ functions?
>
> Because Limit Transformers have finite width $d$, the positional encodings ${\bf p}_i$ in the Limit Transformer can only absorb some parts of the information from the positional encodings ${\bf p}_i$ of the standard transformer, namely those that can be absorbed into finite-width and finite-precision encodings.  It is desirable to assign Limit Transformers a finite width, so their definition is as close to standard transformers as possible.
>
>
> Other parts of the positional information cannot be coded in this way -- for instance, it's not possible in a single transformer operating on unboundedly long inputs, to implement attention from position $t$ to position $t-1$ with a fixed-width positional encoding (cf. line 864). Hence, we introduce functions $\phi_{l,h}$ to absorb such remaining positional information. Due to its bounded width,
> $y_i$
> will generally contain less positional information in the Limit Transformer than in a standard transformer, necessitating the additional term using $\phi_{l,h}$ in Equation 6.
>
>
>
>
> >   (3b) Other than Eq. 6, is the rest of the construction of limit transformer same as standard transformer?
>
>
>
> Yes, everything else is identical to the transformers from Section 2. Limit transformers have the same attention mechanism (Eq. 3); they just differ in adding $\phi(i,j)$ to the attention logits.
>
>
> >  (3c) If we are happy with finitely many positional encodings, then why not just do some periodic encodings in the standard transformer?
>
> Importantly, while the Limit Transformer has only finitely many distinct positional encodings, the standard transformers $T_1, T_2, T_3, ...$ found by the Inference Procedure can have unboundedly many distinct positional encoding vectors, because the width $d$ is not bounded by the definition of the Hypothesis Class. We find it useful to separating, in translating to Limit Transformers, the power of unbounded-width encodings into (i) bounded-width and bounded-precision encodings $p_i$ and (ii) the functions $\phi_{l,h}$. As explained in point 7 of Appendix A (expanded in the revision), $p_i$ capture periodic information, whereas $\phi_{l,h}$ encapsulate local relations. Hence, positional abilities of Limit transformers go beyond the periodic encodings $p_i$.
>
> We study standard transformers with learnable APE encodings, rather than the periodic encodings suggested by the reviewer, to match the setup of Zhou et al 2024 [1]. Extending our theory to hard-coded periodic encodings could be another interesting topic.
>
> We also would like to emphasize that Limit Transformers are a mathematical construct that helps us prove things about standard transformers (as defined in Section 2). There may be other ways of defining these limiting objects that allow proving the same results.
>
> Re *"we can operate with NoPE, express finitely many positional encodings, and simplify the whole story right?"*: Importantly, limit transformers have both bounded-width and bounded-precision positional encodings $p_i$ and $\phi_{l,h}$ functions; jointly, these simulate the power of standard APE transformers, and are substantially more powerful than beyond NoPE, as we explain under point (1).
>
> > (4) The presentation of the translation-invariance and locality constraints imposed on $\phi_{l,h}$ and positional encoding might need to be clearer.
>
> We have rewritten Definition 3 to make this clearer. We would welcome any further advice on how to improve this aspect.
>
> [1] Zhou et al, What algorithms can transformers learn?, ICLR 2024

---

> ### Author Response · Authors · 2024-11-21
> **Part 3**
>
> > (5) Why is translation invariance a reasonable constraint in the definition of the Hypothesis Class? Isn't there a gap between theory and experiments?
>
> We agree that translation invariance merits further consideration.
> We distinguish two relevant properties:
>
> * (A) offset-invariance of the *input-output behavior*; that is, the transformer's output $T(x,o)$ is independent of $o$;
> * (B) translation-invariance of the *product functions*, such as $p_i^T K^T_{l,h} Q_{l,h} p_j$ as assumed in Definition 4 (Hypothesis Class).
>
> Our experimental setup (Section 5) assumes that every training sample is presented with a random offset. This setup is a simplification of the setup in Zhou et al 2024, and aims to mimick how LLMs need to solve reasoning tasks no matter where they appear in a context. Due to this property of the experimental setup, the transformers are trained to be offset-invariant in their input-output behavior (A), in agreement with the theory.
>
> Translation-Invariance (B) is a stronger requirement, and implies (A). We now discuss (B) in Appendix G.6, where we show the following:
>
>
> - We provide theoretical evidence that translation invariance is beneficial for length generalization (Appendix G.6.1). There, we describe a sequence $T_n$ of transformers violating translation invariance, where $n$ operates at lengths $\leq n$, where $\sup_n \mathcal{R}(T_n)<\infty$ and each $T_n$ describes a target function (a simple induction head task) at lengths $\leq n/2$, but no $T_n$ represents the task at length $n$ -- that is, the models fail to length-generalize. This is in contrast to the translation-invariant setup, where such a situation necessarily leads to $T_n$ length-generalizing correctly for large $n$.
>
>
> - Translation invariance is approximately satisfied in trained transformers (Appendix G.6.2), both in small transformers trained on algorithmic problems, and in a real-world LM, GPT-2. This suggests that translation invariance, even though not explicitly enforced, is implicitly favored by standard training, presumably because the target function itself is (at least approximately) offset invariant in these setups.
>
> Overall, we conclude that
>
> * standard training tends to implicitly  favor translation invariance, at least in the setups relevant to our experiments
> * translation invariance is theoretically beneficial for length generalization
>
>
>
> > (6) On the regularizer in definition 5
> >
> >  (6a) How does it enter Lemma 17?
>
>
>
> The term is key to ensuring that $f$ is represented by a single LOCAL limit transformer.
> Specifically, boundedness of the regularizer term, across all context windows n, entails that  $D_0$ (Equation 15) is finite, which is used to derive a contradiction in line 1090. That in turn is used to show that the $\phi$ functions
> are local for a uniform $\tau_\infty$. This is needed for concluding that the function $f$ is representable by a single LOCAL limit transformer.
>
> >  Also, since all positions are equally penalized in this regularizer, why would things far off be more penalized?
>
> The idea is that, when training on lengths $\leq n/2$, the training set constrains the attention behavior between positions at a distance $\leq n/2$, but not at larger distances $>n/2$. The regularizer discourages attention at such greater distances.
> For instance, consider an induction head task requiring attention to the previous token in Layer 1. Without an inductive bias discouraging attention (either through a regularizer, or implicitly through initialization), when tested on longer inputs, the attention head in Layer 1 could end up attending both to the preceding position and to some far-away position at distance $>n/2$, because the training data had no information about such distances.
>
>
> >  Also, the justification based on initialization making the regularizer small is not fully clear.
>
> We refer to our response under Point (2b): Proposition 54 shows that, if one randomly initializes transformers with increasing maximum context length $n$, the regularizer will, in expectation, stay bounded even as $n$ diverges.
>
> > (6b) What is the role of limit transformers and the regularizer in NoPE?
>
> We refer to our response to Point (1), where we explain that APE is substantially more powerful than NoPE. Technically, NoPE is a special case with $p_i \equiv 0$, $\phi(i,j) \equiv 0$ (lines 264-266). Indeed, in this special case, limit transformers are not needed to arrive at our length generalization guarantee. Limit transformers are used to treat APE, which is substantially more powerful than NoPE. We explicitly remark this in Appendix B.2.
>
>
> > (7) Why are the below-diagonal values of $\phi(i,j)$ not constrained?
>
>
> The values below the diagonal are irrelevant as we are considering only causally masked transformers (line 91), in line with standard LLMs and with Zhou et al 2024.
>
>
> >  (8)  Minor remark on line 326/327 Shouldn't the Q_a in the RHS be Q_a(j) and not Q_a(i)?
>
> Agreed, fixed.

---

> ### Author Response · Authors · 2024-11-21
> **Part 4**
>
> >  (9)  Notation remark on Theorem 9. In an unfortunate use of notation, u call Psi function local and Phi function periodic. Earlier u had used phi for local in the definition 3.
>
> Thanks, fixed.
>
> >  (10)  Since the results from the work hold for NoPE positional embeddings. From Theorem 2 in https://proceedings.neurips.cc/paper_files/paper/2023/file/4e85362c02172c0c6567ce593122d31c-Paper-Conference.pdf, the results should extend to relative positional encodings too?
>
> As discussed in our response to Point 1, the simulation of positional encoding in NoPE is limited in its capacity, at least compared to APE. We thus conjecture that, similarly, NoPE cannot fully simulate RPE. We believe that theoretical understanding of RPE, analogous to our results for APE, would take substantial additional technical work and is out of scope.
>
>
> > (11) What is the role of $N_0$? Would it be interesting to explain why, in practice, transformers can generalize well with modest training lengths?
>
> We agree that predicting and validating realistic bounds on $N_0$ is a very important next direction for research. Deriving an $N_0$ from our proof would give a valid and nonvacuous, but likely overly pessimistic estimate. More realistic bounds on $N_0$ will likely require advances in understanding SGD dynamics on transformers, which remains hard to understand, in particular in the multi-layer setup, needed for many of the functions in Figure 1.
>
>
> >    Concerns on the proof of Lemma 17:
>
> We thank the reviewer for the close reading, and have expanded the proof to clarify all these aspects.
>
> > a) In line 976 you say there are only a finitely many settings traversed by the limit transformer $\tilde{T}_i$. Why is this the case? Can you properly justify?
>
> Let $A := sup_i \mathcal{R}_\infty(\tilde{T}_i) < \infty$.
>
> The number of limit transformers $\tilde{T}$ with
> $R_\infty(\tilde{T}) \leq A$
> is finite except for the functions $\phi_{l,h}$, because $A$ bounds  (1) the number of parameters, (2) their magnitudes, (3) the precision at which they are represented.
> We make this explicit in line 993 of the new PDF.
>
>
> >    b) In equation 9, you say that we select an R(Tn) that is less than 1/n + inf (R(T)). In line 981, you argue that R(Tn) should converge because inf R(Tn) is bounded and monotonically increasing. This argument only tells that the RHS in equation (9) converges, which is the upper bound. Why does it imply that the LHS converges? You crucially use the existence of this limit in equation (14).
>
> First, note that
> $$R(T_n) \in [\frac{1}{n} + \inf_{T \in U_n} (R(T)), \inf_{T \in U_n} (R(T))]$$ Due to boundedness and monotonicity, $\inf_{T \in U_n} (R(T)))$ converges to a limit, say $\tilde{R}$.
> Since $1/n \rightarrow 0$, the width of the interval converges to 0. The Squeeze Theorem then implies that $R(T_n) \rightarrow \tilde{R}$. We make this explicit in line 1008.
>
>
> >    c) In line 995, you say that lim D_v_i(v_i) is D_0. Why does this limit exist?
>
> By definition of $R_-$,
> $$D_{\nu_i}(\nu_i) = R(T_{\nu_i}) - R_-(T_{\nu_i})$$ As both terms in the RHS converge, the LHS also has a limit.
> We make this more explicit in the text.
>
>
> >    d) Below equation 17, you state "As this function is monotonically increasing, and as phi_{l,h} has bounded precision, there must be $\tau_{\infty}$..." Why does this hold true? I don't quite follow.
>
>  As $D_\infty(\tau)$ is monotonically increasing and bounded, it converges.  Due to bounded precision, it only takes values in discrete steps (say, only multiples of $2^{-p}$ for some $p$); hence, it must attain the limit at some specific $\tau$, which we refer to as $\tau_\omega$.
> We have made this more explicit in line 1040.
>
> >    e) In line 1010-1017, you construct a sequence of T'n satisfying certain properties. Why does this have to exist? For instance why is $D_{\nu_{i(n)}}(n) = D_{\infty}(n)$ true?
>
> We have expanded (line 1042-1058 of the new PDF). The equality in question is $$D_{\nu_{i(n)}}(n) = \liminf_{j\rightarrow\infty} D_{\nu_j}(n) = D_\infty(n)$$ Regarding the first equality, such a $i(n)$ exists because $\phi_{l,h}$ has fixed precision, which entails that the $\lim\inf$ is attained infinitely often. Regarding the second equality, this is the definition of $D_\infty(n)$.
>
>
>
> >    f) The phi_l,h(i,j) in equation (10) should also bear the index n as it would be different for each limit transformer.
>
> Thanks for the suggestion. We now use a superscript to indicate this, e.g. $\phi_{l,h}^{(\tilde{T}_n)}(i,j)$ (line 989).
>
>
> > In the definition of  $D_n(\tau)$, how can $\tau$ exceed $n$?
>
> We now explicitly restrict summation to $\min(n,\tau)$.

---

> ### Author Response · Authors · 2024-11-21
> **Part 5**
>
> >    g) I do not follow the inequality in line 1044?
>
> This refers to
>
> $$ D_0 =   \limsup_{n\rightarrow \infty} D_{n}(n) \geq \limsup_{n\rightarrow \infty} D_{n}(\tau_\infty) + 2^{-2p} \geq \liminf_{i\rightarrow\infty} D_{\nu_i}(\tau_\infty) + 2^{-2p} = D_0+ 2^{-2p}$$
> The first inequality holds because
> $D_n(n) \geq D_n(\tau_\infty)$
> whenever $n \geq \tau_\infty$, simply because $D_n(\cdot)$ is monotonically increasing for each individual $n$.
>
> The second inequality holds because
> $\nu_1, \nu_2, \dots$
> is a subsequence of
> $1, 2, \dots$;
> hence a $\lim \sup ...$ over the larger sequence upper-bounds the $\lim \inf ...$ over the subsequence.
> We have made the steps more explicit in line 1091 of the new PDF.
>
> >    (12) In line 1049-1052, you say that set of functions traversed becomes stationary, what do u precisely mean here?
>
> We have made this more explicit and rephrased in line 1100-1107 of the revised PDF.
> It means that after $N_0$, all Limit Transformers traversed must be functionally equivalent to $f$.
>
> >  (13)  In line 1121 to 1127, how does the C logN fall out? Do you mean that since you partition stuff into N positions and that takes log N bits to compute, we get C log N?
>
> We have made this more explicit in lines 1210-1223 of the revised PDF.
> Alice can partition the positions into a constant number of set, and for each of them needs to transfer the number of positions in that partition.
>
>
>
> > (14) How does periodicity fall out in the limit transformer?
>
>
> This is an interesting observation. Indeed, periodicity falls out as a consequence of translation-invariance by Lemma 48: translation-invariant positional relations mediated by finite-rank matries are periodic. Hence, we are able to separate the positional relations in a sequence of transformers $T_n$ with bounded $\mathcal{R}(T_n)$ into local and periodic components. We now explain this better in Appendix A, point (7).
>
> > Why not have periodic positional encodings on the original transformer, instead of the offset independence condition?
>
> We refer to our discussion of offset independence and translation invariance independence above, where we show that offset independence is empirically and theoretically well-motivated. Our aim is to describe the behavior of general learnable APE encodings, as in our experiments and the closely related paper, Zhou et al 2024.

---

> > ### Author Response · Authors · 2024-11-21
> > **Part 6**
> >
> > We once again thank the reviewer for the close reading!
> >
> > Please let us know if there are any remaining questions.

---

> ### Author Response · Authors · 2024-11-23
>
> **Addendum:** We have uploaded a new draft version with an added Appendix G.7, in which we report further experiments confirming that **APE generalizes much better than NoPE** on a family of induction head-like tasks, in agreement with our theoretical predictions. On a family of induction head tasks, APE achieves ~99% accuracy in generalizing to 3 times the training length, whereas NoPE shows <5% there. This confirms our theoretical point from Appendix G.4 that the simulation of positional information in Kazemnejad et al 2024 is not powerful enough to allow NoPE to subsume APE. Taken together, we believe that our revision convincingly demonstrates why our theory needs to study the more general (and more complex) case of APE, rather than NoPE.

---

> > ### Comment · Reviewer_H3pw · 2024-11-26
> >
> > I thank the authors for their responses. I do think I understand most things. However, if I have to be honest the previous version was quite sloppy given all the changes. I will increase my rating but decrease my confidence as it is hard to verify all details thoroughly in the long draft.

---

> > > ### Author Response · Authors · 2024-11-26
> > >
> > > We thank the reviewer for their response. Thanks also for taking the time to write an extensive review, which helped us to considerably improve the draft.

---

### Official Review · Reviewer_odFA · 2024-11-03

**Soundness:** 3
**Presentation:** 2
**Contribution:** 3
**Rating:** 8
**Confidence:** 2

**Summary:**

**Update after rebuttal:**
The authors have clarified my questions and made improvements to the presentation of the main manuscript (though it remains a very long paper of course, with some important parts in the appendix, hence I will leave my 'Presentation' rating on 2). I am now more confident that the work is important and is ready to be presented and discussed with the wider ML community. I would now raise my score to a 7, but ICLR does not allow this score this year. I do think that some weaknesses remain, and that the paper does not quite hit an 8. But for the sake of expressing a clear opinion for the decision-making process, I am raising to an 8 (as the only available option), but will keep my confidence low to indirectly indicate that the score is a bit inflated.

---

This paper formally defines a function class, implicitly via the construction of the Limit Transformer, and shows that length generalization with transformers (of arbitrary context length) is provably guaranteed within this function class (this does not include SGD training though). The main part of the proof is the use of an “inference procedure” (Definition 6), which, informally, iterates through all transformers of increasing context size and eliminates the ones that do not fit the data. By construction, the set of functions that these transformers can implement (with increasing long context) is finite, such that eventually, at some finite context size, only the transformer that generalizes correctly to arbitrary length remains (all others have been ruled out by the data at this point; to be precise a complexity regularizer is also required to make the choice unique). This is the main argument in Theorem 7.2 (part of the main result). Keeping the parameters of transformers with increasing context width finite is central to the whole construction, and is reflected in the notions of PERIODIC and LOCAL of the Limit Transformer (informally: the functions implemented via attention only operate on a finite/small context window, and are translation invariant). Finally, the paper shows that a version of C-RASP (with either learned absolute positional encodings or no positional encodings) can be proven to be expressible via Limit Transformers, leading to the conjecture that length generalization with practical transformers is strongly related to whether a solution can be expressed in C-RASP or not. A small set of relevant experiments supports this conjecture well, including a (potential) explanation why some relatively simple regular languages cause trouble with length generalization (for which no C-RASP implementation provably exists).

**Strengths:**

* Very timely question. While frontier models show many surprising and unprecedented capabilities, they fail catastrophically on some really simple problems in length generalization. Understanding the underlying reasons is crucial for Safety and Reliability, and may also pave the way to address these issues in future-generation architectures.
* The expressivity result stating that all C-RASP programs can be expressed by a Limit Transformer, and are thus identifiable in theory via the inference process in Def. 6, is a strong result that bridges the theory to very concrete and empirically testable hypotheses.
* Empirical results show that despite a very elaborate construction of a limit process and complex regularizer, the theoretical predictions have actual practical consequences on standard transformers trained in a standard fashion (that differs significantly from the inference process in Def. 6)

**Weaknesses:**

* The paper is very extensive (72 pages, 62 are appendix, with 3887 lines in total; given the conference timelines and workload I could not review the appendix). The main paper is thus more a summary of the appendix than a standalone paper. While it makes no sense to have the main results and theorems without the proofs, maybe splitting the publication into a journal- / long-format theory paper and a separate more extensive empirical verification for a mainstream ML conference could be better.
* The construction of the Limit Transformer is quite elaborate (or rather the construction to go to the infinite context limit with non-exploding parameter sets / maintaining a finite function class) and the inference procedure in Def. 6 is completely impractical. Without any empirical results I would have been very skeptical about the practical relevance, and to be fully convinced I would like to see further results (though I believe Fig. 1 is significant and very promising). It is a bit unclear whether the theory had to be this complex and the connection to C-RASP dropped out as a lucky coincidence (which is how the paper is currently written), or whether putting C-RASP on a theoretical footing was the original goal that demanded this level of complexity.
* The inference procedure in Def. 6 seems a bit crude (though it does the job theoretically; same applies to the regularizer which is composed of 8 different complexity terms). The standard approach (e.g., in algorithmic information theory / Solomonoff Induction) is to not try to identify the correct function at some point $N_0$ but bound the overall number of mistakes (which is finite for any finite-length program). $N_0$ would be the point where the “last” mistake happens, which is generally unknown, and usually having bounds and generalization guarantees in terms of numbers of (remaining) mistakes are much tighter and shrink much faster with increasing number of observations. This is probably a question for future work, but it is unclear whether the restriction of using Def. 6 (having to identify the correct function) is implicitly limiting the function class where generalization is possible, and whether this class could be extended by focusing on a theoretical scheme that relies on bounding the mistakes (number of mistakes, and/or their cumulative magnitude).

**Verdict:**
Overall I am a bit ambivalent about the paper. On one hand it tackles a very timely and important problem by starting to make good progress from a theoretical angle (rather than adding even more contradicting experiments). On the other hand the current theory and presentation are quite complex and extensive. If the theory had been published in a journal / long-format paper before, and this paper would solely focus on empirically testing the conjecture that C-RASP expressiveness predicts length generalization, then a 10-page conference format seems like a great fit. Similarly, without the empirical results, I would have strongly doubted the practical relevance of the theory. But, the empirical results in Fig. 1 look very promising; though at this point it is unclear whether any results that contradict the main claims can easily be found or not. I do believe that the ML community needs more exposure to good theory, particularly at conferences, though I am not sure that this paper is the best example (due to its excessive length). I am also quite certain that this paper will spark quite a bit of follow-up work to make the theory simpler and/or more complete (expand the function class), and that publishing it will stir the community to conduct more empirical tests of the theory (which will make overall faster progress than asking the authors to perform more experiments). I am therefore currently slightly in favor of accepting the paper, though I would not be upset if others argue that ICLR may be the wrong venue. My confidence is currently on the low end - I did not have time to go through the extensive appendix, and there are a few bits and pieces of the main paper where I am not fully sure how they work out / will be proven. I am very happy to reconsider my opinion based on the other reviews and authors’ responses, and to make my criticism concrete, I leave some suggestions for improvement in the Questions section below.

**Questions:**

**Improvements:**
(I consider all of them optional suggestions, not strict requirements)
 1. Maybe give the reader a better sense of where this is going early in the paper. It should be clear early on that the paper constructs a function class for which generalization can be proven in the theoretical limit, but the theory does not answer how this relates to training actual transformers of fixed context length via SGD (i.e., there may be functions that can theoretically be proven to be length-generalizable, but this may not work in practice). Also state that this function class is likely not complete (i.e., there may be functions where transformers can length-generalize that lie outside this function class).
2. I really liked lines 255-264 in terms of clarifying the paper. Maybe the same information can be qualitatively given early in the paper to prime the reader.
3. Definition 2 can be a bit misleading - it informally may suggest that a Limit Transformer is basically “just a normal transformer with infinite context”. This needs to be clarified. The text already mentions that the Limit Transformer is a *theoretical* construct (maybe consider calling it a Limit Transformer Process or similar, to make sure that the object is not confused with a concrete architecture). The finite precision argument in Def. 2 is theoretically ok, but in practice it just says that the precision is an arbitrarily high natural number (with no upper bound given), which is not implementable “like a standard transformer”. Also state here or earlier in the paper that the Limit Transformer cannot be trained via SGD, but uses a theoretical procedure (Def. 6) that cannot be practically implemented. And finally, the functions $\phi_{l,h}$, whose complexity is not bounded as far as I can tell, do a lot of the heavy lifting and are not just marginal additions to a standard transformer.

**Minor questions**

1. L200-202: This is a very interesting requirement. Together with the requirement for periodicity and locality I am reminded of the pumping lemma for regular languages. But if I understand correctly (some experiments, Fig. 10, and discussion following L 486), C-RASP defines a subset of $TC^0$ and it covers some but not all regular languages and some simple non-regular languages.
2. Showing that C-RASP with a small extension can provably be expressed by a Limit Transformer is very interesting. Can C-RASP potentially still be extended without violating this equivalence, or is the current set of operations (likely) complete?
3. What is the relation of Theorem 7 to standard learnability / language identifiability results (language identifiability is generally not possible from positive examples only)?

---

> ### Author Response · Authors · 2024-11-21
> **Response**
>
> > (1) The paper is very long -- should it be multiple papers?
>
> Our paper makes progress on multiple but tightly interleaved   fronts, both establishing a length generalization guarantee in an idealized setup (Theorem 7), settling the (non)membership of many problems in C-RASP (Section 4 and Figure 1), and validating predictions empirically (Section 5). We believe that these theoretical and empirical components are tightly linked, and will be most convincing and impactful if presented together.
>
> > (2) The construction of the Limit Transformer is quite elaborate -- is this necessary?
>
> Our guiding question was to formalize when there exists a single APE transformer-like ''algorithm'' for a problem across input lengths, which Zhou et al 2024 described as a key intuition behind their RASP-L conjecture. Formalizing this led us to the notion of limit transformers, which allowed us both to derive a length generalization guarantee for an idealized inference procedure (Theorem 7), and a new result on the expressiveness of APE transformers (Corollary 26).
>
> > (3) How does the setup compare to bounding the number of errors, as in Solomonoff Induction?
>
> Thanks for pointing out the link to Solomonoff induction and similar settings. In fact, our setting is quite similar: In our setting, length generalization in the sense of ultimately converging on generalizing correctly when $n$ is large (Theorem 7) is *equivalent* to the number of mistakes being finite. We now make this explicit in Appendix G.5.
>
> > (4) Overall, is this paper too extensive and long for this conference?
>
> We would like to remark that ICLR has published papers with similar length. For instance, [1,2,3] have 66 to 74 pages. This applies similarly across Machine Learning conferences, and there have been highly influential papers longer than our submission, such as  [4] with 84 pages and [5] with 93 pages.
>
> [1] Panigrahi et al, Effect of activation functions on the training of overparametrized neural nets, ICLR 2020, https://openreview.net/pdf?id=rkgfdeBYvH
>
> [2] Li et al, Provable Memory Efficient Self-Play Algorithm for Model-free Reinforcement Learning, ICLR 2024, https://openreview.net/forum?id=vNiI3aGcE6
>
> [3] Li et al, Risk Bounds of Accelerated SGD for Overparameterized Linear Regression, ICLR 2024, https://openreview.net/forum?id=AcoXPIPh4A
>
> [4] Allen-Zhu et al, Learning and Generalization in Overparameterized Neural Networks, Going Beyond Two Layers, NeurIPS, https://arxiv.org/pdf/1811.04918
>
> [5] Bai et al,  Transformers as Statisticians: Provable In-Context Learning with In-Context Algorithm Selection, NeurIPS, https://arxiv.org/abs/2306.04637

---

> > ### Author Response · Authors · 2024-11-21
> > **Answers to Questions**
> >
> > ### Questions
> >
> > > Improvements: (I consider all of them optional suggestions, not strict requirements)
> >
> > > (1) Make more explicit that the theory applies in an idealized limit, and that the function class may not be complete
> >
> > We have made edits in Section 1 to make more explicit that the  theory applies to an idealized theoretical limit, and that the function class may not be complete.
> >
> > > (2) I really liked lines 255-264 in terms of clarifying the paper. Maybe the same information can be qualitatively given early in the paper to prime the reader.
> >
> > We are glad that this content (a high-level sketch of the proof of Theorem 7) is useful. We will add some of this to section 1.
> >
> > > (3) The paper should be clearer that Limit Transformers are just a mathematical construct
> >
> > We agree that it is an important point that Limit Transformers are just a mathematical construct, and are neither trained nor implemented. We have added an explicit statement at line 187.
> >
> > > (4) The functions $\phi_{l,h}$ can have unbounded complexity and do a lot of the heavy lifting.
> >
> > It is true that Definition 2 does not constrain $\phi_{l,h}$. However, Theorem 7 concerns Limit Transformers in which the functions $\phi_{l,h}$ are strongly constrained due to the requirements of translation-invariance and locality defined in Definition 3. Indeed, we could have introduced that constraint directly in Definition 2, but chose not to, in order to make the presentation modular, hoping this makes the reader's job easier.
> >
> > ### Minor questions
> >
> > > (1) Is it true that C-RASP defines a subset of $TC^0$ and covers some but not all regular languages and some simple non-regular languages?
> >
> > This is true. The ability to perform unbounded counting enables representing certain non-regular languages.
> >
> > > (2) Showing that C-RASP with a small extension can provably be expressed by a Limit Transformer is very interesting. Can C-RASP potentially still be extended without violating this equivalence, or is the current set of operations (likely) complete?
> >
> > This is an open question. We have not found functions that are expressed by Limit Transformers but not C-RASP[local, periodic]. We acknowledge this in point 6 of Appendix A.
> >
> > > (3) What is the relation of Theorem 7 to standard learnability / language identifiability results (language identifiability is generally not possible from positive examples only)?
> >
> > In applying Theorem 7 to formal languages (such as the 17 regular languages in Figure 1), we assume that the training data provides the set of possible next tokens at each positions (line 443). This implicitly includes negative examples, as negative examples are strings that at some point include a symbol that is impossible given prior context.

---

> > ### Comment · Reviewer_odFA · 2024-11-25
> > **Thank you for the clarifications and comments**
> >
> > I want to thank the authors for the clarifications and changes and improvements to the paper. I am happy with the authors' responses (though I still believe that very long papers at ML conferences are challenging, due to tight review periods with increased workloads, and should not become the norm; I do acknowledge though that ML has somewhat of a lack of "prestigious" journals, and that authors in general prefer to get a top-tier conference publication out of their work).
> >
> > [Question (4) ] Thank you for correcting me regarding the complexity of $\phi_{l,h}$.
> >
> > As I wrote in my original review, I do believe that the work is important and is ready to be presented and discussed with the wider ML community. I would now raise my score to a 7, but ICLR does not allow this score this year. I do think that some weaknesses remain, and that the paper does not quite hit an 8. But for the sake of expressing a clear opinion for the decision-making process, I am raising to an 8 (as the only available option), but will keep my confidence low to indirectly indicate that the score is a bit inflated.

---

> > > ### Author Response · Authors · 2024-11-26
> > >
> > > We thank the reviewer for their response, and are glad that they are happy with our response.

---

### Official Review · Reviewer_6ZTg · 2024-11-04

**Soundness:** 3
**Presentation:** 3
**Contribution:** 2
**Rating:** 6
**Confidence:** 3

**Summary:**

This paper develops a theoretical framework for transformer length generalization, introducing "Limit Transformers" and proving generalization guarantees for functions satisfying PERIODIC and LOCAL properties. They characterize functions that are identifiable with APE and NOPE, and validate the prediction from theory with experiments on algorithmic and formal language tasks.

**Strengths:**

1. The paper provides theoretical analysis of length generalization in transformers, including both sufficient conditions for generalization and communication complexity bounds showing certain functions cannot exhibit length generalization. This addresses an important open problem in theory. Also, it is the first to include length generalization theories with positional encodings to my best knowledge.
2. The theoretical framework successfully predicts empirical length generalization behavior across diverse tasks. The C-RASP formalism provides an interpretable way to determine whether a function should exhibit length generalization, making the theory practically useful.
3. The experimental evaluation is thorough with both algorithmic tasks and formal languages.

**Weaknesses:**

1. The main contribution compared to the previous C-RASP work and the RASP-L work is to extend the framwork to include positional encodings. However, the positional encodings considered are APE and NOPE, while the more frequently used encodings in practice are relative positional encodings. Also see question 1.
2. The limitation of the proposed theoretical framework is discussed, but it could benefit from more empirical evidence, like where the theory predicts length generalization but empirical performance is poor, or vice versa.

**Questions:**

1. Have you tried the performance of RPE although the current theory does not cover it yet? In [1], NOPE is shown to inherently learn some kind of relative positional encoding. Does RPE behave similarly to NOPE in certain tasks?
2. The model size provided in the appendix is highly dependent on the specific tasks (different tasks use different model depths, embedding space sizes, etc). Do you observe the sensitivity of the generalization performance in terms of model sizes? Or are there other reasons for this?

[1] The Impact of Positional Encoding on Length Generalization in Transformers
Amirhossein Kazemnejad, Inkit Padhi, Karthikeyan Natesan Ramamurthy, Payel Das, Siva Reddy

---

> ### Author Response · Authors · 2024-11-21
> **Response**
>
> > (1) The paper focuses on APE and NoPE, whereas many transformers use relative encodings (RPE)
>
> We closely follow Zhou et al 2024 in using APE, as our aim was to formalize the RASP-L conjecture. We were also motivated by [1], which found NoPE to be empirically competitive with relative positional encodings in various algorithmic problems. We believe that extending our theoretical study to relative positional encodings is an exciting next step for future research, as we also mention in Section 6.
>
>
> [1] Kazemnejad, Amirhossein, et al. "The impact of positional encoding on length generalization in transformers."
>
>
> > (2) The limitation of the proposed theoretical framework is discussed, but it could benefit from more empirical evidence, like where the theory predicts length generalization but empirical performance is poor, or vice versa.
>
> We have not been able to find examples where the theory predicts length generalization but empirical performance is poor, or vice versa. Any such examples would of course be highly interesting for further refining our theory.
>
> ### Questions:
>
> > (1) Have you tried the performance of RPE?
>
> As described above (and in Section 6), we agree that expanding our theoretical treatment to RPE is a very interesting question. While NoPE can simulate some amount of positional encoding as shown by [1], this is strictly weaker than what APE can do, as we explain in Appendix G.4. We conjecture that the same applies to RPE, and that NoPE may not subsume general RPE. Hence, we believe treating RPE theoretically will require substantial additional technical work, out of scope for the present paper.
>
> > (2) Why are the model sizes in the experiments task-dependent?
>
>
> There may be a minimum model size necessary in order to solve a particular algorithmic task. We conjecture that there is a link between model size and the formula complexity (e.g., length, nesting depth) in C-RASP. Zhou et al conjectured that shorter programs are easier to learn, and we thus conjecture that C-RASP complexity could also be used to predict minimum model size. As proving bounds on the smallest program or formula representing a problem is generally nontrivial, we leave rigorous exploration of this idea to future work.

---

> > ### Comment · Reviewer_6ZTg · 2024-11-26
> >
> > Thank you for the response. I have read it and the responses to other reviewers and will keep my score.

---

### Official Review · Reviewer_Vn9i · 2024-11-04

**Soundness:** 4
**Presentation:** 3
**Contribution:** 4
**Rating:** 8
**Confidence:** 3

**Summary:**

The paper studies the length generalization problem in decoder-only Transformers, which refers to the inability of models to deal with longer samples than encountered during the training phase. The paper aims to identify which tasks can achieve length generalization. To this end, the authors introduce Limit Transformer, a theoretical model designed to generalize across varying sequence lengths. Importantly, the authors prove that under an idealized inference procedure, any tasks that can be expressed by Limit Transformer, satisfying Periodic and Local constraints, can provably achieve length generalization. Furthermore, the authors show that any C-RASP programs can be expressed by Limit Transformer, leading to the conclusion that such tasks can also generalize to longer inputs. On the other hand, the paper presents that copying and addition cannot be generalized as these tasks do not exhibit logarithmic communication complexity. Finally, the paper provides experimental results, demonstrating that tasks expressible by C-RASP programs (binary majority, sort, copy unique) easily length-generalize while tasks outside C-RASP (copy repeat, parity, addition) fail to achieve length generalization.

**Strengths:**

- The paper addresses an important question in the literature: which tasks can achieve length generalize and which cannot. Prior work on length generalization has mainly focused on improving empirical performance, with fewer studies providing a theoretical understanding. While [1] introduces RASP-L to identify tasks that can achieve length generalize, their argument still remains conjectural. This study goes one step further than the previous RASP-L conjecture, offering a theoretical framework to determine whether certain tasks will provably achieve length generalization or not. Therefore, I believe this paper makes a significant contribution to the literature.
- The FAQ section in the appendix effectively conveys the paper's intuition and enhances the reader’s understanding.
- Overall, the paper is well-structured and provides a detailed analysis to present their findings.

[1] Zhou, Hattie, et al. "What algorithms can transformers learn? a study in length generalization."

**Weaknesses:**

I don’t see any significant weaknesses in the paper, but there are a few minor limitations, which I don’t consider critical issues in evaluating this paper.

- The scope of the paper is limited to algorithmic tasks and does not cover length generalization problems in natural language processing tasks.
- As explained in the paper, a key assumption in the framework is the idealized inference procedure, which assumes that we can obtain Transformers that are fitted to reproduce a target function while minimizing a specific regularizer $R(T)$, and thus the current framework is not fully end-to-end. Introducing an analysis of training dynamics to replace this assumption would be a promising direction, achieving a truly end-to-end, complete argument.

**Questions:**

- In Figure 1, why do Transformers with NoPE fail even on in-distribution samples (Bin 1) for Copy Unique and Addition? Doesn’t this contradict the observations in [2], which argue that NoPE can length-generalize for these tasks to some extent?

[2] Kazemnejad, Amirhossein, et al. "The impact of positional encoding on length generalization in transformers."

---

> ### Author Response · Authors · 2024-11-21
> **Response**
>
> > I don’t see any significant weaknesses in the paper, but there are a few minor limitations, which I don’t consider critical issues in evaluating this paper.
>
> We thank the reviewer for the positive assessment.
>
> > (1) The scope of the paper is limited to algorithmic tasks and does not cover length generalization problems in natural language processing tasks.
>
> We agree with this, and now make this explicit throughout Section 1. We note that prior empirical work on length generalization (whose results our work aims to give a theoretical foundation to) has largely focused on algorithmic tasks. Expanding to naturalistic language comprehension tasks is an interesting future direction.
>
>
> > (2)  Analysis of training dynamics, instead of an idealized inference procedure, would be desirable.
>
>
> We agree with this limitation, which we acknowledge in Section 6.
>
> ### Questions:
>
> > (1) In Figure 1, why do Transformers with NoPE fail even on in-distribution samples (Bin 1) for Copy Unique and Addition? Doesn’t this contradict the observations in [2], which argue that NoPE can length-generalize for these tasks to some extent?
>
>
> In Copy Unique and Addition (Figure 1), the NoPE results are represented by the red dotted curves, which do start out at 100% in Bin 1 in these tasks [dotted=NoPE, red=not in C-RASP]. Thus, on these tasks, NoPE *does* succeed on in-distribution samples. Please let us know if we should make some aspect of the figure or caption clearer to prevent misunderstanding.
>
> We would also like to point out two differences with [2]. First, [2] report results for Addition on heldout lengths 8-16 (their Figure F.5) , much shorter than our heldout bins 100 and 150. Second, [2] report three "Copy" tasks, where NoPE shows length generalization from length 20 to length 40 only in the first two variants (their Figure F.4), which require matching only the length of the string, but not its character sequence. On their third variant, NoPE does not generalize from length 20 to 40 at all.
>
> [2] Kazemnejad, Amirhossein, et al. "The impact of positional encoding on length generalization in transformers."

---

> > ### Comment · Reviewer_Vn9i · 2024-12-01
> >
> > Sorry for the late comment and thank you for the response. During the extended rebuttal period, I read the paper more carefully and realized that I had some misunderstandings earlier (including the stupid question about Figure 1). I want to ask a few additional questions for further clarification.
> >
> > Questions:
> > - I am slightly confused about the statement in Definition 6. Shouldn't each $T_i$ be an element of $\Theta_i$, not $\Theta_n$? Based on the explanation stated between Lines 228 to 231 of the revised version, I guess that $T_1$ is an element of $\Theta_1$. Furthermore, following Definition 4, it seems that $\Theta_n \subset \Theta_1$ holds, which would imply $T_1 \notin \Theta_n$.
> > - I might have missed this, but can any Limit Transformer be translated into a standard Transformer? If so, can you outline the method?
> > - For a task that is not expressible by the Limit Transformer satisfying Periodic and Local (e.g., n-digit addition), the authors state that any run of the Idealized Inference Procedure will result in a sequence of Transformers with bad properties (Lines 418 to 421). Does this imply that "for Transformers with fixed depth, number of heads, parameter norms, ranks, MLP dimensions, and precision p, there does not exist a length-generalizable solution for n-digit addition"?
> >
> > Minor suggestion for improving clarity:
> > - line 186: R(T) is referenced before it is formally defined.
> >
> > I would appreciate it if the authors provide response for them, but as the rebuttal deadline is approaching, concise answers would be perfectly fine.

---

> ### Author Response · Authors · 2024-12-01
>
> We thank the reviewer for the close reading and further questions. As the rebuttal deadline is approach, we for now provide concise answers for the questions:
>
> > I am slightly confused about the statement in Definition 6. Shouldn't each $T_i$ be an element of $\Theta_i$, not $\Theta_n$?? Based on the explanation stated between Lines 228 to 231 of the revised version, I guess that $T_1$ is an element of $\Theta_1$ . Furthermore, following Definition 4, it seems that $\Theta_n \subset \Theta_1$ holds, which would imply $T_1 \not\in \Theta_n$.
>
> Thanks for the close reading. The phrasing in Definition 6 is indeed not optimal.
> As the reviewer inferred, we intend to say that $T_1 \in \Theta_1$, $T_2 \in \Theta_2$, $T_3 \in \Theta_3$, etc.
> We will fix this in our next version.
> We believe that this addresses the reviewer's question.
>
> >  I might have missed this, but can any Limit Transformer be translated into a standard Transformer? If so, can you outline the method?
>
> Yes, such a translation can be carried out for any finite context length N of the resulting standard transformer, and importantly the translation is uniform in the sense that $\sup_N R(T_N) < \infty$, even though the width of the translation needs to increase. This is formalized in Lemma 47. The outline of the method is as follows; a discussion is also provided in lines 2724-2751 (page 51):
> * Intuitively, the positional encodings of $T_N$ consist of those of the Limit Transformer, concatenated with one-hot vectors for the positions from 1 to N. The overall width is thus $d+N$.
> * Token embeddings just consist of those of the Limit Transformer, concatenated with the N-dimensional zero vector.
> * For each attention head, the $K^T Q$ matrices encode both those of the Limit Transformer (in the first $d$ dimensions) and the $\phi_{l,h}$ functions (in the final $N$ dimensions).
> * Complications arise from the fact that this construction (i) does not ensure translation invariance, (ii) does not keep the regularizer in Eq. (8) bounded. This is solved by (i) attending to SOS and computing position relative to that to ensure translation invariance, (ii) routing some of the positional information through the attention mechanism, effectively making it invisible to Eq. (8). This makes the construction more complex, but allows us to prove that translation invariance and  a bounded value for Eq. (8) can indeed be achieved, which in turn is an important insight leading to Theorem 7. This is summarized in lines 2724-2751.
>
> We will add this high-level explanation before the proof of Lemma 47 in the next version.
>
> > For a task that is not expressible by the Limit Transformer satisfying Periodic and Local (e.g., n-digit addition), the authors state that any run of the Idealized Inference Procedure will result in a sequence of Transformers with bad properties (Lines 418 to 421). Does this imply that "for Transformers with fixed depth, number of heads, parameter norms, ranks, MLP dimensions, and precision p, there does not exist a length-generalizable solution for n-digit addition"?
>
> We believe that this is an accurate statement: intuitively, if such a length-generalizable solution existed, the Idealized Inference Procedure should find it; hence, it cannot exist. We will consider adding an explicit statement and proof in the next version; one thing to take care of is to show that this conclusion holds when bounding depth, number of heads, parameter norms, ranks, MLP dimensions, and precision p, but not necessary explicitly bounding Eq. (8).
>
> >  line 186: R(T) is referenced before it is formally defined.
>
> Thanks for catching this. We will fix this in the next version.

---

> > ### Comment · Reviewer_Vn9i · 2024-12-02
> >
> > Thanks for the prompt response. Your response helped improve my understanding of the paper.
> >
> > I believe this is a good, strong paper for the following reasons:
> > - In prior literature, the feasibility of length generalization for a given task when trained on Transformer could only be roughly conjectured based on the RASP-L hypothesis. This paper makes a significant advancement by proposing a formal criterion to determine whether length generalization is possible or impossible (under the assumption of an "idealized inference procedure").
> > - The paper "almost" proves that we cannot expect length generalization for n-digit addition and copy with duplicate tokens when using APE or NoPE. This result is quite impressive, as it aligns well with trends observed in recent empirical studies, which have shifted away from APE and instead focused on developing specialized input format [1, 2] or novel position embedding method [3, 4].
> >
> > One concern is that, I think the explanation of the Limit Transformer (Section 3.1) is somewhat abstract and difficult to grasp the underlying intuition. The later sections of the paper (section 3.2 and onward) are okay. Maybe the authors can provide additional explanation or example in Section 3.1.
> >
> > Overall, I will maintain my score.
> >
> > ---
> > References
> >
> > [1] Zhou, Hattie, et al. "What algorithms can transformers learn? a study in length generalization." ICLR 2024
> >
> > [2] Zhou, Yongchao, et al. "Transformers can achieve length generalization but not robustly." arXiv preprint arXiv:2402.09371 (2024).
> >
> > [3] McLeish, Sean, et al. "Transformers Can Do Arithmetic with the Right Embeddings." NeurIPS 2024
> >
> > [4] Cho, Hanseul, et al. "Position Coupling: Leveraging Task Structure for Improved Length Generalization of Transformers." NeurIPS 2024

---

> > > ### Author Response · Authors · 2024-12-02
> > >
> > > We thank the reviewer for their feedback and the close reading of the paper. We agree about the link to recent empirical observations regarding addition and copying, and will do our best to increase intuition in Section 3.1.

---

### Official Review · Reviewer_Gpiw · 2024-11-08

**Soundness:** 3
**Presentation:** 2
**Contribution:** 2
**Rating:** 6
**Confidence:** 2

**Summary:**

This work studies the challenge of enabling transformers to generalize to sequences longer than those seen during training. This capability is often inconsistent across tasks and lacking strong theoretical understanding. The authors present a new theoretical framework to examine length generalization in causal-attention transformers with learnable positional encodings. Using this framework, they analyze the conditions under which transformers can generalize to longer sequences under certain conditions. The theoretical findings are further validated through empirical tests on a set of tasks.

**Strengths:**

* Length generalization is a very important problem that is being explored from various perspectives, though primarily through algorithmic and empirical studies, with relatively few theoretical analyses. This work takes a different approach, offering distinct advantages.
* It's encouraging to see that, under certain conditions (as shown in Theorem 7), they identify cases where length generalization is achievable.
* Provides several formalizations (e.g. Limit Transformer) and proofs on this topic that would be valuable for future research.

**Weaknesses:**

* The paper’s current writing seems to focus more on demonstrating how to achieve length generalization under specific conditions than on examining the limitations of existing models. For example, it centers on proving Theorem 7 by introducing various modeling assumptions, such as the Limit Transformer and specific inference conditions (like the proposed regularizer). This approach differs somewhat from the expectations set in the introduction, which suggests a more interpretive analysis of current models.
* The practical impact of this paper feels somewhat limited. It would be valuable if the authors could identify practical applications that leverage their findings.
* C-RASP and formal languages are quite distinct from traditional text corpora or other common scenarios. Even though the aim is to theorize the formalism in this field, there seems to be a significant gap between the formalism presented in this paper and that of other widely used settings.

**Questions:**

* Could the authors more directly relate their theorem to widely used transformer models? For example, which specific conditions do not hold in these models, and how does this align with the empirical observations in Section 5?
* The FAQ in Appendix A actually was quite useful providing valuable insights and gives readers motivation. Could the authors consider integrating some of these discussions more naturally into the introduction or relevant sections?

---

> ### Author Response · Authors · 2024-11-21
> **Response**
>
> Dear reviewer Gpiw,
>
> Thanks for your feedback on our paper!
>
> ### Reply Regarding Weaknesses
>
> > (1) The paper does not include interpretive analysis of existing models
>
> We would like to clarify that our paper focuses on a theoretical framework establishing *general criteria* for when length generalization is achievable when training transformers on some problem, and leave interpretive analysis of existing *specific models* out of scope. We have rephrased the introduction to make explicit that we are interested in the setting where transformers are specifically trained on short inputs from some task.
>
>
> > (2) What is the practical impact of this paper?
>
> We believe that a theoretical understanding explaining empirical findings (Zhou et al. 2024) can improve current and future applications in NLP. A possible practical consequence is deriving new positional encoding or scratchpad schemes using our theoretical insights to enable stronger length-generalization. For instance, RASP and RASP-L have been used to derive practical advances in the design of transformers for various problems, e.g. [1,2]. Additionally, our framework may be used for more refined methods of interpretability due to the formal connection between our RASP variant and length-generalizing transformers - for instance, decompilation of transformer weights into RASP programs [4]. Hence, we believe our theoretical framework is highly valuable for practical work to build upon.
>
> > (3) C-RASP and formal languages are quite distinct from common scenarios
>
> As described in our response to (2), RASP and RASP-L have had substantial impact in research on transformers. Formal languages are a principled way to model sequences of unbounded length - crucial for understanding length generalization. They provide a formal way to analyze the capabilities of transformers to perform practically relevant tasks, such as induction heads (which we have analyzed in this paper). In fact, formal languages have also been used to validate new positional encoding schemes [3].
>
> [1] Fan et al, Looped Transformers for Length Generalization, 2024. https://arxiv.org/abs/2409.15647
>
> [2] Hou et al,
> Universal Length Generalization with Turing Programs, 2024. https://arxiv.org/abs/2407.03310
>
> [3] Ruoss et al, Randomized Positional Encodings Boost Length Generalization of Transformers, 2023. https://arxiv.org/pdf/2305.16843
>
> [4] Friedman et al, Learning Transformer Programs, 2023. https://arxiv.org/abs/2306.01128
>
> ### Reply Regarding Questions
>
> > (1) How does the theory relate to widely used transformer models?
>
> As we describe above, our study focuses on general properties of the transformer architecture. We empirically test the predictions by training models on various problems from scratch in Section 5. We expect that the results also have implications for the algorithmic abilities of commonly used specific models (such as GPT-3 or LLaMa-3), but more detailed exploration of this to future work.
>
> > (2) Can more content from the FAQ be included in the main text?
>
> We are glad to hear that the FAQ is useful. We have made some changes to the main text to make the relevant information (e.g., about the role of limit transformers) more salient.

---

> > ### Comment · Reviewer_Gpiw · 2024-12-02
> >
> > Thank you for your response. I have reviewed it, along with the replies to other reviewers, particularly H3pw, and I will be keeping my score as it is.

---

### Author Response · Authors · 2024-11-23
**Global Response**

We thank all reviewers for their overall positive assessment, extensive comments and constructive criticism. The detailed feedback on both the work's presentation and technical details have allowed us to make the paper more readable and the results more sound. We summarize the reviewers' major concerns and list the changes we have made to address them.

We have uploaded a revised PDF, with changes highlighted in **blue**.

> The treatment of positional encodings produces seemingly contradictory results with Kazemnejad et al. 2023

The cited paper shows that NoPE transformers can simulate APE transformers up to a large input size. Taken at face value this suggests we don't need limit transformers to prove guarantees for APE transformers. However, there is an important distinction: the parameters in a transformer must increase rapidly to perform the cited simulation on larger inputs. Because of this, our theory predicts that NoPE transformers will length-generalize worse than APE transformers on certain tasks. Indeed, our experimental results confirm that APE transformers can length-generalize on several problems that NoPE transformers do not. Furthermore, the experimental setup in the cited paper differs notably from ours. We have written a treatment of this issue in Appendix G.4, as we believe it is quite important (and thank the reviewers for pointing it out). We also provide further experimental results confirming that APE generalizes better than NoPE in agreement with our theory, and even when performing much broader hyperparameter search for NoPE (Appendix G.7): On a family of induction head tasks, APE achieves ~99% accuracy in generalizing to 3 times the training length, whereas NoPE shows <5% there.

> The requirement of translation-invariant product functions is unrealistic

It is true that standard training of transformers does not enforce translation invariance in the product functions. However, we can *prove theoretically* that translation invariance is helpful for length-generalization (on induction heads for instance), and *show empirically* that length-generalizing transformers often learn translation invariant product functions in practice (both in transformers trained from scratch and in GPT-2). As such, we suggest translation-invariance is favored by standard training even though it may not be explicitly enforced. We have written a new section on this in Appendix G.6.

> The paper is too elaborate

Because theoretical understanding of length generalization of transformers is still in its infancy, building a formal framework for it requires the synthesis of many different ideas. We believe that presenting our simultaneous progress in multiple areas - both theoretical and empirical - is necessary to form a solid theoretical framework. In order to make the paper more approachable, we have rewritten the introduction and changed prose throughout the body of the paper.

> The practical impact of this paper is not clear

Indeed, our paper focuses on a theoretical framework explaining the length-generalization on algorithmic problems (rather than analyzing transformers in practice on natural language tasks). Nevertheless, we believe that a solid theoretical understanding can enable future work to have a more immediate practical impact. We have given pointers towards potential avenues for this (new positional encodings, improved interpretability) and modified our introduction to make the limitations of our setting clearer.

Overall, we thank the reviewers for their constructive criticism and support.

---

### Meta-Review · Area_Chair_2iVk · 2024-12-20

**Metareview:**

Summary:
The paper introduces a theoretical framework analyzing length generalization in causal transformers with learnable absolute positional encodings. The authors introduce "Limit Transformer" to handle growing parameters and show that C-RASP programs (under specified periodic and local constraints) can be translated into Limit Transformers, providing a concrete class of problems where length generalization is guaranteed. The theoretical findings are supported by empirical experiments demonstrating that tasks expressible by C-RASP programs succeed in length generalization, while others, such as copying with repetitions and n-digit addition, fail.

Strengths:
- Addresses an important open problem in transformer theory - length generalization capabilities
- Provides rigorous theoretical analysis with concrete proofs and guarantees
- Formalizes and proves aspects of the previously conjectural RASP-L hypothesis
- Strong empirical validation across diverse tasks showing alignment between theory and practice

Weakness:
- Treatment is limited to absolute positional encodings rather than more popular schemes
- The theoretical setup makes some idealized assumptions that do not reflect practice
- Focus is primarily on algorithmic tasks rather than natural language

Decision:
All the reviewers were in consensus that the paper represents an important step forward in understanding transformer capabilities and limitations, providing both theoretical insights and practical implications for future work. It merits publication at ICLR 2025.

**Additional Comments On Reviewer Discussion:**

We thank the authors and reviewers for engaging during the discussion phase towards improving the paper. Below are some of the highlights:

1. NoPE vs APE encodings contradiction:
- Reviewers questioned why NoPE couldn't subsume APE based on prior work and thus alleviate the need for "Limit Transformers"
- Authors clarified with new experiments showing APE's superior performance
- Added Appendix G.7 showing concrete advantages of APE over NoPE
- Convincingly resolved through theoretical and empirical evidence

2. Translation invariance assumptions:
- Concerns about restrictiveness of translation invariance requirement
- Authors added Appendix G.6 showing it emerges naturally in practice
- Demonstrated both theoretical benefits and empirical validation
- Adequately justified the assumption

3. Paper complexity and presentation:
- Reviewers noted elaborate construction and long technical details
- Authors improved introduction and added clarifying content
- While still complex, changes made paper more approachable
- Justified length as necessary for complete treatment

4. Practical impact:
- Questions about real-world applicability
- Authors clarified theoretical foundations enabling future practical work
- Added discussion of potential applications
- Acknowledged limitations while highlighting value for future work

---

### Decision · Program_Chairs · 2025-01-22

Accept (Poster)